# On the Provable Suboptimality of Momentum SGD in Nonstationary Stochastic Optimization

Sharan Sahu [* 1]  Cameron J. Hogan [* 1]  Martin T. Wells [1]

## Abstract

In this paper, we provide a comprehensive theoretical analysis of Stochastic Gradient Descent (SGD) and its momentum variants (Polyak Heavy-Ball and Nesterov) for tracking time-varying optima under strong convexity and smoothness. Our finite-time bounds reveal a sharp decomposition of tracking error into transient, noise-induced, and drift-induced components. This decomposition exposes a fundamental trade-off: while momentum is often used as a gradient-smoothing heuristic, under distribution shift it incurs an explicit drift-amplification penalty that diverges as the momentum parameter $\beta$ approaches 1, yielding systematic tracking lag. We complement these upper bounds with minimax lower bounds under gradient-variation constraints, proving this momentum-induced tracking penalty is not an analytical artifact but an information-theoretic barrier: in drift-dominated regimes, momentum is unavoidably worse because stale-gradient averaging forces systematic lag. Our results provide theoretical grounding for the empirical instability of momentum in dynamic settings and precisely delineate regime boundaries where vanilla SGD provably outperforms its accelerated counterparts.

## 1. Introduction

Consider the optimization problem posed by a strongly convex objective function $G_t : \mathbb{R}^d \to \mathbb{R}$ defined as follows:

$$\boldsymbol{\theta}_t^\star \in \underset{\boldsymbol{\theta} \in \mathbb{R}^d}{\arg\min}\, G_t(\boldsymbol{\theta}), \quad G_t(\boldsymbol{\theta}) = \mathbb{E}_{X_t \sim \Pi_t}[g(\boldsymbol{\theta}, X_t)] \quad \text{(Opt)}$$

Here $g(\boldsymbol{\theta}, X_t)$ is a noise perturbed measurement of $G_t(\boldsymbol{\theta})$ and $X_t$ is a random variable sampled from a time-varying distribution $\Pi_t$. These settings naturally connect to classical stochastic tracking in signal processing (Kushner & Yin, 1997; Sayed, 2003) and concept drift in online learning (Hazan, 2016; Cutler et al., 2021), where the goal is not to converge to a single static solution, but to track an unknown, time-varying sequence of minimizers as closely as possible.

The primary algorithmic tool for this tracking task, provided that $g$ is differentiable, is Stochastic Gradient Descent (SGD) (Robbins & Monro, 1951). Due to their simplicity, scalability for big data and robustness to noise and uncertainty, stochastic gradient methods have become popular in large-scale optimization, machine learning, and data mining applications (Zhang, 2004; Cevher et al., 2014; Szegedy et al., 2015).

Developing methods to accelerate the convergence of SGD has been studied extensively. Momentum methods, such as Polyak's Heavy-Ball (HB) (Polyak, 1963) and Nesterov acceleration (NAG) (Nesterov, 1983), are ubiquitous in deep learning and are widely believed to reduce gradient noise and accelerate training (Sutskever et al., 2013). SGD with momentum (SGDM) is widely used in practice due to its potential benefits in convergence speed and generalization performance in certain settings, though under broad smoothness assumptions it achieves similar convergence rates to standard SGD (Liu et al., 2020c). However, SGDM has been shown to provide provable advantages in specific scenarios (Cutkosky & Mehta, 2020; Liu et al., 2020b; Sato & Iiduka, 2024; Ramezani-Kebrya et al., 2024). However, this notion of acceleration becomes nuanced in nonstationary regimes: when the data distribution $\Pi_t$ drifts, past gradients may become systematically *stale*, so the same temporal averaging that suppresses noise can induce tracking lag and even instability, as observed empirically in online forecasting and suggested by prior steady-state analyses in the slow-adaptation regime.

These observations motivate the following central question in the nonstationary tracking setting:

*When does momentum help under distribution shift and when does it provably hurt for strongly convex, smooth functions $G_t(\boldsymbol{\theta})$?*

**Our contributions.** We answer this question for $\mu$–strongly

*Equal contribution  [1]Department of Statistics and Data Science, Cornell University, Ithaca, NY, USA. Correspondence to: Sharan Sahu <ss4329@cornell.edu>, Cameron J. Hogan <cjh337@cornell.edu>, Martin T. Wells <mtw1@cornell.edu>.

*Proceedings of the 43rd International Conference on Machine Learning*, Seoul, South Korea. PMLR 306, 2026. Copyright 2026 by the author(s).

convex, $L$–smooth risks under predictable distribution shift and make the following advances:

1. **Finite-time tracking with an explicit momentum penalty.** We prove finite-time tracking bounds that separate three terms: initialization decay, a noise floor, and drift-induced tracking lag. Extending to Heavy-Ball and Nesterov yields the same decomposition, but with explicit momentum penalties—large $\beta$ amplifies initialization effects, inflates the noise floor, and imposes stricter stability constraints, particularly in ill-conditioned settings.

2. **Time-resolved high-probability bounds and a drift–noise coupling.** Unlike prior results based on a single worst-case drift parameter, our guarantees are time-resolved, with exponential forgetting ensuring only recent drift matters. Our analysis also reveals a drift–noise interaction whereby tracking mismatch amplifies stochastic fluctuations, motivating restart and windowing schemes that discard stale history.

3. **Minimax lower bounds and an information-theoretic inertia window.** We establish minimax dynamic-regret lower bounds under a gradient-variation budget with an explicit $\beta$-dependent term, yielding an unavoidable *inertia window*: for large momentum, SGDM must lag a drifting optimizer for a nontrivial period regardless of tuning. In the uniformly spread drift regime, our tuned upper bound matches the minimax rate and exhibits the sharp momentum penalty.

4. **Experiments corroborating the regime split.** On drifting quadratics, drifting linear/logistic regression, and a drifting teacher–student MLP, our results match the theory: increasing nonstationarity, momentum, or ill-conditioning reliably worsens HB/NAG tracking, consistent with an inertia-limited lag under drift, while SGD remains comparatively robust across noise levels.

### 1.1. Related Work

Momentum methods—including Polyak's Heavy-Ball (Polyak, 1963) and Nesterov acceleration (Nesterov, 1983)—are ubiquitous in deep learning, where they are widely believed to reduce gradient noise and accelerate training (Sutskever et al., 2013). Indeed, for a variety of strongly convex and nonconvex settings, SGD with momentum provides faster convergence and better generalization than vanilla SGD (Cutkosky & Mehta, 2020; Liu et al., 2020b; Sato & Iiduka, 2024; Ramezani-Kebrya et al., 2024). However, momentum does not universally improve training and can degrade performance in online and adaptive settings (Fu et al., 2023; Wang et al., 2024; Deng et al., 2024). In nonstationary environments, past gradients become stale and may point in the wrong direction, motivating forgetting or discounting of historical information (Zhang et al., 2020). Empirically, momentum can also reduce stability at large learning rates (Aydore et al., 2019). Recent theoretical work reinforces these concerns: in high-dimensional or slow-adaptation regimes, momentum can amplify noise, undermine stability, or collapse to SGD with a rescaled step size (Yuan et al., 2016; Deng et al., 2024; Jagannath et al., 2025).

These findings connect to a broader literature on optimization in time-dependent and nonstationary environments. In stationary settings with constant stepsizes, momentum is essentially equivalent at steady state to SGD with a rescaled learning rate and does not reduce the MSE floor (Yuan et al., 2016; Kovachki & Stuart, 2021). Other work addresses non-stationarity from complementary perspectives: robust and adaptive methods for corrupted or drifting data (Yang et al., 2017; Galashov et al., 2024); non-asymptotic tracking and regret guarantees under time drift or dependent data (Cao et al., 2019; Cutler et al., 2023; Zhao et al., 2024; Shen et al., 2026); and stability analyses of momentum for time-varying objectives (Vidyasagar, 2025). Despite this progress, the interaction between momentum and nonstationarity remains significantly understudied—a gap this paper aims to address. To provide the necessary context, we survey the relevant literature on momentum and acceleration in nonstationary stochastic optimization more extensively in Appendix A.

## 2. Preliminaries

We first summarize the notation used throughout the paper. Scalars, vectors, and matrices are denoted by lowercase, bold lowercase, and bold uppercase letters, respectively; calligraphic letters denote sets, operators, or $\sigma$-algebras. For $\boldsymbol{x} \in \mathbb{R}^d$, $\|\boldsymbol{x}\|$ is the $\ell_2$ norm and $\langle \boldsymbol{x}, \boldsymbol{x}' \rangle$ the inner product. We write $a_m = \mathcal{O}(b_m)$ if $a_m \leq Cb_m$, $a_m = \Omega(b_m)$ if $a_m \geq Cb_m$, and $a_m = \Theta(b_m)$ if both hold, for some $C > 0$. The notation $a_m \lesssim b_m$ (resp. $\gtrsim$, $\asymp$) indicates inequality up to constants independent of $m$ and problem parameters. For $f : \mathbb{R}^d \to \mathbb{R}$, we write $\nabla f$ for the gradient, $\min f$ for the minimum value, and $\boldsymbol{x}^*$ for the minimizer. We use $\mathbb{E}[\cdot]$ for expectation and $\mathcal{F}_t = \sigma(X_0, \ldots, X_t)$ for the natural filtration.

We next introduce conditional $\Psi_\alpha$–Orlicz norms, which we will use throughout our analysis to control random quantities in a filtration-adapted (i.e., history-dependent) manner. Fix $\alpha \geq 1$. For a real-valued random variable $X$, recall the (unconditional) $\Psi_\alpha$–Orlicz norm

$$\|X\|_{\Psi_\alpha} := \inf \left\{ u > 0 : \mathbb{E} \exp \left( (|X|/u)^\alpha \right) \leq 2 \right\}.$$

Given a $\sigma$-algebra $\mathcal{F}$, we write $\|X\|_{\Psi_\alpha | \mathcal{F}} \leq K_\mathcal{F}$ for some

$\mathcal{F}$-measurable $K_{\mathcal{F}} > 0$ if $\mathbb{E}\big[\exp\big((|X|/K_{\mathcal{F}})^{\alpha}\big)\big|\mathcal{F}\big] \leq 2$. Equivalently

$$\|X\|_{\Psi_{\alpha}|\mathcal{F}} := \inf\left\{u > 0 : \mathbb{E}\big[\exp\big((|X|/u)^{\alpha}\big) \mid \mathcal{F}\big] \leq 2\right\},$$

provided $u$ is $\mathcal{F}$ measurable. Vector and matrix conditional Orlicz norms are defined analogously to their unconditional counterparts by taking suprema over one-dimensional projections (see Appendix G.1 for details).

### 2.1. Problem setup

Recall the optimization problem (Opt). Let $(\mathcal{F}_t)_{t \geq 1}$ be the natural filtration $\mathcal{F}_t = \sigma(X_0, \ldots, X_t)$. Throughout the entirety of our analysis, we will be imposing an assumption modeling stochasticity in the non-stationary setting:

**Assumption 2.1** (Stochastic predictability framework). *There exists a filtered probability space $(\Omega, \mathcal{F}, (\mathcal{F}_t)_{t \geq 0}, \mathbb{P})$ with $\mathcal{F}_0 = \{\emptyset, \Omega\}$. Let $(X_t)_{t \geq 0}$ be an $\mathbb{F}$–adapted process, i.e., $X_t$ is $\mathcal{F}_t$–measurable for all $t$. For each $t \geq 0$, let $\Pi_{t+1}$ denote the regular conditional law of $X_{t+1}$ given $\mathcal{F}_t$, i.e., $\Pi_{t+1}(A) = \mathbb{P}(X_{t+1} \in A \mid \mathcal{F}_t)$ a.s. for every measurable set $A$, and note $\Pi_{t+1}$ is $\mathcal{F}_t$–measurable. Define the conditional risk*

$$G_{t+1}(\boldsymbol{\theta}) := \mathbb{E}[g(\boldsymbol{\theta}, X_{t+1}) \mid \mathcal{F}_t] = \mathbb{E}_{X \sim \Pi_{t+1}}[g(\boldsymbol{\theta}, X)],$$

*and let $\boldsymbol{\theta}_{t+1}^{\star} \in \arg\min_{\boldsymbol{\theta} \in \mathbb{R}^d} G_{t+1}(\boldsymbol{\theta})$ denote a (measurable) minimizer. Assume the following hold for all $t \geq 0$:*

1. *(**Predictable minimizer**) $\boldsymbol{\theta}_{t+1}^{\star}$ is $\mathcal{F}_t$–measurable.[1]*

2. *(**Algorithm adaptedness**) The iterate $\boldsymbol{\theta}_t$ is $\mathcal{F}_t$–measurable.*

3. *(**Martingale difference noise**) Define the conditional mean gradient $\boldsymbol{m}_{t+1}(\boldsymbol{\theta}) = \mathbb{E}\big[\nabla_{\boldsymbol{\theta}} g(\boldsymbol{\theta}, X_{t+1}) \mid \mathcal{F}_t\big]$, and the gradient noise $\boldsymbol{\xi}_{t+1}(\boldsymbol{\theta}) = \nabla_{\boldsymbol{\theta}} g(\boldsymbol{\theta}, X_{t+1}) - \boldsymbol{m}_{t+1}(\boldsymbol{\theta})$. Then $\boldsymbol{\xi}_{t+1}(\boldsymbol{\theta})$ is $\mathcal{F}_{t+1}$–measurable and satisfies $\mathbb{E}[\boldsymbol{\xi}_{t+1}(\boldsymbol{\theta}) \mid \mathcal{F}_t] = \mathbf{0}$ a.s. for all $\boldsymbol{\theta} \in \mathbb{R}^d$.*

The time-varying distribution $\Pi_t$ in Assumption 2.1 captures several practical settings. In policy optimization and reinforcement learning, the data distribution is the state-action occupancy induced by the current policy, which evolves as the policy updates (Kakade & Langford, 2002; Schulman et al., 2015). In online recommendation and contextual bandits, user behavior drifts due to seasonality and preference changes (Li et al., 2010). Continual learning requires tracking moving objectives as tasks shift (Parisi et al.,

2019). Federated learning with non-stationary clients exhibits time-varying local data distributions as participating devices change across rounds (Kairouz & McMahan, 2021). In each case, both the noise distribution and the population objective $G_t$ vary with time.

Our parameter updates using the standard SGD update can then be written as

$$\begin{aligned} \boldsymbol{\theta}_{t+1} &= \boldsymbol{\theta}_t - \gamma_t \nabla_{\boldsymbol{\theta}} g(\boldsymbol{\theta}_t, X_{t+1}) \\ &= \boldsymbol{\theta}_t - \gamma_t \boldsymbol{m}_{t+1}(\boldsymbol{\theta}_t) - \gamma_t \boldsymbol{\xi}_{t+1}(\boldsymbol{\theta}_t). \end{aligned} \tag{SGD}$$

For SGD with momentum, we will present a generalized momentum stochastic gradient method, which can capture both Polyak's Heavy-Ball method and Nesterov's acceleration method as special cases. Consider the following with two momentum parameters $\beta_1, \beta_2 \in [0, 1)$:

$$\begin{aligned} \boldsymbol{\psi}_t &= \boldsymbol{\theta}_t + \beta_1(\boldsymbol{\theta}_t - \boldsymbol{\theta}_{t-1}) \\ \boldsymbol{\theta}_{t+1} &= \boldsymbol{\psi}_t - \gamma_t \nabla_{\boldsymbol{\theta}} g(\boldsymbol{\psi}_t, X_{t+1}) + \beta_2(\boldsymbol{\psi}_t - \boldsymbol{\psi}_{t-1}). \end{aligned}$$

When $\beta_1 = 0$ and $\beta_2 = \beta$, we recover Polyak's Heavy-Ball method. If $\beta_2 = 0$ and $\beta_1 = \beta$, we recover Nesterov's accelerated method. Thus to capture both these methods simultaneously, we assume $\beta_1 + \beta_2 = \beta$ and $\beta_1 \beta_2 = 0$ for some constant fixed $\beta \in [0, 1)$. We adopt this formulation from (Yuan et al., 2016). Using this, our parameter updates can be written as follows:

$$\begin{aligned} \boldsymbol{\psi}_t &= \boldsymbol{\theta}_t + \beta_1(\boldsymbol{\theta}_t - \boldsymbol{\theta}_{t-1}) \\ \boldsymbol{\theta}_{t+1} &= \boldsymbol{\psi}_t - \gamma_t \boldsymbol{m}_{t+1}(\boldsymbol{\psi}_t) - \gamma_t \boldsymbol{\xi}_{t+1}(\boldsymbol{\psi}_t) \\ &\quad + \beta_2(\boldsymbol{\psi}_t - \boldsymbol{\psi}_{t-1}). \end{aligned} \tag{SGDM}$$

Note that $\boldsymbol{\theta}_{t+1}$ is $\mathcal{F}_{t+1}$-measurable rather than $\mathcal{F}_t$-measurable. To analyze the convergence of (SGD) and (SGDM), we assume the conditional mean gradient map $\boldsymbol{\theta} \mapsto \boldsymbol{m}_t(\boldsymbol{\theta})$ is uniformly $\mu$-strongly monotone and $L$-Lipschitz. Under mild regularity conditions (e.g. sufficient integrability for the Dominated Convergence Theorem), this is equivalent to assuming $G_t(\boldsymbol{\theta})$ is $\mu$-strongly convex and $L$-smooth, which ensures that the minimizer $\boldsymbol{\theta}_t^{\star}$ is unique.

**Assumption 2.2** (Uniform $\mu$-strongly monotone). *There exists a constant $\mu > 0$ such that for all $t \geq 0$ and all $\boldsymbol{\theta}, \boldsymbol{\theta}' \in \mathbb{R}^d$,*

$$\langle \boldsymbol{m}_{t+1}(\boldsymbol{\theta}) - \boldsymbol{m}_{t+1}(\boldsymbol{\theta}'), \boldsymbol{\theta} - \boldsymbol{\theta}' \rangle \geq \mu \|\boldsymbol{\theta} - \boldsymbol{\theta}'\|^2.$$

**Assumption 2.3** (Uniform Lipschitz continuity). *There exists a constant $L > 0$ such that for all $t \geq 0$ and all $\boldsymbol{\theta}, \boldsymbol{\theta}' \in \mathbb{R}^d$,*

$$\|\boldsymbol{m}_{t+1}(\boldsymbol{\theta}) - \boldsymbol{m}_{t+1}(\boldsymbol{\theta}')\| \leq L \|\boldsymbol{\theta} - \boldsymbol{\theta}'\|.$$

These assumptions are standard in the SGD literature (Yuan et al., 2016) and are satisfied by many problems of interest, especially when regularization is used (e.g. mean-square error risks, logistic risks, etc). We also define the condition number as $\kappa := L/\mu$.

---

[1]Since $G_{t+1}(\boldsymbol{\theta}) = \mathbb{E}[g(\boldsymbol{\theta}, X_{t+1}) \mid \mathcal{F}_t]$ is $\mathcal{F}_t$–measurable for each fixed $\boldsymbol{\theta}$, standard measurable selection conditions ensure existence of an $\mathcal{F}_t$–measurable minimizer (e.g., Carathéodory regularity plus compactness/coercivity yielding a.s. nonempty $\arg\min$). If the minimizer is a.s. unique (e.g., under strong convexity), measurability follows by the measurable maximum theorem.

# 3. Theoretical Results

## 3.1. Bounds in expectation

In this section, we will obtain guarantees on the tracking error $\|\boldsymbol{\theta}_t - \boldsymbol{\theta}_t^\star\|$ in expectation. The intermediate steps used to obtain this bound will be useful in obtaining high-probability bounds, which we discuss later. Before proceeding, we will make a few assumptions regarding the second moments of the $\ell_2$ norm of the minimizer drift $\boldsymbol{\Delta}_t = \boldsymbol{\theta}_t^\star - \boldsymbol{\theta}_{t+1}^\star$ and the gradient noise, simply for presentation. These assumptions are not essential for the result, but we adopt them to simplify the exposition.

**Assumption 3.1** (Second-moment bounds). *There exist constants* $\Delta, \sigma > 0$ *such that for all* $t \geq 0$,

1. *(**Minimizer drift**) The minimizer drift* $\boldsymbol{\Delta}_t$ *satisfies* $\mathbb{E}\big[\|\boldsymbol{\Delta}_t\|^2\big] \leq \Delta^2$ *a.s.*

2. *(**Gradient noise along iterates**) The gradient noise* $\boldsymbol{\xi}_{t+1}$ *satisfies* $\mathbb{E}\big[\|\boldsymbol{\xi}_{t+1}(\boldsymbol{\theta}_t)\|^2\big] \leq \sigma^2$ *and* $\mathbb{E}\big[\|\boldsymbol{\xi}_{t+1}(\boldsymbol{\psi}_t)\|^2\big] \leq \sigma^2$ *a.s.*

We can now state the following theorem that establishes the expected tracking error for (SGD) in nonstationary stochastic environments. We defer the proofs for this section to Appendix C.

**Theorem 3.1** (Tracking error bound in expectation for (SGD)). *Under Assumption 3.1,* $\forall t \geq 0$ *and* $\gamma \leq \min\{\mu/L^2, 1/L\}$, *the following tracking error bound holds in expectation for (SGD):*

$$\mathbb{E}\|\boldsymbol{\theta}_t - \boldsymbol{\theta}_t^\star\|^2 \lesssim \Big(1 - \frac{\gamma\mu}{2}\Big)^t \|\boldsymbol{\theta}_0 - \boldsymbol{\theta}_0^\star\|^2 + \frac{\Delta^2}{\gamma^2\mu^2} + \frac{\sigma^2\gamma}{\mu}.$$

This bound is similar to Cutler et al. (2021), as it consists of a *contraction* term that arises from optimization that decays linearly in $t$, a *drift/tracking* term that depends on the minimizer drift, and a *noise* term. These contributions are *irreducible* for constant stepsize: even as $t \to \infty$, (SGD) cannot converge arbitrarily close to $\boldsymbol{\theta}_t^\star$ because **(i)** the optimizer itself moves over time (drift), and **(ii)** stochastic gradients inject persistent variance (noise). Moreover, the two steady-state terms exhibit an explicit stepsize tradeoff: larger $\gamma$ amplifies the noise floor, while smaller $\gamma$ reduces the noise but worsens tracking of a moving minimizer. Consequently, we can show that after a burn-in period that varies depending on whether we use a constant stepsize or an epoch-wise step-decay schedule, we can reach the irreducible floor of order $\mathcal{O}\big(\sigma^2\gamma/\mu + \Delta^2/\gamma^2\mu^2\big)$.

**Theorem 3.2** (Time to reach the asymptotic tracking error in expectation for (SGD)). *Assume* $\gamma_t \in (0, 1/(2L)]$ *for all* $t \geq 0$. *For any constant* $\gamma \in (0, 1/2L]$, *define*

$$\mathcal{E}(\gamma) := \frac{\sigma^2\gamma}{\mu} + \frac{4\Delta^2}{\mu^2\gamma^2}, \quad \gamma^\star \in \arg\min_{\gamma \in (0, 1/2L]} \mathcal{E}(\gamma)$$

*with* $\mathcal{E} := \mathcal{E}(\gamma^\star)$. *Then we have the following:*

1. *(**Constant learning rate**). If* $\gamma_t \equiv \gamma^\star$, *then for all* $t \geq 0$, $\mathbb{E}\|\boldsymbol{\theta}_t - \boldsymbol{\theta}_t^\star\|^2 \lesssim \mathcal{E}$ *after time* $t \lesssim (1/\mu\gamma^\star) \log\big(\|\boldsymbol{\theta}_0 - \boldsymbol{\theta}_0^\star\|^2/\mathcal{E}\big)$.

2. *(**Step-decay schedule in the low drift-to-noise regime**). Suppose* $\gamma^\star < 1/2L$ *(equivalently, the minimizer of* $\mathcal{E}(\gamma)$ *is not at the smoothness cap), so that*

$$\gamma^\star = \Big(\frac{8\Delta^2}{\mu\sigma^2}\Big)^{1/3}, \quad \mathcal{E} = 3\Big(\frac{\Delta\sigma^2}{\mu^2}\Big)^{2/3}.$$

*Define epochs* $k = 0, 1, \ldots, K - 1$ *with* $\gamma_0 := 1/2L$

$$\gamma_k := \frac{\gamma_{k-1} + \gamma^\star}{2}, \quad K := 1 + \Big\lceil \log_2\Big(\frac{\gamma_0}{\gamma^\star}\Big)\Big\rceil,$$

*and epoch lengths*

$$T_0 := \Big\lceil \frac{2}{\mu\gamma_0} \log\Big(\frac{2\|\boldsymbol{\theta}_0 - \boldsymbol{\theta}_0^\star\|^2}{\mathcal{E}(\gamma_0)}\Big)\Big\rceil, \quad T_k := \Big\lceil \frac{2\log 4}{\mu\gamma_k}\Big\rceil.$$

*Run (SGD) with constant stepsize* $\gamma_k$ *for* $T_k$ *iterations in epoch* $k$, *starting from* $\boldsymbol{\theta}_0$. *Let* $T := \sum_{k=0}^{K-1} T_k$ *be the total horizon. Then the final iterate satisfies* $\mathbb{E}\|\boldsymbol{\theta}_T - \boldsymbol{\theta}_T^\star\|^2 \lesssim \mathcal{E}$ *after time* $T \lesssim (L/\mu) \cdot \log\big(\|\boldsymbol{\theta}_0 - \boldsymbol{\theta}_0^\star\|^2/\mathcal{E}\big) + \sigma^2/\mu^2\mathcal{E}$.

Theorem 3.1 and Theorem 3.2 provide algorithmic guarantees for (SGD) in non-stationary environments, closely matching those of Cutler et al. (2021) and the static setting (Lan, 2011). Extending these results to (SGDM) is substantially more challenging because the update depends on both the current iterate and an auxiliary momentum-dependent evaluation point, precluding a reduction to a single recursion amenable to standard contraction arguments. Instead, we analyze (SGDM) as a 2D dynamical system on extended state vectors via a Lyapunov function capturing both components. We defer the full statement and proof to Appendix C.2.

With this view of (SGDM) as a 2D dynamical system on extended state vectors, we obtain an expectation bound on the tracking error for (SGDM). One will note that we incur extra scaling factors of $(1 - \beta)^{-1}$ and $(1 - \beta)^{-2}$ which explains why momentum can be significantly worse than (SGD) in non-stationary environments.

**Theorem 3.3** (Tracking error bound in expectation for (SGDM)). *Let Assumption 2.2, Assumption 2.3, and Assumption 3.1 hold. Consider the momentum stochastic gradient method (SGDM) and the extended 2D recursion (13). Then when the step-sizes* $\gamma$ *satisfies* $\gamma \leq \mu(1 - \beta)^2/4L^2$, *the following tracking error bound holds in expectation for*

*(SGDM):*

$$\mathbb{E}\|\boldsymbol{\theta}_t - \boldsymbol{\theta}_t^\star\|^2 \lesssim \frac{1}{(1-\beta)^2} \rho^{2t} \|\boldsymbol{\theta}_0 - \boldsymbol{\theta}_0^\star\|^2$$
$$+ \frac{(1+\beta+\gamma L)^2}{(1-\beta)^2} \cdot \frac{\Delta^2}{(1-\rho)^2} + \frac{1}{(1-\beta)^2} \cdot \frac{\sigma^2 \gamma^2}{1-\rho^2}.$$

*In particular taking* $\rho = 1 - \frac{\gamma\mu}{2(1-\beta)}$ *and letting* $c_{\gamma,\beta} = \exp(-\gamma\mu/(1-\beta))$, *we obtain:*

$$\mathbb{E}\|\boldsymbol{\theta}_t - \boldsymbol{\theta}_t^\star\|^2 \lesssim \frac{c_{\gamma,\beta}^t}{(1-\beta)^2} \|\boldsymbol{\theta}_0 - \boldsymbol{\theta}_0^\star\|^2$$
$$+ \frac{(\beta+2)^2}{\gamma^2\mu^2}\Delta^2 + \frac{\sigma^2\gamma}{\mu(1-\beta)}.$$

**Remark 3.1** (Beyond strong convexity). *Our Lyapunov argument relies on a contraction guaranteed by strong convexity. We expect the same proof technique to extend under weaker conditions (e.g., Polyak–Łojasiewicz, quadratic growth, or local strong convexity), where the strong-convexity step would be replaced by a condition guaranteeing sufficient decrease of the Lyapunov potential along the trajectory. In the convex case ($\mu = 0$) where contraction vanishes and minimizers need not be unique, we expect the coupled two-state Lyapunov recursion can still be analyzed via a potential function argument.*

Theorem 3.3 yields the same three-term structure as (SGD): an exponentially decaying *contraction* term plus *irreducible* drift and noise-induced floors, but with explicit momentum penalties. In particular, momentum increases sensitivity to initialization via a $(1-\beta)^{-2}$ prefactor, inflates the steady-state noise floor by $(1-\beta)^{-1}$, and slows the effective decay rate to $\exp(-\gamma\mu(t+1)/(1-\beta))$. Under standard stability tuning, this implies $\gamma\mu \asymp 1/\kappa$ so ill-conditioned problems can exhibit a long burn-in before reaching the steady-state regime. This is especially relevant in deep networks with highly ill-conditioned local curvature (small effective $\mu$) (Sagun et al., 2017; Jastrzębski et al., 2019; Ghorbani et al., 2019; Papyan, 2020). While Nesterov acceleration improves deterministic gradient descent from $\mathcal{O}(\kappa)$ to $\mathcal{O}(\sqrt{\kappa})$ iterations (Nesterov, 1983; 2014), in drifting stochastic settings the dominant error is governed by tracking and variance floors rather than bias decay. Thus the classical $\sqrt{\kappa}$ effect may never materialize, and large $\beta$ primarily raises the noise floor and prolongs burn-in (Theorem 3.4). Note that by strong convexity, these tracking error bounds immediately imply dynamic regret guarantees (see Appendix G.5 for details).

**Theorem 3.4** (Time to reach the asymptotic tracking error in expectation for (SGDM)). *Assume Assumption 2.2, Assumption 2.3, and Assumption 3.1 and let* $\beta \in [0,1)$. *Consider (SGDM), written with momentum buffer* $\boldsymbol{v}_t \in \mathbb{R}^d$ *as*

$$\boldsymbol{v}_{t+1} = \beta\boldsymbol{v}_t - \gamma_t \nabla g(\boldsymbol{\psi}_t, X_{t+1}), \quad \boldsymbol{\theta}_{t+1} = \boldsymbol{\theta}_t + \boldsymbol{v}_{t+1},$$

*where* $\boldsymbol{\psi}_t = \boldsymbol{\theta}_t$ *for heavy-ball momentum and* $\boldsymbol{\psi}_t = \boldsymbol{\theta}_t + \beta\boldsymbol{v}_t$ *for Nesterov momentum. Assume a constant stepsize* $\gamma_t \equiv \gamma$ *satisfying* $\gamma \le \mu(1-\beta)^2/(4L^2) := \gamma_{\max}$, *and set*

$$\rho := 1 - \frac{\gamma\mu}{2(1-\beta)} \in (0,1).$$

*Define the (stepsize-dependent) steady-state tracking error*

$$\mathcal{E}_\beta(\gamma) = \frac{192(2+\beta)^2}{\mu^2\gamma^2}\Delta^2 + \frac{96}{\mu(1-\beta)}\sigma^2\gamma$$
$$\gamma_\beta^\star \in \arg\min_{\gamma \in (0, \gamma_{\max}]} \mathcal{E}_\beta(\gamma)$$

*with* $\mathcal{E}_\beta := \mathcal{E}_\beta(\gamma_\beta^\star)$. *Then we have the following:*

1. *(**Constant learning rate**). If* $\gamma_t \equiv \gamma_\beta^\star$, *then for all* $t \ge 0$, $\mathbb{E}\|\boldsymbol{\theta}_t - \boldsymbol{\theta}_t^\star\|^2 \lesssim \mathcal{E}_\beta$ *after time* $t \lesssim (1-\beta)/(\mu\gamma_\beta^\star) \cdot \log(\|\boldsymbol{\theta}_0 - \boldsymbol{\theta}_0^\star\|^2/(1-\beta)^2\mathcal{E}_\beta)$

2. *(**Step-decay schedule with momentum restart**). Suppose* $\gamma_\beta^\star < \mu(1-\beta)^2/(4L^2)$ *(i.e., the minimizer of* $\mathcal{E}_\beta(\gamma)$ *is not at the stability cap). Define the epoch stepsizes with* $\gamma_0 := \mu(1-\beta)^2/4L^2$

$$\gamma_k := \frac{\gamma_{k-1} + \gamma_\beta^\star}{2}, \quad K := 1 + \left\lceil \log_2\left(\frac{\gamma_0}{\gamma_\beta^\star}\right)\right\rceil.$$

*Define epoch lengths*

$$T_0 := \left\lceil \frac{1-\beta}{\mu\gamma_0} \log\left(\frac{2\|\boldsymbol{\theta}_0 - \boldsymbol{\theta}_0^\star\|^2}{(1-\beta)^2\,\mathcal{E}_\beta(\gamma_0)}\right)\right\rceil,$$
$$T_k := \left\lceil \frac{1-\beta}{\mu\gamma_k} \log 4\right\rceil.$$

*Run (SGDM) with constant stepsize* $\gamma_k$ *for* $T_k$ *iterations in epoch $k$, restarting the momentum buffer at the start of each epoch, i.e., set* $\boldsymbol{v}_{t_k} = \boldsymbol{0}$ *at every epoch boundary* $t_k$. *Let* $T := \sum_{k=0}^{K-1} T_k$ *be the total horizon. Then the final iterate satisfies* $\mathbb{E}\|\boldsymbol{\theta}_T - \boldsymbol{\theta}_T^\star\|^2 \lesssim \mathcal{E}_\beta$ *after time* $T \lesssim (L^2/\mu^2(1-\beta)) \cdot \log(\|\boldsymbol{\theta}_0 - \boldsymbol{\theta}_0^\star\|^2/(1-\beta)^2\mathcal{E}_\beta) + \sigma^2/\mu^2\mathcal{E}_\beta$.

The burn-in time $(1-\beta)/(\mu\gamma)$ suggests resetting the momentum buffer after regime changes. One practical heuristic monitors the alignment

$$S_t := 1 - \frac{\langle\nabla g(\boldsymbol{\psi}_t, X_{t+1}), \boldsymbol{v}_t\rangle}{\|\nabla g(\boldsymbol{\psi}_t, X_{t+1})\| \cdot \|\boldsymbol{v}_t\| + \epsilon},$$

with $\epsilon > 0$, and restarts ($\boldsymbol{v}_t \leftarrow 0$) when $S_t$ stays large over several iterations. This truncates stale velocity that would otherwise persist for $\Omega((1-\beta)/(\mu\gamma))$ steps. Analyzing such restart rules rigorously, particularly whether they provably mitigate stale momentum under nonstationarity, is an interesting direction for future work.

## 3.2. High probability bounds

To obtain high-probability guarantees on the tracking error $\|\boldsymbol{\theta}_t - \boldsymbol{\theta}_t^\star\|$, we will need to make a standard light-tail assumption on the gradient noise (Lan, 2011; Harvey et al., 2019; Cutler et al., 2021):

**Assumption 3.2** (Conditional sub-Gaussian gradient noise along iterates). *There exists a constant $\sigma > 0$ such that for all $t \geq 0$, $\left\|\boldsymbol{\xi}_{t+1}(\boldsymbol{\theta}_t) \mid \mathcal{F}_t\right\|_{\Psi_2} \leq \sigma$ and $\left\|\boldsymbol{\xi}_{t+1}(\boldsymbol{\psi}_t) \mid \mathcal{F}_t\right\|_{\Psi_2} \leq \sigma$ a.s.*

**Remark 3.2** (Relaxing the noise assumption). *The sub-Gaussian assumption can be relaxed. With sub-exponential gradient noise, the concentration step in Theorems 3.5 and 3.6 changes to mixed Bernstein inequalities. With only bounded $q$-th moments for $q > 2$, standard martingale inequalities give polynomial-confidence analogues. In both cases, the three-term bound structure and the momentum-induced $(1 - \beta)^{-1}$ dependence from the coupled iterate–velocity recursion remain.*

Prior high-probability tracking analyses often iterate an MGF recursion for $\|\boldsymbol{\theta}_t - \boldsymbol{\theta}_t^\star\|^2$, which necessitates light-tail assumptions on the drift $\boldsymbol{\Delta}_t$. Our proof instead uses an optional stopping-time argument for weighted martingale difference sums, yielding insights missing from previous bounds for (SGD) in nonstationary settings. We defer the proofs for this section to Appendix D.

**Theorem 3.5** (High probability tracking error bound for (SGD)). *Under Assumption 3.2, for all $t \in [T]$, $\gamma \leq \min\{\mu/L^2, 1/L\}$, and $\delta \in (0, 1)$, the following tracking error bound holds for (SGD) with probability at least $1 - \delta$,*

$$\|\boldsymbol{\theta}_t - \boldsymbol{\theta}_t^\star\|^2 \lesssim \left(1 - \frac{\gamma\mu}{2}\right)^t \|\boldsymbol{\theta}_0 - \boldsymbol{\theta}_0^\star\|^2 + \frac{\mathfrak{D}_t}{\gamma\mu}$$

$$+ \frac{d\sigma^2\gamma}{\mu} + \left(d\sigma^2\gamma^2 + \frac{\sigma^2\gamma}{\mu} + \gamma^2\sigma^2\mathfrak{D}_t^{(2)}\right)\log\frac{2T}{\delta},$$

*where $\mathfrak{D}_t := \sum_{\ell=0}^{t-1}(1 - \gamma\mu/2)^{t-\ell-1}\|\boldsymbol{\Delta}_\ell\|^2$ and $\mathfrak{D}_t^{(2)} := \sum_{\ell=0}^{t-1}(1 - \gamma\mu/2)^{2(t-\ell-1)}\|\boldsymbol{\Delta}_\ell\|^2$.*

Prior high-probability guarantees for (SGD) in drifting environments typically control nonstationarity via a *uniform* drift bound $\Delta$, requiring light-tail assumptions such as conditional sub-exponentiality of $\|\boldsymbol{\Delta}_t\|^2$ (Cutler et al., 2021). Consequently, intermittent or localized nonstationarity (e.g., bursty regime shifts) is indistinguishable from persistent drift of magnitude $\Delta$ at all times, even though the dynamics exponentially forget older perturbations. In contrast, our bound is *drift-adaptive and time-resolved* and captures that **(i)** only recent nonstationarity drift affects the guarantee and **(ii)** drift amplifies stochastic fluctuations by increasing the tracking mismatch. This structure directly motivates restart/windowing/forgetting rules, which truncate or down-weight the history and thereby reduce both drift accumulation and variance inflation after regime changes. As a

consequence of Theorem 3.5, we can obtain guarantees similar to Theorem 3.2 by replicating a similar argument. We exclude these results for (SGD) and (SGDM) for brevity.

**Theorem 3.6** (High probability tracking error bound for (SGDM)). *Under Assumption 3.2, for all $t \in [T]$, $\gamma \leq \min\{1/L, \mu(1 - \beta)^2/4L^2\}$, and $\delta \in (0, 1)$, provided one takes a zero momentum initialization $\boldsymbol{\theta}_{-1} = \boldsymbol{\theta}_0$, the following tracking error bound holds for (SGDM) with probability at least $1 - \delta$,*

$$\|\boldsymbol{\theta}_t - \boldsymbol{\theta}_t^\star\|^2 \lesssim \frac{c_{\gamma,\beta}^t}{(1-\beta)^2}\|\boldsymbol{\theta}_0 - \boldsymbol{\theta}_0^\star\|^2 + \frac{1}{\gamma\mu} \cdot \frac{1}{1-\beta}\mathfrak{D}_t^{\text{lag}}$$

$$+ \frac{d\sigma^2}{\mu^2} + \left(\frac{d\sigma^2\gamma^2}{(1-\beta)^2} + \frac{\sigma^2}{\mu^2} + \frac{\sigma^2\gamma^2}{(1-\beta)^2}\mathfrak{D}_t^{\text{lag},(2)}\right)\log\frac{2T}{\delta},$$

*where $c_{\gamma,\beta} := \exp\left(-\gamma^2\mu^2/(4(1 - \beta)^2)\right)$, $\tilde{\rho} := 1 - \gamma^2\mu^2/4(1 - \beta)^2$, $\mathfrak{D}_t^{\text{lag}} := \sum_{\ell=0}^{t-1}\tilde{\rho}^{t-\ell-1}\|\boldsymbol{b}_\ell\|^2$, and $\mathfrak{D}_t^{\text{lag},(2)} := \sum_{\ell=0}^{t-1}\tilde{\rho}^{2(t-\ell-1)}\|\boldsymbol{b}_\ell\|^2$, with $\boldsymbol{b}_\ell := -(\mathbf{I}_d - \gamma\boldsymbol{H}_\ell)\boldsymbol{\Delta}_\ell - \boldsymbol{K}_\ell\boldsymbol{\Delta}_{\ell-1}$, where $\boldsymbol{H}_\ell$ and $\boldsymbol{K}_\ell$ are defined as in (13).*

**Remark 3.3.** *Theorem 3.6 is proved via a separate pathwise Lyapunov recursion and martingale concentration argument, which is conservative and yields a looser contraction rate than the expectation analysis (Theorem 3.3). We expect that this looser rate is technical and can likely be tightened with a sharper concentration argument.*

Our high-probability bounds show that momentum can become fragile under drift, with volatility that worsens as $\beta \uparrow 1$. (SGDM) exhibits three effects not present for (SGD): **(i)** an *inertia horizon* of order $(1 - \beta)^{-1}$, so tracking depends on higher-order drift (involving $\boldsymbol{\Delta}_{\ell-1}, \boldsymbol{\Delta}_{\ell-2}$) and can overshoot under regime shifts, **(ii)** a *drift–noise coupling* term that appears pathwise in concentration and scales as $(1-\beta)^{-2}$, indicating that systematic misalignment amplifies the impact of stochastic gradient noise on the error direction (cf. (Cutler et al., 2021)), and **(iii)** in ill-conditioned regimes ($\kappa \gg 1$), stability forces $\gamma$ to shrink (e.g. $\gamma \lesssim (1 - \beta)^2/L$), slowing transient decay and preventing the algorithm from quickly "forgetting" initialization or adapting to drift. Together, these mechanisms explain why (SGD) can be strictly more robust in nonstationary, ill-conditioned settings: it avoids the compounded inertia and variance penalties induced by momentum.

## 3.3. Minimax regret bounds

We now turn to minimax lower bounds for dynamic regret and their corresponding implications for tracking error in nonstationary stochastic optimization under distribution shift. To place the lower bound in the appropriate form, we first rephrase the problem in a minimax (worst-case) framework. We consider nonstationary stochastic optimization in which the underlying sample loss is fixed but the data

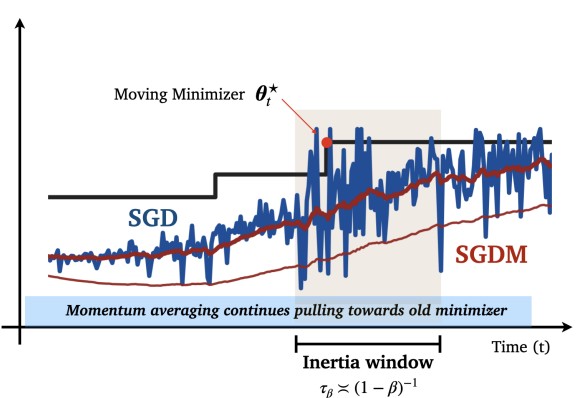
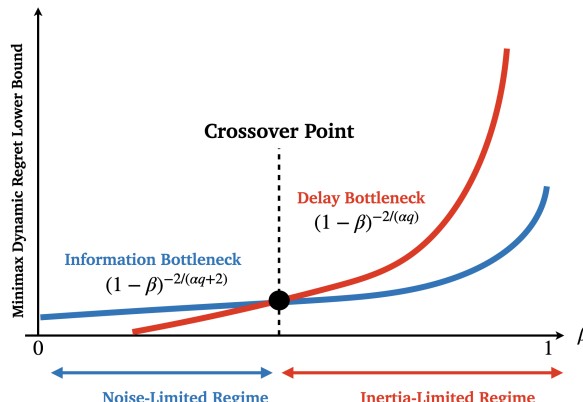

*Figure 1.* **Two minimax regimes and the inertia window. (a)** After a regime change, momentum averages past gradients: this reduces noise but induces *inertia*, so the iterate lags behind the drifting minimizer for an "inertia window" whose length grows with $\beta$. **(b)** The minimax lower bound in Theorem 3.7 is the maximum of two contributions: a noise/variation-limited term (information-limited) and an inertia-limited term (delay-limited) that worsens with momentum and can dominate.

distribution shifts over time. This should be viewed as a specialization of the time-varying-loss framework of (Besbes et al., 2015; Chen et al., 2019): distribution shift induces a time-varying *population risk*, so the learner still faces a sequence of time-dependent objectives arising through the data-generating process rather than an explicitly time-indexed loss.

Let $\Theta \subset \mathbb{R}^d$ be convex. At each time $t \in \{0, \ldots, T-1\}$ the algorithm chooses $\boldsymbol{\theta}_t \in \Theta$, then observes a fresh sample $X_{t+1}$ whose (conditional) law may drift over time. This drift induces a time-varying *population risk*

$$G_{t+1}(\boldsymbol{\theta}) := \mathbb{E}[g(\boldsymbol{\theta}, X_{t+1}) \mid \mathcal{F}_t], \; \boldsymbol{\theta}_{t+1}^\star \in \arg\min_{\boldsymbol{\theta} \in \Theta} G_{t+1}(\boldsymbol{\theta}),$$

where $\mathcal{F}_t = \sigma(X_0, \ldots, X_t)$. We assume $G_t$ is $\mu$-strongly convex and $L$-smooth uniformly in $t$, so $\boldsymbol{\theta}_t^\star$ is a.s. unique. Define the conditional mean gradient $\boldsymbol{m}_{t+1}(\boldsymbol{\theta}) = \mathbb{E}[\nabla g(\boldsymbol{\theta}, X_{t+1}) \mid \mathcal{F}_t]$ and noise $\boldsymbol{\xi}_{t+1}(\boldsymbol{\theta}) = \nabla g(\boldsymbol{\theta}, X_{t+1}) - \boldsymbol{m}_{t+1}(\boldsymbol{\theta})$, so that $\mathbb{E}[\boldsymbol{\xi}_{t+1}(\boldsymbol{\theta}) \mid \mathcal{F}_t] = 0$ a.s. We consider SGD and momentum updates driven by this noisy first-order feedback given by (SGD) and (SGDM). We will measure performance via dynamic regret against the drifting minimizers,

$$\mathcal{R}_T^\pi(G) := \mathbb{E}^\pi \left[ \sum_{t=0}^{T-1} \left( G_{t+1}(\boldsymbol{\theta}_t) - G_{t+1}(\boldsymbol{\theta}_{t+1}^\star) \right) \right].$$

Dynamic regret has been extensively studied in online convex optimization: Jadbabaie et al. (2015) developed dynamic-comparator bounds that adapt to variation in the losses and comparators, Yang et al. (2016) established optimal rates for tracking slowly moving clairvoyant minimizers, and Zhang et al. (2018) obtained adaptive algorithms with matching path-length-dependent upper and lower bounds.

To obtain nontrivial guarantees, given a sequence of differentiable functions $g_1, \ldots, g_T : \Theta \to \mathbb{R}$, define the $L_{p,q}$ gradient-variational functional of $g = (g_1, \ldots, g_T)$ as

$$\text{GVar}_{p,q}(g) := \begin{cases} \left( \dfrac{1}{T} \displaystyle\sum_{t=1}^{T-1} \left\| \nabla g_{t+1} - \nabla g_t \right\|_p^q \right)^{1/q} & q < \infty \\ \displaystyle\max_{1 \le t \le T-1} \left\| \nabla g_{t+1} - \nabla g_t \right\|_p & q = \infty \end{cases}$$

where for any measurable function $g : \Theta \to \mathbb{R}$, we have

$$\|g\|_p := \begin{cases} \left( \displaystyle\int_\Theta \|g(\theta)\|_2^p \, d\theta \right)^{1/p} & p < \infty \\ \displaystyle\sup_{\theta \in \Theta} \|g(\theta)\|_2 & p = \infty. \end{cases}$$

We set a budget constraint defined by the function class

$$\mathcal{G}_{p,q}(\mathbb{V}_T) := \{ g : \text{GVar}_{p,q}(g) \le \mathbb{V}_T \}.$$

This is in contrast to (Besbes et al., 2015; Chen et al., 2019) who place constraints on the function values rather than the gradients. We do this since gradient variation naturally gives rise to distributional shift metrics such as the Wasserstein distance $\mathcal{W}_1(\Pi_{t+1}, \Pi_t)$ via Kantorovich-Rubinstein duality and the Total Variation (TV) distance. We also note that placing a gradient variation budget implies a budget on the minimizer drift by strong convexity. We now establish the following worst-case regret for noisy gradient feedback over $\mathcal{G}_{p,q}(\mathbb{V}_T)$. We defer the proofs to Appendix E. We will denote the minimax risk as $\mathfrak{M}_T(\Pi_\beta, \mathbb{V}_T) := \inf_{\pi \in \Pi_\beta} \sup_{G : \text{GVar}_{p,q}(G) \le \mathbb{V}_T} \mathcal{R}_T^\pi(G)$.

**Theorem 3.7** (Minimax lower bound for strongly-convex function sequences using (SGDM)). *Fix arbitrary $1 \le p \le \infty$ and $1 \le q \le \infty$. Suppose Assumption 2.2 and Assumption 2.3 hold. Consider the class $\Pi_\beta$ of SGDM($\beta$) policies*

*with constant step size* $\gamma \leq c_0(1-\beta)^2/L$. *Then there exists a class* $\mathcal{G}_{p,q}(\mathbb{V}_T)$ *of* $\mu$-*strongly convex, $L$-smooth function sequences whose gradient-variational functional budget satisfies* $\mathrm{GVar}_{p,q}(G) \leq \mathbb{V}_T$ *such that*

$$\mathfrak{M}_T(\Pi_\beta, \mathbb{V}_T) \gtrsim \max\Big\{ A_{\alpha,q}\, \mathbb{V}_T^{2q/(\alpha q+2)}\, T^{\alpha q/(\alpha q+2)},$$
$$B_{\alpha,q}\, \mathbb{V}_T^{2/\alpha}\, T^{1-2/(\alpha q)} \Big\},$$

*where* $\alpha := 1 + d/p$, $A_{\alpha,q} := (1-\beta)^{-2/(\alpha q+2)}$ $\sigma^{4/(\alpha q+2)}\, \mu^{(\alpha q-2q-2)/(\alpha q+2)}$, *and* $B_{\alpha,q} := (1-\beta)^{-2/(\alpha q)}$ $\mu^{(\alpha q-2q-2)/(\alpha q)}\, L^{2/(\alpha q)}$.

Note that by strong convexity, dynamic regret lower bounds cumulative tracking error up to an additive comparator term. Under uniformly spread drift (i.e. $q = 1$, $p = \infty$, and $\Delta \asymp \mathbb{V}_T/(\mu T)$), optimizing the constant stepsize in Theorem 3.4 yields a dynamic regret upper bound of the form $\mathcal{R}_T \lesssim L(1-\beta)^{-2/3}\, \sigma^{4/3}\, \mu^{-2}\, \mathbb{V}_T^{2/3}\, T^{1/3}$ which matches the corresponding minimax lower bound in its dependence on $\mathbb{V}_T, T, \sigma$ and crucially the momentum penalty $(1-\beta)^{-2/3}$ (see Appendix G.5 for details).

The minimax lower bound decomposes into two regimes. The first (statistical) term is driven by noisy gradient feedback under the gradient-variation budget. In this regime, regret is information-limited so algorithmic dynamics matter very little. SGDM's dependence on $(1-\beta)$ in this term is mild and arises only through stability-constrained stepsize tuning. The second (inertia-limited) term captures delay: temporal averaging suppresses noise but induces inertia, so after a regime change the iterate follows stale gradients. Under standard stability restrictions, this yields an unavoidable inertia window of length $\tau_\beta := \Omega(\kappa/(1-\beta))$.

These terms imply **(i)** drift-driven tracking error is amplified as $\beta \uparrow 1$ in our upper bounds, unlike (SGD), and **(ii)** this penalty is information-theoretic: the lower bound constructs blockwise shifts consistent with the variation budget for which any SGDM($\beta$) policy must spend $\tau_\beta$ steps per change in a transient misalignment phase, incurring regret that worsens with $\beta$. Although momentum can, in principle, trade variance reduction for slower adaptation, our numerical results will show that this trade-off is rarely favorable under drift: as nonstationarity grows, the inertia-limited error becomes unavoidable and (SGD) is often preferable even in substantially noisy regimes.

## 4. Numerical Experiments

We evaluate online optimization under drifting optima across four settings: (i) strongly convex quadratics, (ii) linear regression, (iii) logistic regression, and (iv) teacher–student MLP regression. All tasks use time-varying minimizers $\theta_t^\star$ evolving via normalized random walk: $\theta_{t+1}^\star = \theta_t^\star + \delta_{\mathrm{rw}} u_t/\|u_t\|_2$ where $u_t \sim \mathcal{N}(0, I_d)$.

**Setup.** We compare SGD against Polyak Heavy-Ball (HB) and Nesterov acceleration (NAG). Our results (Theorem 3.3 and Theorem 3.6) assume the stability condition $\gamma \leq \mu(1-\beta)^2/(4L^2)$, which can be conservative in practice. We therefore use standard step sizes in the main experiments to test whether drift-induced inertia persists outside the analyzed regime; restricting step sizes recovers the qualitative behavior of (Yuan et al., 2016). For regression, we control conditioning via covariates $x = z\Sigma^{1/2}$ with $z \sim \mathcal{N}(0, I)$ and $\mathrm{cond}(\Sigma) = \kappa \in \{10, 1000\}$, giving $\mu = 1$ and $L = \kappa$. For the MLP, tracking is measured in prediction space to avoid non-identifiability from permutation and scaling symmetries. We set $\delta_{\mathrm{rw}} = 0.01$ and report squared tracking error $e_t = \|\theta_t - \theta_t^\star\|^2$ averaged over 20 runs. Results were robust across dimensions and held under Student-$t$ shifts (Appendix F).

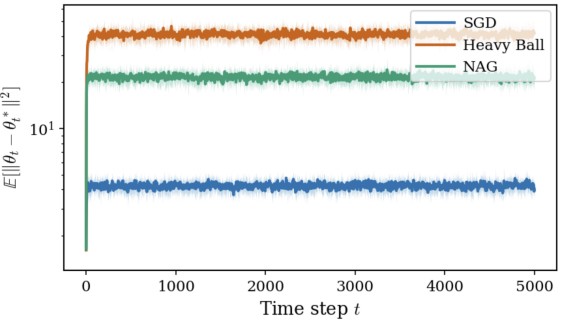

*Figure 2.* **Tracking a drifting minimizer under strong convexity.** Mean squared tracking error versus time for SGD, Heavy-Ball (HB), and Nesterov (NAG) on a strongly convex quadratic with $\gamma = 0.1, \beta = 0.9, \sigma^2 = 0.8$.

*Table 1.* **Mean tracking error after 5000 iterations on a drifting strongly convex quadratic** ($d = 100$). Across regimes, increasing $\beta$ markedly degrades the steady-state tracking of HB/NAG, especially at larger $\gamma$ and $\sigma^2$, highlighting inertia-induced lag under drift, while SGD remains comparatively robust.

| | | $\gamma = 0.01$ | | | | | $\gamma = 0.10$ | | |
|---|---|---|---|---|---|---|---|---|---|
| $\beta$ | $\sigma^2$ | SGD | HB | NAG | $\beta$ | $\sigma^2$ | SGD | HB | NAG |
| 0.50 | 0.1 | 1.04 | **0.34** | 0.35 | 0.50 | 0.1 | **0.53** | 1.05 | 0.91 |
| 0.50 | 0.5 | 1.24 | **0.73** | 0.75 | 0.50 | 0.5 | **2.63** | 5.27 | 4.85 |
| 0.50 | 0.8 | 1.31 | **0.96** | 1.02 | 0.50 | 0.8 | **4.11** | 8.51 | 7.81 |
| 0.90 | 0.1 | 1.03 | 0.50 | **0.45** | 0.90 | 0.1 | **0.53** | 5.17 | 2.60 |
| 0.90 | 0.5 | **1.23** | 2.36 | 2.25 | 0.90 | 0.5 | **2.54** | 27.41 | 13.43 |
| 0.90 | 0.8 | **1.47** | 3.90 | 3.72 | 0.90 | 0.8 | **4.29** | 40.67 | 21.38 |
| 0.95 | 0.1 | 1.04 | 1.05 | **0.81** | 0.95 | 0.1 | **0.53** | 9.98 | 3.33 |
| 0.95 | 0.5 | **1.26** | 5.14 | 4.11 | 0.95 | 0.5 | **2.66** | 51.36 | 17.38 |
| 0.95 | 0.8 | **1.35** | 7.70 | 6.80 | 0.95 | 0.8 | **4.04** | 79.82 | 27.17 |
| 0.99 | 0.1 | **1.04** | 4.84 | 2.62 | 0.99 | 0.1 | **0.55** | 54.25 | 4.41 |
| 0.99 | 0.5 | **1.23** | 23.67 | 13.10 | 0.99 | 0.5 | **2.72** | 243.98 | 24.10 |
| 0.99 | 0.8 | **1.40** | 38.80 | 21.04 | 0.99 | 0.8 | **4.11** | 401.37 | 38.32 |

**Results.** We present the strongly convex quadratic results. Additional experimental results for linear, logistic, and MLP

regression can be found in Appendix F. We evaluate tracking via the squared error $\|\boldsymbol{\theta}_t - \boldsymbol{\theta}_t^\star\|^2$, reporting mean error after 5000 iterations. Across tasks, (SGD) matches or outperforms HB/NAG in nonstationary regimes, with the gap widening as momentum increases, noise rises, or conditioning worsens. At low momentum and small step sizes (e.g., $\beta = 0.5$, $\gamma = 0.01$), HB/NAG can outperform (SGD) through noise suppression via temporal averaging, consistent with the information-limited regime of Theorem 3.7. However, as $\beta$ increases, momentum methods degrade sharply while (SGD) remains stable, and this deterioration is amplified by larger $\sigma^2$, larger $\gamma$, and ill-conditioning. This matches our two-regime minimax bound: in *noise-limited* settings, large $\beta$ can amplify gradient noise, while under *conservative stable tuning*, drift dominates and inertia induces systematic lag scaling as $\tau_\beta = \Omega\big(\kappa/(1-\beta)\big)$. Increasing $\beta$ or $\kappa$ therefore worsens the response to shifts in $\boldsymbol{\theta}_t^\star$. Notably, increasing $\gamma$ beyond conservative regimes does not recover tracking for HB/NAG; instead, performance degrades by orders of magnitude while (SGD) remains comparatively robust.

## 5. Conclusion

We study stochastic optimization with a time-varying population risk, where the goal is to *track* a drifting minimizer rather than converge to a fixed point. For (SGD) and (SGDM), our finite-time and high-probability analyses yield a unified decomposition of tracking error into **(i)** an exponentially decaying transient (forgetting initialization), **(ii)** an irreducible noise floor, and **(iii)** an irreducible drift-induced lag. This makes the core trade-off explicit: temporal averaging can potentially reduce variance, but it also makes gradients stale and slows adaptation to regime changes.

Our main finding is an unavoidable *tracking penalty* from momentum: as $\beta \to 1$, sensitivity to initialization and the effective noise level along the trajectory grow, and stability in ill-conditioned problems can force much smaller step sizes. The high-probability bounds further reveal a drift–noise interaction in which trajectory misalignment amplifies stochastic fluctuations, providing theoretical motivation for explicit forgetting mechanisms (e.g., windowing, restarting) after regime shifts.

We complement these upper bounds with minimax lower bounds for dynamic regret under gradient-variation constraints, showing that in drift-dominated regimes any SGDM($\beta$) policy suffers an information-theoretic *inertia window* during which it necessarily lags the moving minimizer. The experimental results validate our theoretical predictions: momentum induces systematic fragility under distribution shift, with performance degradation amplified by large $\beta$ and poor conditioning, whereas (SGD) maintains stable tracking throughout.

**Future work and extensions.** Our analysis leaves several open questions. The upper and lower bounds hold for fixed-$\beta$ SGDM, so the explicit penalties in $(1-\beta)^{-2}$ reflect fixed geometric memory and do not extend immediately to adaptive weighting schemes with time-varying $\beta_t$, bias correction, or state-dependent averaging. We conjecture that the relevant complexity parameter is the *effective averaging horizon* at each regime shift: methods that place substantial weight on stale gradients incur a lag penalty, while shorter-memory or restart-based schemes can mitigate, but not fully eliminate, this effect. This perspective also motivates the study of adaptive optimizers such as Adam and AdamW, whose first- and second-moment EMAs introduce analogous stale-history effects under drift. This direction is pursued in subsequent work by Sahu et al. (2026). Our Lyapunov analysis targets stochastic nonstationary tracking and does not recover the sharp deterministic accelerated rates known for Nesterov momentum in the stationary noiseless setting. Recent work by Wang & Yurtsever (2026) shows that time-varying parameters and Lyapunov functions built on intermediate iterations can recover accelerated deterministic behavior in generalized SGDM. Adapting such techniques to drifting stochastic settings could recover sharper accelerated rates while preserving our main nonstationary conclusions.

## Impact Statement

This paper presents work whose goal is to advance the field of Optimization. There are many potential societal consequences of our work, none of which we feel must be specifically highlighted here.

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

# Appendices

# A. Related Work

## A.1. Momentum and acceleration in stochastic optimization

Developing methods to accelerate the convergence of SGD has been studied extensively. Momentum methods, such as Polyak's Heavy-Ball (HB) (Polyak, 1963) and Nesterov acceleration (NAG) (Nesterov, 1983), are ubiquitous in deep learning and are widely believed to reduce gradient noise and accelerate training (Sutskever et al., 2013). Adaptive variants such as Adagrad and Adam further modify the effective step-size online (Duchi et al., 2011; Zeiler, 2012; Kingma & Ba, 2015; Loshchilov & Hutter, 2019; Liao & Kyrillidis, 2024), and the general research landscape of second-order and filtering based methods has been studied extensively (Dozat, 2016; Vuckovic, 2018; Zaheer et al., 2018; Gupta et al., 2018; Luo et al., 2019; Zhou et al., 2019; Ollivier, 2019; Liu et al., 2020a; Zhou et al., 2020; Zhuang et al., 2021; Yao et al., 2021; Ziyin et al., 2021; Ilboudo et al., 2022; Davtyan et al., 2022; Yang, 2023; Li et al., 2025; Godichon-Baggioni et al., 2025; Jentzen et al., 2025; Vyas et al., 2025; Yao et al., 2025), with methods being developed specifically for optimizing the training of large neural networks as of late (Bernstein et al., 2018; Liu et al., 2025; Huang et al., 2025). It is widely known for a variety of strongly convex and nonconvex settings that SGD with momentum provides faster convergence and better generalization compared to SGD without momentum (Cutkosky & Mehta, 2020; Liu et al., 2020b; Ramezani-Kebrya et al., 2024; Sato & Iiduka, 2024). While it is relatively well-known that a fixed momentum rate is not as effective as adaptive momentum (Sutskever et al., 2013), addressing how to tune momentum optimally is an active area of research. In practice, it is common to fix the momentum parameter to a standard value (e.g., $\beta = 0.9$) (Kingma & Ba, 2015; Chen et al., 2016), or use a simple scheduler such as exponential momentum decay or cosine momentum.

## A.2. Limitations and instabilities of momentum

While popular, momentum does not always improve training (Fu et al., 2023; Wang et al., 2024; Deng et al., 2024). For online and adaptive learning, prior work suggests that the deterministic acceleration intuition does not directly transfer: in the slow-adaptation regime, momentum can be fundamentally equivalent to standard SGD with a re-scaled step-size, and can even degrade steady-state performance unless parameters are chosen carefully (Yuan et al., 2016). A key challenge in nonstationary stochastic optimization is that *past gradients become stale*: gradients computed under $\Pi_{t-k}$ can systematically point in the wrong direction for the current objective $G_t$. This motivates discounting or forgetting historical gradient information (Zhang et al., 2020). Furthermore, empirical evidence in online forecasting has pointed to SGD with momentum not being as stable as SGD without momentum, especially as learning rates grow (Aydore et al., 2019). Prior work has identified settings in which momentum can be detrimental. In particular, Jagannath et al. (2025) show that in high-dimensional online regimes, adding momentum can amplify gradient noise and worsen performance under a constant stepsize. Relatedly, Deng et al. (2024) demonstrate that momentum can undermine stability as the momentum parameter $\beta$ approaches 1. Methods have been developed to tune momentum throughout the training process in a more stable manner, such as using local quadratic approximations (Zhang & Mitliagkas, 2019), combining different loss planes (Topollai & Choromanska, 2025), momentum decay (Chen et al., 2022), or passive damping (Lucas et al., 2019). Although research has been done on understanding the effects of momentum theoretically (Qian, 1999; Reddi et al., 2018; Gitman et al., 2019; Kovachki & Stuart, 2021; Jelassi & Li, 2022; Liao & Kyrillidis, 2024), the noisy, time-dependent, non-stationary environment is understudied.

## A.3. Optimization under time-dependent and non-stationary environments

A complementary line of work studies the steady-state behavior of momentum under persistent stochastic noise. Yuan et al. (2016) show that, in online stochastic optimization with constant stepsizes and a fixed minimizer, momentum methods are essentially equivalent (at steady state) to standard SGD with a suitably rescaled learning rate. In particular, momentum does not inherently reduce the mean-square error (MSE) floor when the optimum is stationary Kovachki & Stuart (2021). These analyses, however, are primarily confined to stationary regimes and therefore do not capture phenomena that are central under distribution shift, such as tracking lag and stability constraints when the minimizer $\theta_t^\star$ evolves over time. Recent work has begun to address non-stationarity from several perspectives. Yang et al. (2017) propose an online gradient-learning variant aimed at time-dependent data with outliers, modifying Adam to downweight or identify corrupted samples. In a neural-network setting, Galashov et al. (2024) study soft parameter resets under an Ornstein–Uhlenbeck drift model that pulls weights toward a prior with an adaptive drift strength. From an optimization-theoretic standpoint, Cutler et al. (2023) provide non-asymptotic efficiency guarantees for tracking error of proximal stochastic gradient methods under time drift, and Shen et al. (2026) establish optimal estimation and regret bounds for SGD with temporally dependent data. Relatedly,

Cao et al. (2019) analyze online stochastic optimization under time-varying distributions and derive dynamic-regret bounds for SGD, while Zhao et al. (2024) develop problem-dependent dynamic-regret guarantees in online convex optimization that adapt to variation in both the loss sequence and the drifting comparator. Finally, Vidyasagar (2025) study stability and convergence of momentum-based methods for time-varying objectives, but do not explicitly treat distribution shift nor provide tracking-error bounds.

## B. Outline of proofs and assumptions

Before stating auxiliary notation and assumptions, we provide a high-level overview of how the proofs of our main results fit together. Our upper bounds in expectation are proved in Section C, our high-probability guarantees are proved in Section D, and our minimax lower bounds are proved in Section E.

**Section C: tracking bounds in expectation.** Section C proves the expectation guarantees for both (SGD) and (SGDM), Corollaries C.1 and C.2 for (SGD) and Corollaries C.5 and C.6 for (SGDM). We begin in Section C.1 by deriving a stable one-step recursion for the (SGD) tracking error (Lemma C.1). Unrolling it gives the final-iterate decomposition (Proposition C.1), and applying the standing second-moment bounds (Assumption 3.1) yields the closed-form expectation floor and the associated time-to-track statement. In Section C.2 we treat momentum via an augmented-state ($2d$-dimensional) linear recursion: Lemma C.2 rewrites (SGDM) in mode-splitting coordinates (via the transform $V$), isolating how drift and noise enter the dynamics. The stability/contractivity regime for this augmented recursion is characterized in Corollary C.3 as a spectral-radius condition under the stepsize restriction $\gamma_t \leq \mu(1-\beta)^2/(4L^2)$, and Corollary C.4 converts this stability into a bound on the state-transition matrices needed to unroll the recursion. With these results, Corollary C.5 follows by decomposing the unrolled solution into **(i)** an exponentially decaying initialization transient, **(ii)** a drift-induced lag term controlled by $\Delta$, and **(iii)** a noise floor controlled by $\sigma$, with explicit $(1-\beta)^{-1}$ and $(1-\beta)^{-2}$ amplifications arising from the transformation and the stability factor. Finally, Corollaries C.2 and C.6 optimizes the steady-state error over $\gamma$ and provides a restartable step-decay schedule (with momentum-buffer resets at epoch boundaries) achieving the optimized floor within an explicit time horizon.

**Section D: time-resolved high-probability bounds.** Section D proves the high-probability tracking guarantees for (SGD) and (SGDM), stated in Theorems D.1 and D.2. We work throughout under the conditional sub-Gaussian noise condition along the iterates (Assumption D.1). The proof follows analogously to Section C, but now in a pathwise form: we first obtain a final-iterate decomposition whose stochastic terms remain random, and then control those terms via martingale concentration. For (SGD), Section D.1 starts from the unrolled recursion in Proposition C.1 and bounds the two random contributions: a weighted sum of squared noises and a weighted martingale term—using the conditional Orlicz and martingale tools in Sections F.1 to F.3 (e.g., conditional $\Psi_2 \Rightarrow$ second moments using Lemma G.4, Bernstein for sub-exponential MDS using Lemma G.5, and optional stopping with Lemma G.6), yielding Theorem 3.5. The "time-resolved" structure is obtained by keeping the exponentially-weighted history explicit after unrolling, which produces the drift functionals $\mathfrak{D}_t$ and $\mathfrak{D}_t^{(2)}$ that only emphasize recent drift. For (SGDM), Section D.2 first rewrites the method as an augmented-state linear recursion via the mode-splitting transform (Lemma C.2), and unrolls it to a final-iterate inequality (Proposition D.1) whose contraction rate is governed by the stability regime $\gamma \leq \min\{1/L, \mu(1-\beta)^2/(4L^2)\}$. The high-probability control then mirrors the (SGD) case, but with the transformed drift/noise entering through $\mathfrak{D}_t^{\mathrm{lag}}$ and $\mathfrak{D}_t^{\mathrm{lag},(2)}$ and with explicit $(1-\beta)$-dependent amplification through the effective step size $\eta = \gamma/(1-\beta)$. This yields the (SGDM) high-probability theorem Theorem D.2.

**Section E: minimax lower bounds.** Section E proves the minimax lower bound for the class $\Pi_\beta$ of SGDM($\beta$) policies under the nonstationary strongly-convex model and the gradient-variation budget $\mathrm{GVar}_{p,q} \leq \mathbb{V}_T$. The proof has two complementary parts that isolate *statistical* and *algorithmic* obstructions. First, Section E.1 reduces regret minimization to a hypothesis testing problem: Lemma E.1 shows that uniformly small regret would induce a decision rule that identifies the underlying loss sequence in a finite packing with constant probability, and Lemma E.2 (Fano) converts KL control into a constant lower bound on the testing error. Second, Sections E.2 to E.5 implement the information-theoretic hard instance: Section E.2 constructs localized bump losses $g_+, g_-$ via (164) and proves smoothness/strong convexity (Lemma E.3), identifies minimizers and discrepancy (Lemma E.4), and quantifies localization in $L^p$ (Lemma E.5 and Lemma E.6). Section E.3 then injects the sharp $(1-\beta)$ penalty into the noise-dominated term by tuning a rare-visit bump radius: Heavy-Ball noise inflates stationary variance by $(1-\beta)^{-1}$ (Lemma E.7), yielding a $\beta$-dependent occupation bound (Lemma E.8) and hence a tighter KL scaling through the Gaussian chain rule (Lemma E.12). Section E.4 builds the $J$-block packing family (187), lower bounds separation via discrepancy accumulation (Lemma E.11 and Corollary E.1), and upper bounds

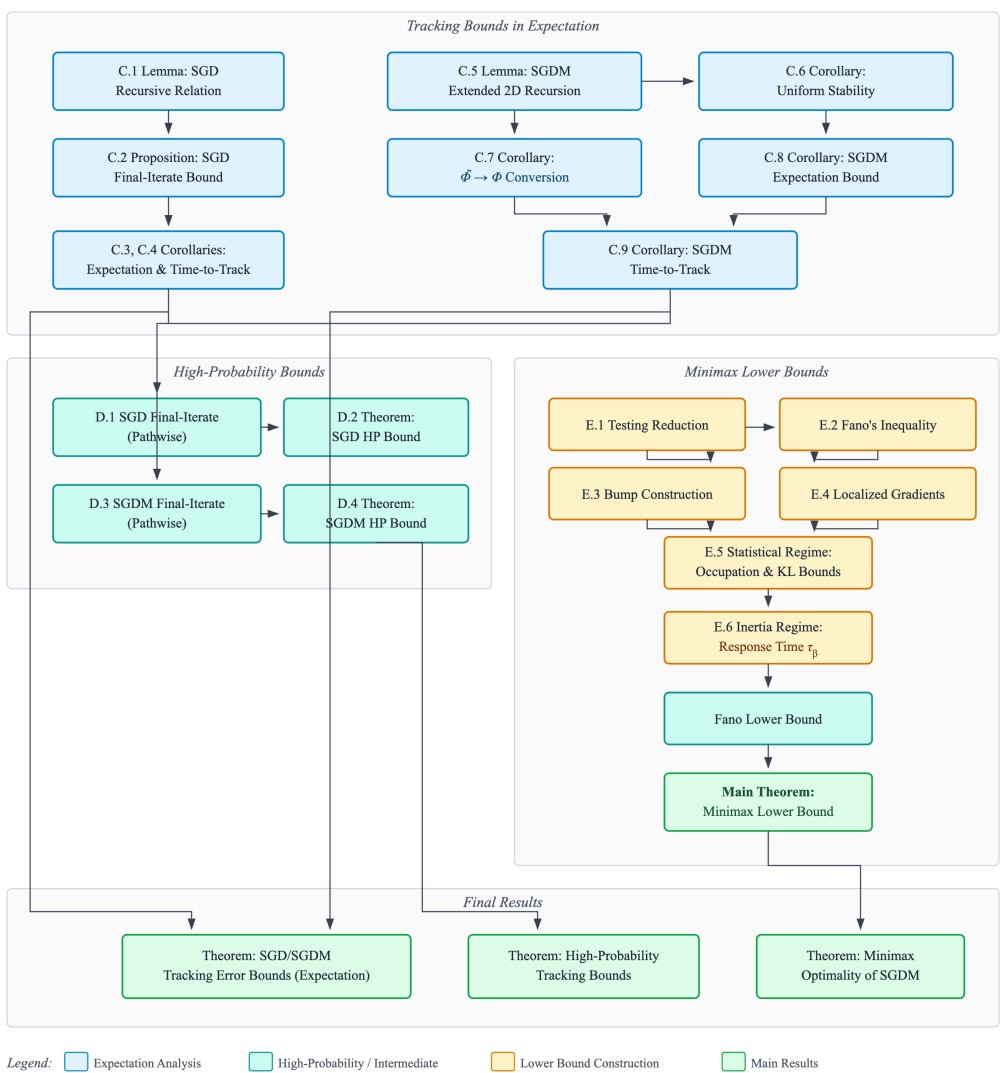

*Figure 3.* **Structure of the proof:** Arrows represent the main logical implications between results. The section *Tracking Bounds in Expectation* develops the core recursive analysis for both (SGD) (C.1–C.4) and (SGDM) (C.5–C.9), including the key $\widetilde{\Phi} \to \Phi$ conversion (C.7). *High-Probability Bounds* lifts these results to pathwise guarantees via concentration arguments (D.1–D.4). The *Minimax Lower Bounds* section constructs the information-theoretic lower bound through a testing reduction and Fano's inequality (E.1–E.2), localized bump functions (E.3–E.4), and separate analyses of the statistical (E.5) and inertia (E.6) regimes. Our main contributions are the unified tracking analysis and the matching lower bound establishing minimax optimality of (SGDM).

the variation budget (Lemma E.10). Section E.5 completes the information-limited lower bound by combining the KL control with Fano inequality (Corollary E.2) and tuning $(a, J)$ to enforce $\mathrm{GVar}_{p,q} \leq \mathbb{V}_T$ while keeping environments statistically indistinguishable (Lemma E.13), yielding the noise/variation term (205). Finally, Section E.6 isolates the *inertia-limited* obstruction: we analyze the (SGDM) deterministic response on a drifting quadratic (Proposition E.1), show the response time obeys $\tau_\beta \gtrsim L/(\mu(1 - \beta))$ under the stability cap (213), and convert this lag into regret under block switching (Theorem E.1), yielding the inertia term (219). Taking the maximum of the statistical and inertia lower bounds gives the claimed minimax scaling for $\Pi_\beta$.

For convenience, the main dependencies between sections of the Appendix are summarized in Figure 3.

## B.1. Problem setup and standing assumptions

We work on a filtered probability space $(\Omega, \mathcal{F}, (\mathcal{F}_t)_{t\geq 0}, \mathbb{P})$ with $\mathcal{F}_0 = \{\emptyset, \Omega\}$, and let $(X_t)_{t\geq 0}$ be an $\mathbb{F}$–adapted process. We take the natural filtration $\mathcal{F}_t := \sigma(X_0, \ldots, X_t)$. For each $t \geq 0$, define the *conditional (predictable) objective* and conditional mean gradient by

$$G_{t+1}(\boldsymbol{\theta}) := \mathbb{E}[g(\boldsymbol{\theta}, X_{t+1}) \mid \mathcal{F}_t], \qquad \boldsymbol{m}_{t+1}(\boldsymbol{\theta}) := \mathbb{E}[\nabla_{\boldsymbol{\theta}} g(\boldsymbol{\theta}, X_{t+1}) \mid \mathcal{F}_t].$$

We write $\boldsymbol{\theta}_{t+1}^\star \in \arg\min_{\boldsymbol{\theta}\in\mathbb{R}^d} G_{t+1}(\boldsymbol{\theta})$ for a (measurable) conditional minimizer (a.s. unique in our strongly convex settings), and define the tracking error and minimizer drift as

$$\boldsymbol{e}_t := \boldsymbol{\theta}_t - \boldsymbol{\theta}_t^\star, \qquad \boldsymbol{\Delta}_t := \boldsymbol{\theta}_t^\star - \boldsymbol{\theta}_{t+1}^\star.$$

**Assumption B.1** (Stochastic predictability framework). *There exists a filtered probability space $(\Omega, \mathcal{F}, (\mathcal{F}_t)_{t\geq 0}, \mathbb{P})$ with $\mathcal{F}_0 = \{\emptyset, \Omega\}$. Let $(X_t)_{t\geq 0}$ be an $\mathbb{F}$–adapted process, i.e., $X_t$ is $\mathcal{F}_t$–measurable for all $t$. For each $t \geq 0$, let $\Pi_{t+1}$ denote the regular conditional law of $X_{t+1}$ given $\mathcal{F}_t$, i.e., $\Pi_{t+1}(A) = \mathbb{P}(X_{t+1} \in A \mid \mathcal{F}_t)$ a.s. for every measurable set $A$, and assume $\Pi_{t+1}$ is $\mathcal{F}_t$–measurable. Define the conditional risk*

$$G_{t+1}(\boldsymbol{\theta}) := \mathbb{E}[g(\boldsymbol{\theta}, X_{t+1}) \mid \mathcal{F}_t] = \mathbb{E}_{X\sim\Pi_{t+1}}[g(\boldsymbol{\theta}, X)],$$

*and let $\boldsymbol{\theta}_{t+1}^\star \in \arg\min_{\boldsymbol{\theta}\in\mathbb{R}^d} G_{t+1}(\boldsymbol{\theta})$ denote a (measurable) minimizer. Assume the following hold for all $t \geq 0$:*

1. *(**Predictable minimizer**) $\boldsymbol{\theta}_{t+1}^\star$ is $\mathcal{F}_t$–measurable.*

2. *(**Algorithm adaptedness**) The iterate $\boldsymbol{\theta}_t$ is $\mathcal{F}_t$–measurable.*

3. *(**Martingale difference noise**) Define the conditional mean gradient $\boldsymbol{m}_{t+1}(\boldsymbol{\theta}) = \mathbb{E}[\nabla_{\boldsymbol{\theta}} g(\boldsymbol{\theta}, X_{t+1}) \mid \mathcal{F}_t]$, and the gradient noise $\boldsymbol{\xi}_{t+1}(\boldsymbol{\theta}) = \nabla_{\boldsymbol{\theta}} g(\boldsymbol{\theta}, X_{t+1}) - \boldsymbol{m}_{t+1}(\boldsymbol{\theta})$. Then $\boldsymbol{\xi}_{t+1}(\boldsymbol{\theta})$ is $\mathcal{F}_{t+1}$–measurable and satisfies $\mathbb{E}[\boldsymbol{\xi}_{t+1}(\boldsymbol{\theta}) \mid \mathcal{F}_t] = \boldsymbol{0}$ a.s. for all $\boldsymbol{\theta} \in \mathbb{R}^d$.*

Under this framework, SGD admits the decomposition

$$\boldsymbol{\theta}_{t+1} = \boldsymbol{\theta}_t - \gamma_t \boldsymbol{m}_{t+1}(\boldsymbol{\theta}_t) - \gamma_t \boldsymbol{\xi}_{t+1}(\theta_t). \tag{SGD}$$

For momentum, we use a two-parameter formulation that captures Heavy-Ball and Nesterov as special cases:

$$\begin{aligned}
\boldsymbol{\psi}_t &= \boldsymbol{\theta}_t + \beta_1(\boldsymbol{\theta}_t - \boldsymbol{\theta}_{t-1}), \\
\boldsymbol{\theta}_{t+1} &= \boldsymbol{\psi}_t - \gamma_t \boldsymbol{m}_{t+1}(\boldsymbol{\psi}_t) - \gamma_t \boldsymbol{\xi}_{t+1}(\boldsymbol{\psi}_t) + \beta_2(\boldsymbol{\psi}_t - \boldsymbol{\psi}_{t-1}),
\end{aligned} \tag{SGDM}$$

with $\beta_1, \beta_2 \in [0, 1)$ satisfying $\beta_1 + \beta_2 = \beta$ and $\beta_1\beta_2 = 0$ for a fixed $\beta \in [0, 1)$.

To obtain contraction and stability, we impose the following uniform regularity on the conditional mean gradient map.

**Assumption B.2** (Uniform $\mu$-strong monotonicity). *There exists a constant $\mu > 0$ such that for all $t \geq 0$ and all $\boldsymbol{\theta}, \boldsymbol{\theta}' \in \mathbb{R}^d$,*

$$\langle \boldsymbol{m}_{t+1}(\boldsymbol{\theta}) - \boldsymbol{m}_{t+1}(\boldsymbol{\theta}'), \, \boldsymbol{\theta} - \boldsymbol{\theta}' \rangle \geq \mu\|\boldsymbol{\theta} - \boldsymbol{\theta}'\|^2.$$

**Assumption B.3** (Uniform $L$-Lipschitz continuity). *There exists a constant $L > 0$ such that for all $t \geq 0$ and all $\boldsymbol{\theta}, \boldsymbol{\theta}' \in \mathbb{R}^d$,*

$$\|\boldsymbol{m}_{t+1}(\boldsymbol{\theta}) - \boldsymbol{m}_{t+1}(\boldsymbol{\theta}')\| \leq L\|\boldsymbol{\theta} - \boldsymbol{\theta}'\|.$$

# C. Proofs of tracking error bounds in expectation

For this section, we will assume the following hold for the $\ell_2$ norm of the minimizer drift and gradient noise second moments (Assumption C.1):

**Assumption C.1** (Second-moment bounds). *There exist constants $\Delta, \sigma > 0$ such that for all $t \geq 0$,*

1. *(**Minimizer drift**) The minimizer drift $\boldsymbol{\Delta}_t$ satisfies $\mathbb{E}[\|\boldsymbol{\Delta_t}\|^2] \leq \Delta^2$ a.s.*

2. *(**Gradient noise along iterates**) The gradient noise $\boldsymbol{\xi}_{t+1}$ satisfies $\mathbb{E}[\|\boldsymbol{\xi}_{t+1}(\boldsymbol{\theta}_t)\|^2] \leq \sigma^2$ and $\mathbb{E}[\|\boldsymbol{\xi}_{t+1}(\boldsymbol{\psi}_t)\|^2] \leq \sigma^2$ a.s.*

## C.1. Proof for SGD tracking error bound

First we will prove a recursive relation for the tracking error that we will subsequently use for our expectation and high-probability bounds:

**Lemma C.1** (Recursive relation for the (SGD) tracking error). *For $\forall t \geq 0$ and $\gamma_t \leq \min\{\mu/L^2, 1/L\}$, the following recursive relation for the tracking error holds*

$$\|\boldsymbol{\theta}_{t+1} - \boldsymbol{\theta}_{t+1}^\star\|^2 \leq \left(1 - \frac{\gamma_t \mu}{2}\right)\|\boldsymbol{\theta}_t - \boldsymbol{\theta}_t^\star\|^2 + \frac{2}{\gamma_t \mu}\|\boldsymbol{\Delta}_t\|^2 + \gamma_t^2 \|\boldsymbol{\xi}_{t+1}(\boldsymbol{\theta}_t)\|^2 + M_{t+1}$$

*where the martingale increment is $M_{t+1} = -2\gamma_t \langle \boldsymbol{d}_t - \gamma_t \boldsymbol{m}_{t+1}(\boldsymbol{\theta}_t), \boldsymbol{\xi}_{t+1}(\boldsymbol{\theta}_t)\rangle$ with $\boldsymbol{d}_t := \boldsymbol{\theta}_t - \boldsymbol{\theta}_{t+1}^\star$.*

*Proof of Lemma C.1.* Before we proceed, let us recall some notation that we will use in our analysis. Let $(\mathcal{F}_t)_{t \geq 1}$ be the natural filtration $\mathcal{F}_t = \sigma(X_0, \ldots, X_t)$. Define $\boldsymbol{m}_{t+1}(\boldsymbol{\theta}) = \mathbb{E}[\nabla_{\boldsymbol{\theta}} g(\boldsymbol{\theta}, X_{t+1}) \mid \mathcal{F}_t]$ to be the conditional expected gradient and the gradient noise as $\boldsymbol{\xi}_{t+1}(\boldsymbol{\theta}) = \nabla_{\boldsymbol{\theta}} g(\boldsymbol{\theta}, X_{t+1}) - \boldsymbol{m}_{t+1}(\boldsymbol{\theta})$ so that $\mathbb{E}[\boldsymbol{\xi}_{t+1}(\boldsymbol{\theta}) \mid \mathcal{F}_t] = \boldsymbol{0}$. Define $\boldsymbol{d}_t = \boldsymbol{\theta}_t - \boldsymbol{\theta}_{t+1}^\star$. Then notice that we can write the minimizer drift as follows:

$$\begin{aligned}
\boldsymbol{d}_t &= \boldsymbol{\theta}_t - \boldsymbol{\theta}_{t+1}^\star \\
&= \boldsymbol{\theta}_t - \boldsymbol{\theta}_t^\star + \boldsymbol{\theta}_t^\star - \boldsymbol{\theta}_{t+1}^\star \\
&= \boldsymbol{e}_t + \boldsymbol{\Delta}_t.
\end{aligned}$$

Then from the (SGD) update rule, we have that

$$\begin{aligned}
\boldsymbol{\theta}_{t+1} - \boldsymbol{\theta}_{t+1}^\star &= \boldsymbol{\theta}_t - \gamma_t\left(\boldsymbol{m}_{t+1}(\boldsymbol{\theta}_t) + \boldsymbol{\xi}_{t+1}(\boldsymbol{\theta}_t)\right) - \boldsymbol{\theta}_{t+1}^\star \\
&= \boldsymbol{d}_t - \gamma_t \boldsymbol{m}_{t+1}(\boldsymbol{\theta}_t) - \gamma_t \boldsymbol{\xi}_{t+1}(\boldsymbol{\theta}_t).
\end{aligned}$$

Taking the $\ell_2$ norm of each side, we find that

$$\|\boldsymbol{\theta}_{t+1} - \boldsymbol{\theta}_{t+1}^\star\|^2 = \|\boldsymbol{d}_t - \gamma_t \boldsymbol{m}_{t+1}(\boldsymbol{\theta}_t)\|^2 - 2\gamma_t \langle \boldsymbol{d}_t - \gamma_t \boldsymbol{m}_{t+1}(\boldsymbol{\theta}_t), \boldsymbol{\xi}_{t+1}(\boldsymbol{\theta}_t)\rangle + \gamma_t^2\|\boldsymbol{\xi}_{t+1}(\boldsymbol{\theta}_t)\|^2.$$

Define $\boldsymbol{\Phi}_{t+1}(\boldsymbol{\theta}_t) = \boldsymbol{d}_t - \gamma_t \boldsymbol{m}_{t+1}(\boldsymbol{\theta}_t)$. By definition we have

$$\|\boldsymbol{\Phi}_{t+1}(\boldsymbol{\theta}_t)\|^2 = \|\boldsymbol{d}_t\|^2 - 2\gamma_t \langle \boldsymbol{d}_t, \boldsymbol{m}_{t+1}(\boldsymbol{\theta}_t)\rangle + \gamma_t^2\|\boldsymbol{m}_{t+1}(\boldsymbol{\theta}_t)\|^2.$$

By $\mu$-strong monotonicity of $\boldsymbol{m}_{t+1}$ (Assumption B.2), we have

$$\begin{aligned}
\langle \boldsymbol{d}_t, \boldsymbol{m}_{t+1}(\boldsymbol{\theta}_t)\rangle &= \langle \boldsymbol{\theta}_t - \boldsymbol{\theta}_{t+1}^\star, \boldsymbol{m}_{t+1}(\boldsymbol{\theta}_t) - \boldsymbol{m}_{t+1}(\boldsymbol{\theta}_{t+1}^\star)\rangle \\
&\geq \mu\|\boldsymbol{\theta}_t - \boldsymbol{\theta}_{t+1}^\star\|^2 \\
&= \mu\|\boldsymbol{d}_t\|^2.
\end{aligned} \tag{1}$$

where the first equality holds since $\boldsymbol{m}_{t+1}(\boldsymbol{\theta}_{t+1}^\star) = \boldsymbol{0}$. We also have by Lipschitz continuity of $\boldsymbol{m}_{t+1}$ (Assumption B.3) that

$$\|\boldsymbol{m}_{t+1}(\boldsymbol{\theta}_t)\|^2 \leq L^2\|\boldsymbol{d}_t\|^2.$$

Combining this with (1), we find that

$$\|\boldsymbol{\Phi}_{t+1}(\boldsymbol{\theta}_t)\|^2 \leq (1 - 2\gamma_t\mu + \gamma_t^2 L^2)\|\boldsymbol{d}_t\|^2. \tag{2}$$

If we take $\gamma_t \leq \mu/L^2$ for $\forall t \geq 0$, then we have that $1 - 2\gamma_t\mu + \gamma_t^2 L^2 \leq 1 - \gamma_t\mu$. Let $\rho_t := 1 - \gamma_t\mu$. Now let the martingale increment be

$$M_{t+1} = -2\gamma_t \langle \boldsymbol{d}_t - \gamma_t \boldsymbol{m}_{t+1}(\boldsymbol{\theta}_t), \boldsymbol{\xi}_{t+1}(\boldsymbol{\theta}_t)\rangle.$$

Then combining our restriction on $\gamma_t$ with (2), we find that

$$\|\boldsymbol{\theta}_{t+1} - \boldsymbol{\theta}_{t+1}^\star\|^2 \leq \rho_t\|\boldsymbol{d}_t\|^2 + \gamma_t^2\|\boldsymbol{\xi}_{t+1}(\boldsymbol{\theta}_t)\|^2 + M_{t+1}. \tag{3}$$

Note that in the literature, one will usually apply Young's inequality to $M_{t+1}$, resulting in the right side being in terms of $\boldsymbol{d}_t$ and $\boldsymbol{\xi}_{t+1}$. However, since $\boldsymbol{d}_t$ can be written in terms of $\boldsymbol{\Delta}_t$ to relate back to the drift, this will result in us needing to apply

an iterating MGF approach (see (Cutler et al., 2021) for details). We instead will apply Young's inequality to $d_t$. This will help us in avoiding any tail-assumptions on the drift, thus yielding a more generalizable high-probability bound. By Young's inequality, for any $\alpha > 0$, we have

$$\|d_t\|^2 \leq (1 + \alpha)\|\theta_t - \theta_t^\star\|^2 + \left(1 + \frac{1}{\alpha}\right)\|\Delta_t\|^2. \tag{4}$$

Combining (3) with (4), we find that

$$\|\theta_{t+1} - \theta_{t+1}^\star\|^2 \leq \rho_t(1 + \alpha)\|\theta_t - \theta_t^\star\|^2 + \rho_t\left(1 + \frac{1}{\alpha}\right)\|\Delta_t\|^2 + \gamma_t^2\|\xi_{t+1}(\theta_t)\|^2 + M_{t+1}. \tag{5}$$

Since we want $\rho_t(1 + \alpha) \leq 1$ for convergence guarantees as $t \to \infty$, we take $\alpha = (1 - \rho_t)/2\rho_t$. Using this, we find

$$\|\theta_{t+1} - \theta_{t+1}^\star\|^2 \leq \left(\frac{1 + \rho_t}{2}\right)\|\theta_t - \theta_t^\star\|^2 + \frac{\rho_t(1 + \rho_t)}{1 - \rho_t}\|\Delta_t\|^2 + \gamma_t^2\|\xi_{t+1}(\theta_t)\|^2 + M_{t+1}. \tag{6}$$

Substituting in $\rho_t := 1 - \gamma_t\mu$, we find

$$\|\theta_{t+1} - \theta_{t+1}^\star\|^2 \leq \left(1 - \frac{\gamma_t\mu}{2}\right)\|\theta_t - \theta_t^\star\|^2 + \frac{(1 - \gamma_t\mu)(2 - \gamma_t\mu)}{\gamma_t\mu}\|\Delta_t\|^2 + \gamma_t^2\|\xi_{t+1}(\theta_t)\|^2 + M_{t+1}.$$

Note that $\mu$-strong monotonicity and Lipschitz continuity of $m_{t+1}$ jointly imply that $\mu \leq L$. As a consequence of taking $\gamma_t \leq \mu/L^2$, we have $\gamma_t\mu \leq 1$ which implies $-3 + \gamma_t\mu \leq 0$. This implies that

$$\frac{(1 - \gamma_t\mu)(2 - \gamma_t\mu)}{\gamma_t\mu} \leq \frac{2}{\gamma_t\mu}.$$

Thus we can conclude that

$$\|\theta_{t+1} - \theta_{t+1}^\star\|^2 \leq \left(1 - \frac{\gamma_t\mu}{2}\right)\|\theta_t - \theta_t^\star\|^2 + \frac{2}{\gamma_t\mu}\|\Delta_t\|^2 + \gamma_t^2\|\xi_{t+1}(\theta_t)\|^2 + M_{t+1}.$$

$\square$

Applying Lemma C.1 recursively and setting a constant step-size $\gamma_t = \gamma$, we obtain the following result:

**Proposition C.1** (Final-iterate tracking error bound for (SGD)). *Let $\gamma \leq \min\left\{\mu/L^2, 1/L\right\}$. Then for $\forall t \geq 0$, the following bound holds:*

$$\|\theta_{t+1} - \theta_{t+1}^\star\|^2 \leq \left(1 - \frac{\gamma\mu}{2}\right)^{t+1}\|\theta_0 - \theta_0^\star\|^2 + \frac{2}{\gamma\mu}\sum_{\ell=0}^{t}\left(1 - \frac{\gamma\mu}{2}\right)^{t-\ell}\|\Delta_\ell\|^2$$

$$+ \gamma^2\sum_{\ell=0}^{t}\left(1 - \frac{\gamma\mu}{2}\right)^{t-\ell}\|\xi_{\ell+1}(\theta_\ell)\|^2 + \sum_{\ell=0}^{t}\left(1 - \frac{\gamma\mu}{2}\right)^{t-\ell}M_{\ell+1}$$

*where $M_{t+1} := -2\gamma\langle d_t - \gamma m_{t+1}(\theta_t), \xi_{t+1}(\theta_t)\rangle$ with $d_t = \theta_t - \theta_{t+1}^\star$.*

We can finally obtain a tracking error bound in expectation for (SGD) using Assumption 3.1.

**Corollary C.1** (Tracking error bound in expectation for (SGD)). *Under Assumption 3.1, for $\forall t \geq 0$ and $\gamma \leq \min\left\{\mu/L^2, 1/L\right\}$, the following tracking error bound holds in expectation for (SGD):*

$$\mathbb{E}\|\theta_{t+1} - \theta_{t+1}^\star\|^2 \leq \left(1 - \frac{\gamma\mu}{2}\right)^{t+1}\|\theta_0 - \theta_0^\star\|^2 + \frac{4\Delta^2}{\gamma^2\mu^2} + \frac{\sigma^2\gamma}{\mu}.$$

*Proof of Corollary C.1.* Recall that $M_{t+1} = -2\gamma\langle d_t - \gamma m_{t+1}(\theta_t), \xi_{t+1}(\theta_t)\rangle$. Since $m_{t+1}(\theta_t)$ is $\mathcal{F}_t$ measurable and $\mathbb{E}[\xi_{t+1}(\theta_t) \mid \mathcal{F}_t] = 0$, we have that

$$\mathbb{E}[M_{t+1} \mid \mathcal{F}_t] = -2\gamma\langle d_t - \gamma m_{t+1}(\theta_t), \mathbb{E}[\xi_{t+1}(\theta_t) \mid \mathcal{F}_t]\rangle = 0.$$

This implies that $M_{t+1}$ is a martingale difference sequence (MDS). Thus by iterated expectation and using the fact that $\sum_{\ell\geq 0}\delta^\ell = 1/(1 - \delta)$, we are done and the proof is complete. $\square$

Using Corollary C.1, we can obtain the following result which gives us an algorithmic guarantee for (SGD):

**Corollary C.2** (Time to reach the asymptotic tracking error in expectation for (SGD))**.** *Assume $\gamma_t \in (0, 1/(2L)]$ for all $t \geq 0$. For any constant $\gamma \in (0, 1/2L]$, define*

$$\mathcal{E}(\gamma) := \frac{\sigma^2 \gamma}{\mu} + \frac{4\Delta^2}{\mu^2 \gamma^2}, \qquad \gamma^\star \in \arg \min_{\gamma \in (0, 1/2L]} \mathcal{E}(\gamma), \qquad \mathcal{E} := \mathcal{E}(\gamma^\star).$$

*Then we have the following:*

1. *(**Constant learning rate**). If $\gamma_t \equiv \gamma^\star$, then for all $t \geq 0$,*

$$\mathbb{E}\|\boldsymbol{\theta}_{t+1} - \boldsymbol{\theta}_{t+1}^\star\|^2 \lesssim \mathcal{E} \text{ after time } t \lesssim \frac{1}{\mu \gamma^\star} \log\Big(\frac{\|\boldsymbol{\theta}_0 - \boldsymbol{\theta}_0^\star\|^2}{\mathcal{E}}\Big).$$

2. *(**Step-decay schedule in the low drift-to-noise regime**). Suppose $\gamma^\star < 1/2L$ (equivalently, the minimizer of $\mathcal{E}(\gamma)$ is not at the smoothness cap), so that*

$$\gamma^\star = \Big(\frac{8\Delta^2}{\mu \sigma^2}\Big)^{1/3}, \qquad \mathcal{E} = 3\Big(\frac{\Delta \sigma^2}{\mu^2}\Big)^{2/3}.$$

*Define epochs $k = 0, 1, \ldots, K-1$ with*

$$\gamma_0 := \frac{1}{2L}, \qquad \gamma_k := \frac{\gamma_{k-1} + \gamma^\star}{2} \quad (k \geq 1), \qquad K := 1 + \Big\lceil \log_2 \Big(\frac{\gamma_0}{\gamma^\star}\Big) \Big\rceil,$$

*and epoch lengths*

$$T_0 := \Big\lceil \frac{2}{\mu \gamma_0} \log\Big(\frac{2\|\boldsymbol{\theta}_0 - \boldsymbol{\theta}_0^\star\|^2}{\mathcal{E}(\gamma_0)}\Big) \Big\rceil, \qquad T_k := \Big\lceil \frac{2 \log 4}{\mu \gamma_k} \Big\rceil \quad (k \geq 1).$$

*Run (SGD) with constant stepsize $\gamma_k$ for $T_k$ iterations in epoch $k$, starting from $\boldsymbol{\theta}_0$. Let $T := \sum_{k=0}^{K-1} T_k$ be the total horizon. Then the final iterate satisfies*

$$\mathbb{E}\|\boldsymbol{\theta}_T - \boldsymbol{\theta}_T^\star\|^2 \lesssim \mathcal{E} \text{ after time } T \lesssim \frac{L}{\mu} \log\Big(\frac{\|\boldsymbol{\theta}_0 - \boldsymbol{\theta}_0^\star\|^2}{\mathcal{E}}\Big) + \frac{\sigma^2}{\mu^2 \mathcal{E}}.$$

*Proof of Corollary C.2.* From Corollary C.1, we have the following

$$\mathbb{E}\|\boldsymbol{\theta}_{t+1} - \boldsymbol{\theta}_{t+1}^\star\|^2 \leq \Big(1 - \frac{\gamma \mu}{2}\Big)^{t+1} \|\boldsymbol{\theta}_0 - \boldsymbol{\theta}_0^\star\|^2 + \frac{4\Delta^2}{\gamma^2 \mu^2} + \frac{\sigma^2 \gamma}{\mu}. \tag{7}$$

**Proof of (i).** By (7) and $\gamma = \gamma^\star$, using $\mathcal{E}(\gamma) := \sigma^2 \gamma / \mu + 4\Delta^2 / \mu^2 \gamma^2$, we have:

$$\mathbb{E}\|\boldsymbol{\theta}_{t+1} - \boldsymbol{\theta}_{t+1}^\star\|^2 \leq \Big(1 - \frac{\mu \gamma^\star}{2}\Big)^{t+1} \|\boldsymbol{\theta}_0 - \boldsymbol{\theta}_0^\star\|^2 + \mathcal{E}(\gamma^\star).$$

If $t \geq (2/\mu\gamma^\star) \log(\|\boldsymbol{\theta}_0 - \boldsymbol{\theta}_0^\star\|^2/\mathcal{E})$, then $\left(1 - \mu\gamma^\star/2\right)^{t+1} \leq \exp(-(t+1)\mu\gamma^\star/2) \leq \mathcal{E}/\|\boldsymbol{\theta}_0 - \boldsymbol{\theta}_0^\star\|^2$, and hence $\mathbb{E}\|\boldsymbol{\theta}_{t+1} - \boldsymbol{\theta}_{t+1}^\star\|^2 \leq 2\mathcal{E}$.

**Proof of (ii).** Let $t_0 := 0$ and $t_{k+1} := t_k + T_k$. Define epoch iterates $\boldsymbol{x}_k := \boldsymbol{\theta}_{t_k}$ and corresponding minimizers $\boldsymbol{x}_k^\star := \boldsymbol{\theta}_{t_k}^\star$. Applying (7) inside epoch $k$ with stepsize $\gamma_k$ and length $T_k$ yields

$$\mathbb{E}\|\boldsymbol{x}_{k+1} - \boldsymbol{x}_{k+1}^\star\|^2 \leq \Big(1 - \frac{\mu \gamma_k}{2}\Big)^{T_k} \mathbb{E}\|\boldsymbol{x}_k - \boldsymbol{x}_k^\star\|^2 + \mathcal{E}(\gamma_k).$$

For $k \geq 1$, by the choice of $T_k$ and $\log(1 - x) \leq -x$,

$$\Big(1 - \frac{\mu \gamma_k}{2}\Big)^{T_k} \leq \exp\Big(-\frac{\mu \gamma_k}{2} T_k\Big) \leq \exp(-\log 4) = \frac{1}{4},$$

so for $k \geq 1$,

$$\mathbb{E}\|\boldsymbol{x}_{k+1} - \boldsymbol{x}_{k+1}^\star\|^2 \leq \frac{1}{4}\mathbb{E}\|\boldsymbol{x}_k - \boldsymbol{x}_k^\star\|^2 + \mathcal{E}(\gamma_k). \tag{8}$$

*Base epoch.* By the definition of $T_0$, we have $\left(1 - \mu\gamma_0/2\right)^{T_0}\|\boldsymbol{x}_0 - \boldsymbol{x}_0^\star\|^2 \leq \mathcal{E}(\gamma_0)$, hence

$$\mathbb{E}\|\boldsymbol{x}_1 - \boldsymbol{x}_1^\star\|^2 \leq \left(1 - \frac{\mu\gamma_0}{2}\right)^{T_0}\|\boldsymbol{x}_0 - \boldsymbol{x}_0^\star\|^2 + \mathcal{E}(\gamma_0) \leq 2\mathcal{E}(\gamma_0).$$

*Induction.* We claim that for all $k \geq 1$,

$$\mathbb{E}\|\boldsymbol{x}_k - \boldsymbol{x}_k^\star\|^2 \leq 2\mathcal{E}(\gamma_{k-1}).$$

Assume this holds for some $k \geq 1$. Using (8),

$$\mathbb{E}\|\boldsymbol{x}_{k+1} - \boldsymbol{x}_{k+1}^\star\|^2 \leq \frac{1}{4} \cdot 2\mathcal{E}(\gamma_{k-1}) + \mathcal{E}(\gamma_k) = \frac{1}{2}\mathcal{E}(\gamma_{k-1}) + \mathcal{E}(\gamma_k).$$

Since $\gamma_k = (\gamma_{k-1} + \gamma^\star)/2$, we have $\gamma_{k-1} \leq 2\gamma_k$. For $\mathcal{E}(\gamma) = A/\gamma^2 + B\gamma$ with $A = 4\Delta^2/\mu^2$ and $B = \sigma^2/\mu$, one checks that for all $\gamma \geq \gamma^\star$,

$$\mathcal{E}(2\gamma) \leq 2\mathcal{E}(\gamma).$$

Because $\gamma_k \geq \gamma^\star$, this gives $\mathcal{E}(\gamma_{k-1}) \leq \mathcal{E}(2\gamma_k) \leq 2\mathcal{E}(\gamma_k)$, and thus

$$\mathbb{E}\|\boldsymbol{x}_{k+1} - \boldsymbol{x}_{k+1}^\star\|^2 \leq \frac{1}{2} \cdot 2\mathcal{E}(\gamma_k) + \mathcal{E}(\gamma_k) = 2\mathcal{E}(\gamma_k),$$

closing the induction. Hence $\mathbb{E}\|\boldsymbol{x}_K - \boldsymbol{x}_K^\star\|^2 \leq 2\mathcal{E}(\gamma_{K-1})$.

Next, by the definition of $K$,

$$\gamma_{K-1} - \gamma^\star = \frac{\gamma_0 - \gamma^\star}{2^{K-1}} \leq \frac{\gamma_0}{2^{K-1}} \leq \gamma^\star, \quad \text{so} \quad \gamma_{K-1} \leq 2\gamma^\star.$$

Therefore $\mathcal{E}(\gamma_{K-1}) \leq \mathcal{E}(2\gamma^\star) \leq 2\mathcal{E}(\gamma^\star) = 2\mathcal{E}$, yielding

$$\mathbb{E}\|\boldsymbol{e}_T\|^2 = \mathbb{E}\|\boldsymbol{x}_K - \boldsymbol{x}_K^\star\|^2 \leq 2\mathcal{E}(\gamma_{K-1}) \leq 4\mathcal{E}.$$

*Time bound.* Since $\mathcal{E}(\gamma_0) \geq \mathcal{E}$, we have

$$T_0 \leq \frac{2}{\mu\gamma_0}\log\left(\frac{2\|\boldsymbol{x}_0 - \boldsymbol{x}_0^\star\|^2}{\mathcal{E}}\right) + 1 = \frac{4L}{\mu}\log\left(\frac{2\|\boldsymbol{x}_0 - \boldsymbol{x}_0^\star\|^2}{\mathcal{E}}\right) + 1.$$

Moreover, for $k \geq 1$, $\gamma_k \geq \gamma_0/2^{k+1}$, hence

$$\sum_{k=1}^{K-1}\frac{1}{\gamma_k} \leq \sum_{k=1}^{K-1}\frac{2^{k+1}}{\gamma_0} \leq \frac{2^{K+1}}{\gamma_0} \leq \frac{4}{\gamma^\star}.$$

Thus

$$\sum_{k=1}^{K-1}T_k \leq \sum_{k=1}^{K-1}\left(\frac{2\log 4}{\mu\gamma_k} + 1\right) \leq \frac{8\log 4}{\mu}\sum_{k=1}^{K-1}\frac{1}{\gamma_k} + \mathcal{O}(K) \leq \frac{32\log 4}{\mu\gamma^\star} + \mathcal{O}(K).$$

In the low drift-to-noise regime, $\gamma^\star = (8\Delta^2/(\mu\sigma^2))^{1/3}$ and $\mathcal{E} = 3(\Delta\sigma^2/\mu^2)^{2/3}$, so $1/\mu\gamma^\star = 3/2 \cdot \sigma^2/\mu^2\mathcal{E}$. Substituting yields

$$\sum_{k=1}^{K-1}T_k \leq 32\log 4 \cdot \frac{3}{2} \cdot \frac{\sigma^2}{\mu^2\mathcal{E}} + \mathcal{O}(1) = 48\log 4 \cdot \frac{\sigma^2}{\mu^2\mathcal{E}} + \mathcal{O}(1).$$

Combining with the bound on $T_0$ proves the stated horizon bound up to universal constants. $\square$

## C.2. Proof for SGDM tracking error bound

For (SGDM), we will take a different approach to prove bounds on the tracking error in expectation and with high-probability. We will instead work with a 2D state-space view of (SGDM) by defining extended state vectors. We will first introduce the transformation matrices:

$$
\mathbf{V} = \begin{bmatrix} \mathbf{I}_d & -\beta\mathbf{I}_d \\ \mathbf{I}_d & -\mathbf{I}_d \end{bmatrix}, \; \mathbf{V}^{-1} = \frac{1}{1-\beta} \begin{bmatrix} \mathbf{I}_d & -\beta\mathbf{I}_d \\ \mathbf{I}_d & -\mathbf{I}_d \end{bmatrix}. \tag{9}
$$

Recall the following (SGDM) updates:

$$
\begin{aligned}
\boldsymbol{\psi}_t &= \boldsymbol{\theta}_t + \beta_1(\boldsymbol{\theta}_t - \boldsymbol{\theta}_{t-1}) \\
\boldsymbol{\theta}_{t+1} &= \boldsymbol{\psi}_t - \gamma_t \boldsymbol{m}_{t+1}(\boldsymbol{\psi}_t) - \gamma_t \boldsymbol{\xi}_{t+1}(\boldsymbol{\psi}_t) + \beta_2(\boldsymbol{\psi}_t - \boldsymbol{\psi}_{t-1}).
\end{aligned} \tag{10}
$$

Define $\widetilde{\boldsymbol{\theta}}_t = \boldsymbol{\theta}_t^\star - \boldsymbol{\theta}_t$. Define the transformed error vectors, each of size $2d \times 1$:

$$
\begin{bmatrix} \widehat{\boldsymbol{\theta}}_t \\ \check{\boldsymbol{\theta}}_t \end{bmatrix} \triangleq \mathbf{V}^{-1} \begin{bmatrix} \widetilde{\boldsymbol{\theta}}_t \\ \widetilde{\boldsymbol{\theta}}_{t-1} \end{bmatrix} = \frac{1}{1-\beta} \begin{bmatrix} \widetilde{\boldsymbol{\theta}}_t - \beta\widetilde{\boldsymbol{\theta}}_{t-1} \\ \widetilde{\boldsymbol{\theta}}_t - \widetilde{\boldsymbol{\theta}}_{t-1} \end{bmatrix}. \tag{11}
$$

We can obtain an extended recursion 2D state-space matrix that captures the dynamics of (SGDM). This will prove very useful in our subsequent analysis to obtain expectation and high probability bounds without needing to recurse on the velocity vector $\boldsymbol{v}_t$ or using a Lyapunov stability function argument. Before we proceed with obtaining this extended recursion, we state an assumption that typically holds under sufficient conditions for the dominated convergence theorem to hold:

**Assumption C.2** (Interchanging conditional expectation and gradient). *For each $t \geq 0$, define the conditional objective*

$$
F_{t+1}(\boldsymbol{\theta}) := \mathbb{E}[g(\boldsymbol{\theta}, X_{t+1}) \mid \mathcal{F}_t].
$$

*Assume $F_{t+1}$ is differentiable and*

$$
\nabla F_{t+1}(\boldsymbol{\theta}) = \mathbb{E}[\nabla g(\boldsymbol{\theta}, X_{t+1}) \mid \mathcal{F}_t] = \boldsymbol{m}_{t+1}(\boldsymbol{\theta}) \quad \forall \boldsymbol{\theta} \in \mathbb{R}^d,
$$

*almost surely. (Sufficient conditions are standard dominated-convergence hypotheses.)*

Note that since we assumed $\boldsymbol{m}_{t+1}$ is $\mu$-strongly monotone and Lipschitz continuous (Assumption B.2 and Assumption B.3), it follows that $F_{t+1}$ is $\mu$-strongly convex and $L$-smooth. Equivalently, this implies

$$
\mu\mathbf{I}_d \preceq \nabla^2 F_{t+1}(\boldsymbol{\theta}) \preceq L\mathbf{I}_d \quad \forall \boldsymbol{\theta} \in \mathbb{R}^d. \tag{12}
$$

We now prove the extended 2D recursion. This proof largely follows (Yuan et al., 2016) with a similar form except for a matrix decoupling that arises from the minimizer drift.

**Lemma C.2** (Extended 2D recursion for SGD with momentum). *Under Assumption 2.2, Assumption 2.3, and $\beta_1 + \beta_2 = \beta$ and $\beta_1\beta_2 = 0$ for fixed $\beta \in [0, 1)$, the SGD with momentum update equations can be transformed into the following extended recursion:*

$$
\begin{bmatrix} \widehat{\boldsymbol{\theta}}_{t+1} \\ \check{\boldsymbol{\theta}}_{t+1} \end{bmatrix} = \begin{bmatrix} \mathbf{I}_d - \frac{\gamma_t}{1-\beta}\mathbf{H}_t & \frac{\gamma_t\beta'}{1-\beta}\mathbf{H}_t \\ -\frac{\gamma_t}{1-\beta}\mathbf{H}_t & \beta\mathbf{I}_d + \frac{\gamma_t\beta'}{1-\beta}\mathbf{H}_t \end{bmatrix} \begin{bmatrix} \widehat{\boldsymbol{\theta}}_t \\ \check{\boldsymbol{\theta}}_t \end{bmatrix} + \frac{1}{1-\beta} \begin{bmatrix} -(\mathbf{I}_d - \gamma_t\mathbf{H}_t)\boldsymbol{\Delta}_t - \mathbf{K}_t\boldsymbol{\Delta}_{t-1} \\ -(\mathbf{I}_d - \gamma_t\mathbf{H}_t)\boldsymbol{\Delta}_t - \mathbf{K}_t\boldsymbol{\Delta}_{t-1} \end{bmatrix} + \frac{\gamma_t}{1-\beta} \begin{bmatrix} \boldsymbol{\xi}_{t+1}(\boldsymbol{\psi}_t) \\ \boldsymbol{\xi}_{t+1}(\boldsymbol{\psi}_t) \end{bmatrix} \tag{13}
$$

*where*

$$
\beta' \triangleq \beta\beta_1 + \beta_2 \tag{14}
$$

$$
\mathbf{H}_t \triangleq \int_0^1 \nabla^2 F_{t+1}(\boldsymbol{\theta}_{t+1}^\star + s(\boldsymbol{\psi}_t - \boldsymbol{\theta}_{t+1}^\star))ds \tag{15}
$$

$$
\mathbf{K}_t = -\beta\mathbf{I}_d + \gamma_t\beta_1\mathbf{H}_t. \tag{16}
$$

*Proof of Lemma C.2.* From the (SGDM) update, we have

$$
\boldsymbol{\theta}_{t+1} = \boldsymbol{\psi}_t - \gamma_t\boldsymbol{m}_{t+1}(\boldsymbol{\psi}_t) - \gamma_t\boldsymbol{\xi}_{t+1}(\boldsymbol{\psi}_t) + \beta_2(\boldsymbol{\psi}_t - \boldsymbol{\psi}_{t-1}). \tag{17}
$$

Let $\widetilde{\boldsymbol{\theta}}_t = \boldsymbol{\theta}_t^\star - \boldsymbol{\theta}_t$ and $\widetilde{\boldsymbol{\psi}}_t = \boldsymbol{\theta}_{t+1}^\star - \boldsymbol{\psi}_t$. Subtracting both sides of (17) from $\boldsymbol{\theta}_{t+1}^\star$, we get:

$$\widetilde{\boldsymbol{\theta}}_{t+1} = \widetilde{\boldsymbol{\psi}}_t + \gamma_t \boldsymbol{m}_{t+1}(\boldsymbol{\psi}_t) + \gamma_t \boldsymbol{\xi}_{t+1}(\boldsymbol{\psi}_t) - \beta_2(\boldsymbol{\psi}_t - \boldsymbol{\psi}_{t-1}). \tag{18}$$

We can now appeal to the mean-value theorem:

$$\nabla F_{t+1}(\boldsymbol{\psi}_t) - \nabla F_{t+1}(\boldsymbol{\theta}_{t+1}^\star) = \left( \int_0^1 \nabla^2 F_{t+1}(\boldsymbol{\theta}_{t+1}^\star + s(\boldsymbol{\psi}_t - \boldsymbol{\theta}_{t+1}^\star)) ds \right) (\boldsymbol{\psi}_t - \boldsymbol{\theta}_{t+1}^\star) \triangleq -\boldsymbol{H}_t \widetilde{\boldsymbol{\psi}}_t. \tag{19}$$

Since we have that $\boldsymbol{m}_{t+1}(\boldsymbol{\theta}_{t+1}^\star) = \boldsymbol{0}$, this implies that $\boldsymbol{m}_{t+1}(\boldsymbol{\psi}_t) = -\boldsymbol{H}_t \widetilde{\boldsymbol{\psi}}_t$. Now for any $k \leq t$, let us define

$$\widetilde{\boldsymbol{\theta}}_k^{(t+1)} := \boldsymbol{\theta}_{t+1}^\star - \boldsymbol{\theta}_k. \tag{20}$$

Then we have that $\widetilde{\boldsymbol{\theta}}_{t+1}^{(t+1)} = \widetilde{\boldsymbol{\theta}}_{t+1}$. Using the lookahead from (SGDM), we have

$$\widetilde{\boldsymbol{\psi}}_t = \boldsymbol{\theta}_{t+1}^\star - \boldsymbol{\psi}_t = (\boldsymbol{\theta}_{t+1}^\star - \boldsymbol{\theta}_t) - \beta_1 (\boldsymbol{\theta}_t - \boldsymbol{\theta}_{t-1}) = \widetilde{\boldsymbol{\theta}}_t^{(t+1)} - \beta_1 (\boldsymbol{\theta}_t - \boldsymbol{\theta}_{t-1}). \tag{21}$$

However also note that

$$\boldsymbol{\theta}_t - \boldsymbol{\theta}_{t-1} = -(\boldsymbol{\theta}_{t+1}^\star - \boldsymbol{\theta}_t) + (\boldsymbol{\theta}_{t+1}^\star - \boldsymbol{\theta}_{t-1}) = -\widetilde{\boldsymbol{\theta}}_t^{(t+1)} + \widetilde{\boldsymbol{\theta}}_{t-1}^{(t+1)}. \tag{22}$$

Combining (21) and (22), we find

$$\widetilde{\boldsymbol{\psi}}_t = \widetilde{\boldsymbol{\theta}}_t^{(t+1)} - \beta_1 \left( -\widetilde{\boldsymbol{\theta}}_t^{(t+1)} + \widetilde{\boldsymbol{\theta}}_{t-1}^{(t+1)} \right) = (1 + \beta_1) \widetilde{\boldsymbol{\theta}}_t^{(t+1)} - \beta_1 \widetilde{\boldsymbol{\theta}}_{t-1}^{(t+1)}. \tag{23}$$

Plugging in (23) into (18), we obtain the following:

$$\widetilde{\boldsymbol{\theta}}_{t+1}^{(t+1)} = \boldsymbol{J}_t \widetilde{\boldsymbol{\theta}}_t^{(t+1)} + \boldsymbol{K}_t \widetilde{\boldsymbol{\theta}}_{t-1}^{(t+1)} + \boldsymbol{L}_t \widetilde{\boldsymbol{\theta}}_{t-2}^{(t+1)} + \gamma_t \boldsymbol{\xi}_{t+1}(\boldsymbol{\psi}_t), \tag{24}$$

where

$$\begin{aligned}
\boldsymbol{J}_t &\triangleq (1 + \beta_1)(1 + \beta_2)\mathbf{I}_d - \gamma_t(1 + \beta_1)\boldsymbol{H}_t \\
\boldsymbol{K}_t &\triangleq -(\beta_1 + \beta_2 + \beta_1\beta_2)\mathbf{I}_d + \gamma_t\beta_1\boldsymbol{H}_t \\
\boldsymbol{L}_t &\triangleq \beta_1\beta_2\mathbf{I}_d.
\end{aligned} \tag{25}$$

Since we have that $\beta_1 + \beta_2 = \beta$ and $\beta_1\beta_2 = 0$, we can simplify this as

$$\begin{aligned}
\boldsymbol{J}_t &= (1 + \beta)\mathbf{I}_d - \gamma_t(1 + \beta_1)\boldsymbol{H}_t \\
\boldsymbol{K}_t &= -\beta\mathbf{I}_d + \gamma_t\beta_1\boldsymbol{H}_t \\
\boldsymbol{L}_t &= \boldsymbol{0}.
\end{aligned} \tag{26}$$

This gives us a recursive bound based on an augmented shifted-error state. That is, it is written in a coordinate system relative to the minimizer $\boldsymbol{\theta}_{t+1}^\star$. We now convert this into a recursion for the tracking error $\widetilde{\boldsymbol{\theta}}_t$. Recall that the minimizer drift is defined as $\boldsymbol{\Delta}_t = \boldsymbol{\theta}_t^\star - \boldsymbol{\theta}_{t+1}^\star$. Then we can write

$$\widetilde{\boldsymbol{\theta}}_t^{(t+1)} = \boldsymbol{\theta}_{t+1}^\star - \boldsymbol{\theta}_t = (\boldsymbol{\theta}_t^\star - \boldsymbol{\theta}_t) + (\boldsymbol{\theta}_{t+1}^\star - \boldsymbol{\theta}_t^\star) = \widetilde{\boldsymbol{\theta}}_t - \boldsymbol{\Delta}_t. \tag{27}$$

Similarly we have

$$\widetilde{\boldsymbol{\theta}}_{t-1}^{(t+1)} = (\boldsymbol{\theta}_{t-1}^\star - \boldsymbol{\theta}_{t-1}) + (\boldsymbol{\theta}_{t+1}^\star - \boldsymbol{\theta}_t^\star + \boldsymbol{\theta}_t^\star - \boldsymbol{\theta}_{t-1}^\star) = \widetilde{\boldsymbol{\theta}}_{t-1} - (\boldsymbol{\Delta}_{t-1} + \boldsymbol{\Delta}_t). \tag{28}$$

Plugging (27) and (28) into (24), we get

$$\widetilde{\boldsymbol{\theta}}_{t+1} = \boldsymbol{J}_t \widetilde{\boldsymbol{\theta}}_t + \boldsymbol{K}_t \widetilde{\boldsymbol{\theta}}_{t-1} + \boldsymbol{b}_t + \gamma_t \boldsymbol{\xi}_{t+1}(\boldsymbol{\psi}_t), \tag{29}$$

where

$$b_t \triangleq -(J_t + K_t)\Delta_t - K_t\Delta_{t-1} = -(\mathbf{I}_d - \gamma_t H_t)\Delta_t - K_t\Delta_{t-1}. \tag{30}$$

It follows that we can write the extended recursion relation as:

$$\underbrace{\begin{bmatrix} \widetilde{\theta}_{t+1} \\ \widetilde{\theta}_t \end{bmatrix}}_{\triangleq\, z_{t+1}} = \underbrace{\begin{bmatrix} J_t & K_t \\ \mathbf{I}_d & 0 \end{bmatrix}}_{\triangleq\, B_t} \underbrace{\begin{bmatrix} \widetilde{\theta}_t \\ \widetilde{\theta}_{t-1} \end{bmatrix}}_{\triangleq\, z_t} + \underbrace{\begin{bmatrix} b_t \\ 0 \end{bmatrix}}_{\triangleq\, u_t} + \gamma_t \underbrace{\begin{bmatrix} \xi_{t+1}(\psi_t) \\ 0 \end{bmatrix}}_{\triangleq\, \eta_{t+1}}. \tag{31}$$

Note that we can write $B_t = P - M_t$ where

$$P = \begin{bmatrix} (1+\beta)\mathbf{I}_d & -\beta\mathbf{I}_d \\ \mathbf{I}_d & 0 \end{bmatrix}, \quad M_t = \begin{bmatrix} \gamma_t(1+\beta_1)H_t & -\gamma_t\beta_1 H_t \\ 0 & 0 \end{bmatrix}. \tag{32}$$

Now $P$ has the following eigenvalue decomposition $P = VDV^{-1}$ where

$$V = \begin{bmatrix} \mathbf{I}_d & -\beta\mathbf{I}_d \\ \mathbf{I}_d & -\mathbf{I}_d \end{bmatrix}, \quad V^{-1} = \frac{1}{1-\beta}\begin{bmatrix} \mathbf{I}_d & -\beta\mathbf{I}_d \\ \mathbf{I}_d & -\mathbf{I}_d \end{bmatrix}, \quad D = \begin{bmatrix} \mathbf{I}_d & 0 \\ 0 & \beta\mathbf{I}_d \end{bmatrix}. \tag{33}$$

Define the transformed "mode-splitting" coordinates as follows:

$$y_t := V^{-1}z_t = \begin{bmatrix} \widehat{\theta}_t \\ \check{\theta}_t \end{bmatrix} = \frac{1}{1-\beta}\begin{bmatrix} \widetilde{\theta}_t - \beta\widetilde{\theta}_{t-1} \\ \widetilde{\theta}_t - \widetilde{\theta}_{t-1} \end{bmatrix}. \tag{34}$$

Using this transform with (29) gives us:

$$\begin{aligned} y_{t+1} &= V^{-1}B_t V y_t + V^{-1}u_t + \gamma_t V^{-1}\eta_{t+1} \\ &= \underbrace{(D - V^{-1}M_t V)}_{\triangleq\, \widetilde{B}_t} y_t + V^{-1}u_t + \gamma_t V^{-1}\eta_{t+1}. \end{aligned} \tag{35}$$

Finally we note that

$$D - V^{-1}M_t V = \begin{bmatrix} \mathbf{I}_d - \frac{\gamma_t}{1-\beta}H_t & \frac{\gamma_t\beta'}{1-\beta}H_t \\ -\frac{\gamma_t}{1-\beta}H_t & \beta\mathbf{I}_d + \frac{\gamma_t\beta'}{1-\beta}H_t \end{bmatrix}, \quad V^{-1}\eta_{t+1} = \frac{1}{1-\beta}\begin{bmatrix} \xi_{t+1}(\psi_t) \\ \xi_{t+1}(\psi_t) \end{bmatrix}. \tag{36}$$

This concludes the proof. $\qquad\square$

In order to establish a convergence guarantee of (SGDM), we will need to analyze the dynamics of the norm of the extended state vectors. What we aim to do is show a path-wise inequality (analogous to a Lyapunov stability function) of the form $s_t \le \Gamma_t s_t$ and then get a uniform $\Gamma$ by upper bounding $\Gamma_t, \forall t \ge 0$.

**Corollary C.3** (Uniform stability for (SGDM)). *Let Assumption 2.2, Assumption 2.3 hold, and $\beta_1 + \beta_2 = \beta$ and $\beta_1\beta_2 = 0$ for fixed $\beta \in [0,1)$. Consider (SGDM) and the extended 2D recursion (13). Then when the step-sizes $\gamma_t$ satisfies*

$$\gamma_t \le \frac{\mu(1-\beta)^2}{4L^2} \tag{37}$$

*it holds that the mean-square values of the transformed error vectors evolve according to the recursive inequality below:*

$$\begin{bmatrix} \|\widehat{\theta}_{t+1}\|^2 \\ \|\check{\theta}_{t+1}\|^2 \end{bmatrix} \le \begin{bmatrix} a & b \\ c & d \end{bmatrix} \begin{bmatrix} \|\widehat{\theta}_t\|^2 \\ \|\check{\theta}_t\|^2 \end{bmatrix}, \tag{38}$$

*where*

$$a = 1 - \frac{\gamma_t\mu}{1-\beta}, \quad b = \frac{\gamma_t\beta'^2 L^2}{\mu(1-\beta)}, \quad c = \frac{2\gamma_t^2 L^2}{(1-\beta)^3}, \quad d = \beta + \frac{2\gamma_t^2\beta'^2 L^2}{(1-\beta)^3} \tag{39}$$

*and the coefficient matrix is stable i.e.*

$$\rho\left(\begin{bmatrix} a & b \\ c & d \end{bmatrix}\right) < 1. \tag{40}$$

*Proof of Corollary C.3.* Define the energy vector as follows:

$$s_t = \begin{bmatrix} \|\widehat{\boldsymbol{\theta}}_t\|^2 \\ \|\check{\boldsymbol{\theta}}_t\|^2 \end{bmatrix} \in \mathbb{R}^2_{\geq 0}. \tag{41}$$

We will show the pathwise inequality $s_{t+1} \leq \boldsymbol{\Gamma}_t s_t$ with explicit scalar entries $a_t, b_t, c_t, d_t$, and then get a uniform $\boldsymbol{\Gamma}$ by upper bounded $\boldsymbol{\Gamma}_t$ over $\forall t \geq 0$. From Lemma C.2, we find the first row can be written as

$$\widehat{\boldsymbol{\theta}}_{t+1} = \boldsymbol{A}_t \widehat{\boldsymbol{\theta}}_t + \boldsymbol{B}_t \check{\boldsymbol{\theta}}_t, \tag{42}$$

with $\boldsymbol{A}_t = \mathbf{I}_d - \eta_t \boldsymbol{H}_t$ and $\boldsymbol{B}_t = \eta_t \beta' \boldsymbol{H}_t$ where $\eta_t = \gamma_t/(1-\beta)$ and $\beta' = \beta\beta_1 + \beta_2$. By Young's inequality, for $\tau > 0$

$$\|\widehat{\boldsymbol{\theta}}_{t+1}\|^2 \leq \frac{1}{1-\tau} \|\boldsymbol{A}_t \widehat{\boldsymbol{\theta}}_t\|^2 + \frac{1}{\tau} \|\boldsymbol{B}_t \check{\boldsymbol{\theta}}_t\|^2 \leq \frac{\|\boldsymbol{A}_t\|^2}{1-\tau} \|\widehat{\boldsymbol{\theta}}_t\|^2 + \frac{\|\boldsymbol{B}_t\|^2}{\tau} \|\check{\boldsymbol{\theta}}_t\|^2. \tag{43}$$

We will first impose that $\eta_t \leq 1/L$. Also recall that since $\mu \mathbf{I}_d \preceq \nabla^2 F_{t+1}(\boldsymbol{\theta}) \preceq L \mathbf{I}_d$, we immediately have that $\mu \mathbf{I}_d \preceq \boldsymbol{H}_t \preceq L \mathbf{I}_d$. Since we have defined $\boldsymbol{A}_t = \mathbf{I}_d - \eta_t \boldsymbol{H}_t$, these facts imply the eigenvalues of $\boldsymbol{A}_t$ lie in $[1 - \eta_t L, 1 - \eta_t \mu] \subset [0, 1 - \eta_t \mu]$. Hence we find that $\|\boldsymbol{A}_t\|_{\mathrm{op}} \leq 1 - \eta_t \mu$. Likewise we find that $\|\boldsymbol{B}_t\|_{\mathrm{op}} \leq \eta_t \beta' L$. Thus we get

$$\|\widehat{\boldsymbol{\theta}}_{t+1}\|^2 \leq \frac{(1 - \eta_t \mu)^2}{1 - \tau} \|\widehat{\boldsymbol{\theta}}_t\|^2 + \frac{\eta_t^2 \beta'^2 L^2}{\tau} \|\check{\boldsymbol{\theta}}_t\|^2. \tag{44}$$

Take $\tau = \eta_t \mu \in (0, 1)$. Indeed this is true since $\eta_t \leq 1/L$ and $\mu \leq L$. Then we find that

$$\|\widehat{\boldsymbol{\theta}}_{t+1}\|^2 \leq \underbrace{(1 - \eta_t \mu)}_{\triangleq a} \|\widehat{\boldsymbol{\theta}}_t\|^2 + \underbrace{\left( \eta_t \frac{\beta'^2 L^2}{\mu} \right)}_{\triangleq b} \|\check{\boldsymbol{\theta}}_t\|^2. \tag{45}$$

We will now repeat this for the second row. We can write the second row as

$$\check{\boldsymbol{\theta}}_{t+1} = -\eta_t \boldsymbol{H}_t \widehat{\boldsymbol{\theta}}_t + (\beta \mathbf{I}_d + \eta_t \beta' \boldsymbol{H}_t) \check{\boldsymbol{\theta}}_t = \beta \check{\boldsymbol{\theta}}_t + \boldsymbol{r}_t, \tag{46}$$

where $\boldsymbol{r}_t := \eta_t \boldsymbol{H}_t (\beta' \check{\boldsymbol{\theta}}_t - \widehat{\boldsymbol{\theta}}_t)$. By the convexity of the map $\boldsymbol{x} \mapsto \|\boldsymbol{x}\|^2$, we have that

$$\begin{aligned} \|\check{\boldsymbol{\theta}}_{t+1}\|^2 &= \|\beta \check{\boldsymbol{\theta}}_t + \boldsymbol{r}_t\|^2 \\ &= \|\beta \check{\boldsymbol{\theta}}_t + (1-\beta)\boldsymbol{r}_t/(1-\beta)\|^2 \\ &\leq \beta \|\check{\boldsymbol{\theta}}_t\|^2 + \frac{1}{1-\beta} \|\boldsymbol{r}_t\|^2. \end{aligned} \tag{47}$$

Now we bound $\|\boldsymbol{r}_t\|$:

$$\|\boldsymbol{r}_t\| \leq \eta_t \|\boldsymbol{H}_t\| \|\beta' \cdot \check{\boldsymbol{\theta}}_t - \widehat{\boldsymbol{\theta}}_t\| \leq \eta_t L \left( \beta' \|\check{\boldsymbol{\theta}}_t\| + \|\widehat{\boldsymbol{\theta}}_t\| \right). \tag{48}$$

Using the inequality $(a+b)^2 \lesssim a^2 + b^2$, we conclude that

$$\|\boldsymbol{r}_t\|^2 \leq 2\eta_t^2 L^2 \left( \beta'^2 \|\check{\boldsymbol{\theta}}_t\|^2 + \|\widehat{\boldsymbol{\theta}}_t\|^2 \right). \tag{49}$$

Therefore we find that

$$\|\check{\boldsymbol{\theta}}_{t+1}\|^2 \leq \underbrace{\left( \frac{2\eta_t^2 L^2}{1-\beta} \right)}_{\triangleq c} \|\widehat{\boldsymbol{\theta}}_t\|^2 + \underbrace{\left( \beta + \frac{2\eta_t^2 \beta'^2 L^2}{1-\beta} \right)}_{\triangleq d} \|\check{\boldsymbol{\theta}}_t\|^2. \tag{50}$$

Collecting (45) and (50), we find

$$\begin{bmatrix} \|\widehat{\boldsymbol{\theta}}_{t+1}\|^2 \\ \|\check{\boldsymbol{\theta}}_{t+1}\|^2 \end{bmatrix} \leq \boldsymbol{\Gamma}_t \begin{bmatrix} \|\widehat{\boldsymbol{\theta}}_t\|^2 \\ \|\check{\boldsymbol{\theta}}_t\|^2 \end{bmatrix}, \tag{51}$$

with

$$\boldsymbol{\Gamma}_t := \begin{bmatrix} a & b \\ c & d \end{bmatrix} = \begin{bmatrix} 1 - \eta_t \mu & \eta_t \frac{\beta'^2 L^2}{\mu} \\ \frac{2\eta_t^2 L^2}{1-\beta} & \beta + \frac{2\eta_t^2 \beta'^2 L^2}{1-\beta} \end{bmatrix}. \tag{52}$$

We will now examine the stability of the 2x2 coefficient matrix $\boldsymbol{\Gamma}_t$. First assume a constant upper bound

$$\eta := \sup_t \eta_t = \sup_t \frac{\gamma_t}{1 - \beta}.$$

Define the uniform dominating matrix

$$\boldsymbol{\Gamma} := \begin{bmatrix} 1 - \eta\mu & \eta \frac{\beta'^2 L^2}{\mu} \\ \frac{2\eta^2 L^2}{1-\beta} & \beta + \frac{2\eta^2 \beta'^2 L^2}{1-\beta} \end{bmatrix}. \tag{53}$$

Then $\boldsymbol{\Gamma}_t \le \boldsymbol{\Gamma}$ entrywise $\forall t \ge 0$. Hence we have that $\boldsymbol{s}_{t+1} \le \boldsymbol{\Gamma} \boldsymbol{s}_t \le \boldsymbol{\Gamma}^\alpha \boldsymbol{s}_{t-\alpha}$ for any $\alpha \le t$. Since the spectral radius of a matrix is upper bounded by its 1-norm, we have that $\rho(\boldsymbol{\Gamma}) \le \max\{a + c, b + d\}$, it suffices to compute the column sums and impose conditions on $\eta$ to ensure that $\rho(\boldsymbol{\Gamma}) < 1$. Computing this, we find

$$\rho(\boldsymbol{\Gamma}) \le \max\left\{ 1 - \eta\mu + \frac{2\eta^2 L^2}{1-\beta}, \beta + \eta \frac{\beta'^2 L^2}{\mu} + \frac{2\eta^2 \beta'^2 L^2}{1-\beta} \right\}. \tag{54}$$

We can choose a step-size $\eta$ small enough to satisfy:

$$\begin{cases} \frac{\eta\mu}{2} > \frac{2\eta^2 L^2}{1-\beta} \\ \frac{\eta\beta'^2 L^2}{\mu} > \frac{2\eta^2 \beta'^2 L^2}{1-\beta} \\ 1 - \beta > \frac{2\eta\beta'^2 L^2}{\mu}, \end{cases} \tag{55}$$

which is equivalent to

$$\eta < \min\left\{ \frac{\mu(1-\beta)}{4L^2}, \frac{1-\beta}{2\mu}, \frac{\mu(1-\beta)}{\beta'^2 L^2} \right\} = \frac{\mu(1-\beta)}{4L^2}. \tag{56}$$

Converting this back to $\gamma$ using the relation $\eta = \gamma/(1-\beta)$, we find

$$\gamma < \min\left\{ \frac{\mu(1-\beta)^2}{4L^2}, \frac{(1-\beta)^2}{2\mu}, \frac{\mu(1-\beta)^2}{\beta'^2 L^2} \right\} = \frac{\mu(1-\beta)^2}{4L^2}. \tag{57}$$

It then holds that

$$\rho(\boldsymbol{\Gamma}) < \max\left\{ 1 - \frac{\gamma\mu}{2(1-\beta)}, \beta + \frac{2\gamma L^2}{\mu(1-\beta)} \right\} \le 1. \tag{58}$$

In this case, $\boldsymbol{\Gamma}$ is a stable matrix. This concludes the proof. $\qquad\square$

From Lemma C.2, we established that we have a recursive relation of the form $\boldsymbol{y}_{t+1} = \widetilde{\boldsymbol{B}}_t \boldsymbol{y}_t + \boldsymbol{V}^{-1}\boldsymbol{u}_t + \gamma_t \boldsymbol{V}^{-1}\boldsymbol{\eta}_{t+1}$. Define the original state matrices as $\boldsymbol{\Phi}(t, s) := \boldsymbol{B}_{t-1}\boldsymbol{B}_{t-2}\dots\boldsymbol{B}_s$ and the transformed state transition matrices as $\widetilde{\boldsymbol{\Phi}}(t, s) := \widetilde{\boldsymbol{B}}_{t-1}\widetilde{\boldsymbol{B}}_{t-2}\dots\widetilde{\boldsymbol{B}}_s$ for $\forall t > s$ and $\boldsymbol{\Phi}(s, s) := \widetilde{\boldsymbol{\Phi}}(s, s) := \boldsymbol{I}_{2d}$. With the above established, we can convert the fact that $\boldsymbol{\Gamma}$ is stable into a bound on $\widetilde{\boldsymbol{\Phi}}$ and subsequently $\boldsymbol{\Phi}$. We will establish this below:

**Corollary C.4** (Conversion from $\widetilde{\boldsymbol{\Phi}}$ to $\boldsymbol{\Phi}$). *Let Assumption 2.2, Assumption 2.3 hold and $\beta_1 + \beta_2 = \beta$ with $\beta_1\beta_2 = 0$ for fixed $\beta \in [0, 1)$. Consider (SGDM) and the extended 2D recursion (13). Assume $\eta_t = \gamma_t/(1-\beta)$ and*

$$\gamma_t \le \frac{\mu(1-\beta)^2}{4L^2}.$$

*Let $\boldsymbol{\Gamma}_t$ be defined by*

$$\boldsymbol{\Gamma}_t = \begin{bmatrix} a_t & b_t \\ c_t & d_t \end{bmatrix} = \begin{bmatrix} 1 - \eta_t \mu & \eta_t \frac{\beta'^2 L^2}{\mu} \\ \frac{2\eta_t^2 L^2}{1-\beta} & \beta + \frac{2\eta_t^2 \beta'^2 L^2}{1-\beta} \end{bmatrix}, \qquad \beta' = \beta\beta_1 + \beta_2.$$

*Define*

$$\bar{\rho} := \sup_{t \geq 0} \sqrt{\|\mathbf{\Gamma}_t\|_1}, \qquad where \quad \|\mathbf{\Gamma}_t\|_1 = \max\{a_t + c_t, \ b_t + d_t\}.$$

*Then $\bar{\rho} < 1$ and for all $t \geq \alpha$,*

$$\|\mathbf{\Phi}(t, \alpha)\|_{op} \leq \frac{4}{1 - \beta} \, \bar{\rho}^{\,t-\alpha}.$$

*Proof of Corollary C.4.* First note that $\mathbf{y}_t = \widetilde{\mathbf{\Phi}}(t, \alpha)\mathbf{y}_\alpha$. Recall from Lemma C.2 that we have

$$\|\mathbf{y}_t\|^2 = \|\widehat{\boldsymbol{\theta}}_t\|^2 + \|\check{\boldsymbol{\theta}}_t\|^2 = \mathbf{1}^\top \mathbf{s}_t, \tag{59}$$

where $\mathbf{s}_t := [\|\widehat{\boldsymbol{\theta}}_t\|^2; \|\check{\boldsymbol{\theta}}_t\|^2] \in \mathbb{R}^2_+$. In Corollary C.3, we showed that for all $k \geq 0$,

$$\mathbf{s}_{k+1} \leq \mathbf{\Gamma}_k \mathbf{s}_k \qquad \text{(entrywise)}. \tag{60}$$

Iterating (60) from $\alpha$ to $t - 1$ gives

$$\mathbf{s}_t \leq \mathbf{\Gamma}_{t-1}\mathbf{\Gamma}_{t-2} \cdots \mathbf{\Gamma}_\alpha \, \mathbf{s}_\alpha \qquad \text{(entrywise)}. \tag{61}$$

Taking $\mathbf{1}^\top(\cdot)$ and using nonnegativity yields

$$\|\mathbf{y}_t\|^2 = \mathbf{1}^\top \mathbf{s}_t \leq \mathbf{1}^\top \left(\mathbf{\Gamma}_{t-1} \cdots \mathbf{\Gamma}_\alpha\right) \mathbf{s}_\alpha \leq \|\mathbf{\Gamma}_{t-1} \cdots \mathbf{\Gamma}_\alpha\|_1 \|\mathbf{s}_\alpha\|_1. \tag{62}$$

Since $\|\mathbf{s}_\alpha\|_1 = \mathbf{1}^\top \mathbf{s}_\alpha = \|\mathbf{y}_\alpha\|^2$, we obtain

$$\|\mathbf{y}_t\|^2 \leq \|\mathbf{\Gamma}_{t-1} \cdots \mathbf{\Gamma}_\alpha\|_1 \|\mathbf{y}_\alpha\|^2. \tag{63}$$

By submultiplicativity of the induced 1-norm,

$$\|\mathbf{\Gamma}_{t-1} \cdots \mathbf{\Gamma}_\alpha\|_1 \leq \prod_{k=\alpha}^{t-1} \|\mathbf{\Gamma}_k\|_1 \leq \left(\sup_{k \geq 0} \|\mathbf{\Gamma}_k\|_1\right)^{t-\alpha}. \tag{64}$$

Define

$$\bar{\rho} := \sqrt{\sup_{k \geq 0} \|\mathbf{\Gamma}_k\|_1}. \tag{65}$$

Combining (63) and (65) yields

$$\|\mathbf{y}_t\| \leq \bar{\rho}^{\,t-\alpha}\|\mathbf{y}_\alpha\|. \tag{66}$$

Therefore,

$$\|\widetilde{\mathbf{\Phi}}(t, \alpha)\|_{op} \leq \bar{\rho}^{\,t-\alpha}. \tag{67}$$

Since we have $\mathbf{z}_t = \mathbf{V}\mathbf{y}_t$, it follows that

$$\mathbf{\Phi}(t, \alpha) = \mathbf{B}_{t-1} \cdots \mathbf{B}_\alpha = \mathbf{V}\widetilde{\mathbf{\Phi}}(t, \alpha)\mathbf{V}^{-1}.$$

Hence

$$\|\mathbf{\Phi}(t, \alpha)\|_{op} \leq \|\mathbf{V}\|_{op}\|\mathbf{V}^{-1}\|_{op}\|\widetilde{\mathbf{\Phi}}(t, \alpha)\|_{op} \leq \|\mathbf{V}\|_{op}\|\mathbf{V}^{-1}\|_{op} \bar{\rho}^{\,t-\alpha}. \tag{68}$$

From (9), one can show that $\|\mathbf{V}[\mathbf{x}; \mathbf{y}]\| \leq \sqrt{2}(\|\mathbf{x}\| + \|\mathbf{y}\|) \leq 2\|[\mathbf{x}; \mathbf{y}]\|$, hence $\|\mathbf{V}\|_{op} \leq 2$. This also implies $\|\mathbf{V}^{-1}\|_{op} \leq 2/(1 - \beta)$. Plugging into (68) yields

$$\|\mathbf{\Phi}(t, \alpha)\|_{op} \leq \frac{4}{1 - \beta} \, \bar{\rho}^{\,t-\alpha}. \tag{69}$$

It remains to show that $\bar{\rho} < 1$. Note that

$$\|\mathbf{\Gamma}_t\|_1 = \max\{a_t + c_t, \ b_t + d_t\},$$

where

$$a_t + c_t = 1 - \eta_t \mu + \frac{2\eta_t^2 L^2}{1 - \beta} \leq 1 - \frac{\eta_t \mu}{2}, \qquad \text{since } \frac{2\eta_t^2 L^2}{1 - \beta} \leq \frac{\eta_t \mu}{2} \text{ under } \gamma_t \leq \frac{\mu(1 - \beta)^2}{4L^2}.$$

Moreover, using $\beta'^2 \leq 1$,

$$b_t + d_t = \beta + \eta_t \frac{\beta'^2 L^2}{\mu} + \frac{2\eta_t^2 \beta'^2 L^2}{1 - \beta} \leq \beta + \eta_t \frac{L^2}{\mu} + \frac{2\eta_t^2 L^2}{1 - \beta}.$$

With $\eta_t = \gamma_t/(1 - \beta)$ and $\gamma_t \leq \mu(1 - \beta)^2/(4L^2)$, we have

$$\eta_t \frac{L^2}{\mu} \leq \frac{1 - \beta}{4}, \qquad \frac{2\eta_t^2 L^2}{1 - \beta} \leq \frac{1 - \beta}{8},$$

and therefore

$$b_t + d_t \leq \beta + \frac{3}{8}(1 - \beta) = \frac{3}{8} + \frac{5}{8}\beta < 1.$$

Consequently $\sup_{t \geq 0} \|\mathbf{\Gamma}_t\|_1 < 1$, and thus $\bar{\rho} < 1$ by (65). This completes the proof. $\qquad \square$

In light of these results, we can finally prove an expectation bound on the tracking error of (SGDM):

**Corollary C.5** (Tracking error bound in expectation for (SGDM)). *Let Assumption 2.2, Assumption 2.3, and Assumption 3.1. Consider (SGDM) and the extended 2D recursion (13). Then when the step-sizes $\gamma_t$ satisfies*

$$\gamma_t \leq \frac{\mu(1 - \beta)^2}{4L^2}, \tag{70}$$

*the following tracking error bound holds in expectation for SGD with momentum with constant stepsize $\gamma_t = \gamma$:*

$$\mathbb{E}\|\boldsymbol{\theta}_{t+1} - \boldsymbol{\theta}_{t+1}^\star\|^2 \leq \frac{48}{(1 - \beta)^2}\rho^{2(t+1)}\|\boldsymbol{\theta}_0 - \boldsymbol{\theta}_0^\star\|^2 + \frac{48(1 + \beta + \gamma L)^2}{(1 - \beta)^2} \cdot \frac{\Delta^2}{(1 - \rho)^2} + \frac{48}{(1 - \beta)^2} \cdot \frac{\sigma^2 \gamma^2}{1 - \rho^2}. \tag{71}$$

*In particular taking $\rho = 1 - \gamma\mu/2(1 - \beta)$, we obtain:*

$$\mathbb{E}\|\boldsymbol{\theta}_{t+1} - \boldsymbol{\theta}_{t+1}^\star\|^2 \leq \frac{48}{(1 - \beta)^2} \exp\left(-\frac{\gamma\mu}{1 - \beta}(t + 1)\right)\|\boldsymbol{\theta}_0 - \boldsymbol{\theta}_0^\star\|^2 + \frac{192(2 + \beta)^2}{\gamma^2\mu^2}\Delta^2 + \frac{96\sigma^2\gamma}{\mu(1 - \beta)}. \tag{72}$$

*Proof of Corollary C.5.* For the analysis, we will denote $C_\beta := 4/(1 - \beta)$. From Lemma C.2, we showed that (SGDM) follows a 2D recursive system as follows:

$$\underbrace{\begin{bmatrix} \widetilde{\boldsymbol{\theta}}_{t+1} \\ \widetilde{\boldsymbol{\theta}}_t \end{bmatrix}}_{\triangleq \, \boldsymbol{z}_{t+1}} = \underbrace{\begin{bmatrix} \boldsymbol{J}_t & \boldsymbol{K}_t \\ \mathbf{I}_d & \mathbf{0} \end{bmatrix}}_{\triangleq \, \boldsymbol{B}_t} \underbrace{\begin{bmatrix} \widetilde{\boldsymbol{\theta}}_t \\ \widetilde{\boldsymbol{\theta}}_{t-1} \end{bmatrix}}_{\triangleq \, \boldsymbol{z}_t} + \underbrace{\begin{bmatrix} \boldsymbol{b}_t \\ \mathbf{0} \end{bmatrix}}_{\triangleq \, \boldsymbol{u}_t} + \gamma_t \underbrace{\begin{bmatrix} \boldsymbol{\xi}_{t+1}(\boldsymbol{\psi}_t) \\ \mathbf{0} \end{bmatrix}}_{\triangleq \, \boldsymbol{\eta}_{t+1}} \tag{73}$$

where

$$\begin{aligned} \boldsymbol{J}_t &= (1 + \beta)\mathbf{I}_d - \gamma_t(1 + \beta_1)\boldsymbol{H}_t \\ \boldsymbol{K}_t &= -\beta\mathbf{I}_d + \gamma_t\beta_1\boldsymbol{H}_t \\ \boldsymbol{b}_t &= -(\mathbf{I}_d - \gamma_t\boldsymbol{H}_t)\boldsymbol{\Delta}_t - \boldsymbol{K}_t\boldsymbol{\Delta}_{t-1}. \end{aligned} \tag{74}$$

Iterating (73) gives us the identity:

$$\boldsymbol{z}_{t+1} = \boldsymbol{\Phi}(t + 1, 0)\boldsymbol{z}_0 + \sum_{k=0}^{t} \boldsymbol{\Phi}(t + 1, k + 1)\boldsymbol{u}_k + \sum_{k=0}^{t} \boldsymbol{\Phi}(t + 1, k + 1)\gamma_k\boldsymbol{\eta}_{k+1}. \tag{75}$$

Define the following quantities:

$$\boldsymbol{A}_{t+1} := \boldsymbol{\Phi}(t + 1, 0)\boldsymbol{z}_0, \quad \boldsymbol{D}_{t+1} := \sum_{k=0}^{t} \boldsymbol{\Phi}(t + 1, k + 1)\boldsymbol{u}_k, \quad \boldsymbol{N}_{t+1} := \sum_{k=0}^{t} \boldsymbol{\Phi}(t + 1, k + 1)\gamma_k\boldsymbol{\eta}_{k+1}. \tag{76}$$

Then we have that

$$z_{t+1} = A_{t+1} + D_{t+1} + N_{t+1}. \tag{77}$$

Using the inequality $\|a + b + c\|^2 \leq 3\|a\|^2 + 3\|b\|^2 + 3\|c\|^2$, we have:

$$\mathbb{E}\|z_{t+1}\|^2 \leq 3\mathbb{E}\|A_{t+1}\|^2 + 3\mathbb{E}\|D_{t+1}\|^2 + 3\mathbb{E}\|N_{t+1}\|^2. \tag{78}$$

We will now proceed with bounding each of these quantities.

**Bounding $\mathbb{E}\|A_{t+1}\|^2$:** From the stability established in Corollary C.4, we have

$$\|A_{t+1}\| = \|\Phi(t + 1, 0)z_0\| \leq C_\beta \rho^{t+1}\|z_0\|. \tag{79}$$

Thus we can conclude that $\mathbb{E}\|A_{t+1}\|^2 \leq C_\beta^2 \rho^{2(t+1)}\|z_0\|^2$.

**Bounding $\mathbb{E}\|D_{t+1}\|^2$:** First simply note that $\|u_t\| = \|b_t\|$. Thus it suffices to bound $\|b_t\|$. By definition, we have that $b_t = -(\mathbf{I}_d - \gamma_t H_t)\Delta_t - K_t \Delta_{t-1}$. Since $\mu \mathbf{I}_d \preceq H_t \preceq L\mathbf{I}_d$ and $\gamma_t \leq 1/L$, the eigenvalues of $\mathbf{I}_d - \gamma_t H_t$ lie in $[1 - \gamma_t L, 1 - \gamma_t \mu] \subset [0, 1]$ so we have that $\|\mathbf{I}_d - \gamma_t H_t\| \leq 1$. Also note that since $K_t = -\beta \mathbf{I}_d + \gamma_t \beta_1 H_t$, we have $\|K_t\| \leq \beta + \gamma_t L$. Thus putting everything together, we find

$$\|b_t\| \leq \|\mathbf{I}_d - \gamma_t H_t\|\|\Delta_t\| + \|K_t\|\|\Delta_{t-1}\| \leq (1 + \beta + \gamma_t L)\Delta. \tag{80}$$

Using triangle inequality and the bound on $\|b_t\|$, we have

$$\|D_{t+1}\|^2 \leq \sum_{k=0}^{t} \|\Phi(t + 1, k + 1)\|^2 \|b_k\|^2 \leq C_\beta^2 \Delta^2 (1 + \beta + \gamma_t L)^2 \left(\sum_{k=0}^{t} \rho^{t-k}\right)^2. \tag{81}$$

Using the fact that $\sum_{j \geq 0} \rho^j \leq 1/(1 - \rho)$, we conclude that

$$\mathbb{E}\|D_{t+1}\|^2 \leq C_\beta^2 (1 + \beta + \gamma_t L)^2 \frac{\Delta^2}{(1 - \rho)^2}. \tag{82}$$

**Bounding $\mathbb{E}\|N_{t+1}\|^2$:** We will work with the recursion that $N_{t+1}$ satisfies:

$$N_{t+1} = B_t N_t + \gamma_t \eta_{t+1} \tag{83}$$

with $N_0 = 0$. Multiplying by $V^{-1}$, we find that

$$n_{t+1} = \widetilde{B}_t n_t + \gamma_t \widetilde{\eta}_{t+1} \tag{84}$$

where $n_{t+1} = V^{-1}N_{t+1}$, $\widetilde{B}_t = V^{-1}B_t V$, and $\widetilde{\eta}_{t+1} = V^{-1}\eta_{t+1}$. Now since $\widetilde{B}_t n_t$ is $\mathcal{F}_t$ measurable and $\mathbb{E}[\widetilde{\eta}_{t+1} \mid \mathcal{F}_t] = 0$, we have by conditioning and then iterated expectation that

$$\mathbb{E}[\|n_{t+1}\|^2 \mid \mathcal{F}_t] = \|\widetilde{B}_t n_t\|^2 + \gamma_t^2 \mathbb{E}[\|\widetilde{\eta}_{t+1}\|^2 \mid \mathcal{F}_t]. \tag{85}$$

From the stability established in Corollary C.3, we have that $\|\widetilde{B}_t n_t\|^2 \leq \rho^2 \|n_t\|^2$. Also note that from Lemma C.2, we have an explicit form for $\widetilde{\eta}_{t+1}$:

$$\widetilde{\eta}_{t+1} = \frac{1}{1 - \beta}\begin{bmatrix} \xi_{t+1}(\psi_t) \\ \xi_{t+1}(\psi_t) \end{bmatrix}. \tag{86}$$

Under the boundedness assumption (Assumption 3.1), we have that

$$\mathbb{E}[\|\widetilde{\eta}_{t+1}\|^2 \mid \mathcal{F}_t] \leq \frac{2\sigma^2}{(1 - \beta)^2}. \tag{87}$$

Thus we find that

$$\mathbb{E}[\|n_{t+1}\|^2] \leq \rho^2 \mathbb{E}[\|n_t\|^2] + \frac{2\gamma_t^2 \sigma^2}{(1 - \beta)^2}. \tag{88}$$

Unrolling this expression and using the fact that $N_0 = 0$, we find that

$$\mathbb{E}[\|n_{t+1}\|^2] \leq \frac{2\gamma_t^2 \sigma^2}{(1-\beta)^2(1-\rho^2)}. \tag{89}$$

Thus we can conclude with:

$$\mathbb{E}\|N_{t+1}\|^2 = \|V\|^2 \mathbb{E}[\|n_{t+1}\|^2] \leq \frac{8\gamma_t^2 \sigma^2}{(1-\beta)^2(1-\rho^2)}. \tag{90}$$

Combining everything and noting that $\|\theta_{t+1} - \theta_{t+1}^\star\|^2 \leq \|z_{t+1}\|^2$, we conclude that

$$\mathbb{E}\|\theta_{t+1} - \theta_{t+1}^\star\|^2 \leq \frac{48}{(1-\beta)^2} \rho^{2(t+1)} \|\theta_0 - \theta_0^\star\|^2 + \frac{48(1+\beta+\gamma L)^2}{(1-\beta)^2} \cdot \frac{\Delta^2}{(1-\rho)^2} + \frac{48}{(1-\beta)^2} \cdot \frac{\sigma^2 \gamma^2}{1-\rho^2}. \tag{91}$$

From Corollary C.3, we can take $\rho = 1 - \gamma\mu/2(1-\beta)$ and use the inequality $(1-a)^{2(t+1)} \leq e^{-2a(t+1)}$ to conclude:

$$\mathbb{E}\|\theta_{t+1} - \theta_{t+1}^\star\|^2 \leq \frac{48}{(1-\beta)^2} \exp\left(-\frac{\gamma\mu}{1-\beta}(t+1)\right) \|\theta_0 - \theta_0^\star\|^2 + \frac{192(2+\beta)^2}{\gamma^2\mu^2}\Delta^2 + \frac{96\sigma^2\gamma}{\mu(1-\beta)}. \tag{92}$$

$\square$

Using Corollary C.5, we can similarly obtain an algorithmic guarantee for (SGDM):

**Corollary C.6** (Time to reach the asymptotic tracking error in expectation for (SGDM)). *Assume Assumption 2.2 Assumption 2.3, and Assumption 3.1 and let $\beta \in [0,1)$. Consider (SGDM), written with momentum buffer $v_t \in \mathbb{R}^d$ as*

$$v_{t+1} = \beta v_t - \gamma_t \nabla g(\psi_t, X_{t+1}), \quad \theta_{t+1} = \theta_t + v_{t+1},$$

*where $\psi_t = \theta_t$ for heavy-ball momentum and $\psi_t = \theta_t + \beta v_t$ for Nesterov momentum. Assume a constant stepsize $\gamma_t \equiv \gamma$ satisfying $\gamma \leq \mu(1-\beta)^2/(4L^2) =: \gamma_{\max}$, and set*

$$\rho := 1 - \frac{\gamma\mu}{2(1-\beta)} \in (0,1).$$

*Define the (stepsize-dependent) steady-state tracking error*

$$\mathcal{E}_\beta(\gamma) = \frac{192(2+\beta)^2}{\mu^2\gamma^2}\Delta^2 + \frac{96}{\mu(1-\beta)}\sigma^2\gamma, \quad \gamma_\beta^\star \in \arg\min_{\gamma \in (0,\, \mu(1-\beta)^2/(4L^2)]} \mathcal{E}_\beta(\gamma), \quad \mathcal{E}_\beta := \mathcal{E}_\beta(\gamma_\beta^\star). \tag{93}$$

*Then:*

1. **(Constant learning rate).** *If $\gamma_t \equiv \gamma_\beta^\star$, then for all $t \geq 0$,*

$$\mathbb{E}\|\theta_{t+1} - \theta_{t+1}^\star\|^2 \lesssim \mathcal{E}_\beta \quad \text{after time} \quad t \lesssim \frac{1-\beta}{\mu\gamma_\beta^\star}\log\left(\frac{\|\theta_0 - \theta_0^\star\|^2}{(1-\beta)^2\mathcal{E}_\beta}\right).$$

2. **(Step-decay schedule with momentum restart).** *Suppose $\gamma_\beta^\star < \mu(1-\beta)^2/(4L^2)$ (i.e., the minimizer of $\mathcal{E}_\beta(\gamma)$ is not at the stability cap). Define the epoch stepsizes*

$$\gamma_0 := \frac{\mu(1-\beta)^2}{4L^2}, \qquad \gamma_k := \frac{\gamma_{k-1} + \gamma_\beta^\star}{2} \quad (k \geq 1), \qquad K := 1 + \left\lceil \log_2\left(\frac{\gamma_0}{\gamma_\beta^\star}\right)\right\rceil.$$

*Define epoch lengths*

$$T_0 := \left\lceil\frac{1-\beta}{\mu\gamma_0}\log\left(\frac{2\|\theta_0 - \theta_0^\star\|^2}{(1-\beta)^2\mathcal{E}_\beta(\gamma_0)}\right)\right\rceil, \qquad T_k := \left\lceil\frac{1-\beta}{\mu\gamma_k}\log 4\right\rceil \quad (k \geq 1).$$

*Run (SGDM) with constant stepsize $\gamma_k$ for $T_k$ iterations in epoch $k$, restarting the momentum buffer at the start of each epoch, i.e., set $v_{t_k} = 0$ at every epoch boundary $t_k$. Let $T := \sum_{k=0}^{K-1} T_k$ be the total horizon. Then the final iterate satisfies*

$$\mathbb{E}\|\theta_T - \theta_T^\star\|^2 \lesssim \mathcal{E}_\beta \quad \text{after time} \quad T \lesssim \frac{L^2}{\mu^2(1-\beta)}\log\left(\frac{\|\theta_0 - \theta_0^\star\|^2}{(1-\beta)^2\mathcal{E}_\beta}\right) + \frac{\sigma^2}{\mu^2\mathcal{E}_\beta},$$

*Proof of Corollary C.6.* Fix any stepsize $\gamma \in (0, \gamma_{\max}]$ and define

$$\rho(\gamma) := 1 - \frac{\gamma\mu}{2(1-\beta)} \in (0, 1).$$

By Corollary C.5 with this choice of $\rho$ and the inequality $\rho^{2(t+1)} \leq \exp(-\gamma\mu(t+1)/(1-\beta))$, we have for all $t \geq 0$:

$$\mathbb{E}\|\boldsymbol{\theta}_{t+1} - \boldsymbol{\theta}_{t+1}^\star\|^2 \leq \underbrace{\frac{48}{(1-\beta)^2}}_{\triangleq\ C_\beta} \exp\Big(-\frac{\mu\gamma}{1-\beta}(t+1)\Big) \|\boldsymbol{\theta}_0 - \boldsymbol{\theta}_0^\star\|^2 + \mathcal{E}_\beta(\gamma). \tag{94}$$

Since the assumptions are *uniform in time* (same $\mu, L, \Delta, \sigma$ for all $t$), the same inequality applies when the method is started at any time $s \geq 0$ with initial point $\boldsymbol{\theta}_s$ and initial minimizer $\boldsymbol{\theta}_s^\star$, *provided the momentum buffer is reset at time $s$.* Concretely, if we set $\boldsymbol{v}_s = \boldsymbol{0}$ (equivalently $\boldsymbol{\theta}_{s-1} = \boldsymbol{\theta}_s$), and run for $T$ steps with constant stepsize $\gamma$, then (94) (with $t$ replaced by $T-1$ and $(\boldsymbol{\theta}_0, \boldsymbol{\theta}_0^\star)$ replaced by $(\boldsymbol{\theta}_s, \boldsymbol{\theta}_s^\star)$) yields:

$$\mathbb{E}\|\boldsymbol{\theta}_{s+T} - \boldsymbol{\theta}_{s+T}^\star\|^2 \leq C_\beta \exp\Big(-\frac{\mu\gamma}{1-\beta}T\Big) \mathbb{E}\|\boldsymbol{\theta}_s - \boldsymbol{\theta}_s^\star\|^2 + \mathcal{E}_\beta(\gamma). \tag{95}$$

**Proof of (i).** Apply (94) with $\gamma = \gamma_\beta^\star$:

$$\mathbb{E}\|\boldsymbol{\theta}_{t+1} - \boldsymbol{\theta}_{t+1}^\star\|^2 \leq C_\beta \exp\Big(-\frac{\mu\gamma_\beta^\star}{1-\beta}(t+1)\Big) \|\boldsymbol{\theta}_0 - \boldsymbol{\theta}_0^\star\|^2 + \mathcal{E}_\beta.$$

If

$$t \geq \frac{1-\beta}{\mu\gamma_\beta^\star} \log\Big(\frac{C_\beta\|\boldsymbol{\theta}_0 - \boldsymbol{\theta}_0^\star\|^2}{\mathcal{E}_\beta}\Big),$$

then the transient term is at most $\mathcal{E}_\beta$ and hence the total is at most $2\mathcal{E}_\beta$. Substituting $C_\beta = 48/(1-\beta)^2$ yields the stated sufficient condition.

**Proof of (ii).** Let $t_0 := 0$ and $t_{k+1} := t_k + T_k$, and define the epoch iterates and epoch minimizers

$$\boldsymbol{x}_k := \boldsymbol{\theta}_{t_k}, \qquad \boldsymbol{x}_k^\star := \boldsymbol{\theta}_{t_k}^\star.$$

By construction, at the start of each epoch $k$ we restart the momentum buffer (set $\boldsymbol{v}_{t_k} = \boldsymbol{0}$), so we may apply the one-epoch bound (95) with $s = t_k$, $T = T_k$, and $\gamma = \gamma_k$:

$$\mathbb{E}\|\boldsymbol{x}_{k+1} - \boldsymbol{x}_{k+1}^\star\|^2 \leq C_\beta \exp\Big(-\frac{\mu\gamma_k}{1-\beta}T_k\Big) \mathbb{E}\|\boldsymbol{x}_k - \boldsymbol{x}_k^\star\|^2 + \mathcal{E}_\beta(\gamma_k). \tag{96}$$

*Contraction for epochs $k \geq 1$.* For $k \geq 1$, by the definition of $T_k$ we have

$$T_k \geq \frac{1-\beta}{\mu\gamma_k} \log(4C_\beta),$$

and therefore

$$C_\beta \exp\Big(-\frac{\mu\gamma_k}{1-\beta}T_k\Big) \leq C_\beta \exp(-\log(4C_\beta)) = \frac{1}{4}.$$

Substituting this into (96) yields, for all $k \geq 1$,

$$\mathbb{E}\|\boldsymbol{x}_{k+1} - \boldsymbol{x}_{k+1}^\star\|^2 \leq \frac{1}{4} \mathbb{E}\|\boldsymbol{x}_k - \boldsymbol{x}_k^\star\|^2 + \mathcal{E}_\beta(\gamma_k). \tag{97}$$

*Base epoch ($k = 0$).* By the definition of $T_0$,

$$T_0 \geq \frac{1-\beta}{\mu\gamma_0} \log\Big(\frac{2C_\beta\|\boldsymbol{x}_0 - \boldsymbol{x}_0^\star\|^2}{\mathcal{E}_\beta(\gamma_0)}\Big),$$

so

$$C_\beta \exp\Big(-\frac{\mu\gamma_0}{1-\beta}T_0\Big)\|\boldsymbol{x}_0 - \boldsymbol{x}_0^\star\|^2 \;\leq\; \frac{1}{2}\,\mathcal{E}_\beta(\gamma_0).$$

Plugging into (96) at $k = 0$ gives

$$\mathbb{E}\|\boldsymbol{x}_1 - \boldsymbol{x}_1^\star\|^2 \;\leq\; \frac{1}{2}\,\mathcal{E}_\beta(\gamma_0) \;+\; \mathcal{E}_\beta(\gamma_0) \;=\; \frac{3}{2}\,\mathcal{E}_\beta(\gamma_0) \;\leq\; 2\,\mathcal{E}_\beta(\gamma_0). \tag{98}$$

*Induction.* We claim that for all $k \geq 1$,
$$\mathbb{E}\|\boldsymbol{x}_k - \boldsymbol{x}_k^\star\|^2 \;\leq\; 2\,\mathcal{E}_\beta(\gamma_{k-1}). \tag{99}$$
The base case $k = 1$ follows from (98). Assume (99) holds for some $k \geq 1$. Applying (97) yields

$$\mathbb{E}\|\boldsymbol{x}_{k+1} - \boldsymbol{x}_{k+1}^\star\|^2 \leq \frac{1}{4} \cdot 2\mathcal{E}_\beta(\gamma_{k-1}) + \mathcal{E}_\beta(\gamma_k) = \frac{1}{2}\,\mathcal{E}_\beta(\gamma_{k-1}) + \mathcal{E}_\beta(\gamma_k).$$

Since $\gamma_k = (\gamma_{k-1} + \gamma_\beta^\star)/2$, we have $\gamma_{k-1} \leq 2\gamma_k$. Write $\mathcal{E}_\beta(\gamma) = A_\beta/\gamma^2 + B_\beta\gamma$ with $A_\beta, B_\beta > 0$. For any $\gamma > 0$,

$$\mathcal{E}_\beta(2\gamma) = \frac{A_\beta}{4\gamma^2} + 2B_\beta\gamma \;\leq\; 2\Big(\frac{A_\beta}{\gamma^2} + B_\beta\gamma\Big) = 2\mathcal{E}_\beta(\gamma),$$

so in particular

$$\mathcal{E}_\beta(\gamma_{k-1}) \;\leq\; \mathcal{E}_\beta(2\gamma_k) \;\leq\; 2\mathcal{E}_\beta(\gamma_k).$$

Therefore,

$$\mathbb{E}\|\boldsymbol{x}_{k+1} - \boldsymbol{x}_{k+1}^\star\|^2 \leq \frac{1}{2} \cdot 2\mathcal{E}_\beta(\gamma_k) + \mathcal{E}_\beta(\gamma_k) = 2\,\mathcal{E}_\beta(\gamma_k),$$

which proves (99) at $k + 1$ and closes the induction. Hence $\mathbb{E}\|\boldsymbol{x}_K - \boldsymbol{x}_K^\star\|^2 \leq 2\mathcal{E}_\beta(\gamma_{K-1})$. By the definition of $K$,

$$\gamma_{K-1} - \gamma_\beta^\star = \frac{\gamma_0 - \gamma_\beta^\star}{2^{K-1}} \leq \frac{\gamma_0}{2^{K-1}} \leq \gamma_\beta^\star, \quad \text{so} \quad \gamma_{K-1} \leq 2\gamma_\beta^\star.$$

Thus,

$$\mathcal{E}_\beta(\gamma_{K-1}) \leq \mathcal{E}_\beta(2\gamma_\beta^\star) \leq 2\mathcal{E}_\beta(\gamma_\beta^\star) = 2\mathcal{E}_\beta,$$

and therefore

$$\mathbb{E}\|\boldsymbol{\theta}_T - \boldsymbol{\theta}_T^\star\|^2 = \mathbb{E}\|\boldsymbol{x}_K - \boldsymbol{x}_K^\star\|^2 \leq 2\mathcal{E}_\beta(\gamma_{K-1}) \leq 4\mathcal{E}_\beta.$$

*Time bound.* Since $\mathcal{E}_\beta(\gamma_0) \geq \mathcal{E}_\beta$, we have

$$T_0 \leq \frac{1-\beta}{\mu\gamma_0} \log\Big(\frac{2C_\beta\|\boldsymbol{\theta}_0 - \boldsymbol{\theta}_0^\star\|^2}{\mathcal{E}_\beta}\Big) + 1.$$

Moreover, for $k \geq 1$, $\gamma_k \geq \gamma_0/2^{k+1}$ (by the same argument as in Corollary C.2), hence

$$\sum_{k=1}^{K-1} \frac{1}{\gamma_k} \leq \sum_{k=1}^{K-1} \frac{2^{k+1}}{\gamma_0} \leq \frac{2^{K+1}}{\gamma_0} \leq \frac{4}{\gamma_\beta^\star}.$$

Therefore,

$$\sum_{k=1}^{K-1} T_k \leq \sum_{k=1}^{K-1} \Big(\frac{1-\beta}{\mu\gamma_k}\log(4C_\beta) + 1\Big) \leq \frac{(1-\beta)\log(4C_\beta)}{\mu}\sum_{k=1}^{K-1}\frac{1}{\gamma_k} + K \leq \frac{4(1-\beta)\log(4C_\beta)}{\mu\gamma_\beta^\star} + K.$$

Combining with the bound on $T_0$ yields the stated explicit horizon bound. Finally, in the interior regime (unconstrained minimizer), $\mathcal{E}_\beta(\gamma) = A_\beta/\gamma^2 + B_\beta\gamma$ satisfies $\mathcal{E}_\beta = \frac{3}{2}B_\beta\gamma_\beta^\star$, so

$$\frac{1-\beta}{\mu\gamma_\beta^\star} = \frac{1-\beta}{\mu} \cdot \frac{B_\beta}{(2/3)\mathcal{E}_\beta} = \frac{1-\beta}{\mu} \cdot \frac{96\sigma^2/(\mu(1-\beta))}{(2/3)\mathcal{E}_\beta} = \frac{144\sigma^2}{\mu^2\mathcal{E}_\beta}.$$

Also $\gamma_0 = \mu(1-\beta)^2/(4L^2)$ implies $\frac{1-\beta}{\mu\gamma_0} = \frac{4L^2}{\mu^2(1-\beta)}$. Substituting yields the final bound on $T$. $\qquad\square$

# D. Proofs of tracking error bounds with high probability

For this section, we will assume a light-tailed assumption on the gradient noise (Assumption D.1)

**Assumption D.1** (Conditional sub-Gaussian gradient noise along iterates). *There exists a constant $\sigma > 0$ such that for all $t \geq 0$, $\left\|\boldsymbol{\xi}_{t+1}(\boldsymbol{\theta}_t) \mid \mathcal{F}_t\right\|_{\Psi_2} \leq \sigma$ and $\left\|\boldsymbol{\xi}_{t+1}(\boldsymbol{\psi}_t) \mid \mathcal{F}_t\right\|_{\Psi_2} \leq \sigma$ a.s.*

## D.1. Proof for SGD high probability tracking error bound

We will prove the following high probability bound holds for the tracking error in (SGD):

**Theorem D.1** (High probability tracking error bound for (SGD)). *Under Assumption 3.2, for all $t \in [T]$, $\gamma \leq \min\left\{\mu/L^2, 1/L\right\}$, and $\delta \in (0,1)$, the following tracking error bound holds for (SGD) with probability at least $1 - \delta$,*

$$\|\boldsymbol{\theta}_t - \boldsymbol{\theta}_t^\star\|^2 \lesssim \left(1 - \frac{\gamma\mu}{2}\right)^t \|\boldsymbol{\theta}_0 - \boldsymbol{\theta}_0^\star\|^2 + \frac{\mathfrak{D}_t}{\gamma\mu} + \frac{d\sigma^2\gamma}{\mu} + d\sigma^2\gamma^2 \log\frac{2T}{\delta} + \left(\frac{\sigma^2\gamma}{\mu} + \gamma^2\sigma^2\mathfrak{D}_t^{(2)}\right)\log\frac{2T}{\delta},$$

*where $\mathfrak{D}_t := \sum_{\ell=0}^{t-1}(1 - \gamma\mu/2)^{t-\ell-1}\|\boldsymbol{\Delta}_\ell\|^2$ and $\mathfrak{D}_t^{(2)} := \sum_{\ell=0}^{t-1}(1 - \gamma\mu/2)^{2(t-\ell-1)}\|\boldsymbol{\Delta}_\ell\|^2$.*

*Proof of Theorem D.1.* First recall by Proposition C.1 that we have

$$\|\boldsymbol{\theta}_t - \boldsymbol{\theta}_t^\star\|^2 \leq \left(1 - \frac{\gamma\mu}{2}\right)^t \|\boldsymbol{\theta}_0 - \boldsymbol{\theta}_0^\star\|^2 + \frac{2\mathfrak{D}_t}{\gamma\mu}$$

$$+ \underbrace{\gamma^2 \sum_{\ell=0}^{t-1}\left(1 - \frac{\gamma\mu}{2}\right)^{t-\ell-1}\|\boldsymbol{\xi}_{\ell+1}(\boldsymbol{\theta}_\ell)\|^2}_{(a)} + \underbrace{\sum_{\ell=0}^{t-1}\left(1 - \frac{\gamma\mu}{2}\right)^{t-\ell-1}M_{\ell+1}}_{(b)} \tag{100}$$

where $M_{t+1} := -2\gamma\langle \boldsymbol{d}_t - \gamma\boldsymbol{m}_{t+1}(\boldsymbol{\theta}_t), \boldsymbol{\xi}_{t+1}(\boldsymbol{\theta}_t)\rangle$ with $\boldsymbol{d}_t = \boldsymbol{\theta}_t - \boldsymbol{\theta}_{t+1}^\star$. It remains to bound (a) and (b).

**Bounding part (a):** First notice the following equivalence:

$$\|\boldsymbol{\xi}_{\ell+1}(\boldsymbol{\theta}_\ell)\|^2 = \mathbb{E}[\|\boldsymbol{\xi}_{\ell+1}(\boldsymbol{\theta}_\ell)\|^2 \mid \mathcal{F}_\ell] + V_{\ell+1}, \quad V_{\ell+1} := \|\boldsymbol{\xi}_{\ell+1}(\boldsymbol{\theta}_\ell)\|^2 - \mathbb{E}[\|\boldsymbol{\xi}_{\ell+1}(\boldsymbol{\theta}_\ell)\|^2 \mid \mathcal{F}_\ell]. \tag{101}$$

First we bound $V_{\ell+1}$. Note that $V_{\ell+1}$ is a martingale difference sequence (MDS) and sub-exponential with $\|V_{\ell+1} \mid \mathcal{F}_\ell\|_{\Psi_1} \lesssim d\sigma^2$ (Lemma G.1). Define $\rho := (1 - \gamma\mu/2)$. Then for fixed $t \leq T$, let $Z_{\ell+1}^{(t)} := \gamma^2\rho^{t-\ell-1}V_{\ell+1}$. Then we have that $\|Z_{\ell+1}^{(t)} \mid \mathcal{F}_\ell\|_{\Psi_1} \lesssim \gamma^2\rho^{t-\ell-1}d\sigma^2$. We also have the following that hold:

$$\sum_{\ell=0}^{t-1}\gamma^4\rho^{2(t-\ell-1)}d^2\sigma^4 \lesssim \frac{d^2\sigma^4\gamma^3}{\mu}, \quad \max_{0\leq\ell\leq t-1}\gamma^2\rho^{t-\ell-1}d\sigma^2 \lesssim d\sigma^2\gamma^2. \tag{102}$$

By Bernstein's inequality for conditionally sub-exponential martingale differences (Lemma G.5), there exists an absolute constant $c > 0$ such that for every $s > 0$,

$$\mathbb{P}\left(\sum_{\ell=0}^{t-1}Z_{\ell+1}^{(t)} \geq s\right) \lesssim \exp\left(-\min\left\{\frac{s^2\mu}{d^2\sigma^4\gamma^3}, \frac{s}{\gamma^2 d\sigma^2}\right\}\right). \tag{103}$$

Take $s = C\left(d\sigma^2\sqrt{\frac{\gamma^3}{\mu}\log(2T/\delta)} + d\sigma^2\gamma^2\log(2T/\delta)\right)$ for a sufficiently large absolute constant $C > 0$. Furthermore since we have $\|\boldsymbol{\xi}_{\ell+1}(\boldsymbol{\theta}_\ell) \mid \mathcal{F}_\ell\|_{\Psi_2} \leq \sigma$, we have $\mathbb{E}[\|\boldsymbol{\xi}_{\ell+1}(\boldsymbol{\theta}_\ell)\|^2 \mid \mathcal{F}_\ell] \leq d\sigma^2$ (Lemma G.4). Thus we have:

$$\gamma^2\sum_{\ell=0}^{t-1}\rho^{t-\ell-1}\mathbb{E}[\|\boldsymbol{\xi}_{\ell+1}(\boldsymbol{\theta}_\ell)\|^2 \mid \mathcal{F}_\ell] \lesssim \frac{d\sigma^2\gamma}{\mu}. \tag{104}$$

Combining everything and applying AM-GM inequality to the first term of $s$, we obtain the event $\mathcal{E}_{\boldsymbol{\xi}}(t)$ that with probability at least $1 - \delta/2T$,

$$\gamma^2\sum_{\ell=0}^{t-1}\left(1 - \frac{\gamma\mu}{2}\right)^{t-\ell-1}\|\boldsymbol{\xi}_{\ell+1}(\boldsymbol{\theta}_\ell)\|^2 \lesssim \frac{d\sigma^2\gamma}{\mu} + d\sigma^2\gamma^2\log\frac{2T}{\delta}. \tag{105}$$

A union bound over $t = 1, \ldots, T$ gives the event $\mathcal{E}_{\boldsymbol{\xi}} = \cap_{t\leq T}\mathcal{E}_{\boldsymbol{\xi}}(t)$ with $\mathbb{P}(\mathcal{E}_{\boldsymbol{\xi}}) \geq 1 - \delta/2$. It remains to bound (b).

**Bounding part (b):** Recall that $M_{\ell+1} := -2\gamma\langle d_\ell - \gamma m_{\ell+1}(\theta_\ell), \xi_{\ell+1}(\theta_\ell)\rangle$ with $d_\ell = \theta_\ell - \theta_{\ell+1}^\star$. Define $a_\ell := d_\ell - \gamma m_{\ell+1}(\theta_\ell)$ Since $a_\ell$ is $\mathcal{F}_\ell$–measurable and $\mathbb{E}[\xi_{\ell+1}(\theta_\ell) \mid \mathcal{F}_\ell] = 0$, we have $\mathbb{E}[M_{\ell+1} \mid \mathcal{F}_\ell] = 0$ and thus $(M_{\ell+1})_{\ell\geq 0}$ is a martingale difference sequence (MDS).

By Assumption 3.2, for any $\mathcal{F}_\ell$–measurable unit vector $u$, $\|u^\top \xi_{\ell+1}(\theta_\ell) \mid \mathcal{F}_\ell\|_{\Psi_2} \leq \sigma$ a.s. Hence, for any $\mathcal{F}_\ell$–measurable vector $v$, $\langle v, \xi_{\ell+1}(\theta_\ell)\rangle$ is conditionally sub-Gaussian given $\mathcal{F}_\ell$, $\|\langle v, \xi_{\ell+1}(\theta_\ell)\rangle \mid \mathcal{F}_\ell\|_{\Psi_2} \leq \sigma\|v\|$. Moreover, by the contraction inequality proved in Lemma C.1 (using $\gamma \leq \mu/L^2$), $\|a_\ell\|^2 = \|d_\ell - \gamma m_{\ell+1}(\theta_\ell)\|^2 \leq \|d_\ell\|^2$. Combining these yields

$$\|M_{\ell+1} \mid \mathcal{F}_\ell\|_{\Psi_2} \leq 2\gamma\sigma\|a_\ell\| \leq 2\gamma\sigma\|d_\ell\|. \tag{106}$$

Recall $\Delta_\ell := \theta_\ell^\star - \theta_{\ell+1}^\star$ and $e_\ell := \theta_\ell - \theta_\ell^\star$ so that $d_\ell = \theta_\ell - \theta_{\ell+1}^\star = e_\ell + \Delta_\ell$. By the predictability/measurability condition on $\theta_{\ell+1}^\star$ (Assumption 2.1), $\Delta_\ell$ is $\mathcal{F}_\ell$–measurable.

Fix $\rho := 1 - \gamma\mu/2 \in (0, 1)$ and define for $t \geq 1$

$$\mathfrak{D}_t := \sum_{\ell=0}^{t-1} \rho^{t-\ell-1}\|\Delta_\ell\|^2, \qquad \mathfrak{D}_t^{(2)} := \sum_{\ell=0}^{t-1} \rho^{2(t-\ell-1)}\|\Delta_\ell\|^2,$$

with $\mathfrak{D}_0 = \mathfrak{D}_0^{(2)} := 0$. Define the adapted, nondecreasing radius process

$$\mathfrak{B}_t := C_\star\left[\|\theta_0 - \theta_0^\star\|^2 + \frac{1}{\gamma\mu}\max_{s\in\{0,1,\ldots,t\}}\mathfrak{D}_s + \frac{d\sigma^2\gamma}{\mu} + d\sigma^2\gamma^2\log\frac{2T}{\delta} + \left(\frac{\sigma^2\gamma}{\mu} + \gamma^2\sigma^2\max_{s\in\{0,1,\ldots,t\}}\mathfrak{D}_s^{(2)}\right)\log\frac{2T}{\delta}\right],$$

where $C_\star > 0$ is a sufficiently large absolute constant. Since $\Delta_\ell$ is $\mathcal{F}_\ell$–measurable, each $\mathfrak{D}_s, \mathfrak{D}_s^{(2)}$ depends only on $(\Delta_0, \ldots, \Delta_{s-1})$ and hence $\mathfrak{B}_t$ is $\mathcal{F}_t$–measurable. Moreover, the max over $s \leq t$ makes $\mathfrak{B}_t$ nondecreasing in $t$.

Define the stopping time

$$\tau := \inf\{t \in \{0, 1, \ldots, T\} : \|e_t\|^2 > \mathfrak{B}_t\} \wedge (T+1). \tag{107}$$

Define the stopped increments $M_{\ell+1}^\tau := M_{\ell+1}\mathbf{1}\{\ell < \tau\}$. Since $\mathbf{1}\{\ell < \tau\}$ is $\mathcal{F}_\ell$–measurable and $\mathbb{E}[M_{\ell+1} \mid \mathcal{F}_\ell] = 0$,

$$\mathbb{E}[M_{\ell+1}^\tau \mid \mathcal{F}_\ell] = \mathbf{1}\{\ell < \tau\}\mathbb{E}[M_{\ell+1} \mid \mathcal{F}_\ell] = 0,$$

so $(M_{\ell+1}^\tau)_{\ell\geq 0}$ is also an MDS. On the event $\{\ell < \tau\}$ we have $\|e_\ell\|^2 \leq \mathfrak{B}_\ell$, hence by $(a+b)^2 \leq 2a^2 + 2b^2$,

$$\|d_\ell\|^2 = \|e_\ell + \Delta_\ell\|^2 \leq 2\|e_\ell\|^2 + 2\|\Delta_\ell\|^2 \leq 2\mathfrak{B}_\ell + 2\|\Delta_\ell\|^2 \qquad \text{on } \{\ell < \tau\}. \tag{108}$$

Plugging (108) into (106) yields the localized conditional sub-Gaussian bound:

$$\|M_{\ell+1}^\tau \mid \mathcal{F}_\ell\|_{\Psi_2} \leq 2\gamma\sigma\|d_\ell\|\mathbf{1}\{\ell < \tau\} \lesssim \gamma\sigma\sqrt{\mathfrak{B}_\ell + \|\Delta_\ell\|^2}\,\mathbf{1}\{\ell < \tau\}. \tag{109}$$

Fix an evaluation time $t \in \{1, \ldots, T\}$ and define the deterministic weights $w_\ell^{(t)} := \rho^{t-\ell-1}$ for $\ell = 0, \ldots, t-1$ and the weighted stopped increments

$$Z_{\ell+1}^{(t)} := w_\ell^{(t)} M_{\ell+1}^\tau = \rho^{t-\ell-1}M_{\ell+1}^\tau, \quad \ell = 0, \ldots, t-1.$$

Then $(Z_{\ell+1}^{(t)})_{\ell=0}^{t-1}$ is an MDS and by (109) we have

$$\|Z_{\ell+1}^{(t)} \mid \mathcal{F}_\ell\|_{\Psi_2} \lesssim v_\ell^{(t)}, \qquad v_\ell^{(t)} := \gamma\sigma\rho^{t-\ell-1}\sqrt{\mathfrak{B}_\ell + \|\Delta_\ell\|^2}\,\mathbf{1}\{\ell < \tau\}.$$

Since $\mathfrak{B}_\ell$ and $\Delta_\ell$ are $\mathcal{F}_\ell$–measurable, $v_\ell^{(t)}$ is predictable. Thus there exists an absolute constant $c > 0$ such that for all $\lambda \in \mathbb{R}$,

$$\mathbb{E}\left[\exp\left(\lambda Z_{\ell+1}^{(t)}\right) \,\middle|\, \mathcal{F}_\ell\right] \leq \exp\left(c\lambda^2(v_\ell^{(t)})^2\right) \qquad \text{a.s.}$$

Define $S_k^{(t)} := \sum_{\ell=0}^{k-1} Z_{\ell+1}^{(t)}$ and $V_k^{(t)} := \sum_{\ell=0}^{k-1}(v_\ell^{(t)})^2$, and

$$Y_k(\lambda) := \exp\left(\lambda S_k^{(t)} - c\lambda^2 V_k^{(t)}\right), \qquad k = 0, 1, \ldots, t.$$

Then $(Y_k(\lambda))_{k=0}^t$ is a nonnegative $(\mathcal{F}_k)$–supermartingale, so optional sampling at $\tau \wedge t$ (Lemma G.6) gives us $\mathbb{E}\big[Y_{\tau \wedge t}(\lambda)\big] \leq 1$. Since $Z_{\ell+1}^{(t)}$ already includes $\mathbf{1}\{\ell < \tau\}$, we have $S_{\tau \wedge t}^{(t)} = S_t^{(t)}$ and therefore, for any $s > 0$,

$$\mathbb{P}\left(\sum_{\ell=0}^{t-1} \rho^{t-\ell-1} M_{\ell+1}^{\tau} \geq s\right) \leq \exp\left(-\lambda s + c\lambda^2 V_t^{(t)}\right).$$

Optimizing over $\lambda$ yields

$$\mathbb{P}\left(\sum_{\ell=0}^{t-1} \rho^{t-\ell-1} M_{\ell+1}^{\tau} \geq s\right) \lesssim \exp\left(-\frac{s^2}{c V_t^{(t)}}\right). \tag{110}$$

It remains to upper bound $V_t^{(t)}$. Using $\mathbf{1}\{\ell < \tau\} \leq 1$ and that $\mathfrak{B}_\ell$ is nondecreasing,

$$V_t^{(t)} \lesssim \gamma^2 \sigma^2 \sum_{\ell=0}^{t-1} \rho^{2(t-\ell-1)}\left(\mathfrak{B}_\ell + \|\Delta_\ell\|^2\right) \leq \gamma^2 \sigma^2 \left(\mathfrak{B}_{t-1} \sum_{k=0}^{t-1} \rho^{2k} + \sum_{\ell=0}^{t-1} \rho^{2(t-\ell-1)} \|\Delta_\ell\|^2\right).$$

Since $\sum_{k=0}^{t-1} \rho^{2k} \leq \frac{1}{1-\rho^2} \leq \frac{2}{\gamma\mu}$, we obtain

$$V_t^{(t)} \lesssim \gamma^2 \sigma^2 \left(\frac{\mathfrak{B}_{t-1}}{\gamma\mu} + \mathfrak{D}_t^{(2)}\right). \tag{111}$$

Plugging (111) into (110) and choosing $s$ so that the RHS equals $\delta/(2T)$, we obtain the event

$$\mathcal{E}_M(t) := \left\{\sum_{\ell=0}^{t-1} \rho^{t-\ell-1} M_{\ell+1}^{\tau} \lesssim \gamma\sigma \sqrt{\left(\frac{\mathfrak{B}_{t-1}}{\gamma\mu} + \mathfrak{D}_t^{(2)}\right) \log \frac{2T}{\delta}}\right\},$$

holding with $\mathbb{P}(\mathcal{E}_M(t)) \geq 1 - \delta/(2T)$. A union bound over $t = 1, \ldots, T$ yields $\mathbb{P}(\cap_{t=1}^T \mathcal{E}_M(t)) \geq 1 - \delta/2$. Finally, applying Young's inequality to the RHS and using $\mathfrak{B}_{t-1} \leq \mathfrak{B}_T$, we obtain that on $\cap_{t=1}^T \mathcal{E}_M(t)$,

$$\sum_{\ell=0}^{t-1} \rho^{t-\ell-1} M_{\ell+1}^{\tau} \lesssim \mathfrak{B}_{t-1} + \frac{\sigma^2\gamma}{\mu} \log \frac{2T}{\delta} + \gamma^2 \sigma^2 \mathfrak{D}_t^{(2)} \log \frac{2T}{\delta}, \qquad \forall t \in [T]. \tag{112}$$

**Concluding the bound:** Let $\mathcal{E} := \mathcal{E}_{\boldsymbol{\xi}} \cap \bigcap_{t=1}^T \mathcal{E}_M(t)$. Then we have that $\mathbb{P}(\mathcal{E}) \geq 1 - \delta$. Since the martingale term was localized using the stopped increments $M_{\ell+1}^{\tau}$, the recursive inequality must first be applied to the stopped process. Combining the bound from $\mathcal{E}_{\boldsymbol{\xi}}$ for term (a) with (112) for term (b), and using the stopped version of Lemma C.1, we obtain for every $t \in [T]$,

$$\|e_{t \wedge \tau}\|^2 \lesssim \left(1 - \frac{\gamma\mu}{2}\right)^t \|e_0\|^2 + \frac{1}{\gamma\mu} \mathfrak{D}_t + \frac{d\sigma^2\gamma}{\mu} + d\sigma^2\gamma^2 \log \frac{2T}{\delta} + \left(\frac{\sigma^2\gamma}{\mu} + \gamma^2 \sigma^2 \mathfrak{D}_t^{(2)}\right) \log \frac{2T}{\delta}. \tag{113}$$

By the definition of $\mathfrak{B}_t$, the right-hand side of (113) is bounded by $\mathfrak{B}_t$. Hence, on $\mathcal{E}$,

$$\|e_{t \wedge \tau}\|^2 \leq \mathfrak{B}_t, \qquad \forall t \in [T]. \tag{114}$$

We now release the stopping time. Suppose for contradiction that $\tau \leq T$ on $\mathcal{E}$. Then taking $t = \tau$ in (114) gives

$$\|e_\tau\|^2 = \|e_{\tau \wedge \tau}\|^2 \leq \mathfrak{B}_\tau.$$

But this contradicts the definition of the first exit time

$$\tau := \inf\{t \in \{0, 1, \ldots, T\} : \|e_t\|^2 > \mathfrak{B}_t\} \wedge (T+1),$$

which requires $\|e_\tau\|^2 > \mathfrak{B}_\tau$ whenever $\tau \leq T$. Therefore $\tau = T + 1$ on $\mathcal{E}$. Consequently, on $\mathcal{E}$, the stopped and original processes coincide up to time $T$, i.e. $M_{\ell+1}^{\tau} = M_{\ell+1}$ for all $\ell \leq T - 1$, and (113) becomes the desired bound for $\|e_t\|^2$ for every $t \in [T]$. Since $\mathbb{P}(\mathcal{E}) \geq 1 - \delta$, the claimed high-probability result follows. $\qquad \square$

## D.2. Proof for SGDM high probability tracking error bound

Before we prove the (SGDM) high probability bound, we will first establish a recursive relation for the (SGDM) tracking error that we will use to prove the high probability bound, analogous to Proposition C.1.

**Proposition D.1** (Final iterate recursive relation for the (SGDM) tracking error)**.** *For all $t \geq 0$ and fixed $\gamma = \gamma_t \leq \min\left\{1/L, \mu(1-\beta)^2/4L^2\right\}$, the following recursive relation for the tracking error holds provided one takes a zero momentum initialization $\boldsymbol{\theta}_{-1} = \boldsymbol{\theta}_0$:*

$$\|\boldsymbol{\theta}_{t+1} - \boldsymbol{\theta}_{t+1}^\star\|^2 \lesssim \frac{2}{(1-\beta)^2}\tilde{\rho}^{t+1}\|\boldsymbol{\theta}_0 - \boldsymbol{\theta}_0^\star\|^2 + \frac{1}{\gamma\mu}\cdot\frac{1}{1-\beta}\mathfrak{D}_t^{\mathrm{lag}} + \frac{2\gamma^2}{(1-\beta)^2}\sum_{\ell=0}^t \tilde{\rho}^{t-\ell}\|\boldsymbol{\xi}_{\ell+1}(\boldsymbol{\psi}_\ell)\|^2 + \sum_{\ell=0}^t \tilde{\rho}^{t-\ell}M_{\ell+1}$$

*where $\tilde{\rho} = 1 - \eta^2\mu^2/4$, $\eta = \gamma/(1-\beta)$, $\mathfrak{D}_t^{\mathrm{lag}} := \sum_{\ell=0}^t \tilde{\rho}^{t-\ell}\|\boldsymbol{b}_\ell\|^2$, and the martingale increment is $M_{\ell+1} := 2\eta\langle\widetilde{\boldsymbol{B}}_\ell\boldsymbol{y}_\ell + \boldsymbol{r}_\ell, \boldsymbol{\zeta}_{\ell+1}\rangle$ where $\boldsymbol{b}_\ell, \widetilde{\boldsymbol{B}}_\ell, \boldsymbol{y}_\ell, \boldsymbol{r}_\ell, \boldsymbol{\zeta}_{\ell+1}$ are defined in Lemma C.2.*

*Proof of Proposition D.1.* Define $\eta := \gamma/(1-\beta)$. From Lemma C.2, we have the transformed "mode-splitting" coordinates as follows:

$$\boldsymbol{y}_t := \boldsymbol{V}^{-1}\boldsymbol{z}_t = \begin{bmatrix}\widehat{\boldsymbol{\theta}}_t \\ \check{\boldsymbol{\theta}}_t\end{bmatrix} = \frac{1}{1-\beta}\begin{bmatrix}\widetilde{\boldsymbol{\theta}}_t - \beta\widetilde{\boldsymbol{\theta}}_{t-1} \\ \widetilde{\boldsymbol{\theta}}_t - \widetilde{\boldsymbol{\theta}}_{t-1}\end{bmatrix}. \tag{115}$$

Using this transform with (29) gives us:

$$\underbrace{\begin{bmatrix}\widehat{\boldsymbol{\theta}}_{t+1} \\ \check{\boldsymbol{\theta}}_{t+1}\end{bmatrix}}_{\triangleq\ \boldsymbol{y}_{t+1}} = \underbrace{\begin{bmatrix}\mathbf{I}_d - \frac{\gamma_t}{1-\beta}\boldsymbol{H}_t & \frac{\gamma_t\beta'}{1-\beta}\boldsymbol{H}_t \\ -\frac{\gamma_t}{1-\beta}\boldsymbol{H}_t & \beta\mathbf{I}_d + \frac{\gamma_t\beta'}{1-\beta}\boldsymbol{H}_t\end{bmatrix}}_{\triangleq\ \widetilde{\boldsymbol{B}}_t}\begin{bmatrix}\widehat{\boldsymbol{\theta}}_t \\ \check{\boldsymbol{\theta}}_t\end{bmatrix} + \underbrace{\frac{1}{1-\beta}\begin{bmatrix}-(\mathbf{I}_d - \gamma_t\boldsymbol{H}_t)\boldsymbol{\Delta}_t - \boldsymbol{K}_t\boldsymbol{\Delta}_{t-1} \\ -(\mathbf{I}_d - \gamma_t\boldsymbol{H}_t)\boldsymbol{\Delta}_t - \boldsymbol{K}_t\boldsymbol{\Delta}_{t-1}\end{bmatrix}}_{\triangleq\ \boldsymbol{r}_t} + \underbrace{\frac{\gamma_t}{1-\beta}\begin{bmatrix}\boldsymbol{\xi}_{t+1}(\boldsymbol{\psi}_t) \\ \boldsymbol{\xi}_{t+1}(\boldsymbol{\psi}_t)\end{bmatrix}}_{\triangleq\ \eta\boldsymbol{\zeta}_{t+1}} \tag{116}$$

where

$$\beta' \triangleq \beta\beta_1 + \beta_2 \tag{117}$$

$$\boldsymbol{H}_t \triangleq \int_0^1 \nabla^2 F_{t+1}(\boldsymbol{\theta}_{t+1}^\star + s(\boldsymbol{\psi}_t - \boldsymbol{\theta}_{t+1}^\star))ds \tag{118}$$

$$\boldsymbol{K}_t = -\beta\mathbf{I}_d + \gamma_t\beta_1\boldsymbol{H}_t. \tag{119}$$

Define $E_t := \|\boldsymbol{y}_t\|^2 = \|\widehat{\boldsymbol{\theta}}_t\|^2 + \|\check{\boldsymbol{\theta}}_t\|^2$. From the momentum initialization, it follows that $\widehat{\boldsymbol{\theta}}_0 = (\widetilde{\boldsymbol{\theta}}_0 - \beta\widetilde{\boldsymbol{\theta}}_0)/(1-\beta)$ and $\check{\boldsymbol{\theta}}_0 = 0$. This implies that

$$E_0 := \|\boldsymbol{y}_0\|^2 \leq \frac{2}{(1-\beta)^2}\|\widetilde{\boldsymbol{\theta}}_0\|^2 \tag{120}$$

where the inequality holds from triangle inequality and $\beta \in (0,1)$. Another fact to note is that by definition of $\boldsymbol{y}_t$, one sees that $\widetilde{\boldsymbol{\theta}}_t = \widehat{\boldsymbol{\theta}}_t - \beta\check{\boldsymbol{\theta}}_t$. This implies

$$\|\widetilde{\boldsymbol{\theta}}_t\|^2 \lesssim \|\widehat{\boldsymbol{\theta}}_t\|^2 + \beta^2\|\check{\boldsymbol{\theta}}_t\|^2 \leq E_t \tag{121}$$

where the first inequality holds from triangle inequality and $\beta \in (0,1)$. Thus it suffices to upper bound $E_t$.

From (116), we have that

$$\boldsymbol{y}_{t+1} = \widetilde{\boldsymbol{B}}_t\boldsymbol{y}_t + \boldsymbol{r}_t + \eta\boldsymbol{\zeta}_{t+1}. \tag{122}$$

Expanding the square, we get

$$\|\boldsymbol{y}_{t+1}\|^2 = \|\widetilde{\boldsymbol{B}}_t\boldsymbol{y}_t + \boldsymbol{r}_t\|^2 + 2\eta\langle\widetilde{\boldsymbol{B}}_t\boldsymbol{y}_t + \boldsymbol{r}_t, \boldsymbol{\zeta}_{t+1}\rangle + \eta^2\|\boldsymbol{\zeta}_{t+1}\|^2. \tag{123}$$

We will first upper bound $\|\widetilde{\boldsymbol{B}}_t\boldsymbol{y}_t + \boldsymbol{r}_t\|^2$. Applying Young's inequality with $\alpha = \eta\mu/2 \in (0,1)$, we have

$$\begin{aligned}\|\widetilde{\boldsymbol{B}}_t\boldsymbol{y}_t + \boldsymbol{r}_t\|^2 &\leq \left(1 + \frac{\eta\mu}{2}\right)\|\widetilde{\boldsymbol{B}}_t\boldsymbol{y}_t\|^2 + \left(1 + \frac{2}{\eta\mu}\right)\|\boldsymbol{r}_t\|^2 \\ &\leq \left(1 + \frac{\eta\mu}{2}\right)\rho\|\boldsymbol{y}_t\|^2 + \left(1 + \frac{2}{\eta\mu}\right)\|\boldsymbol{r}_t\|^2.\end{aligned} \tag{124}$$

where the last inequality holds from the stability condition (Corollary C.3). Now observe that

$$\left(1 + \frac{\eta\mu}{2}\right)\rho = \left(1 + \frac{\eta\mu}{2}\right)\left(1 - \frac{\eta\mu}{2}\right) = 1 - \frac{\eta^2\mu^2}{4} \in (0, 1). \tag{125}$$

Now define $M_{t+1} := 2\eta\langle \widetilde{B}_t y_t + r_t, \zeta_{t+1}\rangle$. Since $\widetilde{B}_t y_t + r_t$ is $\mathcal{F}_t$ measurable and $\mathbb{E}[\zeta_{t+1} \mid \mathcal{F}_t] = \mathbf{0}$, we see that $M_{t+1}$ is a MDS. Furthermore note that since $\|\zeta_{t+1}\|^2 = 2\|\xi_{t+1}(\psi_t)\|^2$, we have

$$\eta^2\|\zeta_{t+1}\|^2 = 2\eta^2\|\xi_{t+1}(\psi_t)\|^2. \tag{126}$$

Combining everything, we get the following recursive relation:

$$E_{t+1} \leq \tilde{\rho}E_t + \left(1 + \frac{2}{\eta\mu}\right)\|r_t\|^2 + 2\eta^2\|\xi_{t+1}(\psi_t)\|^2 + M_{t+1}. \tag{127}$$

Unrolling this recursion, we find

$$E_{t+1} \leq \tilde{\rho}^{t+1}E_0 + \left(1 + \frac{2}{\eta\mu}\right)\sum_{\ell=0}^{t}\tilde{\rho}^{t-\ell}\|r_\ell\|^2 + 2\eta^2\sum_{\ell=0}^{t}\tilde{\rho}^{t-\ell}\|\xi_{\ell+1}(\psi_\ell)\|^2 + \sum_{\ell=0}^{t}\tilde{\rho}^{t-\ell}M_{\ell+1}. \tag{128}$$

We can now simplify the drift term using $r_\ell = [b_\ell; b_\ell]/(1 - \beta)$ (13). This gives us

$$\sum_{\ell=0}^{t}\tilde{\rho}^{t-\ell}\|r_\ell\|^2 = \frac{2}{(1-\beta)^2}\sum_{\ell=0}^{t}\tilde{\rho}^{t-\ell}\|b_\ell\|^2 := \frac{2}{(1-\beta)^2}\mathfrak{D}_t^{\mathrm{lag}}. \tag{129}$$

We can also use $\eta = \gamma/(1 - \beta)$ and (120) to get

$$E_{t+1} \leq \frac{2}{(1-\beta)^2}\tilde{\rho}^{t+1}\|\widetilde{\theta}_0\|^2 + \left(1 + \frac{2}{\eta\mu}\right)\frac{2}{(1-\beta)^2}\mathfrak{D}_t^{\mathrm{lag}} + \frac{2\gamma^2}{(1-\beta)^2}\sum_{\ell=0}^{t}\tilde{\rho}^{t-\ell}\|\xi_{\ell+1}(\psi_\ell)\|^2 + \sum_{\ell=0}^{t}\tilde{\rho}^{t-\ell}M_{\ell+1}. \tag{130}$$

Since $1 + 2/\eta\mu \lesssim 1/\eta\mu$, we have

$$\left(1 + \frac{2}{\eta\mu}\right)\frac{2}{(1-\beta)^2}\mathfrak{D}_t^{\mathrm{lag}} \lesssim \frac{1-\beta}{\gamma\mu}\frac{1}{(1-\beta)^2}\mathfrak{D}_t^{\mathrm{lag}} = \frac{1}{\gamma\mu}\cdot\frac{1}{1-\beta}\mathfrak{D}_t^{\mathrm{lag}}. \tag{131}$$

This lets us conclude using (121). $\qquad\square$

We can now establish a high-probability bound on the tracking error for (SGDM). This proof will be similar to Theorem 3.5.

**Theorem D.2** (High probability tracking error bound for (SGDM)). *Under Assumption 3.2, for all $t \in [T]$, $\gamma \leq \min\left\{1/L, \mu(1-\beta)^2/(4L^2)\right\}$, and $\delta \in (0, 1)$, provided one takes a zero momentum initialization $\theta_{-1} = \theta_0$, the following tracking error bound holds for (SGDM) with probability at least $1 - \delta$,*

$$\|\theta_t - \theta_t^\star\|^2 \lesssim \frac{1}{(1-\beta)^2}\exp\left(-\frac{\gamma^2\mu^2}{4(1-\beta)^2}t\right)\|\theta_0 - \theta_0^\star\|^2 + \frac{1}{\gamma\mu}\cdot\frac{1}{1-\beta}\mathfrak{D}_t^{\mathrm{lag}} + \frac{d\sigma^2}{\mu^2}$$

$$+ \frac{d\sigma^2\gamma^2}{(1-\beta)^2}\log\frac{2T}{\delta} + \left(\frac{\sigma^2}{\mu^2} + \frac{\sigma^2\gamma^2}{(1-\beta)^2}\mathfrak{D}_t^{\mathrm{lag},(2)}\right)\log\frac{2T}{\delta},$$

*where $\tilde{\rho} := 1 - \frac{\gamma^2\mu^2}{4(1-\beta)^2}$, $\mathfrak{D}_t^{\mathrm{lag}} := \sum_{\ell=0}^{t-1}\tilde{\rho}^{t-\ell-1}\|b_\ell\|^2$, and $\mathfrak{D}_t^{\mathrm{lag},(2)} := \sum_{\ell=0}^{t-1}\tilde{\rho}^{2(t-\ell-1)}\|b_\ell\|^2$ with $b_\ell := -(\mathbf{I}_d - \gamma H_\ell)\Delta_\ell - K_\ell\Delta_{\ell-1}$ and $H_\ell, K_\ell$ defined as in Lemma C.2.*

*Proof of Theorem D.2.* First recall by Proposition D.1 that we have,

$$\|\theta_t - \theta_t^\star\|^2 \lesssim \frac{2}{(1-\beta)^2}\tilde{\rho}^t\|\theta_0 - \theta_0^\star\|^2 + \frac{1}{\gamma\mu}\cdot\frac{1}{1-\beta}\mathfrak{D}_t^{\mathrm{lag}} + \underbrace{\frac{2\gamma^2}{(1-\beta)^2}\sum_{\ell=0}^{t-1}\tilde{\rho}^{t-\ell-1}\|\xi_{\ell+1}(\psi_\ell)\|^2}_{(a)} + \underbrace{\sum_{\ell=0}^{t-1}\tilde{\rho}^{t-\ell-1}M_{\ell+1}}_{(b)} \tag{132}$$

where $\tilde{\rho} = 1 - \eta^2\mu^2/4$, $\eta = \gamma/(1 - \beta)$, $\mathfrak{D}_t^{\mathrm{lag}} := \sum_{\ell=0}^{t-1}\tilde{\rho}^{t-\ell-1}\|b_\ell\|^2$, and the martingale increment is $M_{\ell+1} := 2\eta\langle\widetilde{B}_\ell y_\ell + r_\ell, \zeta_{\ell+1}\rangle$ where $b_\ell, \widetilde{B}_\ell, y_\ell, r_\ell, \zeta_{\ell+1}$ are defined in Lemma C.2. It remains to bound (a) and (b).

**Bounding part (a):** First notice the following equivalence:

$$\|\boldsymbol{\xi}_{\ell+1}(\boldsymbol{\psi}_\ell)\|^2 = \mathbb{E}[\|\boldsymbol{\xi}_{\ell+1}(\boldsymbol{\psi}_\ell)\|^2 \mid \mathcal{F}_\ell] + V_{\ell+1}, \quad V_{\ell+1} := \|\boldsymbol{\xi}_{\ell+1}(\boldsymbol{\psi}_\ell)\|^2 - \mathbb{E}[\|\boldsymbol{\xi}_{\ell+1}(\boldsymbol{\psi}_\ell)\|^2 \mid \mathcal{F}_\ell]. \tag{133}$$

First we bound $V_{\ell+1}$. Note that $V_{\ell+1}$ is a martingale difference sequence (MDS) and sub-exponential with $\|V_{\ell+1} \mid \mathcal{F}_\ell\|_{\Psi_1} \lesssim d\sigma^2$ (Lemma G.1). Then for fixed $t \leq T$, let $Z_{\ell+1}^{(t)} := 2\gamma^2 \tilde{\rho}^{t-\ell-1} V_{\ell+1}/(1-\beta)^2$. Then we have that $\|Z_{\ell+1}^{(t)} \mid \mathcal{F}_\ell\|_{\Psi_1} \lesssim 2\gamma^2 \tilde{\rho}^{t-\ell-1} d\sigma^2/(1-\beta)^2$. We also have the following:

$$\sum_{\ell=0}^{t-1} \frac{\gamma^4}{(1-\beta)^4} \tilde{\rho}^{2(t-\ell-1)} d^2\sigma^4 \lesssim \frac{d^2\sigma^4\gamma^2}{\mu^2(1-\beta)^2}, \qquad \max_{0 \leq \ell \leq t-1} \frac{\gamma^2}{(1-\beta)^2} \tilde{\rho}^{t-\ell-1} d\sigma^2 \lesssim \frac{d\sigma^2\gamma^2}{(1-\beta)^2}. \tag{134}$$

By Bernstein's inequality for conditionally sub-exponential martingale differences (Lemma G.5), there exists an absolute constant $c > 0$ such that for every $s > 0$,

$$\mathbb{P}\left(\sum_{\ell=0}^{t-1} Z_{\ell+1}^{(t)} \geq s\right) \lesssim \exp\left(-\min\left\{\frac{s^2\mu^2(1-\beta)^2}{d^2\sigma^4\gamma^2}, \frac{s(1-\beta)^2}{d\sigma^2\gamma^2}\right\}\right). \tag{135}$$

Take $s = C\left(d\sigma^2 \frac{\gamma}{\mu(1-\beta)}\sqrt{\log(2T/\delta)} + \frac{d\sigma^2\gamma^2}{(1-\beta)^2}\log(2T/\delta)\right)$ for a sufficiently large absolute constant $C > 0$. Then with probability at least $1 - \delta/(2T)$, we have

$$\sum_{\ell=0}^{t-1} Z_{\ell+1}^{(t)} \lesssim d\sigma^2 \frac{\gamma}{\mu(1-\beta)}\sqrt{\log\frac{2T}{\delta}} + \frac{d\sigma^2\gamma^2}{(1-\beta)^2}\log\frac{2T}{\delta}. \tag{136}$$

Furthermore since we have $\|\boldsymbol{\xi}_{\ell+1}(\boldsymbol{\psi}_\ell) \mid \mathcal{F}_\ell\|_{\Psi_2} \leq \sigma$, we have $\mathbb{E}[\|\boldsymbol{\xi}_{\ell+1}(\boldsymbol{\psi}_\ell)\|^2 \mid \mathcal{F}_\ell] \leq d\sigma^2$ (Lemma G.4). Thus we have:

$$\frac{2\gamma^2}{(1-\beta)^2}\sum_{\ell=0}^{t-1} \tilde{\rho}^{t-\ell-1}\mathbb{E}[\|\boldsymbol{\xi}_{\ell+1}(\boldsymbol{\psi}_\ell)\|^2 \mid \mathcal{F}_\ell] \lesssim \frac{\gamma^2}{(1-\beta)^2(1-\tilde{\rho})}d\sigma^2. \tag{137}$$

Since $1 - \tilde{\rho} = \eta^2\mu^2/4 = \gamma^2\mu^2/4(1-\beta)^2$, we have

$$\frac{\gamma^2}{(1-\beta)^2} \cdot \frac{1}{1-\tilde{\rho}} \asymp \frac{1}{\mu^2}. \tag{138}$$

Combining everything and applying AM-GM inequality to the first term of $s$, we obtain the event $\mathcal{E}_{\boldsymbol{\xi}}(t)$ that with probability at least $1 - \delta/2T$,

$$\frac{2\gamma^2}{(1-\beta)^2}\sum_{\ell=0}^{t-1} \tilde{\rho}^{t-\ell-1}\|\boldsymbol{\xi}_{\ell+1}(\boldsymbol{\psi}_\ell)\|^2 \lesssim \frac{d\sigma^2}{\mu^2} + \frac{d\sigma^2\gamma^2}{(1-\beta)^2}\log\frac{2T}{\delta}. \tag{139}$$

A union bound over $t = 1, \ldots, T$ gives the event $\mathcal{E}_{\boldsymbol{\xi}} = \cap_{t \leq T}\mathcal{E}_{\boldsymbol{\xi}}(t)$ with $\mathbb{P}(\mathcal{E}_{\boldsymbol{\xi}}) \geq 1 - \delta/2$. It remains to bound (b).

**Bounding part (b):** Recall that $M_{\ell+1} := 2\eta\langle \widetilde{\boldsymbol{B}}_\ell \boldsymbol{y}_\ell + \boldsymbol{r}_\ell, \boldsymbol{\zeta}_{\ell+1}\rangle$ where $\boldsymbol{b}_\ell$, $\widetilde{\boldsymbol{B}}_\ell$, $\boldsymbol{y}_\ell$, $\boldsymbol{r}_\ell$, $\boldsymbol{\zeta}_{\ell+1}$ are defined in Lemma C.2. Define $\boldsymbol{a}_\ell := \widetilde{\boldsymbol{B}}_\ell \boldsymbol{y}_\ell + \boldsymbol{r}_\ell \in \mathbb{R}^{2d}$. Since $\boldsymbol{a}_\ell$ is $\mathcal{F}_\ell$–measurable and $\mathbb{E}[\boldsymbol{\zeta}_{\ell+1} \mid \mathcal{F}_\ell] = \boldsymbol{0}$, we have $\mathbb{E}[M_{\ell+1} \mid \mathcal{F}_\ell] = 0$ and thus $(M_{\ell+1})_{\ell \geq 0}$ is a martingale difference sequence (MDS). By Assumption 3.2, for any $\mathcal{F}_\ell$–measurable unit vector $\boldsymbol{u}$, $\|\boldsymbol{u}^\top\boldsymbol{\zeta}_{\ell+1} \mid \mathcal{F}_\ell\|_{\Psi_2} \leq \sigma$ a.s. Hence, for any $\mathcal{F}_\ell$–measurable vector $\boldsymbol{v}$, $\langle \boldsymbol{v}, \boldsymbol{\zeta}_{\ell+1}\rangle$ is conditionally sub-Gaussian given $\mathcal{F}_\ell$, $\|\langle \boldsymbol{v}, \boldsymbol{\zeta}_{\ell+1}\rangle \mid \mathcal{F}_\ell\|_{\Psi_2} \leq \sigma\|\boldsymbol{v}\|$. This yields

$$\|M_{\ell+1} \mid \mathcal{F}_\ell\|_{\Psi_2} \leq 2\eta\,\sigma\,\|\boldsymbol{a}_\ell\|. \tag{140}$$

Since $\|\boldsymbol{a}_\ell\|$ depends on $\boldsymbol{y}_\ell$, which is random and can, a priori, be unbounded, we cannot directly apply a uniform sub-Gaussian martingale concentration inequality. We therefore localize via a predictable (time-varying) stopping radius, which remains valid under stochastic drift.

Recall $\eta := \gamma/(1-\beta)$ and

$$\tilde{\rho} := 1 - \frac{\eta^2\mu^2}{4} = 1 - \frac{\gamma^2\mu^2}{4(1-\beta)^2} \in (0, 1).$$

Define for $t \geq 1$ the drift-forcing aggregates

$$\mathfrak{D}_t^{\text{lag}} := \sum_{\ell=0}^{t-1} \tilde{\rho}^{t-\ell-1} \|\boldsymbol{b}_\ell\|^2, \qquad \mathfrak{D}_t^{\text{lag},(2)} := \sum_{\ell=0}^{t-1} \tilde{\rho}^{2(t-\ell-1)} \|\boldsymbol{b}_\ell\|^2,$$

with $\mathfrak{D}_0^{\text{lag}} = \mathfrak{D}_0^{\text{lag},(2)} := 0$. Define the adapted, nondecreasing radius process $(\mathfrak{B}_t)_{t=0}^T$ by

$$
\begin{aligned}
\mathfrak{B}_t := C_\star \Bigg[ &\frac{1}{(1-\beta)^2} \|\boldsymbol{\theta}_0 - \boldsymbol{\theta}_0^\star\|^2 + \frac{1}{\gamma\mu} \cdot \frac{1}{1-\beta} \max_{s \in \{0,1,\ldots,t\}} \mathfrak{D}_s^{\text{lag}} + \frac{d\sigma^2}{\mu^2} \\
&+ \frac{d\sigma^2 \gamma^2}{(1-\beta)^2} \log \frac{2T}{\delta} + \left( \frac{\sigma^2}{\mu^2} + \frac{\sigma^2 \gamma^2}{(1-\beta)^2} \max_{s \in \{0,1,\ldots,t\}} \mathfrak{D}_s^{\text{lag},(2)} \right) \log \frac{2T}{\delta} \Bigg],
\end{aligned}
\tag{141}
$$

where $C_\star > 0$ is a sufficiently large absolute constant. Since each $\boldsymbol{b}_\ell$ is $\mathcal{F}_\ell$–measurable (it depends only on predictable matrices and the drift increments up to time $\ell$), $\mathfrak{B}_t$ is $\mathcal{F}_t$–measurable and nondecreasing in $t$.

Define the stopping time

$$\tau := \inf \left\{ t \in \{0,1,\ldots,T\} : \|\boldsymbol{\theta}_t - \boldsymbol{\theta}_t^\star\|^2 > \mathfrak{B}_t \right\} \wedge (T+1). \tag{142}$$

Define the stopped increments $M_{\ell+1}^\tau := M_{\ell+1} \mathbf{1}\{\ell < \tau\}$. Since $\mathbf{1}\{\ell < \tau\}$ is $\mathcal{F}_\ell$–measurable and $\mathbb{E}[M_{\ell+1} \mid \mathcal{F}_\ell] = 0$,

$$\mathbb{E}[M_{\ell+1}^\tau \mid \mathcal{F}_\ell] = \mathbf{1}\{\ell < \tau\} \mathbb{E}[M_{\ell+1} \mid \mathcal{F}_\ell] = 0, \tag{143}$$

so $(M_{\ell+1}^\tau)_{\ell \geq 0}$ is also an MDS. On the event $\{\ell < \tau\}$ we have $\|\boldsymbol{e}_\ell\|^2 \leq \mathfrak{B}_\ell$ and also $\|\boldsymbol{e}_{\ell-1}\|^2 \leq \mathfrak{B}_{\ell-1} \leq \mathfrak{B}_\ell$. Moreover, by the definition of the $2d$-state $\boldsymbol{y}_\ell$ in Lemma C.2,

$$\|\boldsymbol{y}_\ell\|^2 \lesssim \|\boldsymbol{e}_\ell\|^2 + \|\boldsymbol{e}_{\ell-1}\|^2 \lesssim \mathfrak{B}_\ell \qquad \text{on } \{\ell < \tau\}. \tag{144}$$

Hence, using $\|\boldsymbol{a}_\ell\| = \|\widetilde{\boldsymbol{B}}_\ell \boldsymbol{y}_\ell + \boldsymbol{r}_\ell\| \leq \|\widetilde{\boldsymbol{B}}_\ell \boldsymbol{y}_\ell\| + \|\boldsymbol{r}_\ell\| \lesssim \|\boldsymbol{y}_\ell\| + \|\boldsymbol{r}_\ell\|$, we obtain on $\{\ell < \tau\}$ that

$$\|\boldsymbol{a}_\ell\| \lesssim \sqrt{\mathfrak{B}_\ell} + \|\boldsymbol{r}_\ell\|.$$

Plugging into (140) yields the localized $\Psi_2$ bound

$$\|M_{\ell+1}^\tau \mid \mathcal{F}_\ell\|_{\Psi_2} \lesssim 2\eta\sigma \left( \sqrt{\mathfrak{B}_\ell} + \|\boldsymbol{r}_\ell\| \right) \mathbf{1}\{\ell < \tau\}. \tag{145}$$

Fix an evaluation time $t \in \{1,\ldots,T\}$. Define the deterministic weights $w_\ell^{(t)} := \tilde{\rho}^{t-\ell-1}$ for $\ell = 0,\ldots,t-1$ and the weighted stopped increments

$$Z_{\ell+1}^{(t)} := w_\ell^{(t)} M_{\ell+1}^\tau = \tilde{\rho}^{t-\ell-1} M_{\ell+1}^\tau, \qquad \ell = 0,\ldots,t-1.$$

By (143) and the fact that $w_\ell^{(t)}$ is deterministic, $\mathbb{E}[Z_{\ell+1}^{(t)} \mid \mathcal{F}_\ell] = 0$, i.e., $(Z_{\ell+1}^{(t)})_{\ell=0}^{t-1}$ is an MDS. Let $S_k^{(t)} := \sum_{\ell=0}^{k-1} Z_{\ell+1}^{(t)}$ denote its partial sums ($k = 0,1,\ldots,t$). Define the predictable scale

$$v_\ell^{(t)} := \eta\sigma \tilde{\rho}^{t-\ell-1} \left( \sqrt{\mathfrak{B}_\ell} + \|\boldsymbol{r}_\ell\| \right) \mathbf{1}\{\ell < \tau\}, \qquad \ell = 0,\ldots,t-1,$$

so that (145) implies $\|Z_{\ell+1}^{(t)} \mid \mathcal{F}_\ell\|_{\Psi_2} \leq v_\ell^{(t)}$. A standard implication of conditional $\Psi_2$ control is the conditional mgf bound: there exists an absolute constant $c > 0$ such that for all $\lambda \in \mathbb{R}$,

$$\mathbb{E}\left[ \exp\left( \lambda Z_{\ell+1}^{(t)} \right) \Big| \mathcal{F}_\ell \right] \leq \exp\left( c\lambda^2 (v_\ell^{(t)})^2 \right) \qquad \text{a.s.} \tag{146}$$

Define the predictable variance proxy $V_k^{(t)} := \sum_{\ell=0}^{k-1} (v_\ell^{(t)})^2$ and the exponential process

$$Y_k(\lambda) := \exp\left( \lambda S_k^{(t)} - c\lambda^2 V_k^{(t)} \right), \qquad k = 0,1,\ldots,t.$$

Using (146) we have

$$\mathbb{E}[Y_{k+1}(\lambda) \mid \mathcal{F}_k] = Y_k(\lambda)\,\mathbb{E}\Big[\exp\big(\lambda Z_{k+1}^{(t)} - c\lambda^2 (v_k^{(t)})^2\big) \,\Big|\, \mathcal{F}_k\Big] \le Y_k(\lambda),$$

so $(Y_k(\lambda))_{k=0}^t$ is a nonnegative $(\mathcal{F}_k)$–supermartingale. Now apply optional sampling at the bounded stopping time $\tau \wedge t$ (Lemma G.6):

$$\mathbb{E}\big[Y_{\tau \wedge t}(\lambda)\big] \le \mathbb{E}[Y_0(\lambda)] = 1. \tag{147}$$

Since $Z_{\ell+1}^{(t)}$ already includes the factor $\mathbf{1}\{\ell < \tau\}$, we have $S_{\tau \wedge t}^{(t)} = S_t^{(t)} = \sum_{\ell=0}^{t-1} \tilde{\rho}^{t-\ell-1} M_{\ell+1}^{\tau}$. Therefore, by Markov's inequality and (147), for any $s > 0$,

$$\mathbb{P}\left(\sum_{\ell=0}^{t-1} \tilde{\rho}^{t-\ell-1} M_{\ell+1}^{\tau} \ge s\right) \le \exp\Big(-\lambda s + c\lambda^2 V_t^{(t)}\Big).$$

Optimizing over $\lambda$ yields the sub-Gaussian tail

$$\mathbb{P}\left(\sum_{\ell=0}^{t-1} \tilde{\rho}^{t-\ell-1} M_{\ell+1}^{\tau} \ge s\right) \lesssim \exp\left(-\frac{s^2}{c\,V_t^{(t)}}\right). \tag{148}$$

By definition of $v_\ell^{(t)}$ and using $\mathbf{1}\{\ell < \tau\} \le 1$ and monotonicity of $\mathfrak{B}_\ell$,

$$V_t^{(t)} = \sum_{\ell=0}^{t-1} (v_\ell^{(t)})^2 \lesssim \eta^2 \sigma^2 \sum_{\ell=0}^{t-1} \tilde{\rho}^{2(t-\ell-1)}\big(\mathfrak{B}_\ell + \|\mathbf{r}_\ell\|^2\big) \le \eta^2 \sigma^2 \left(\mathfrak{B}_{t-1} \sum_{k=0}^{t-1} \tilde{\rho}^{2k} + \sum_{\ell=0}^{t-1} \tilde{\rho}^{2(t-\ell-1)}\|\mathbf{r}_\ell\|^2\right).$$

Using $\sum_{k=0}^{t-1} \tilde{\rho}^{2k} \le \frac{1}{1-\tilde{\rho}^2} \le \frac{1}{1-\tilde{\rho}} \asymp \frac{(1-\beta)^2}{\gamma^2 \mu^2}$ and $\eta = \gamma/(1-\beta)$ gives

$$V_t^{(t)} \lesssim \frac{\sigma^2}{\mu^2}\mathfrak{B}_{t-1} + \frac{\sigma^2 \gamma^2}{(1-\beta)^2} \sum_{\ell=0}^{t-1} \tilde{\rho}^{2(t-\ell-1)}\|\mathbf{r}_\ell\|^2. \tag{149}$$

Moreover, by the linear relation between $\mathbf{r}_\ell$ and the drift forcing $\mathbf{b}_\ell$ in Lemma C.2, we have $\|\mathbf{r}_\ell\|^2 \lesssim \|\mathbf{b}_\ell\|^2$ and thus

$$\sum_{\ell=0}^{t-1} \tilde{\rho}^{2(t-\ell-1)}\|\mathbf{r}_\ell\|^2 \lesssim \mathfrak{D}_t^{\mathrm{lag},(2)}.$$

Plugging into (149) yields

$$V_t^{(t)} \lesssim \frac{\sigma^2}{\mu^2}\mathfrak{B}_{t-1} + \frac{\sigma^2 \gamma^2}{(1-\beta)^2}\mathfrak{D}_t^{\mathrm{lag},(2)}.$$

Choose $s > 0$ so that $\exp\Big(-s^2/c\,V_t^{(t)}\Big) = \delta/(2T)$. Then with probability at least $1 - \delta/(2T)$,

$$\sum_{\ell=0}^{t-1} \tilde{\rho}^{t-\ell-1} M_{\ell+1}^{\tau} \lesssim \sqrt{V_t^{(t)} \log \frac{2T}{\delta}}. \tag{150}$$

Using the above bound on $V_t^{(t)}$ and $\sqrt{a+b} \le \sqrt{a} + \sqrt{b}$ gives

$$\sum_{\ell=0}^{t-1} \tilde{\rho}^{t-\ell-1} M_{\ell+1}^{\tau} \lesssim \sqrt{\frac{\sigma^2}{\mu^2}\mathfrak{B}_{t-1} \log \frac{2T}{\delta}} + \sqrt{\frac{\sigma^2 \gamma^2}{(1-\beta)^2}\mathfrak{D}_t^{\mathrm{lag},(2)} \log \frac{2T}{\delta}}. \tag{151}$$

Applying Young's inequality $\sqrt{\mathcal{B}}\,u \le \frac{1}{4}\mathcal{B} + u^2$ to the first term with $u := \sqrt{\frac{\sigma^2}{\mu^2}\log\frac{2T}{\delta}}$, and to the second term in the same way, yields

$$\sum_{\ell=0}^{t-1} \tilde{\rho}^{t-\ell-1} M_{\ell+1}^{\tau} \lesssim \mathfrak{B}_{t-1} + \frac{\sigma^2}{\mu^2}\log\frac{2T}{\delta} + \frac{\sigma^2 \gamma^2}{(1-\beta)^2}\mathfrak{D}_t^{\mathrm{lag},(2)} \log \frac{2T}{\delta}. \tag{152}$$

Call this event $\mathcal{E}_M(t)$. A union bound over $t \in [T]$ yields

$$\mathcal{E}_M := \bigcap_{t=1}^{T} \mathcal{E}_M(t) \qquad \text{with} \qquad \mathbb{P}(\mathcal{E}_M) \ge 1 - \delta/2. \tag{153}$$

**Concluding the bound:** Let $\mathcal{E} := \mathcal{E}_{\boldsymbol{\xi}} \cap \mathcal{E}_M$, so that $\mathbb{P}(\mathcal{E}) \geq 1 - \delta$. Since the martingale term in part (b) was localized through the stopped increments $M_{\ell+1}^{\tau}$, the recursive estimate must first be written for the stopped process. Combining the bound for part (a) from $\mathcal{E}_{\boldsymbol{\xi}}$ with (152) for part (b), and applying the stopped version of Proposition D.1, we obtain for every $t \in [T]$ that

$$
\|\boldsymbol{\theta}_{t \wedge \tau} - \boldsymbol{\theta}_{t \wedge \tau}^{\star}\|^2 \lesssim \frac{2}{(1-\beta)^2} \tilde{\rho}^t \|\boldsymbol{\theta}_0 - \boldsymbol{\theta}_0^{\star}\|^2 + \frac{1}{\gamma\mu} \cdot \frac{1}{1-\beta} \mathfrak{D}_t^{\text{lag}} + \frac{d\sigma^2}{\mu^2} \\
+ \frac{d\sigma^2 \gamma^2}{(1-\beta)^2} \log \frac{2T}{\delta} + \left( \frac{\sigma^2}{\mu^2} + \frac{\sigma^2 \gamma^2}{(1-\beta)^2} \mathfrak{D}_t^{\text{lag},(2)} \right) \log \frac{2T}{\delta}.
\tag{154}
$$

By the definition of $\mathfrak{B}_t$ in (141), the right-hand side of (154) is bounded by $\mathfrak{B}_t$. Hence, on $\mathcal{E}$,

$$
\|\boldsymbol{\theta}_{t \wedge \tau} - \boldsymbol{\theta}_{t \wedge \tau}^{\star}\|^2 \leq \mathfrak{B}_t, \qquad \forall t \in [T].
\tag{155}
$$

We now release the stopping time. Suppose for contradiction that $\tau \leq T$ on $\mathcal{E}$. Then taking $t = \tau$ in (155) gives

$$
\|\boldsymbol{\theta}_{\tau} - \boldsymbol{\theta}_{\tau}^{\star}\|^2 \leq \mathfrak{B}_{\tau}.
$$

But this contradicts the definition of the first exit time

$$
\tau := \inf \left\{ t \in \{0, 1, \ldots, T\} : \|\boldsymbol{\theta}_t - \boldsymbol{\theta}_t^{\star}\|^2 > \mathfrak{B}_t \right\} \wedge (T+1),
$$

which requires $\|\boldsymbol{\theta}_{\tau} - \boldsymbol{\theta}_{\tau}^{\star}\|^2 > \mathfrak{B}_{\tau}$ whenever $\tau \leq T$. Therefore $\tau = T + 1$ on $\mathcal{E}$. Consequently, on $\mathcal{E}$, the stopped and original processes coincide throughout the horizon, i.e. $M_{\ell+1}^{\tau} = M_{\ell+1}$ for all $\ell \leq T - 1$. Thus (154) becomes the desired bound for the original iterates for every $t \in [T]$. Taking $t = T$ yields

$$
\|\boldsymbol{\theta}_T - \boldsymbol{\theta}_T^{\star}\|^2 \lesssim \mathfrak{B}_T \qquad \text{on } \mathcal{E}.
$$

Since $\mathbb{P}(\mathcal{E}) \geq 1 - \delta$, the claimed high-probability bound follows. $\qquad\square$

## E. Minimax lower bounds for SGDM under nonstationary strongly-convex losses

Before we proceed with obtaining the minimax lower bound, we must first reduce the problem of lower bounding regret to the problem of lower bounding the success probability of testing a sequence of functions. We do this as there are information theoretic tools such as Fano's inequality (Cover & Thomas, 2012) that be used. We first give the definition that measures the "discrepancy" between two functions $g, \widetilde{g} : \Theta \to \mathbb{R}$:

$$
\chi(g, \widetilde{g}) := \inf_{\boldsymbol{\theta} \in \Theta} \max\{g(\boldsymbol{\theta}) - g^{\star}, \widetilde{g}(\boldsymbol{\theta}) - \widetilde{g}^{\star}\} \quad \text{where} \quad g^{\star} = \inf_{\boldsymbol{\theta} \in \Theta} g(\boldsymbol{\theta}), \ \widetilde{g}^{\star} = \inf_{\boldsymbol{\theta} \in \Theta} \widetilde{g}(\boldsymbol{\theta}).
$$

This measures characterizes the best regret one can achieve without knowing whether $g$ or $\widetilde{g}$ is the underlying function. This allows us to obtain a reduction from regret minimization to testing problems. This is analogous to the reduction used by (Besbes et al., 2015; Chen et al., 2019).

### E.1. From regret minimization to testing

Fix $1 \leq p < \infty$, $1 \leq q \leq \infty$, and $V_T > 0$. Let $\phi^G$ denote the noisy gradients feedback model. Consider any finite packing $\mathcal{P} = \{g^{(1)}, \ldots, g^{(M)}\} \subseteq \mathcal{G}_{p,q}(V_T)$ where each $g^{(i)} = (g_1^{(i)}, \ldots, g_T^{(i)})$ is a length-$T$ sequence of convex losses. The following reduction formalizes that *uniformly small regret* over $\mathcal{G}_{p,q}(V_T)$ implies the existence of a *hypothesis test* that identifies the true sequence $g \in \mathcal{P}$ with constant probability. We include a proof for completeness but note that this result is standard in (Besbes et al., 2015; Chen et al., 2019).

**Lemma E.1** (Reduction from regret to testing). *Let $\mathcal{P} \subseteq \mathcal{G}_{p,q}(V_T)$ be finite, and let $\chi(\cdot, \cdot)$ be a nonnegative per-round separation functional. Suppose there exists an admissible policy $\pi$ such that*

$$
\sup_{g \in \mathcal{G}_{p,q}(V_T)} \mathcal{R}_T^{\pi, \phi^G}(g) \leq \frac{1}{9} \inf_{\substack{g, \tilde{g} \in \mathcal{P} \\ g \neq \tilde{g}}} \sum_{t=1}^{T} \chi(g_t, \tilde{g}_t).
\tag{156}
$$

*Then there exists an estimator (measurable decision rule) mapping the interactions and feedback $\{(\boldsymbol{\theta}_{t-1}, \phi_t^G)\}_{t=1}^T \longmapsto \widehat{g} \in \mathcal{P}$ such that*

$$\sup_{g \in \mathcal{P}} \mathbb{P}_g[\widehat{g} \neq g] \leq \frac{1}{3}, \tag{157}$$

*where $\mathbb{P}_g$ denotes the distribution induced by the policy $\pi$ and the feedback model $\phi^G$ when the underlying function sequence is $g$.*

*Proof of Lemma E.1.* Fix any admissible policy $\pi$ satisfying (156). For a realized actions and feedback produced by $\pi$, write the (realized) dynamic regret of $\pi$ on a sequence $g$ as

$$\mathcal{R}_T^\pi(g) = \sum_{t=1}^T \big(g_t(\boldsymbol{\theta}_{t-1}) - g_t^\star\big), \qquad g_t^\star := \inf_{\boldsymbol{\theta} \in \Theta} g_t(\boldsymbol{\theta}),$$

where $\{\boldsymbol{\theta}_{t-1}\}_{t=1}^T$ are the actions generated by $\pi$ and the expectation in $\sup_{g \in \mathcal{G}_{p,q}(V_T)} \mathcal{R}_T^\pi(g)$ is over the feedback noise under $\mathbb{P}_g$. Let

$$\Delta := \inf_{\substack{g, \tilde{g} \in \mathcal{P} \\ g \neq \tilde{g}}} \sum_{t=1}^T \chi(g_t, \tilde{g}_t).$$

By the assumption in (156), for every $g \in \mathcal{G}_{p,q}(V_T)$ we have

$$\mathbb{E}_g\big[\mathcal{R}_T^\pi(g)\big] \leq \frac{1}{9}\Delta.$$

Applying Markov's inequality to the nonnegative random variable $\mathcal{R}_T^\pi(g)$ yields, for every $g \in \mathcal{P}$,

$$\mathbb{P}_g\left(\mathcal{R}_T^\pi(g) \leq \frac{1}{3}\Delta\right) \geq 1 - \frac{\mathbb{E}_g[\mathcal{R}_T^\pi(g)]}{\Delta/3} \geq 1 - \frac{(1/9)\Delta}{\Delta/3} = \frac{2}{3}. \tag{158}$$

Define the estimator $\widehat{g}$ as the empirical risk minimizer over $\mathcal{P}$ evaluated on the realized action sequence $\{\boldsymbol{\theta}_{t-1}\}_{t=1}^T$:

$$\widehat{g} \in \arg\min_{g \in \mathcal{P}} \sum_{t=1}^T \big(g_t(\boldsymbol{\theta}_{t-1}) - g_t^\star\big). \tag{159}$$

By definition of $\widehat{g}$,

$$\sum_{t=1}^T \big(\widehat{g}_t(\boldsymbol{\theta}_{t-1}) - \widehat{g}_t^\star\big) \leq \sum_{t=1}^T \big(g_t(\boldsymbol{\theta}_{t-1}) - g_t^\star\big) = \mathcal{R}_T^\pi(g). \tag{160}$$

Now condition on the event $\mathcal{E}_g := \big\{\mathcal{R}_T^\pi(g) \leq \frac{1}{3}\Delta\big\}$ which has probability at least $2/3$ under $\mathbb{P}_g$ by (158). On $\mathcal{E}_g$, we can upper bound the separation between $\widehat{g}$ and $g$ as follows. First, by the definition of $\chi$ assumed in the lemma statement (nonnegative per-round separation),

$$\chi(\widehat{g}_t, g_t) \leq \inf_{\boldsymbol{\theta} \in \Theta} \max\left\{\widehat{g}_t(\boldsymbol{\theta}) - \widehat{g}_t^\star, \, g_t(\boldsymbol{\theta}) - g_t^\star\right\} \leq \max\left\{\widehat{g}_t(\boldsymbol{\theta}_{t-1}) - \widehat{g}_t^\star, \, g_t(\boldsymbol{\theta}_{t-1}) - g_t^\star\right\}, \tag{161}$$

and hence, summing over $t$ and using $\max\{a, b\} \leq a + b$ for $a, b \geq 0$,

$$\begin{aligned}
\sum_{t=1}^T \chi(\widehat{g}_t, g_t) &\leq \sum_{t=1}^T \max\left\{\widehat{g}_t(\boldsymbol{\theta}_{t-1}) - \widehat{g}_t^\star, \, g_t(\boldsymbol{\theta}_{t-1}) - g_t^\star\right\} \\
&\leq \sum_{t=1}^T \big(\widehat{g}_t(\boldsymbol{\theta}_{t-1}) - \widehat{g}_t^\star\big) + \sum_{t=1}^T \big(g_t(\boldsymbol{\theta}_{t-1}) - g_t^\star\big) \\
&\leq 2\sum_{t=1}^T \big(g_t(\boldsymbol{\theta}_{t-1}) - g_t^\star\big) = 2\,\mathcal{R}_T^\pi(g),
\end{aligned} \tag{162}$$

where in the last step we used (160). On $\mathcal{E}_g$, (162) gives

$$\sum_{t=1}^{T} \chi(\widehat{g}_t, g_t) \ \leq \ 2\,\mathcal{R}_T^{\pi}(g) \ \leq \ \frac{2}{3}\Delta. \tag{163}$$

Finally, we show that (163) forces $\widehat{g} = g$. Indeed, if $\widehat{g} \neq g$, then by definition of $\Delta$,

$$\sum_{t=1}^{T} \chi(\widehat{g}_t, g_t) \ \geq \ \inf_{\substack{g,\tilde{g}\in\mathcal{P} \\ g\neq\tilde{g}}} \sum_{t=1}^{T} \chi(g_t, \tilde{g}_t) \ = \ \Delta,$$

which contradicts (163). Therefore, on the event $\mathcal{E}_g$ we must have $\widehat{g} = g$. Combining with (158), we conclude

$$\mathbb{P}_g(\widehat{g} \neq g) \ \leq \ \mathbb{P}_g(\mathcal{E}_g^c) \ \leq \ \frac{1}{3},$$

and taking the supremum over $g \in \mathcal{P}$ proves (157). $\qquad\qquad\square$

Now let $D_{\mathrm{KL}}(P \,\|\, Q) = \int \log \frac{dP}{dQ} dP$ denote the Kullback-Leibler divergence between two distributions $P$ and $Q$. We now introduce the following version of Fano's inequality:

**Lemma E.2** (Fano's inequality). *Let $\mathcal{P} = \{\vartheta_1, \ldots, \vartheta_M\}$ be a finite parameter set with $|\mathcal{P}| = M$. For each $\vartheta \in \mathcal{P}$, let $P_\vartheta$ denote the distribution of the observations under parameter $\vartheta$. Suppose there exists a constant $0 < \alpha < \infty$ such that $D_{\mathrm{KL}}(P_\vartheta \,\|\, P_{\vartheta'}) \leq \alpha$ for all $\vartheta, \vartheta' \in \mathcal{P}$. Then*

$$\inf_{\widehat{\vartheta}} \sup_{\vartheta\in\mathcal{P}} \mathbb{P}_\vartheta\left[\widehat{\vartheta} \neq \vartheta\right] \ \geq \ 1 - \frac{\alpha + \log 2}{\log M}.$$

With Lemma E.1 and Lemma E.2, obtaining a minimax lower bound on the regret reduces to finding a "hard" subset $\mathcal{P} \subseteq \mathcal{G}_{p,q}(\mathbb{V}_T)$ such that we can lower bound $\inf_{g,\tilde{g}\in\mathcal{P}} \sum_{t=1}^{T} \chi(g_t, \tilde{g}_t)$ and upper bound $\sup_{g,\tilde{g}\in\mathcal{P}} D_{\mathrm{KL}}(P_g \,\|\, P_{\tilde{g}})$. We proceed by constructing two $\mu$-strongly convex, $\mathcal{O}(\mu)$-smooth losses $g_+$ and $g_-$ whose minimizers are separated by $2a$ along $\boldsymbol{e}_1$, but whose gradients differ only inside a ball of radius $\asymp a$. This localization makes $\|\nabla g_+ - \nabla g_-\|_{L^p(\Theta)}$ of order $\mu a^{1+d/p}$ so we can switch between $g_+$ and $g_-$ many times while staying within the gradient-variation budget. We then build a $J$-block family of sequences $g_{1:T}^{\boldsymbol{u}}$ indexed by $\boldsymbol{u} \in \{\pm 1\}^J$ where each block uses either $g_+$ or $g_-$ and $\Delta_T := \lfloor T/J \rfloor$ for $1 \leq J \leq T$ to be determined later. If an algorithm cannot reliably infer $\boldsymbol{u}$ from the noisy gradients, it must play near the wrong minimizer on a constant fraction of the time indices, costing $\Omega(\mu a^2)$ loss per round. Hence the regret is $\Omega(\mu a^2 T)$ times the testing error. Finally we can upper bound the mutual information using Gaussian KL chain rule, apply Fano's inequality, and choose $\Delta_T$ to conclude.

### E.2. Constructing hard smooth convex functions with localized gradients

We will proceed with constructing hard $\mu$-strongly convex, $\mathcal{O}(\mu)$-smooth losses $g_+$ and $g_-$. Fix a smooth bump $\psi : \mathbb{R}^d \to [0,1]$ such that $\psi(\boldsymbol{\theta}) = 1$ for $\|\boldsymbol{\theta}\| \leq 1/2$ and $\psi(\boldsymbol{\theta}) = 0$ for $\|\boldsymbol{\theta}\| \geq 1$ with $\|\nabla\psi(\boldsymbol{\theta})\| \leq C_\psi$ and $\|\nabla^2\psi(\boldsymbol{\theta})\|_{\mathrm{op}} \leq C_\psi$ for all $\boldsymbol{\theta} \in \Theta$. For scale $r > 0$, define $\psi_r(\boldsymbol{\theta}) := \psi(\boldsymbol{\theta}/r)$. Then we have that $\|\nabla\psi_r(\boldsymbol{\theta})\| \leq C_\psi/r$ and $\|\nabla^2\psi_r(\boldsymbol{\theta})\|_{\mathrm{op}} \leq C_\psi/r^2$. Let $\boldsymbol{e}_1 = (1, 0, \ldots, 0)$. Choose parameters $a > 0$ and $r > 0$ with $a \leq r/4$. Define on $\mathbb{R}^d$ the following function:

$$g_u(\boldsymbol{\theta}) := \frac{\mu}{2}\|\boldsymbol{\theta}\|^2 - u \cdot \mu a\langle\boldsymbol{\theta}, \boldsymbol{e}_1\rangle\psi_r(\boldsymbol{\theta}), \quad u \in \{+1, -1\}. \tag{164}$$

We now show that this function is strongly convex and smooth on all of $\mathbb{R}^d$.

**Lemma E.3** (Smoothness and strong convexity of $g_u$). *There exists a universal constant $c_1 > 0$ such that if $a/r \leq c_1$, then each $g_u$ is $\mu/4$ strongly convex and $2\mu$-smooth on all of $\mathbb{R}^d$.*

*Proof of Lemma E.3.* We can compute the gradient as follows:

$$\nabla_{\boldsymbol{\theta}} g_u(\boldsymbol{\theta}) = \mu\boldsymbol{\theta} - u \cdot \mu a\left(\boldsymbol{e}_1\psi_r(\boldsymbol{\theta}) + \langle\boldsymbol{\theta}, \boldsymbol{e}_1\rangle\nabla\psi_r(\boldsymbol{\theta})\right). \tag{165}$$

Similarly we can compute the Hessian as:

$$\nabla^2 g_u(\boldsymbol{\theta}) = \mu \mathbf{I}_d - u \cdot \mu a \left( \boldsymbol{e}_1 (\nabla \psi_r(\boldsymbol{\theta}))^\top + (\nabla \psi_r(\boldsymbol{\theta})) \boldsymbol{e}_1^\top + \langle \boldsymbol{\theta}, \boldsymbol{e}_1 \rangle \nabla^2 \psi_r(\boldsymbol{\theta}) \right). \tag{166}$$

Using $\|\boldsymbol{e}_1 (\nabla \psi_r(\boldsymbol{\theta}))^\top\| \le \|\nabla \psi_r(\boldsymbol{\theta})\|$, the derivative bounds for the bump function, and $|\langle \boldsymbol{\theta}, \boldsymbol{e}_1 \rangle| \le \|\boldsymbol{\theta}\| \le r$ whenever $\psi_r(\boldsymbol{\theta}) \ne 0$, we get for all $\boldsymbol{\theta} \in \Theta$:

$$\|\nabla^2 g_u(\boldsymbol{\theta}) - \mu \mathbf{I}_d\|_{\mathrm{op}} \le \mu a \left( 2\|\nabla \psi_r(\boldsymbol{\theta})\| + |\langle \boldsymbol{\theta}, \boldsymbol{e}_1 \rangle| \|\nabla^2 \psi_r(\boldsymbol{\theta})\|_{\mathrm{op}} \right) \le \frac{3\mu a C_\psi}{r}. \tag{167}$$

So if $a/r \le c_1 := 1/(12 C_\psi)$, we have $\|\nabla^2 g_u(\boldsymbol{\theta}) - \mu \mathbf{I}_d\|_{\mathrm{op}} \le \mu/4$. This completes the proof. $\qquad\square$

We now show that with this construction, the discrepancy between these functions is lower bounded by $\mu a^2$.

**Lemma E.4** (Minimizers and discrepancy of the two-point construction). *Assume $a \le r/4$ and $a/r \le c_1$, where $c_1 > 0$ is the constant from Lemma E.3. Let $g_u$ be defined in (164) for $u \in \{+1, -1\}$, and assume that $\Theta$ contains the line segment $\{t a \boldsymbol{e}_1 : t \in [-1, 1]\}$.*

1. (***Minimizers.***) *Each $g_u$ admits a unique minimizer $\boldsymbol{\theta}_u^\star$, and $\boldsymbol{\theta}_u^\star = u a \boldsymbol{e}_1$.*

2. (***Discrepancy.***) *The two-point discrepancy satisfies $\chi(g_+, g_-) \ge \mu a^2 / 8$.*

*Proof of Lemma E.4.* On the Euclidean ball $\{\boldsymbol{\theta} : \|\boldsymbol{\theta}\|_2 \le r/2\}$, we have $\psi_r(\boldsymbol{\theta}) = 1$ and $\nabla \psi_r(\boldsymbol{\theta}) = 0$ by construction of the bump. Therefore, for all $\|\boldsymbol{\theta}\|_2 \le r/2$, $\nabla g_u(\boldsymbol{\theta}) = \mu \boldsymbol{\theta} - u \mu a \boldsymbol{e}_1$. Setting $\nabla g_u(\boldsymbol{\theta}) = 0$ yields $\boldsymbol{\theta} = u a \boldsymbol{e}_1$, and this point indeed lies in the region where the above simplification holds since $\|u a \boldsymbol{e}_1\|_2 = a \le r/4 < r/2$. By Lemma E.3, each $g_u$ is $\mu/4$–strongly convex on $\mathbb{R}^d$, hence admitting a unique global minimizer, which must coincide with its unique stationary point. Thus $\boldsymbol{\theta}_u^\star = u a \boldsymbol{e}_1$ and this proves the first claim.

To prove the second claim, by $\mu/4$–strong convexity of $g_u$, for every $\boldsymbol{\theta} \in \Theta$,

$$g_u(\boldsymbol{\theta}) - g_u(\boldsymbol{\theta}_u^\star) \ge \frac{\mu}{8} \|\boldsymbol{\theta} - \boldsymbol{\theta}_u^\star\|_2^2.$$

Define $d_+(\boldsymbol{\theta}) := \|\boldsymbol{\theta} - a \boldsymbol{e}_1\|_2$ and $d_-(\boldsymbol{\theta}) := \|\boldsymbol{\theta} + a \boldsymbol{e}_1\|_2$. Then

$$\max\{d_+(\boldsymbol{\theta})^2, d_-(\boldsymbol{\theta})^2\} \ge \frac{1}{2}\left(d_+(\boldsymbol{\theta})^2 + d_-(\boldsymbol{\theta})^2\right).$$

Moreover, by the parallelogram identity,

$$d_+(\boldsymbol{\theta})^2 + d_-(\boldsymbol{\theta})^2 = \|\boldsymbol{\theta} - a \boldsymbol{e}_1\|_2^2 + \|\boldsymbol{\theta} + a \boldsymbol{e}_1\|_2^2 = 2\|\boldsymbol{\theta}\|_2^2 + 2\|a \boldsymbol{e}_1\|_2^2 \ge 2a^2,$$

so $\max\{d_+(\boldsymbol{\theta})^2, d_-(\boldsymbol{\theta})^2\} \ge a^2$. Combining with the strong-convexity lower bound gives, for all $\boldsymbol{\theta} \in \Theta$,

$$\max\{g_+(\boldsymbol{\theta}) - g_+(\boldsymbol{\theta}_+^\star), \ g_-(\boldsymbol{\theta}) - g_-(\boldsymbol{\theta}_-^\star)\} \ge \frac{\mu}{8} \max\{d_+(\boldsymbol{\theta})^2, d_-(\boldsymbol{\theta})^2\} \ge \frac{\mu a^2}{8}.$$

Taking $\inf_{\boldsymbol{\theta} \in \Theta}$ of the left-hand side yields $\chi(g_+, g_-) \ge \mu a^2 / 8$. Finally, since $\boldsymbol{\theta}_u^\star = u a \boldsymbol{e}_1 \in \Theta$ by the segment assumption, we have $\min_{\boldsymbol{\theta} \in \Theta} g_u(\boldsymbol{\theta}) = g_u(\boldsymbol{\theta}_u^\star)$ for each $u$. This proves the second claim. $\qquad\square$

Next we show that this localization makes $\|\nabla g_+ - \nabla g_-\|_{L^p(\Theta)}$ of order $\mu a^{1+d/p}$ so we can switch between $g_+$ and $g_-$ many times while staying within the gradient-variation budget.

**Lemma E.5** (Localized gradient difference). *Assume $r \asymp a$ (i.e., $c_- a \le r \le c_+ a$ for universal constants $c_-, c_+ > 0$). Then there exists a universal constant $C > 0$ such that*

$$\|\nabla g_+ - \nabla g_-\|_{L^p(\Theta)} \le C \mu a \, r^{d/p} \asymp C \mu a^{1+d/p}.$$

*Equivalently, writing $\alpha := 1 + d/p$, we have $\|\nabla g_+ - \nabla g_-\|_{L^p(\Theta)} \lesssim \mu a^\alpha$.*

*Proof of Lemma E.5.* From Lemma E.3, we explicitly computed $\nabla_{\boldsymbol{\theta}} g_u$. Using this, we find

$$\nabla g_+(\boldsymbol{\theta}) - \nabla g_-(\boldsymbol{\theta}) = -2\mu a \Big( \boldsymbol{e}_1 \, \psi_r(\boldsymbol{\theta}) + \langle \boldsymbol{\theta}, \boldsymbol{e}_1 \rangle \, \nabla \psi_r(\boldsymbol{\theta}) \Big).$$

The right-hand side vanishes whenever $\psi_r(\boldsymbol{\theta}) = 0$, hence it is supported on $B(0, r)$. On this support, $|\langle \boldsymbol{\theta}, \boldsymbol{e}_1 \rangle| \leq \|\boldsymbol{\theta}\|_2 \leq r$, and by the bump derivative bounds $\|\nabla \psi_r(\boldsymbol{\theta})\|_2 \leq C_\psi / r$. Therefore, for all $\boldsymbol{\theta} \in \Theta$,

$$\|\nabla g_+(\boldsymbol{\theta}) - \nabla g_-(\boldsymbol{\theta})\|_2 \leq 2\mu a \Big( \|\boldsymbol{e}_1\|_2 \, |\psi_r(\boldsymbol{\theta})| + |\langle \boldsymbol{\theta}, \boldsymbol{e}_1 \rangle| \, \|\nabla \psi_r(\boldsymbol{\theta})\|_2 \Big) \leq 2\mu a \Big( 1 + r \cdot \frac{C_\psi}{r} \Big) \leq C_0 \, \mu a,$$

for a universal constant $C_0 > 0$, and the quantity is $0$ outside $B(0, r)$. Hence

$$\|\nabla g_+ - \nabla g_-\|_{L^p(\Theta)} = \Big( \int_\Theta \|\nabla g_+(\boldsymbol{\theta}) - \nabla g_-(\boldsymbol{\theta})\|_2^p \, d\boldsymbol{\theta} \Big)^{1/p} \leq (C_0 \mu a) \, |\Theta \cap B(0, r)|^{1/p} \leq (C_0 \mu a) \, |B(0, r)|^{1/p}.$$

Using $|B(0, r)| = c_d r^d$ yields

$$\|\nabla g_+ - \nabla g_-\|_{L^p(\Theta)} \leq (C_0 \mu a) \, (c_d r^d)^{1/p} = C \, \mu \, a \, r^{d/p}.$$

If additionally $r \asymp a$, then $a \, r^{d/p} \asymp a^{1+d/p}$, giving the claimed scaling. $\qquad\square$

A simple consequence of Lemma E.5 and constructing $g_u$ through a smooth bump function is that we can obtain pointwise localization as stated in the following result:

**Lemma E.6** (Pointwise localization)**.** *Let $g_u$ be defined by (164). Then for all $\boldsymbol{\theta} \in \Theta$,*

$$\big\|\nabla g_+(\boldsymbol{\theta}) - \nabla g_-(\boldsymbol{\theta})\big\|_2^2 \ \leq \ 4C\mu^2 a^2 \mathbf{1}\{\|\boldsymbol{\theta}\|_2 < r\}, \tag{168}$$

*for some universal $C > 0$.*

*Proof of Lemma E.6.* From Lemma E.5, we found that

$$\nabla g_+(\boldsymbol{\theta}) - \nabla g_-(\boldsymbol{\theta}) = -2\mu a \Big( \boldsymbol{e}_1 \, \psi_r(\boldsymbol{\theta}) + \langle \boldsymbol{\theta}, \boldsymbol{e}_1 \rangle \, \nabla \psi_r(\boldsymbol{\theta}) \Big).$$

The right-hand side vanishes whenever $\psi_r(\boldsymbol{\theta}) = 0$, hence it is supported on $B(0, r)$. On this support, $|\langle \boldsymbol{\theta}, \boldsymbol{e}_1 \rangle| \leq \|\boldsymbol{\theta}\|_2 \leq r$, and by the bump derivative bounds $\|\nabla \psi_r(\boldsymbol{\theta})\|_2 \leq C_\psi / r$. Using the triangle inequality, we find

$$\big\|\nabla g_+(\boldsymbol{\theta}) - \nabla g_-(\boldsymbol{\theta})\big\|_2^2 \leq 4C\mu^2 a^2 \mathbf{1}\{\|\boldsymbol{\theta}\|_2 < r\}.$$

This concludes the proof. $\qquad\square$

### E.3. Inducing sharp momentum penalty into the noise-dominated term

We now show that the noise-dominated (information-theoretic) minimax term inherits sharp dependence on the momentum parameter $\beta$. The key is to exploit that, under noisy gradient feedback, the KL divergence between two environments is proportional to the cumulative squared mean shift observed along the algorithm's trajectory. For our localized bump construction, this mean shift is supported on a small region. Define $\mathrm{Occ}(r) := \sum_{t=0}^{T-1} \mathbf{1}\{\|\boldsymbol{\theta}_t\|_2 \leq r\}$. We will show that by tuning the bump radius into a rare-visit regime whose visitation probability depends on $\beta$, the expected occupation time $\mathbb{E}^\pi[\mathrm{Occ}(r)]$ scales with $(1 - \beta)$ which will in turn cause the KL to scale as if the noise variance were inflated from $\sigma^2$ to $\sigma^2 / (1 - \beta)$.

Before we prove this claim, recall our localized construction (164). Since $\psi_r(\boldsymbol{\theta}) = 0$ and $\nabla \psi_r(\boldsymbol{\theta}) = 0$ whenever $\|\boldsymbol{\theta}\|_2 \geq r$, we have the identity $\nabla g_u(\boldsymbol{\theta}) = \mu \, \boldsymbol{\theta}$ for all $\|\boldsymbol{\theta}\|_2 \geq r$ and all $u \in \{+1, -1\}$. Thus, outside the bump region, the (SGDM) iterate evolves according to the same linear recursion (independent of $u$). This is the mechanism we exploit: the visitation frequency of the bump is controlled by the stationary spread of this linear system, and this spread depends sharply on $\beta$. We will focus on Polyak Heavy-Ball but we note that this analysis can be extended to Nesterov acceleration.

We now quantify the fundamental reason the occupation time depends on momentum: in the presence of gradient noise, Heavy-Ball (SGDM) exhibits a stationary covariance that scales as $(1 - \beta)^{-1}$.

**Lemma E.7** (Exact stationary variance of Heavy-Ball on a quadratic). *Fix $\mu > 0$, $\gamma > 0$, $\beta \in [0, 1)$, and $\sigma > 0$. Consider the one-dimensional Heavy-Ball recursion on the quadratic $x \mapsto \frac{\mu}{2}x^2$ with additive Gaussian gradient noise:*

$$x_{t+1} = (1 + \beta - \gamma\mu)\, x_t - \beta\, x_{t-1} - \gamma\, \zeta_{t+1}, \qquad \zeta_{t+1} \overset{\text{i.i.d.}}{\sim} \mathcal{N}(0, \sigma^2), \tag{169}$$

*for $t \geq 0$ with deterministic initialization $(x_0, x_{-1}) \in \mathbb{R}^2$. Define the state vector $s_t := (x_t, x_{t-1})^\top \in \mathbb{R}^2$ and matrices*

$$A := \begin{pmatrix} 1 + \beta - \gamma\mu & -\beta \\ 1 & 0 \end{pmatrix}, \qquad B := \begin{pmatrix} -\gamma \\ 0 \end{pmatrix}. \tag{170}$$

*Then (169) becomes the linear system $s_{t+1} = As_t + B\zeta_{t+1}$. Assume the stability condition $\rho(A) < 1$, where $\rho(\cdot)$ denotes spectral radius (equivalently, the roots of $\lambda^2 - (1 + \beta - \gamma\mu)\lambda + \beta = 0$ lie strictly inside the unit disk). Then:*

1. *The Markov chain $(s_t)_{t \geq 0}$ admits a unique stationary distribution $\pi_\infty = \mathcal{N}(0, \Sigma_\infty)$ on $\mathbb{R}^2$.*

2. *The stationary covariance $\Sigma_\infty$ is the unique positive semidefinite solution of the Lyapunov equation*

$$\Sigma = A\Sigma A^\top + \sigma^2 BB^\top. \tag{171}$$

3. *Writing $\Sigma_\infty = \left(\begin{smallmatrix} v & c \\ c & v \end{smallmatrix}\right)$ with $v = \mathrm{Var}_{\pi_\infty}(x_t)$, we have*

$$v = \frac{(1+\beta)\gamma^2\sigma^2}{(1-\beta)\big((1+\beta)^2 - (1+\beta-\gamma\mu)^2\big)} = \frac{(1+\beta)\gamma\sigma^2}{(1-\beta)\mu\big(2(1+\beta) - \gamma\mu\big)}. \tag{172}$$

4. *Stability requires $\gamma\mu < 2(1+\beta)$, so the denominator in (172) is positive. Moreover, if $\gamma\mu \leq 1 + \beta$, then $v \geq \gamma\sigma^2/(4\mu(1-\beta))$.*

*Proof of Lemma E.7.* We proceed in four steps: (1) existence/uniqueness of a stationary law, (2) Lyapunov characterization, (3) explicit solution, and (4) the lower bound.

**Step 1: Existence and uniqueness under $\rho(A) < 1$.** Fix deterministic $s_0 \in \mathbb{R}^2$. Iterating the linear system gives

$$s_t = A^t s_0 + \sum_{k=1}^{t} A^{t-k} B\zeta_k, \qquad t \geq 1. \tag{173}$$

Since $(\zeta_k)_{k \geq 1}$ are independent Gaussians, each $s_t$ is Gaussian in $\mathbb{R}^2$ with $\mathbb{E}[s_t] = A^t s_0$. Under $\rho(A) < 1$, there exist constants $C \geq 1$, $\rho \in (0, 1)$ such that $\|A^t\| \leq C\rho^t$ for all $t \geq 0$, hence $\mathbb{E}[s_t] \to 0$. Let $\Sigma_t := \mathrm{Cov}(s_t)$. From the linear system and independence,

$$\Sigma_{t+1} = A\Sigma_t A^\top + \sigma^2 BB^\top, \qquad t \geq 0, \tag{174}$$

with $\Sigma_0 = 0$ if $s_0$ is deterministic. Iterating yields

$$\Sigma_t = \sum_{j=0}^{t-1} A^j\, (\sigma^2 BB^\top)\, (A^j)^\top \; + \; A^t \Sigma_0 (A^t)^\top. \tag{175}$$

The second term vanishes as $t \to \infty$ since $\|A^t\| \to 0$. The first term converges absolutely because $\|A^j(\sigma^2 BB^\top)(A^j)^\top\| \leq \sigma^2 C^2 \rho^{2j}\|B\|^2$ and $\sum_{j \geq 0} \rho^{2j} < \infty$. Therefore $\Sigma_t \to \Sigma_\infty$ where

$$\Sigma_\infty = \sum_{j=0}^{\infty} A^j\, (\sigma^2 BB^\top)\, (A^j)^\top. \tag{176}$$

Define $\pi_\infty := \mathcal{N}(0, \Sigma_\infty)$. The law of $s_t$ converges weakly to $\pi_\infty$, which is stationary. For uniqueness, if $\pi$ is any stationary distribution with finite second moment, then $\Sigma_\pi := \mathrm{Cov}_\pi(s)$ solves (171). Under $\rho(A) < 1$, this equation has a unique positive semidefinite solution (Step 2), hence $\Sigma_\pi = \Sigma_\infty$, identifying $\pi = \mathcal{N}(0, \Sigma_\infty)$.

**Step 2: Lyapunov characterization.** Vectorizing (171) gives $(\mathbf{I}_4 - \boldsymbol{A} \otimes \boldsymbol{A})\text{vec}(\boldsymbol{\Sigma}) = \sigma^2 \text{vec}(\boldsymbol{BB}^\top)$. Since $\rho(\boldsymbol{A} \otimes \boldsymbol{A}) = \rho(\boldsymbol{A})^2 < 1$, the operator $\mathbf{I}_4 - \boldsymbol{A} \otimes \boldsymbol{A}$ is invertible with unique solution

$$\text{vec}(\boldsymbol{\Sigma}_\infty) = \sum_{j=0}^\infty (\boldsymbol{A} \otimes \boldsymbol{A})^j \sigma^2 \text{vec}(\boldsymbol{BB}^\top) = \sigma^2 \sum_{j=0}^\infty \text{vec}\big(\boldsymbol{A}^j \boldsymbol{BB}^\top (\boldsymbol{A}^j)^\top\big),$$

which upon unvectorizing recovers (176). This proves uniqueness of the PSD solution.

**Step 3: Explicit computation of $v = \text{Var}_{\pi_\infty}(x_t)$.** By stationarity, $x_t$ and $x_{t-1}$ have identical marginals, so $\boldsymbol{\Sigma}_\infty = \left(\begin{smallmatrix} v & c \\ c & v \end{smallmatrix}\right)$ where $v = \mathbb{E}_{\pi_\infty}[x_t^2]$ and $c = \mathbb{E}_{\pi_\infty}[x_t x_{t-1}]$. Note that $\sigma^2 \boldsymbol{BB}^\top = \sigma^2 \left(\begin{smallmatrix} \gamma^2 & 0 \\ 0 & 0 \end{smallmatrix}\right)$. Let $a := 1 + \beta - \gamma\mu$. Direct calculation gives

$$\boldsymbol{A}\boldsymbol{\Sigma}_\infty\boldsymbol{A}^\top = \begin{pmatrix} a^2 v - 2a\beta c + \beta^2 v & av - \beta c \\ av - \beta c & v \end{pmatrix}.$$

Equating with $\boldsymbol{\Sigma}_\infty - \sigma^2 \boldsymbol{BB}^\top$ in (171), the $(1,2)$ entry yields $c = av - \beta c$, so

$$c = \frac{a}{1+\beta} v. \tag{177}$$

The $(1,1)$ entry gives $v = (a^2 + \beta^2)v - 2a\beta c + \gamma^2 \sigma^2$. Substituting (177) and simplifying yields

$$v = \frac{(1+\beta)\gamma^2 \sigma^2}{(1-\beta)\big((1+\beta)^2 - a^2\big)}.$$

Since $a = 1 + \beta - \gamma\mu$ and $(1+\beta)^2 - a^2 = \gamma\mu(2(1+\beta) - \gamma\mu)$, this gives (172).

**Step 4: Positivity and lower bound.** Stability requires $(1+\beta)^2 - a^2 > 0$, equivalently $\gamma\mu < 2(1+\beta)$, making the denominator in (172) positive. If $\gamma\mu \leq 1 + \beta$, then $2(1+\beta) - \gamma\mu \leq 2(1+\beta)$, so

$$v = \frac{(1+\beta)\gamma\sigma^2}{(1-\beta)\mu\big(2(1+\beta) - \gamma\mu\big)} \geq \frac{(1+\beta)\gamma\sigma^2}{(1-\beta)\mu \cdot 2(1+\beta)} \geq \frac{\gamma\sigma^2}{4\mu(1-\beta)}.$$

This completes the proof. $\qquad\square$

Fix $u \in \{+1, -1\}$ and let $(\boldsymbol{\theta}_t)$ denote Heavy–Ball iterates on $g_u$ with Gaussian gradient noise (as in (SGDM)). Define the *quadratic-core proxy* $(\tilde{\boldsymbol{\theta}}_t)$ driven by the *same* noises and the *same* initialization by

$$\tilde{\boldsymbol{\theta}}_{t+1} = \tilde{\boldsymbol{\theta}}_t + \beta(\tilde{\boldsymbol{\theta}}_t - \tilde{\boldsymbol{\theta}}_{t-1}) - \gamma(\mu\tilde{\boldsymbol{\theta}}_t + \boldsymbol{\zeta}_{t+1}), \qquad (\tilde{\boldsymbol{\theta}}_0, \tilde{\boldsymbol{\theta}}_{-1}) = (\boldsymbol{\theta}_0, \boldsymbol{\theta}_{-1}), \tag{178}$$

where $\boldsymbol{\zeta}_{t+1} \overset{\text{i.i.d.}}{\sim} \mathcal{N}(\mathbf{0}, \sigma^2 \mathbf{I}_d)$. Since (178) is a linear affine function of $(\boldsymbol{\zeta}_1, \ldots, \boldsymbol{\zeta}_t)$, every coordinate of $\tilde{\boldsymbol{\theta}}_t$ is Gaussian for each fixed $t$. We use $(\tilde{\boldsymbol{\theta}}_t)$ to justify a Gaussian small-ball bound. Fix any orthogonal index $j \in \{2, \ldots, d\}$ and define the projections

$$x_t := \langle \boldsymbol{\theta}_t - \boldsymbol{\theta}_u^\star, \boldsymbol{e}_j \rangle, \qquad \tilde{x}_t := \langle \tilde{\boldsymbol{\theta}}_t - \boldsymbol{\theta}_u^\star, \boldsymbol{e}_j \rangle, \qquad d_t := x_t - \tilde{x}_t.$$

Subtracting the $\boldsymbol{e}_j$-coordinates of the true and proxy recursions gives, for all $t \geq 0$,

$$d_{t+1} = (1 + \beta - \gamma\mu)\, d_t - \beta\, d_{t-1} + \eta_t^{(j)}, \qquad \eta_t^{(j)} := \gamma u \cdot \mu a \langle \boldsymbol{\theta}_t, \boldsymbol{e}_1 \rangle \partial_j \psi_r(\boldsymbol{\theta}_t), \tag{179}$$

with $(d_0, d_{-1}) = (0, 0)$. Since $\psi_r(\boldsymbol{\theta}) = 0$ for $\|\boldsymbol{\theta}\|_2 \geq r$ and $\|\nabla\psi_r(\boldsymbol{\theta})\|_2 \leq C_\psi/r$, we have on the support of $\eta_t^{(j)}$ that $|\langle \boldsymbol{\theta}_t, \boldsymbol{e}_1 \rangle| \leq \|\boldsymbol{\theta}_t\|_2 \leq r$. This results in

$$|\eta_t^{(j)}| \leq C_\psi \gamma\mu a \qquad \forall t \geq 0. \tag{180}$$

Let $\boldsymbol{A} = \left(\begin{smallmatrix} 1+\beta-\gamma\mu & -\beta \\ 1 & 0 \end{smallmatrix}\right)$ be the companion matrix from (170). In the small-step regime

$$0 < \gamma\mu \leq \frac{1-\beta}{4}, \tag{181}$$

the eigenvalues of $\boldsymbol{A}$ are real and lie in $(0, 1)$, and a standard diagonalization argument yields $\sum_{k=0}^{\infty} \|\boldsymbol{A}^k\|_2 \leq C(1 - \beta)/(\gamma\mu)$ for a universal constant $C > 0$. Iterating (179) from $(d_0, d_{-1}) = (0, 0)$ and using (180) therefore gives the uniform bound

$$\sup_{t \geq 0} |d_t| \leq \Delta, \qquad \Delta := C_\Delta \, a(1 + C_\psi)(1 - \beta), \tag{182}$$

for a universal constant $C_\Delta > 0$. Finally, for every $t$ we have deterministically

$$\{\|\boldsymbol{\theta}_t - \boldsymbol{\theta}_u^\star\|_2 \leq r\} \subseteq \{|x_t| \leq r\} \subseteq \{|\tilde{x}_t| \leq r + \Delta\},$$

where the first inclusion uses $|\langle \boldsymbol{v}, \boldsymbol{e}_j \rangle| \leq \|\boldsymbol{v}\|_2$ and the second uses $|\tilde{x}_t| \leq |x_t| + |d_t|$ with (182). Consequently,

$$\mathrm{Occ}_T(r) := \sum_{t=0}^{T-1} \mathbf{1}\{\|\boldsymbol{\theta}_t - \boldsymbol{\theta}_u^\star\|_2 \leq r\} \; \leq \; \sum_{t=0}^{T-1} \mathbf{1}\{|\tilde{x}_t| \leq r + \Delta\}. \tag{183}$$

Thus it suffices to bound Gaussian small-ball probabilities for the proxy coordinate $\tilde{x}_t$.

**Lemma E.8** ($\beta$-dependent occupation bound in the rare-visit regime)**.** *Assume $\rho(\boldsymbol{A}) < 1$ and $\gamma\mu \leq 1 + \beta$ so that Lemma E.7 applies. Let $\tilde{x}_t$ be the proxy coordinate and write $v_\infty := \mathrm{Var}_{\pi_\infty}(\tilde{x})$ for the stationary variance of the one-dimensional quadratic-core recursion. By Lemma E.7, $v_\infty \geq \gamma\sigma^2/(4\mu(1 - \beta))$. Then for every $R > 0$,*

$$\mathbb{E}\left[\sum_{t=0}^{T-1} \mathbf{1}\{|\tilde{x}_t| \leq R\}\right] \; \leq \; T\sqrt{\frac{2}{\pi}} \frac{R}{\sqrt{v_\infty}} \; + \; T_0, \tag{184}$$

*where $T_0 \in \mathbb{N}$ is a finite burn-in index (depending on $(\beta, \gamma\mu)$ and initialization) such that $\mathrm{Var}(\tilde{x}_t) \geq v_\infty/2$ for all $t \geq T_0$. Combining (183) with (184) at $R = r + \Delta$ yields*

$$\mathbb{E}^u[\mathrm{Occ}_T(r)] \lesssim C_1 \, T \cdot (r + \Delta)\sqrt{\frac{\mu(1 - \beta)}{\gamma\sigma^2}} \tag{185}$$

*for a universal constant $C_1 > 0$. Consequently, choosing the bump radius $r := c_r \sigma \sqrt{\gamma(1 - \beta)/\mu}$ in the rare-visit regime and calibrating the bump amplitude $\alpha$ so that $\Delta \leq r$ (equivalently, $C_A C_\psi \gamma\mu\alpha \leq r$) gives*

$$\mathbb{E}^u[\mathrm{Occ}_T(r)] \lesssim C_2 \, T(1 - \beta) \tag{186}$$

*for a universal constant $C_2 > 0$.*

*Proof of Lemma E.8.* Fix $R > 0$. For any Gaussian $Z \sim \mathcal{N}(m, v)$, we have

$$\mathbb{P}(|Z| \leq R) = \int_{-R}^{R} \frac{1}{\sqrt{2\pi v}} \exp\left(-\frac{(z - m)^2}{2v}\right) dz \leq \frac{2R}{\sqrt{2\pi v}} = \sqrt{\frac{2}{\pi}} \frac{R}{\sqrt{v}}.$$

Since $\tilde{x}_t$ is Gaussian, $\mathbb{P}(|\tilde{x}_t| \leq R) \leq \sqrt{2/\pi} \, R/\sqrt{\mathrm{Var}(\tilde{x}_t)}$ for all $t$. By Lemma E.7, the covariance converges to its stationary value, so there exists finite $T_0$ such that $\mathrm{Var}(\tilde{x}_t) \geq v_\infty/2$ for all $t \geq T_0$. Therefore,

$$\mathbb{E}\left[\sum_{t=0}^{T-1} \mathbf{1}\{|\tilde{x}_t| \leq R\}\right] = \sum_{t=0}^{T-1} \mathbb{P}(|\tilde{x}_t| \leq R) \leq T_0 + \sum_{t=T_0}^{T-1} \sqrt{\frac{2}{\pi}} \frac{R}{\sqrt{v_\infty/2}} \leq T_0 + T\sqrt{\frac{2}{\pi}} \frac{R}{\sqrt{v_\infty}},$$

proving (184). Taking $R = r + \Delta$ and combining with (183) gives

$$\mathbb{E}^u[\mathrm{Occ}_T(r)] \leq T\sqrt{\frac{2}{\pi}} \frac{r + \Delta}{\sqrt{v_\infty}} + T_0.$$

By Lemma E.7, $v_\infty \geq \gamma\sigma^2/(4\mu(1 - \beta))$, hence $1/\sqrt{v_\infty} \leq 2\sqrt{\mu(1 - \beta)/(\gamma\sigma^2)}$, yielding (185). Substituting $r = c_r \sigma \sqrt{\gamma(1 - \beta)/\mu}$ and enforcing $\Delta \leq r$ gives $r + \Delta \leq 2r = 2c_r \sigma \sqrt{\gamma(1 - \beta)/\mu}$, which yields (186). $\qquad\square$

### E.4. Constructing a $J$-block hard family

We now build a $J$-block packing family of nonstationary sequences $G^{\boldsymbol{u}}_{1:T}$ indexed by sign vectors $\boldsymbol{u} \in \{\pm 1\}^J$, where each block uses one of the two base losses $\{g_+, g_-\}$. Fix an integer $1 \leq J \leq T$ (to be tuned later) and set $\Delta_T := \lfloor T/J \rfloor$. Partition the horizon $[T] := \{1, \ldots, T\}$ into $J$ disjoint consecutive batches (blocks) $B_1, \ldots, B_J$ such that each $B_j$ has cardinality either $\Delta_T$ or $\Delta_T + 1$ and $\bigcup_{j=1}^J B_j = [T]$. For each block $j \in [J]$, let $|B_j|$ denote its length and write its indices as

$$B_j = \{t_j(1), t_j(2), \ldots, t_j(|B_j|)\}, \qquad t_j(1) < \cdots < t_j(|B_j|).$$

Let $\{\pm 1\}^J$ be the class of sign vectors of length $J$, and let $\mathcal{U} \subset \{\pm 1\}^J$. For any $\boldsymbol{u} = (u_1, \ldots, u_J) \in \mathcal{U}$, define the nonstationary loss sequence $G^{\boldsymbol{u}}_{1:T}$ by holding the loss fixed within each block,

$$G^{\boldsymbol{u}}_t(\cdot) := g_{u_j}(\cdot) \qquad \text{for all } t \in B_j, \ j \in [J]. \tag{187}$$

Thus $G^{\boldsymbol{u}}$ can change only at block boundaries while encoding one bit per block. To apply Fano's method we require a large subset $\mathcal{U} \subset \{\pm 1\}^J$ whose elements are well separated in Hamming distance. We obtain such a set via a constant-weight code construction. This is the same construction used by (Chen et al., 2019) and originates from (Graham & Sloane, 1980) and (Wang & Singh, 2016) gave an explicit lower bound which we cite below. For simplicity assume $J$ is even (the odd case follows by restricting to $J - 1$ coordinates).

**Lemma E.9** ((Wang & Singh, 2016), Lemma 9). *Suppose $J \geq 2$ is even. There exists a subset $\mathcal{I} \subset \{0,1\}^J$ such that (i) every $\boldsymbol{i} \in \mathcal{I}$ has exactly $J/2$ ones, i.e. $\sum_{j=1}^J i_j = J/2$; and (ii) every pair $\boldsymbol{i} \neq \boldsymbol{i}' \in \mathcal{I}$ satisfies $d_H(\boldsymbol{i}, \boldsymbol{i}') \geq J/16$, where $d_H$ is Hamming distance. Moreover, $\log |\mathcal{I}| \geq 0.0625\,J$.*

By Lemma E.9, any two indices $\boldsymbol{u} \neq \boldsymbol{v}$ disagree on at least $J/16$ blocks; equivalently, the sequences $G^{\boldsymbol{u}}$ and $G^{\boldsymbol{v}}$ differ on at least $\sum_{j:\,u_j \neq v_j} |B_j| \geq (J/16)\Delta_T \gtrsim T$ time indices, which is precisely the separation needed to convert a constant testing error (via Fano) into an $\Omega(T)$ regret gap. We next bound the gradient-variation functional $\mathrm{GVar}_{p,q}$ for the $J$-block construction.

**Lemma E.10** ($\mathrm{GVar}_{p,q}$ bound for the $J$-block construction). *Let $1 \leq J \leq T$ and set $\Delta_T := \lfloor T/J \rfloor$. Consider the $J$ disjoint consecutive blocks $B_1, \ldots, B_J$ with $|B_j| \in \{\Delta_T, \Delta_T + 1\}$ and the blockwise-constant sequence $G^{\boldsymbol{u}}_{1:T}$ defined by (187) for some $\boldsymbol{u} \in \{\pm 1\}^J$. Then, for every such $\boldsymbol{u}$,*

$$\mathrm{GVar}_{p,q}(G^{\boldsymbol{u}}_{1:T}) \ \leq \ C\,\mu\,a^\alpha\,J^{1/q}, \qquad \alpha := 1 + \frac{d}{p}, \tag{188}$$

*where $C > 0$ is a universal constant (depending only on the bump construction in $\psi_r$ through constants such as $C_\psi$ in Lemma E.5).*

*Proof of Lemma E.10.* By construction, $G^{\boldsymbol{u}}_t$ is constant on each block $B_j$, hence the gradient can change only at a boundary between consecutive blocks. More precisely, if $t$ is a boundary time with $t \in B_j$ and $t + 1 \in B_{j+1}$, then by (187),

$$\nabla G^{\boldsymbol{u}}_{t+1} - \nabla G^{\boldsymbol{u}}_t \ = \ \nabla g_{u_{j+1}} - \nabla g_{u_j}.$$

If $u_{j+1} = u_j$, the right-hand side is $\boldsymbol{0}$. If $u_{j+1} \neq u_j$, then $\nabla g_{u_{j+1}} - \nabla g_{u_j} = \pm(\nabla g_+ - \nabla g_-)$ and therefore, by Lemma E.5,

$$\left\| \nabla G^{\boldsymbol{u}}_{t+1} - \nabla G^{\boldsymbol{u}}_t \right\|_{L^p(\Theta)} \ \leq \ \left\| \nabla g_+ - \nabla g_- \right\|_{L^p(\Theta)} \ \leq \ C\,\mu\,a^\alpha.$$

There are at most $J - 1$ such boundaries, and all other times contribute zero. Hence, for $1 \leq q < \infty$,

$$\mathrm{GVar}_{p,q}(G^{\boldsymbol{u}}_{1:T}) = \left( \sum_{t=1}^{T-1} \left\| \nabla G^{\boldsymbol{u}}_{t+1} - \nabla G^{\boldsymbol{u}}_t \right\|^q_{L^p(\Theta)} \right)^{1/q} \ \leq \ \left( (J-1)\,(C\mu a^\alpha)^q \right)^{1/q} \ \leq \ C\,\mu\,a^\alpha\,J^{1/q},$$

after adjusting $C$ by a universal factor. The case $q = \infty$ is analogous and yields $\mathrm{GVar}_{p,\infty}(G^{\boldsymbol{u}}_{1:T}) \leq C\,\mu\,a^\alpha$ since the maximum increment occurs at a boundary. $\square$

We now show that if the algorithm cannot identity $\boldsymbol{u}$, it must incur regret $\gtrsim \mu a^2$. This proof is rather elementary and follows simply from the definition of the discrepancy measure and Lemma E.4. We include it for completeness.

**Lemma E.11** (Per-round separation). *Let $g_+, g_- : \Theta \to \mathbb{R}$ be the two base losses, with minimizers $\boldsymbol{\theta}_+^\star \in \arg\min_\Theta g_+$ and $\boldsymbol{\theta}_-^\star \in \arg\min_\Theta g_-$. Define the discrepancy*

$$\chi(g_+, g_-) := \inf_{\boldsymbol{\theta} \in \Theta} \max\left\{ g_+(\boldsymbol{\theta}) - g_+(\boldsymbol{\theta}_+^\star), \; g_-(\boldsymbol{\theta}) - g_-(\boldsymbol{\theta}_-^\star) \right\}.$$

*Then for every $\boldsymbol{\theta} \in \Theta$,*

$$\big(g_+(\boldsymbol{\theta}) - g_+(\boldsymbol{\theta}_+^\star)\big) + \big(g_-(\boldsymbol{\theta}) - g_-(\boldsymbol{\theta}_-^\star)\big) \geq \chi(g_+, g_-). \tag{189}$$

*In particular, under the conditions of Lemma E.4, we have*

$$\big(g_+(\boldsymbol{\theta}) - g_+(\boldsymbol{\theta}_+^\star)\big) + \big(g_-(\boldsymbol{\theta}) - g_-(\boldsymbol{\theta}_-^\star)\big) \geq \chi(g_+, g_-) \geq \frac{\mu a^2}{8} \qquad \forall \boldsymbol{\theta} \in \Theta. \tag{190}$$

*Proof of Lemma E.11.* Fix any $\boldsymbol{\theta} \in \Theta$. By the definition of $\chi(g_+, g_-)$,

$$\max\left\{ g_+(\boldsymbol{\theta}) - g_+(\boldsymbol{\theta}_+^\star), \; g_-(\boldsymbol{\theta}) - g_-(\boldsymbol{\theta}_-^\star) \right\} \geq \chi(g_+, g_-).$$

Using the elementary inequality $A + B \geq \max\{A, B\}$ for all real numbers $A, B$, we obtain (189). The bound (190) follows immediately by combining (189) with Lemma E.4, which gives $\chi(g_+, g_-) \geq \mu a^2/8$. $\qquad \square$

This immediately gives us a lower bound on the quantity $\inf_{\boldsymbol{u}, \boldsymbol{v} \in \mathcal{U}} \sum_{t=1}^T \chi(G_t^{\boldsymbol{u}}, G_t^{\boldsymbol{v}})$.

**Corollary E.1** (Blockwise accumulation of discrepancy). *Fix $1 \leq J \leq T$ and set $\Delta_T := \lfloor T/J \rfloor$. Let $B_1, \ldots, B_J$ be a partition of $[T]$ into consecutive blocks with $|B_j| \in \{\Delta_T, \Delta_T + 1\}$. For any $\boldsymbol{u}, \boldsymbol{v} \in \{\pm 1\}^J$, define the blockwise-constant sequences $G_{1:T}^{\boldsymbol{u}}$ and $G_{1:T}^{\boldsymbol{v}}$ via (187), i.e.,*

$$G_t^{\boldsymbol{u}} \equiv g_{u_j} \quad \text{and} \quad G_t^{\boldsymbol{v}} \equiv g_{v_j} \qquad \text{for all } t \in B_j, \; j \in [J].$$

*Assume the per-round separation bound holds:*

$$\chi(g_+, g_-) \geq \frac{\mu a^2}{8}. \tag{191}$$

*Then for any $\boldsymbol{u}, \boldsymbol{v} \in \{\pm 1\}^J$,*

$$\sum_{t=1}^T \chi(G_t^{\boldsymbol{u}}, G_t^{\boldsymbol{v}}) \geq \frac{\mu a^2}{8} \sum_{j=1}^J |B_j| \, \mathbf{1}\{u_j \neq v_j\} \geq \frac{\mu a^2}{8} \Delta_T \, d_H(\boldsymbol{u}, \boldsymbol{v}), \tag{192}$$

*where $d_H(\boldsymbol{u}, \boldsymbol{v}) := \sum_{j=1}^J \mathbf{1}\{u_j \neq v_j\}$ is the Hamming distance. In particular, if $\mathcal{U} \subset \{\pm 1\}^J$ satisfies $d_H(\boldsymbol{u}, \boldsymbol{v}) \geq J/16$ for all distinct $\boldsymbol{u}, \boldsymbol{v} \in \mathcal{U}$, then for all $\boldsymbol{u} \neq \boldsymbol{v}$ in $\mathcal{U}$,*

$$\sum_{t=1}^T \chi(G_t^{\boldsymbol{u}}, G_t^{\boldsymbol{v}}) \geq c_0 \, \mu a^2 \, T, \tag{193}$$

*for a universal constant $c_0 > 0$.*

*Proof of Corollary E.1.* Fix two distinct codewords $\boldsymbol{u}, \boldsymbol{v} \in \mathcal{U}$ and recall that the horizon is partitioned into disjoint blocks $B_1, \ldots, B_J$, each of size $|B_j| \in \{\Delta_T, \Delta_T + 1\}$ with $\Delta_T := \lfloor T/J \rfloor$. Fix an arbitrary block $B_j$ and (w.l.o.g.) assume $|B_j| = \Delta_T$.

If $u_j = v_j$, then $G_t^{\boldsymbol{u}} \equiv G_t^{\boldsymbol{v}}$ throughout $B_j$, hence $\chi(G_t^{\boldsymbol{u}}, G_t^{\boldsymbol{v}}) = 0$ for all $t \in B_j$. If instead $u_j \neq v_j$, then along the entire block we have $(G_t^{\boldsymbol{u}}, G_t^{\boldsymbol{v}}) \in \{(g_+, g_-), (g_-, g_+)\}$, and therefore by Lemma E.11,

$$\chi(G_t^{\boldsymbol{u}}, G_t^{\boldsymbol{v}}) = \chi(g_{u_j}, g_{v_j}) \geq \chi(g_+, g_-) \geq \frac{\mu a^2}{8} \qquad \text{for all } t \in B_j.$$

Summing over the block yields the blockwise contribution

$$\sum_{t \in B_j} \chi(G_t^{\boldsymbol{u}}, G_t^{\boldsymbol{v}}) \geq |B_j| \cdot \frac{\mu a^2}{8} \qquad \text{whenever } u_j \neq v_j.$$

Consequently, summing over all mismatched blocks gives

$$\sum_{t=1}^{T} \chi(G_t^{\boldsymbol{u}}, G_t^{\boldsymbol{v}}) = \sum_{j:\, u_j \neq v_j} \sum_{t \in B_j} \chi(G_t^{\boldsymbol{u}}, G_t^{\boldsymbol{v}}) \geq \Delta_T \cdot d_H(\boldsymbol{u}, \boldsymbol{v}) \cdot \frac{\mu a^2}{8},$$

where $d_H(\boldsymbol{u}, \boldsymbol{v}) := \big|\{j \in [J] : u_j \neq v_j\}\big|$ is the Hamming distance. Finally, by the Varshamov–Gilbert packing property of $\mathcal{U}$ (Lemma E.9), any distinct $\boldsymbol{u} \neq \boldsymbol{v}$ satisfy $d_H(\boldsymbol{u}, \boldsymbol{v}) \geq J/16$, and since $\Delta_T = \lfloor T/J \rfloor$ we obtain

$$\sum_{t=1}^{T} \chi(G_t^{\boldsymbol{u}}, G_t^{\boldsymbol{v}}) \geq \frac{J}{16} \cdot \left\lfloor \frac{T}{J} \right\rfloor \cdot \frac{\mu a^2}{8} \geq c_0 \, \mu a^2 \, T,$$

for a universal constant $c_0 > 0$. In words, two sequences that disagree on a constant fraction of blocks incur a constant per-round discrepancy on each such block, and this discrepancy accumulates over $\Theta(T)$ rounds. $\qquad \square$

It remains to upper bound $\sup_{\boldsymbol{u}, \boldsymbol{v} \in \mathcal{U}} D_{\mathrm{KL}}(P_{\boldsymbol{u}} \| P_{\boldsymbol{v}})$. Using the noisy gradient feedback model, this follows simply from a Gaussian KL chain rule.

**Lemma E.12** (KL control under Gaussian gradient noise). *Fix any admissible policy $\pi$ and two environments $\boldsymbol{u}, \boldsymbol{v} \in \mathcal{U}$. Let $P_{\boldsymbol{u}}^{\pi}$ and $P_{\boldsymbol{v}}^{\pi}$ denote the joint laws of $\boldsymbol{Z}_{1:T} := (\boldsymbol{Y}_1, \dots, \boldsymbol{Y}_T)$ generated under environments $\boldsymbol{u}$ and $\boldsymbol{v}$, respectively, when the learner follows $\pi$. Assume the (conditional) Gaussian gradient feedback model*

$$\boldsymbol{Y}_t = \nabla G_t(\boldsymbol{\theta}_{t-1}) + \boldsymbol{\varepsilon}_t, \qquad \boldsymbol{\varepsilon}_t \overset{i.i.d.}{\sim} \mathcal{N}(\boldsymbol{0}, \sigma^2 \mathbf{I}_d),$$

*where $\boldsymbol{\theta}_{t-1}$ is $\sigma(\boldsymbol{Y}_{1:t-1})$–measurable under $\pi$. Then the KL divergence satisfies the chain-rule identity*

$$D_{\mathrm{KL}}(P_{\boldsymbol{u}}^{\pi} \| P_{\boldsymbol{v}}^{\pi}) = \frac{1}{2\sigma^2} \sum_{t=1}^{T} \mathbb{E}_{\boldsymbol{u}} \Big[ \big\| \nabla G_t^{\boldsymbol{u}}(\boldsymbol{\theta}_{t-1}) - \nabla G_t^{\boldsymbol{v}}(\boldsymbol{\theta}_{t-1}) \big\|_2^2 \Big]. \tag{194}$$

*Moreover, for the block construction with base losses $\{g_+, g_-\}$, we have the uniform bound*

$$D_{\mathrm{KL}}(P_{\boldsymbol{u}}^{\pi} \| P_{\boldsymbol{v}}^{\pi}) \leq C \frac{\mu^2 a^2}{\sigma^2/(1-\beta)} T, \tag{195}$$

*for a universal constant $C > 0$.*

*Proof of Lemma E.12.* We compute $D_{\mathrm{KL}}(P_{\boldsymbol{u}}^{\pi} \| P_{\boldsymbol{v}}^{\pi})$ by conditioning on the learner's past. Let $\mathcal{F}_{t-1} := \sigma(\boldsymbol{Y}_1, \dots, \boldsymbol{Y}_{t-1})$ denote the natural filtration. By admissibility of $\pi$, the iterate $\boldsymbol{\theta}_{t-1}$ is $\mathcal{F}_{t-1}$–measurable, and under environment $\boldsymbol{w} \in \{\boldsymbol{u}, \boldsymbol{v}\}$, the conditional law of $\boldsymbol{Y}_t$ given $\mathcal{F}_{t-1}$ is Gaussian:

$$\boldsymbol{Y}_t \,\big|\, \mathcal{F}_{t-1} \sim \mathcal{N}\big(\nabla G_t^{\boldsymbol{w}}(\boldsymbol{\theta}_{t-1}),\, \sigma^2 \mathbf{I}_d\big).$$

Using the chain rule for KL divergence,

$$D_{\mathrm{KL}}(P_{\boldsymbol{u}}^{\pi} \| P_{\boldsymbol{v}}^{\pi}) = \sum_{t=1}^{T} \mathbb{E}_{\boldsymbol{u}} \big[ D_{\mathrm{KL}}\big( P_{\boldsymbol{u}}^{\pi}(\boldsymbol{Y}_t \,|\, \mathcal{F}_{t-1}) \,\big\|\, P_{\boldsymbol{v}}^{\pi}(\boldsymbol{Y}_t \,|\, \mathcal{F}_{t-1}) \big) \big]. \tag{196}$$

Since the two conditional distributions in (196) are Gaussians with the same covariance $\sigma^2 \mathbf{I}_d$, their conditional KL is

$$D_{\mathrm{KL}}\big( \mathcal{N}(\boldsymbol{m}_1, \sigma^2 \mathbf{I}_d) \,\big\|\, \mathcal{N}(\boldsymbol{m}_2, \sigma^2 \mathbf{I}_d) \big) = \frac{1}{2\sigma^2} \|\boldsymbol{m}_1 - \boldsymbol{m}_2\|_2^2.$$

Substituting $\boldsymbol{m}_1 = \nabla G_t^{\boldsymbol{u}}(\boldsymbol{\theta}_{t-1})$ and $\boldsymbol{m}_2 = \nabla G_t^{\boldsymbol{v}}(\boldsymbol{\theta}_{t-1})$ into (196) yields (194). By Lemma E.6, we have that $\left\|\nabla g_+(\boldsymbol{\theta}) - \nabla g_-(\boldsymbol{\theta})\right\|_2^2 \leq 4C\mu^2 a^2 \mathbf{1}\{\|\boldsymbol{\theta}\|_2 < r\}$. This results in

$$D_{\mathrm{KL}}(P_{\boldsymbol{u}}^\pi \,\|\, P_{\boldsymbol{v}}^\pi) \leq \frac{2C\mu^2 a^2}{\sigma^2} \sum_{t=1}^{T} \mathbb{E}_{\boldsymbol{u}}[\mathbf{1}\{\|\boldsymbol{\theta}_{t-1}\|_2 < r\}] = \frac{2C\mu^2 a^2}{\sigma^2} \mathbb{E}_{\boldsymbol{u}}[\mathrm{Occ}_T(r)].$$

Combining this with Lemma E.8, we conclude that

$$D_{\mathrm{KL}}(P_{\boldsymbol{u}}^\pi \,\|\, P_{\boldsymbol{v}}^\pi) \leq \frac{2C'\mu^2 a^2}{\sigma^2/(1-\beta)} T,$$

for a universal constant $C' > 0$. $\qquad\square$

### E.5. Obtaining a minimax lower bound in the statistical/variation-limited regime

We can get a bound on $\inf_{\widehat{\boldsymbol{u}}} \sup_{\boldsymbol{u} \in \mathcal{U}} \mathbb{P}_{\boldsymbol{u}}[\widehat{\boldsymbol{u}} \neq \boldsymbol{u}]$ via Fano's inequality:

**Corollary E.2** (Fano lower bound via KL control). *Fix any admissible policy $\pi$ and let $\mathcal{U} \subset \{\pm 1\}^J$ be a packing set with*

$$\log|\mathcal{U}| \geq \frac{J}{8}\log 2. \tag{197}$$

*Let $\boldsymbol{U}$ be uniformly distributed on $\mathcal{U}$, and let $\boldsymbol{Z}_{1:T} := (\boldsymbol{Y}_1, \ldots, \boldsymbol{Y}_T)$ denote the transcript generated under environment $\boldsymbol{U}$ when the learner follows policy $\pi$; write $P_{\boldsymbol{u}}^\pi$ for the law of $\boldsymbol{Z}_{1:T}$ under environment $\boldsymbol{u} \in \mathcal{U}$. Then for any estimator $\widehat{\boldsymbol{U}} = \widehat{\boldsymbol{U}}(\boldsymbol{Z}_{1:T})$,*

$$\mathbb{P}(\widehat{\boldsymbol{U}} \neq \boldsymbol{U}) \geq 1 - \frac{I(\boldsymbol{U}; \boldsymbol{Z}_{1:T}) + \log 2}{\log|\mathcal{U}|}. \tag{198}$$

*In particular, if the Gaussian-gradient KL bound of Lemma E.12 holds so that*

$$\sup_{\boldsymbol{u},\boldsymbol{v} \in \mathcal{U}} D_{\mathrm{KL}}(P_{\boldsymbol{u}}^\pi \,\|\, P_{\boldsymbol{v}}^\pi) \leq C \frac{\mu^2 a^2}{\sigma^2/(1-\beta)} T, \tag{199}$$

*then $I(\boldsymbol{U}; \boldsymbol{Z}_{1:T}) \leq C\mu^2 a^2 T/(\sigma^2/(1-\beta))$. Consequently, if*

$$C\frac{\mu^2 a^2}{\sigma^2/(1-\beta)} T \leq \frac{1}{16} J \log 2, \tag{200}$$

*then every estimator $\widehat{\boldsymbol{U}}$ obeys the constant error lower bound*

$$\mathbb{P}(\widehat{\boldsymbol{U}} \neq \boldsymbol{U}) \geq \frac{1}{2}. \tag{201}$$

*Proof of Corollary E.2.* Combining Lemma G.7 with (199) yields

$$I(\boldsymbol{U}; \boldsymbol{Z}_{1:T}) \leq \frac{1}{|\mathcal{U}|^2} \sum_{\boldsymbol{u},\boldsymbol{v} \in \mathcal{U}} C\frac{\mu^2 a^2}{\sigma^2/(1-\beta)} T = C\frac{\mu^2 a^2}{\sigma^2/(1-\beta)} T.$$

Substituting this into (198) and using (197) gives

$$\mathbb{P}(\widehat{\boldsymbol{U}} \neq \boldsymbol{U}) \geq 1 - \frac{C\mu^2 a^2 T/(\sigma^2/(1-\beta)) + \log 2}{(J/8)\log 2}.$$

Under the condition (200), the numerator is at most $(J/16)\log 2 + \log 2$, hence for all $J \geq 16$ the right-hand side is at least $1/2$. This yields (201). $\qquad\square$

We can finally conclude with Lemma E.1 and the upper and lower bounds we have on the KL divergence and discrepancy measure $\chi$, respectively.

**Lemma E.13** (Parameter tuning for the information-limited lower bound). *Fix* $1 \leq p \leq \infty$, $1 \leq q \leq \infty$, *and set* $\alpha := 1 + d/p$. *Consider the $J$-block construction* $\{G_{1:T}^{\boldsymbol{u}} : \boldsymbol{u} \in \mathcal{U}\}$ *with block length* $\Delta_T := \lfloor T/J \rfloor$ *and base losses* $\{g_+, g_-\}$. *Assume:*

1. **Gradient-variation bound:** *from Lemma E.10, we have* $\mathrm{GVar}_{p,q}(G_{1:T}^{\boldsymbol{u}}) \leq C \mu a^{\alpha} J^{1/q}$, $\forall \boldsymbol{u} \in \mathcal{U}$;

2. **Gaussian-gradient KL bound:** *from Lemma E.12, we have* $\sup_{\boldsymbol{u},\boldsymbol{v} \in \mathcal{U}} D_{\mathrm{KL}}(P_{\boldsymbol{u}}^{\pi} \,\|\, P_{\boldsymbol{v}}^{\pi}) \leq C \mu^2 a^2 T/(\sigma^2/(1-\beta))$, $\forall \pi$;

3. **Packing size lower bound:** *from Lemma E.9, we have the packing size lower bound* $\log |\mathcal{U}| \geq c J$ *for a universal constant* $c > 0$ *(e.g., from Varshamov–Gilbert packing).*

*Let* $\mathbb{V}_T > 0$ *be the variation budget. Choose*

$$a := \left( \frac{\mathbb{V}_T}{C \mu J^{1/q}} \right)^{1/\alpha}. \tag{202}$$

*Then* $\mathrm{GVar}_{p,q}(G_{1:T}^{\boldsymbol{u}}) \leq \mathbb{V}_T$ *for all* $\boldsymbol{u} \in \mathcal{U}$. *Moreover, if $J$ additionally satisfies*

$$C \frac{\mu^2 a^2}{\sigma^2/(1-\beta)} T \leq \frac{c}{2} J, \tag{203}$$

*then Fano's method yields a constant testing error (hence a constant separation in regret) for the family* $\{G_{1:T}^{\boldsymbol{u}} : \boldsymbol{u} \in \mathcal{U}\}$ *under any policy. In particular, substituting (202) into (203) shows it suffices to take*

$$J \gtrsim \left( \frac{\mu^{2-2/\alpha}}{\sigma^2/(1-\beta)} \mathbb{V}_T^{2/\alpha} T \right)^{\alpha q/(\alpha q+2)}, \tag{204}$$

*and for this choice one obtains the scaling*

$$\inf_{\pi \in \Pi_{\beta}} \sup_{G: \, \mathrm{GVar}_{p,q}(G) \leq \mathbb{V}_T} \mathcal{R}_T^{\pi}(G) \gtrsim (1-\beta)^{-2/(\alpha q+2)} \sigma^{4/(\alpha q+2)} \mu^{(\alpha q-2q-2)/(\alpha q+2)} \mathbb{V}_T^{2q/(\alpha q+2)} T^{\alpha q/(\alpha q+2)}, \tag{205}$$

*where* $\alpha = 1 + d/p$.

*Proof of Lemma E.13.* First note that by Corollary E.2, we have that $\mathbb{P}(\widehat{U} \neq U) \geq \frac{1}{2}$. Subsequently invoking Lemma E.1, we conclude that there does not exist an admissible policy $\pi \in \Pi_{\beta}$ such that $\sup_{g \in \mathcal{G}_{p,q}(\mathbb{V}_T)} \mathcal{R}_T^{\pi, \phi^G}(g) \leq \frac{1}{9} \inf_{\substack{\boldsymbol{u},\boldsymbol{v} \in \mathcal{U} \\ \boldsymbol{u} \neq \boldsymbol{v}}} \sum_{t=1}^{T} \chi(G_t^{\boldsymbol{u}}, G_t^{\boldsymbol{v}})$. This gives us a lower bound for the minimax regret. Now we must just enforce the gradient variational budget and the information constraint. To enforce the gradient variational budget, we have by Lemma E.10 that the condition $\mathrm{GVar}_{p,q}(G_{1:T}^{\boldsymbol{u}}) \leq \mathbb{V}_T$ for all $\boldsymbol{u} \in \mathcal{U}$ is ensured whenever $C \mu a^{\alpha} J^{1/q} \leq \mathbb{V}_T$, which is exactly achieved by the choice (202).

To enforce the information (Fano/KL) constraint,, we have by Lemma E.12, for any policy $\pi$ and any $\boldsymbol{u}, \boldsymbol{v} \in \mathcal{U}$,

$$D_{\mathrm{KL}}(P_{\boldsymbol{u}}^{\pi} \,\|\, P_{\boldsymbol{v}}^{\pi}) \leq C \frac{\mu^2 a^2}{\sigma^2/(1-\beta)} T.$$

Combining this uniform pairwise bound with the standard mutual-information upper bound (Lemma G.7), we get

$$I(\boldsymbol{U}; \boldsymbol{Z}_{1:T}) \leq C \frac{\mu^2 a^2}{\sigma^2/(1-\beta)} T.$$

Since $\log |\mathcal{U}| \geq cJ$, imposing (203) makes $I(\boldsymbol{U}; \boldsymbol{Z}_{1:T})$ a small constant fraction of $\log |\mathcal{U}|$. Fano's inequality then yields a constant lower bound on the minimax probability of misidentifying $\boldsymbol{U}$, uniformly over all estimators $\widehat{\boldsymbol{U}}(\boldsymbol{Z}_{1:T})$.

Substituting (202) into (203) gives

$$C \frac{\mu^2 T}{\sigma^2/(1-\beta)} \left( \frac{\mathbb{V}_T}{C \mu J^{1/q}} \right)^{2/\alpha} \lesssim J,$$

or equivalently,

$$\frac{\mu^{2-2/\alpha}}{\sigma^2/(1-\beta)}\,\mathbb{V}_T^{2/\alpha}\,T \;\lesssim\; J^{1+\frac{2}{\alpha q}}.$$

Rearranging yields (204). Finally, the quantity that drives the regret separation in the information-limited regime is $\mu a^2 T$. Using (202), plugging $J \asymp J_\star$ from (204), and simplifying exponents yields we get

$$\inf_{\pi\in\Pi_\beta}\ \sup_{G:\,\mathrm{GVar}_{p,q}(G)\leq\mathbb{V}_T} \mathcal{R}_T^\pi(G) \;\gtrsim\; (1-\beta)^{-2/(\alpha q+2)}\sigma^{4/(\alpha q+2)}\mu^{(\alpha q-2q-2)/(\alpha q+2)}\mathbb{V}_T^{2q/(\alpha q+2)}T^{\alpha q/(\alpha q+2)}.$$

$\square$

## E.6. Obtaining a minimax lower bound in the inertia-limited regime

Our information-theoretic construction yields a lower bound that is driven by *statistical indistinguishability* of environments and therefore captures the dependence on the noise level $\sigma$ and the variation budget $\mathbb{V}_T$. However, it does not by itself explain the empirically dominant failure mode of momentum under drift: *inertia*. To isolate this mechanism, we analyze (SGDM) (Heavy-Ball) on the simplest strongly convex and smooth objective, $\phi_u(z) = \frac{\mu}{2}(z - ua)^2$, where the minimizer jumps between $\pm a$. We prove the momentum-specific lower bound by explicitly analyzing (SGDM) on a single block switch and showing it takes $\tau_\beta := \Omega(\kappa/(1-\beta))$ steps to reduce the tracking error by a constant factor under a step size restriction. Crucially, this part does not rely on hiding information; the learner may instantly know the new function. The lower bound comes from the algorithmic constraint of (SGDM) under stability. We will focus on the Polyak Heavy-Ball method of momentum ($\beta_1 = 0, \beta_2 = \beta$). However a similar analysis can be carried out for Nesterov and yield a analogous result.

**Proposition E.1** ((SGDM) on a 1D quadratic). *Fix $u \in \{\pm1\}$ and consider the one-dimensional quadratic $\phi_u(z) := \frac{\mu}{2}(z - ua)^2$ for $z \in \mathbb{R}$. Let Heavy-Ball (SGDM) with step size $\gamma > 0$ and momentum $\beta \in [0, 1)$ evolve as*

$$z_{t+1} = z_t + \beta(z_t - z_{t-1}) - \gamma\big(\mu(z_t - ua) + \eta_t\big), \qquad \eta_t \sim \mathcal{N}(0, \sigma^2) \ i.i.d. \tag{206}$$

*Writing $e_t := z_t - ua$, the mean error satisfies*

$$\begin{pmatrix} \mathbb{E}[e_{t+1}] \\ \mathbb{E}[e_t] \end{pmatrix} = \underbrace{\begin{pmatrix} 1 + \beta - \gamma\mu & -\beta \\ 1 & 0 \end{pmatrix}}_{\triangleq\,A} \begin{pmatrix} \mathbb{E}[e_t] \\ \mathbb{E}[e_{t-1}] \end{pmatrix}, \tag{207}$$

*hence $\left\|\begin{pmatrix}\mathbb{E}[e_{t+1}]\\\mathbb{E}[e_t]\end{pmatrix}\right\|_2 \leq \|A\|_2^t \left\|\begin{pmatrix}\mathbb{E}[e_1]\\\mathbb{E}[e_0]\end{pmatrix}\right\|_2$. Let $\lambda_{\max}$ denote the eigenvalue of $A$ with maximal modulus. If*

$$0 < \gamma\mu \leq \min\left\{(1 - \sqrt{\beta})^2, \frac{1-\beta}{4}\right\}, \tag{208}$$

*then $A$ has two real eigenvalues in $(0, 1)$ and there exists a universal constant $c > 0$ such that*

$$|\lambda_{\max}| \geq 1 - c\frac{\gamma\mu}{1-\beta}. \tag{209}$$

*Consequently, for any initialization with $\mathbb{E}[e_0] = \mathbb{E}[e_{-1}] = a$, there exists a universal constant $C \geq 1$ such that for all $t \geq 0$,*

$$|\mathbb{E}[e_t]| \geq C^{-1}|\lambda_{\max}|^t a \geq C^{-1}\left(1 - c\frac{\gamma\mu}{1-\beta}\right)^t a. \tag{210}$$

*Defining the response time $\tau_\beta := \min\{t \geq 0 : (1 - c\gamma\mu/(1-\beta))^t \leq 1/2\}$, we have $\tau_\beta \asymp (1-\beta)/(\gamma\mu)$. Finally, Jensen's inequality yields*

$$\mathbb{E}\big[\phi_u(z_t) - \phi_u(ua)\big] = \frac{\mu}{2}\mathbb{E}[e_t^2] \geq \frac{\mu}{2}\big(\mathbb{E}[e_t]\big)^2 \gtrsim \mu a^2\left(1 - c\frac{\gamma\mu}{1-\beta}\right)^{2t}. \tag{211}$$

*Proof of Proposition E.1.* Subtracting $ua$ from (206) gives

$$e_{t+1} = (1 + \beta - \gamma\mu)e_t - \beta e_{t-1} - \gamma\eta_t. \tag{212}$$

Taking expectations yields the homogeneous recursion $\mathbb{E}[e_{t+1}] = (1 + \beta - \gamma\mu)\mathbb{E}[e_t] - \beta\mathbb{E}[e_{t-1}]$, equivalently the linear system (207) with matrix $A$. The eigenvalues of $A$ are the roots of $\lambda^2 - (1 + \beta - \gamma\mu)\lambda + \beta = 0$. Under (208), the discriminant is nonnegative so $\lambda_{\pm} \in \mathbb{R} \cap (0, 1)$. Let $\lambda_{\max} := \max\{\lambda_+, \lambda_-\}$. Writing $\delta := 1 - \beta$ and $\varepsilon := \gamma\mu$, we have

$$\lambda_{\max} = 1 - \frac{\delta + \varepsilon}{2} + \frac{1}{2}\sqrt{(\delta + \varepsilon)^2 - 4\varepsilon}.$$

Setting $a := \delta + \varepsilon$ and $b := 4\varepsilon$, the discriminant condition $0 \le b \le a^2$ gives $\sqrt{a^2 - b} \ge a - b/a$, hence

$$\lambda_{\max} \ge 1 - \frac{a}{2} + \frac{1}{2}\left(a - \frac{b}{a}\right) = 1 - \frac{b}{2a} = 1 - \frac{2\varepsilon}{\delta + \varepsilon} \ge 1 - \frac{2\varepsilon}{\delta} = 1 - 2\frac{\gamma\mu}{1 - \beta},$$

proving (209) with $c = 2$. Since $A$ is diagonalizable with eigenvalues in $(0, 1)$, the solution of (207) is a linear combination of $\lambda_+^t$ and $\lambda_-^t$. For the symmetric initialization $\mathbb{E}[e_0] = \mathbb{E}[e_{-1}] = a$, the dominant mode aligns with $\lambda_{\max}$, so there exists a universal constant $C \ge 1$ (depending on the eigenbasis conditioning in regime (208)) such that (210) holds. The response time scaling $\tau_\beta \asymp (1 - \beta)/(\gamma\mu)$ follows from $\log(1 - x) \asymp -x$ for $x \in (0, 1/2)$, applied with $x = c\gamma\mu/(1 - \beta)$. Finally, since $\phi_u(z_t) - \phi_u(ua) = \frac{\mu}{2}e_t^2$ and $\mathbb{E}[e_t^2] \ge (\mathbb{E}[e_t])^2$, plugging (210) yields (211). $\qquad\square$

We now use this as the block length $\Delta_T \asymp \tau_\beta$. Since in our $J$-block construction, the minimizer is constant within a block and flips between blocks. If the blocks are much longer than $\tau_\beta$, (SGDM) has time to settle near the new minimizer after each switch. Then the regret per block is mostly transient and doesn't accumulate strongly. On the other hand, if the block size is much shorter than $\tau_\beta$, (SGDM) will never be able to catch up. Taking $\Delta_T \asymp \tau_\beta$ will give us a constant per-round suboptimality throughout the block (up to constants), hence resulting in $\Omega(\mu a^2 \Delta_T)$ regret contribution per block. We formalize this in the following result:

**Theorem E.1** (From response time to regret under block switching). *Consider the $J$-block construction of nonstationary losses $G_{1:T}^{\boldsymbol{u}}$ over $[T]$ with blocks $B_1, \ldots, B_J$ of lengths $|B_j| \in \{\Delta_T, \Delta_T + 1\}$, $\Delta_T = \lfloor T/J \rfloor$, and $G_t^{\boldsymbol{u}} \equiv g_{u_j}$ for $t \in B_j$, where $\boldsymbol{u} \in \{\pm 1\}^J$. Assume the base losses are the translated quadratics $\phi_\pm$ above (embedded along $\boldsymbol{e}_1$ if $d > 1$), so that within each block the unique minimizer is $x_{u_j}^\star = u_j a$. Run Heavy-Ball (SGDM) with parameters $(\gamma, \beta)$ satisfying the stability cap*

$$\gamma \le c_0 \frac{(1 - \beta)^2}{L}, \tag{213}$$

*for a universal constant $c_0 > 0$. Let $\tau_\beta$ be the response time from Proposition E.1, so in particular*

$$\tau_\beta \gtrsim \frac{L}{\mu(1 - \beta)}. \tag{214}$$

*Choose the block length on the order of the response time,*

$$\Delta_T \asymp \tau_\beta, \qquad \text{equivalently} \qquad J \asymp \frac{T}{\tau_\beta}. \tag{215}$$

*Then there exists a universal constant $c > 0$ such that for any $\boldsymbol{u} \in \{\pm 1\}^J$ with a sign flip at each block boundary (e.g., $u_{j+1} = -u_j$), the expected dynamic regret of Heavy-Ball (SGDM) satisfies*

$$\mathcal{R}_T^{\pi_{\mathrm{HB}}}(G^{\boldsymbol{u}}) \ge c\,\mu a^2\,T. \tag{216}$$

*Moreover, if the block family is tuned to satisfy the gradient-variation budget $\mathrm{GVar}_{p,q}(G_{1:T}^{\boldsymbol{u}}) \le \mathbb{V}_T$ via Lemma E.10, i.e.*

$$a \asymp \left(\frac{\mathbb{V}_T}{\mu\,J^{1/q}}\right)^{1/\alpha}, \qquad \alpha := 1 + \frac{d}{p}, \tag{217}$$

*then substituting $J \asymp T/\tau_\beta$ into (216) yields*

$$\mathcal{R}_T^{\pi_{\mathrm{HB}}}(G^{\boldsymbol{u}}) \gtrsim \mu^{1-2/\alpha}\,\mathbb{V}_T^{2/\alpha}\,\tau_\beta^{2/(\alpha q)}\,T^{1-(2/\alpha q)}. \tag{218}$$

*Finally, using (214) gives the explicit inertia-dependent lower bound*

$$\mathcal{R}_T^{\pi_{\mathrm{HB}}}(G^{\boldsymbol{u}}) \gtrsim \mu^{1-2/\alpha}\,\mathbb{V}_T^{2/\alpha}\left(\frac{L}{\mu(1 - \beta)}\right)^{2/(\alpha q)}\,T^{1-(2/\alpha q)}. \tag{219}$$

*Proof of Theorem E.1.* The proof has three steps: (i) a single sign flip creates an $\Omega(a)$ initialization error (in the extended state) relative to the *new* minimizer, (ii) over a time window of length $\Theta(\tau_\beta)$ this error cannot contract by more than a constant factor, and (iii) strong convexity converts this persistent distance into $\Omega(\mu a^2)$ per-round regret, which then accumulates across blocks.

**Step 1: A flip induces an $\Omega(a)$ error at the start of the next block.** Consider a boundary between two consecutive blocks $B_j$ and $B_{j+1}$ at which the sign flips, $u_{j+1} = -u_j$. Let

$$x_j^\star := u_j a, \qquad x_{j+1}^\star := u_{j+1} a = -u_j a, \qquad \text{so that} \qquad |x_{j+1}^\star - x_j^\star| = 2a.$$

Let the first time index in block $B_{j+1}$ be $t_0$ and define the post-switch error variables

$$e_t := x_t - x_{j+1}^\star, \qquad t \in B_{j+1},$$

(and analogously for $e_{t_0-1} = x_{t_0-1} - x_{j+1}^\star$). By the reverse triangle inequality,

$$|e_{t_0-1}| = |x_{t_0-1} - x_{j+1}^\star| \geq |x_j^\star - x_{j+1}^\star| - |x_{t_0-1} - x_j^\star| = 2a - |x_{t_0-1} - x_j^\star|. \tag{220}$$

Thus, whenever the iterate has achieved even a moderate accuracy on the previous block, say $|x_{t_0-1} - x_j^\star| \leq a$, we obtain $|e_{t_0-1}| \geq a$. In particular, for the alternating-sign choice in the theorem (a flip at every boundary), either the algorithm fails to track the previous minimizer by time $t_0 - 1$—which already incurs regret on block $B_j$—or else it necessarily starts block $B_{j+1}$ with initial error at least $a$ relative to the new minimizer. This is what allows us to lower bound regret block-by-block. To make this quantitative in a way compatible with the Heavy-Ball state, define the extended state

$$s_t := \begin{bmatrix} e_t \\ e_{t-1} \end{bmatrix}.$$

There exists a universal constant $c_{\text{init}} > 0$ such that at the beginning of each block with a flip,

$$\|s_{t_0-1}\|_2 \geq c_{\text{init}} a. \tag{221}$$

We will use (221) as the initial condition for the lag argument in Step 2.

**Step 2: Over $\Theta(\tau_\beta)$ steps the *mean* error cannot shrink by more than a constant factor.** On block $B_{j+1}$ the loss is the fixed quadratic $g_{u_{j+1}}(x) = \frac{\mu}{2}(x - x_{j+1}^\star)^2$, so with $e_t := x_t - x_{j+1}^\star$ the Heavy-Ball recursion gives

$$e_{t+1} = (1 + \beta - \gamma\mu)e_t - \beta e_{t-1} - \gamma\eta_t, \qquad t \in B_{j+1}.$$

Let $\bar{e}_t := \mathbb{E}[e_t]$ and $\bar{s}_t := \begin{bmatrix} \bar{e}_t \\ \bar{e}_{t-1} \end{bmatrix}$. Taking expectations yields the homogeneous linear system

$$\bar{s}_{t+1} = A\,\bar{s}_t, \qquad A = \begin{bmatrix} 1 + \beta - \gamma\mu & -\beta \\ 1 & 0 \end{bmatrix}. \tag{222}$$

Fix a flipped boundary between $B_j$ and $B_{j+1}$, and let $t_0$ be the first index of $B_{j+1}$. By Step 1, either the previous block already contributes $\gtrsim \mu a^2 \Delta_T$ regret (and we are done for that block), or else the algorithm is at least moderately aligned with the previous minimizer in the sense that $|\mathbb{E}[x_{t_0-1}] - x_j^\star| \leq a$ and $|\mathbb{E}[x_{t_0-2}] - x_j^\star| \leq a$. In this latter case, using $x_{j+1}^\star = -x_j^\star$ and the reverse triangle inequality for scalars gives

$$|\bar{e}_{t_0-1}| = |\mathbb{E}[x_{t_0-1}] - x_{j+1}^\star| \geq |x_j^\star - x_{j+1}^\star| - |\mathbb{E}[x_{t_0-1}] - x_j^\star| \geq 2a - a = a,$$

and similarly $|\bar{e}_{t_0-2}| \geq a$. Thus $\bar{s}_{t_0-1}$ satisfies $|\bar{e}_{t_0-1}|, |\bar{e}_{t_0-2}| \geq a$.

Now apply Proposition E.1 to the shifted mean trajectory on this block. Using (210) and the definition of $\tau_\beta$, we obtain that there exists a universal constant $c_{\text{lag}} \in (0, 1)$ such that for all $k = 0, 1, \ldots, \lfloor \tau_\beta \rfloor$,

$$|\bar{e}_{t_0+k}| \geq c_{\text{lag}} a. \tag{223}$$

Consequently, Jensen implies that on the same window

$$\mathbb{E}[e_{t_0+k}^2] \geq (\mathbb{E}[e_{t_0+k}])^2 \geq c_{\text{lag}}^2 a^2. \tag{224}$$

**Step 3: Strong convexity turns persistent distance into regret and accumulates across blocks.** For the quadratic loss on block $B_{j+1}$,

$$g_{u_{j+1}}(x_t) - g_{u_{j+1}}(x_{j+1}^\star) = \frac{\mu}{2}e_t^2 \geq \frac{\mu}{2}\Big(|e_t|\Big)^2. \tag{225}$$

By (224), for each $k \leq \lfloor c_{\mathrm{lag}}\tau_\beta \rfloor$ we have $|e_{t_0+k}| \gtrsim a$, hence the per-round regret on that window is

$$g_{t_0+k}(x_{t_0+k}) - g_{t_0+k}(x_{t_0+k}^\star) = g_{u_{j+1}}(x_{t_0+k}) - g_{u_{j+1}}(x_{j+1}^\star) \gtrsim \mu a^2. \tag{226}$$

Summing (226) over $k = 0, \ldots, \lfloor c_{\mathrm{lag}}\tau_\beta \rfloor$ gives an expected regret contribution of order $\mu a^2 \tau_\beta$ per flipped block:

$$\sum_{t \in B_{j+1}} \big(g_t(x_t) - g_t(x_t^\star)\big) \geq \sum_{k=0}^{\lfloor c_{\mathrm{lag}}\tau_\beta \rfloor} \big(g_{t_0+k}(x_{t_0+k}) - g_{t_0+k}(x_{t_0+k}^\star)\big) \gtrsim \mu a^2 \tau_\beta. \tag{227}$$

With the block-length choice $\Delta_T \asymp \tau_\beta$ in (215), this becomes $\gtrsim \mu a^2 \Delta_T$ per block. Since the construction flips at every boundary, a constant fraction of the $J$ blocks contribute this amount, and therefore

$$\mathcal{R}_T^{\pi_{\mathrm{HB}}}(G^{\boldsymbol{u}}) = \sum_{t=1}^{T} \big(g_t(x_t) - g_t(x_t^\star)\big) \gtrsim J \cdot \mu a^2 \Delta_T \asymp \mu a^2 T,$$

which proves (216).

Under the $J$-block construction, Lemma E.10 gives $\mathrm{GVar}_{p,q}(G_{1:T}^{\boldsymbol{u}}) \lesssim \mu a^\alpha J^{1/q}$. Enforcing $\mathrm{GVar}_{p,q}(G_{1:T}^{\boldsymbol{u}}) \leq \mathbb{V}_T$ yields (217). Plugging (217) into (216) gives

$$\mathcal{R}_T^{\pi_{\mathrm{HB}}}(G^{\boldsymbol{u}}) \gtrsim \mu T \Big(\frac{\mathbb{V}_T}{\mu J^{1/q}}\Big)^{2/\alpha} = \mu^{1-2/\alpha} \mathbb{V}_T^{2/\alpha} T J^{-2/(\alpha q)}.$$

Using $J \asymp T/\tau_\beta$ from (215) yields (218). Finally, the stability cap (213) implies $\gamma \lesssim (1-\beta)^2/L$, and combining this with $\tau_\beta \asymp (1-\beta)/(\gamma\mu)$ from Proposition E.1 yields $\tau_\beta \gtrsim L/(\mu(1-\beta))$, i.e. (214). Substituting this lower bound into (218) gives (219). $\qquad\square$

Since both the statistical and inertia constructions produce $\mu$-strongly convex, $\mu$-smooth functions satisfying $\mathrm{GVar}_{p,q} \leq \mathbb{V}_T$, the restricted minimax regret over $\Pi_\beta$ is at least the maximum of the two lower bounds. This completes the proof of the result.

# F. Additional experimental details and results

We provide complete details for reproducibility. All experiments compare SGD, Heavy-Ball (HB), and Nesterov (NAG) under identical drift conditions, with step sizes selected to be fixed as described below.

## F.1. General setup

**Step size constraints and drift model.** Our results (Theorem 3.3 and Theorem 3.6) assumes the sufficient stability condition $\gamma \leq \mu(1-\beta)^2/(4L^2)$ for HB/NAG to obtain clean finite-time bounds. In the main experimental setup, we *do not* enforce this conservative cap: we evaluate all methods at a standard set of base step sizes $\gamma$ reported in the tables (and used uniformly across methods within each task/regime) to test whether the drift-induced inertia phenomenon persists beyond the theoretically analyzed regime. For reference, when we choose step sizes within the stability region, we observe the same qualitative behavior as (Yuan et al., 2016). Across all tasks, we evolve the population minimizer $\boldsymbol{\theta}_t^\star \in \mathbb{R}^d$ via a normalized random walk

$$\boldsymbol{\theta}_{t+1}^\star = \boldsymbol{\theta}_t^\star + \delta_{\mathrm{rw}} \cdot \frac{\mathbf{u}_t}{\|\mathbf{u}_t\|_2}, \qquad \mathbf{u}_t \sim \mathcal{N}(0, I_d),$$

so that $\|\boldsymbol{\theta}_{t+1}^\star - \boldsymbol{\theta}_t^\star\|_2 = \delta_{\mathrm{rw}}$ for each step. We set $\delta_{\mathrm{rw}} = 0.01$ unless otherwise stated, and additionally report robustness to heavier-tailed Student-$t$ drift (Table 3).

**Initialization and training.** All optimizers are initialized at $\boldsymbol{\theta}_0 = \boldsymbol{\theta}_0^\star$ with velocity buffer $\mathbf{v}_0 = \mathbf{0}$, yielding zero initial tracking error to isolate the effect of drift. All experiments use mini-batch stochastic gradients with batch size $B = 256$. The reported $\sigma^2$ values refer to label-noise variance for the teacher–student MLP). We use fixed base step sizes $\gamma$ for each task/regime (listed in the captions of Table 3 and Table 4) rather than per-method tuning. For the teacher–student MLP, teacher and student weights are initialized i.i.d. from $\mathcal{N}(0, 0.04)$, and the student is warm-started to match the teacher at $t = 0$.

**Evaluation.** We report the squared tracking error $e_t = \|\boldsymbol{\theta}_t - \boldsymbol{\theta}_t^\star\|^2$ averaged over $N = 20$ runs, with final performance measured at $T = 5000$ iterations. Results were consistent across dimensions $d \in \{50, 100, 200\}$. The shaded regions in figures denote $\pm 1$ standard deviation.

## F.2. Task-specific details

**Strongly convex quadratic.** We consider the time-varying objective

$$f_t(\boldsymbol{\theta}) = \frac{\mu}{2} \|\boldsymbol{\theta} - \boldsymbol{\theta}_t^\star\|^2, \tag{228}$$

with $\mu = 1$, so that $f_t$ is 1-strongly convex with Lipschitz constant $L = \mu = 1$ (condition number $\kappa = 1$). The gradient is $\nabla f_t(\boldsymbol{\theta}) = \mu(\boldsymbol{\theta} - \boldsymbol{\theta}_t^\star)$. At each iteration, the algorithm receives a stochastic gradient

$$G_t(\boldsymbol{\theta}) = \nabla f_t(\boldsymbol{\theta}) + \boldsymbol{\xi}_t, \qquad \boldsymbol{\xi}_t \sim \mathcal{N}(0, \sigma^2 I_d), \tag{229}$$

with noise variance $\sigma^2 \in \{0.1, 0.5, 0.8\}$. This task isolates the effect of gradient noise and momentum under drift in the simplest strongly convex setting.

**Linear regression.** We consider streaming least-squares with drifting ground truth:

$$y_t = \langle \mathbf{x}_t, \boldsymbol{\theta}_t^\star \rangle + \epsilon_t, \qquad \epsilon_t \sim \mathcal{N}(0, \sigma^2), \tag{230}$$

where covariates are drawn as $\mathbf{x}_t = \Sigma^{1/2} \mathbf{z}_t$ with $\mathbf{z}_t \sim \mathcal{N}(0, I_d)$ and $d = 50$. The covariance $\Sigma$ is constructed with eigenvalues log-spaced between 1 and $\kappa$, yielding strong convexity parameter $\mu = 1$ and smoothness $L = \kappa$. We test two conditioning regimes: well-conditioned ($\kappa = 10$) and ill-conditioned ($\kappa = 1000$).

**Logistic regression.** We consider streaming binary classification with a drifting decision boundary:

$$y_t \sim \text{Bernoulli}\big(\varsigma(\langle \mathbf{x}_t, \boldsymbol{\theta}_t^\star \rangle)\big), \tag{231}$$

where $\varsigma(z) = 1/(1 + e^{-z})$ is the sigmoid function. Covariates are generated identically to linear regression, with the same conditioning regimes ($\kappa \in \{10, 1000\}$) and dimension $d = 50$. A small $\ell_2$ regularization term ($\lambda = 10^{-3}$) is added to ensure strong convexity. We report tracking error $\|\boldsymbol{\theta}_t - \boldsymbol{\theta}_t^\star\|$, training loss, and classification accuracy evaluated on held-out samples.

**Teacher–student MLP.** We consider a two-layer ReLU network of the form

$$f_{\boldsymbol{\theta}}(\mathbf{x}) = \mathbf{W}_2 \operatorname{ReLU}(\mathbf{W}_1 \mathbf{x} + \mathbf{b}_1) + b_2, \tag{232}$$

with architecture and training details summarized in Table 2. The teacher parameters $\boldsymbol{\theta}_t^\star = (\mathbf{W}_{1,t}^\star, \mathbf{b}_{1,t}^\star, \mathbf{W}_{2,t}^\star, b_{2,t}^\star)$ drift according to the same normalized random walk described above. Minimizers are generated as $y_t = f_{\boldsymbol{\theta}_t^\star}(\mathbf{x}_t) + \epsilon_t$ with $\epsilon_t \sim \mathcal{N}(0, 1)$.

*Table 2.* Teacher–student MLP architecture and training configuration.

| Property | Value |
|---|---|
| *Architecture* | |
| Input dimension ($d_{\text{in}}$) | 100 |
| Hidden dimension ($h$) | 128 |
| Output dimension | 1 (scalar) |
| Activation | ReLU |
| Total parameters ($d_\theta$) | 13,057 |
| *Initialization* | |
| Weight distribution | $\mathcal{N}(0, 0.04)$ |
| Bias initialization | $\mathcal{N}(0, 0.04)$ |
| Student warm-start | Matched to teacher at $t = 0$ |
| *Training* | |
| Batch size ($B$) | 256 |
| Label noise ($\sigma$) | 1.0 |
| Horizon ($T$) | 5000 |
| Drift rate ($\delta_{\text{rw}}$) | 0.02 |
| *Evaluation* | |
| Validation set size ($|\mathcal{V}|$) | 2048 |
| Tracking metric | Prediction-space MSE |
| Seeds | 20 |

Since parameter-space distances are not identifiable under the permutation and scaling symmetries of ReLU networks, we evaluate tracking in *prediction space*:

$$e_t^{\text{pred}} = \frac{1}{|\mathcal{V}|} \sum_{\mathbf{x} \in \mathcal{V}} \|f_{\boldsymbol{\theta}_t}(\mathbf{x}) - f_{\boldsymbol{\theta}_t^\star}(\mathbf{x})\|^2, \tag{233}$$

where $\mathcal{V}$ is a fixed validation set held constant across all time steps. Conditioning is controlled via the input covariance $\Sigma$ as in the regression tasks.

## F.3. Tables and figures

*Table 3.* **Mean tracking error after 5000 iterations on a drifting strongly convex quadratic ($d = 100$) with Student-t distribution drift.** Boldface indicates the best (lowest) method within each $(\beta, \sigma^2, \gamma)$ setting.

| $\beta$ | $\sigma^2$ | $\gamma = 0.01$ | | | $\gamma = 0.05$ | | | $\gamma = 0.10$ | | |
|---|---|---|---|---|---|---|---|---|---|---|
| | | SGD | HB | NAG | SGD | HB | NAG | SGD | HB | NAG |
| 0.50 | 0.1 | **0.056** | 0.103 | 0.100 | **0.256** | 0.509 | 0.459 | **0.573** | 1.079 | 0.942 |
| 0.50 | 0.5 | **0.245** | 0.498 | 0.509 | **1.286** | 2.597 | 2.491 | **2.464** | 5.206 | 4.750 |
| 0.50 | 0.8 | **0.383** | 0.792 | 0.760 | **2.043** | 4.151 | 3.611 | **4.226** | 8.010 | 7.785 |
| 0.90 | 0.1 | **0.054** | 0.501 | 0.471 | **0.257** | 2.397 | 1.831 | **0.532** | 5.015 | 2.689 |
| 0.90 | 0.5 | **0.241** | 2.585 | 2.282 | **1.232** | 13.022 | 8.099 | **2.607** | 24.212 | 13.258 |
| 0.90 | 0.8 | **0.400** | 3.947 | 3.637 | **1.999** | 21.154 | 14.204 | **4.236** | 40.316 | 21.831 |
| 0.95 | 0.1 | **0.051** | 1.015 | 0.830 | **0.255** | 5.350 | 2.514 | **0.507** | 9.995 | 3.378 |
| 0.95 | 0.5 | **0.237** | 4.930 | 4.062 | **1.256** | 23.270 | 12.708 | **2.703** | 50.767 | 16.846 |
| 0.95 | 0.8 | **0.404** | 7.718 | 6.624 | **2.101** | 40.283 | 20.850 | **3.947** | 79.354 | 28.313 |
| 0.99 | 0.1 | **0.052** | 4.767 | 2.433 | **0.268** | 26.093 | 4.490 | **0.520** | 51.724 | 4.656 |
| 0.99 | 0.5 | **0.247** | 26.309 | 12.036 | **1.226** | 131.242 | 21.845 | **2.673** | 245.091 | 23.187 |
| 0.99 | 0.8 | **0.391** | 40.600 | 21.424 | **1.968** | 210.841 | 33.517 | **4.143** | 405.148 | 38.306 |

*Table 4.* **Benchmark summary across linear regression, logistic regression, and teacher–student MLP under streaming drift.** Mean performance after 5000 iterations for SGD, Heavy-Ball (HB), and Nesterov (NAG) under two conditioning regimes (well-conditioned: $\kappa = 10$; ill-conditioned: $\kappa = 1000$). Arrows indicate the preferred direction (lower is better for loss/tracking/val MSE; higher is better for accuracy). For the teacher–student MLP, tracking is computed in *prediction space*. We report the training settings ($\sigma^2, \beta, \gamma$) used in each regime (with $\gamma$ the base step size). Overall, SGD is the most robust across conditioning, while HB/NAG deteriorate sharply in ill-conditioned regimes which is consistent with inertia amplifying lag under nonstationary drift.

### Linear Regression
**Well-conditioned ($\kappa = 10$)**
*Settings:* $\sigma^2 = 0.5$, $\beta = 0.9$, $\gamma = 0.1$

| Method | Loss ↓ | Track ↓ |
|---|---|---|
| SGD | **1.192** | **0.174** |
| HB | 1.449 | 0.493 |
| NAG | 1.447 | 0.494 |

**Ill-conditioned ($\kappa = 1000$)**
*Settings:* $\sigma^2 = 0.5$, $\beta = 0.9$, $\gamma = 0.001$

| Method | Loss ↓ | Track ↓ |
|---|---|---|
| SGD | **1.262** | **0.168** |
| HB | 56.910 | 0.898 |
| NAG | 56.902 | 0.898 |

### Logistic Regression
**Well-conditioned ($\kappa = 10$)**
*Settings:* $\sigma^2 = 0.5$, $\beta = 0.9$, $\gamma = 0.5$

| Method | Loss ↓ | Track ↓ | Acc ↑ |
|---|---|---|---|
| SGD | **0.509** | **0.154** | **0.741** |
| HB | 0.609 | 0.763 | 0.693 |
| NAG | 0.609 | 0.763 | 0.693 |

**Ill-conditioned ($\kappa = 1000$)**
*Settings:* $\sigma^2 = 0.5$, $\beta = 0.9$, $\gamma = 0.1$

| Method | Loss ↓ | Track ↓ | Acc ↑ |
|---|---|---|---|
| SGD | **0.173** | **0.518** | **0.924** |
| HB | 0.690 | 0.902 | 0.758 |
| NAG | 0.690 | 0.902 | 0.759 |

### Teacher–Student MLP
**Well-conditioned ($\kappa = 10$)**
*Settings:* $\sigma^2 = 0.5$, $\beta = 0.9$, $\gamma = 0.06$

| Method | Loss ↓ | Pred. Track ↓ | Val MSE ↓ |
|---|---|---|---|
| SGD | **14.47** | **28.31** | **14.15** |
| HB | 15.62 | 30.19 | 15.09 |
| NAG | 15.59 | 30.18 | 15.09 |

**Ill-conditioned ($\kappa = 1000$)**
*Settings:* $\sigma^2 = 0.5$, $\beta = 0.9$, $\gamma = 0.02$

| Method | Loss ↓ | Pred. Track ↓ | Val MSE ↓ |
|---|---|---|---|
| SGD | **525.1** | **1064.5** | **532.3** |
| HB | 640.5 | 1275.3 | 637.6 |
| NAG | 639.8 | 1275.3 | 637.6 |

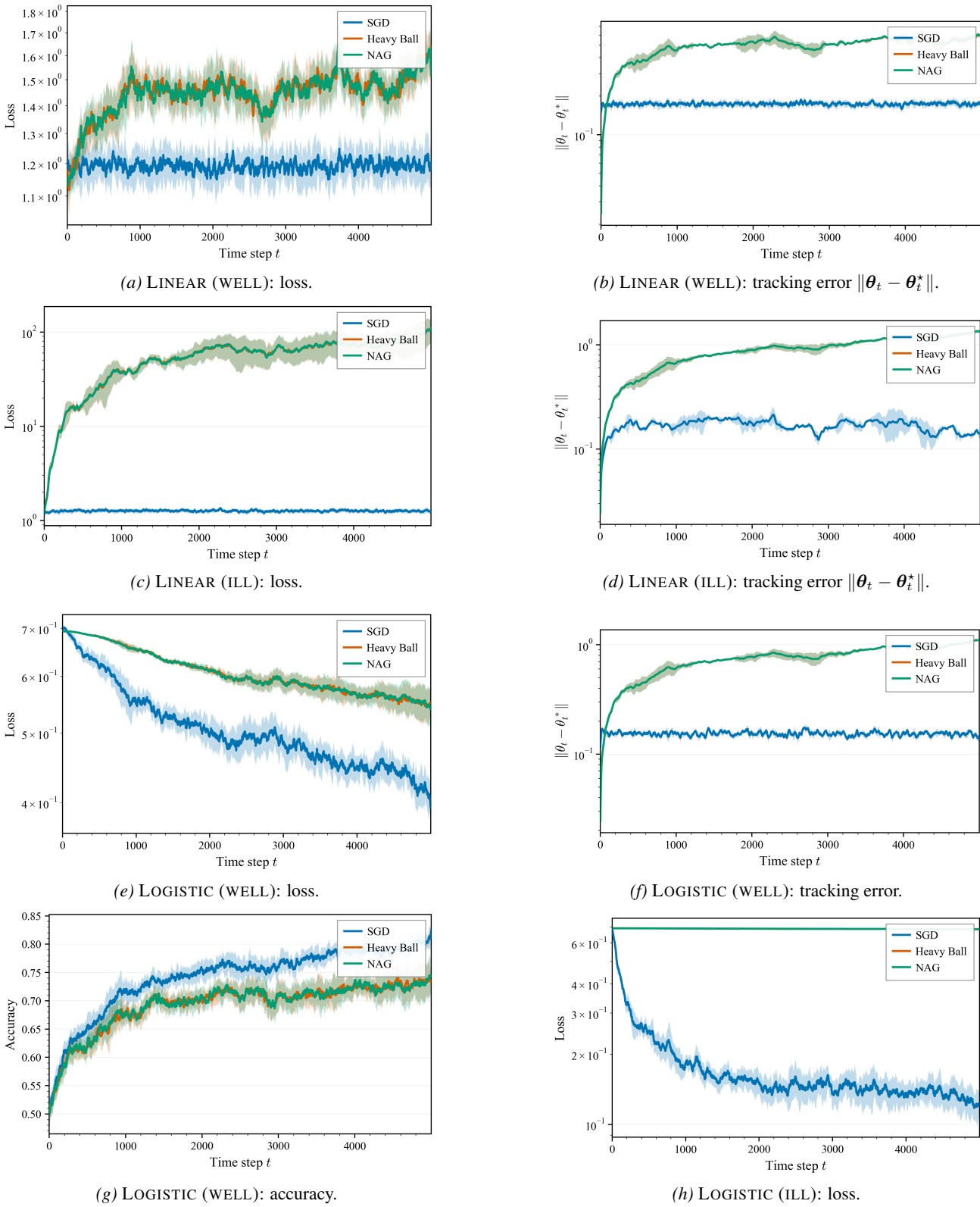

*(a)* LINEAR (WELL): loss.

*(b)* LINEAR (WELL): tracking error $\|\boldsymbol{\theta}_t - \boldsymbol{\theta}_t^\star\|$.

*(c)* LINEAR (ILL): loss.

*(d)* LINEAR (ILL): tracking error $\|\boldsymbol{\theta}_t - \boldsymbol{\theta}_t^\star\|$.

*(e)* LOGISTIC (WELL): loss.

*(f)* LOGISTIC (WELL): tracking error.

*(g)* LOGISTIC (WELL): accuracy.

*(h)* LOGISTIC (ILL): loss.

*Figure 4.* **Non-stationary benchmarks.** Results for Linear and Logistic tasks. *(Continued on next page.)*

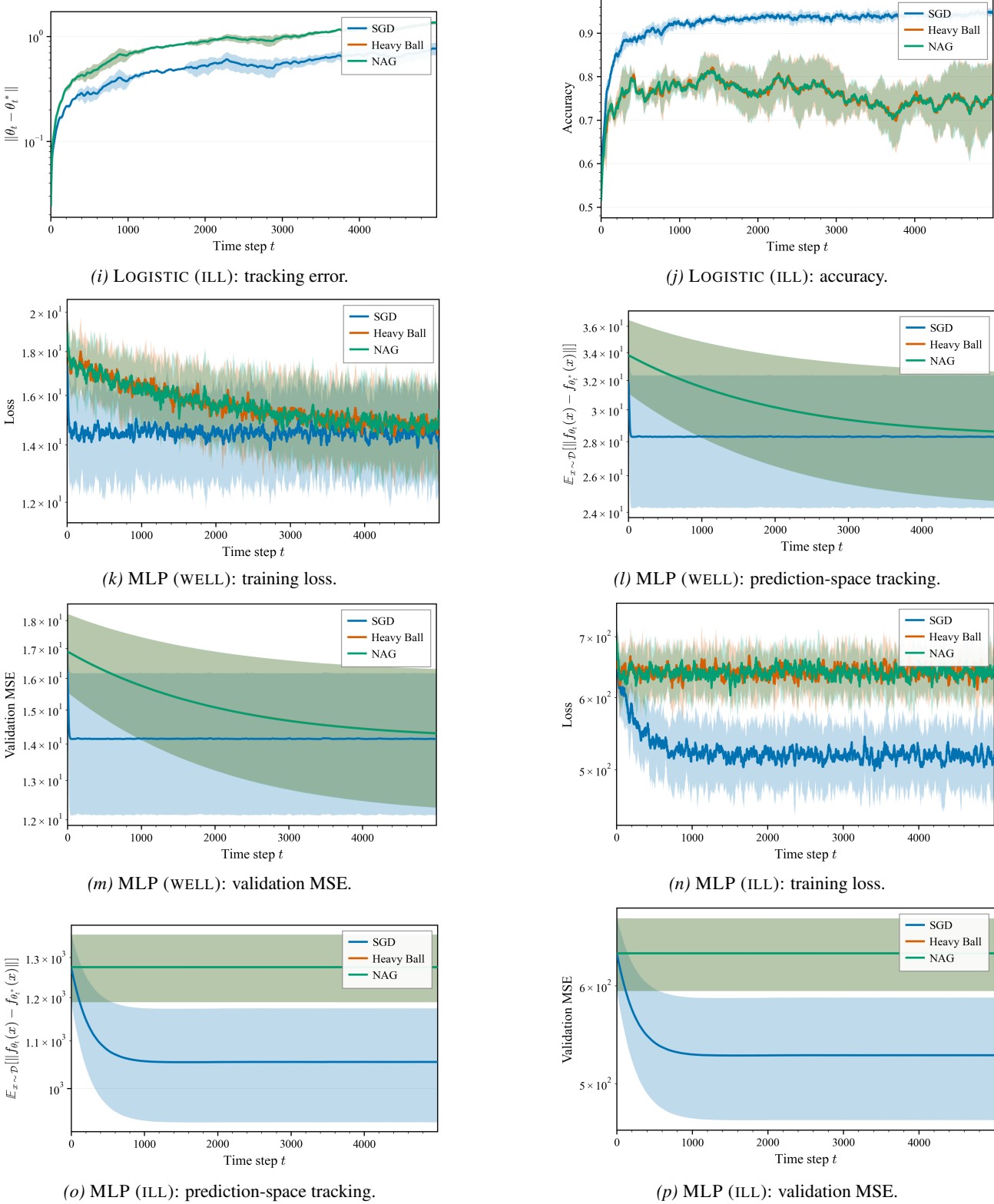

*(i)* LOGISTIC (ILL): tracking error.

*(j)* LOGISTIC (ILL): accuracy.

*(k)* MLP (WELL): training loss.

*(l)* MLP (WELL): prediction-space tracking.

*(m)* MLP (WELL): validation MSE.

*(n)* MLP (ILL): training loss.

*(o)* MLP (ILL): prediction-space tracking.

*(p)* MLP (ILL): validation MSE.

*Figure 4.* **Non-stationary benchmarks.** Mean ± 1 std over 20 seeds for well-conditioned ($\kappa = 10$) and ill-conditioned ($\kappa = 1000$) regimes. Tracking error: parameter-space for linear/logistic, prediction-space for MLP. Step sizes respect the stability bounds from Theorem 3.3.

# G. Technical Lemmas

## G.1. Conditional Orlicz norms

In this section, we will introduce the definition of conditional Orlicz norm. These properties follow from (Shen et al., 2026).

**Essential supremum/infimum.** Let $(\Omega, \mathcal{F}, \mathbb{P})$ be a probability space and let $\Phi$ be a collection of real-valued random variables on it. A random variable $\phi^\star$ is called the *essential supremum* of $\Phi$, denoted $\operatorname{ess\,sup} \Phi$, if: (i) $\phi^\star \geq \phi$ a.s. for all $\phi \in \Phi$, and (ii) whenever $\psi$ satisfies $\psi \geq \phi$ a.s. for all $\phi \in \Phi$, then $\psi \geq \phi^\star$ a.s. The *essential infimum* is defined by $\operatorname{ess\,inf} \Phi := -\operatorname{ess\,sup}(-\Phi)$. If $\Phi$ is *directed upward* (i.e., for any $\phi, \tilde{\phi} \in \Phi$ there exists $\psi \in \Phi$ with $\psi \geq \phi \vee \tilde{\phi}$ a.s.), then there exists an increasing sequence $\phi_1 \leq \phi_2 \leq \cdots$ in $\Phi$ such that $\operatorname{ess\,sup} \Phi = \lim_{n\to\infty} \phi_n$ a.s.

**Theorem G.1** (Basic properties). *Fix $\alpha \geq 1$ and let $X$ be a real-valued random variable. For a sub-$\sigma$-field $\mathcal{G} \subseteq \mathcal{F}$, define the set of admissible (random) scales*

$$\Phi_\alpha(X \mid \mathcal{G}) := \left\{ \phi : \Omega \to (0, \infty) \ \mathcal{G}\text{-measurable} \ : \ \mathbb{E}\big[\exp\big(|X/\phi|^\alpha\big) \mid \mathcal{G}\big] \leq 2 \text{ a.s.} \right\}.$$

*The* conditional $\Psi_\alpha$*–Orlicz norm of $X$ given $\mathcal{G}$ is*

$$\|X \mid \mathcal{G}\|_{\Psi_\alpha} := \operatorname{ess\,inf} \Phi_\alpha(X \mid \mathcal{G}).$$

*Let $\alpha \geq 1$ and $\mathcal{G} \subseteq \mathcal{F}$. Then $\|\cdot \mid \mathcal{G}\|_{\Psi_\alpha}$ is well-defined (a.s. finite whenever $\Phi_\alpha(X \mid \mathcal{G}) \neq \emptyset$) and satisfies:*

1. ***Positive homogeneity:*** *for any scalar $a$, $\|aX \mid \mathcal{G}\|_{\Psi_\alpha} = |a| \cdot \|X \mid \mathcal{G}\|_{\Psi_\alpha}$ a.s.*

2. ***Definiteness:*** $\|X \mid \mathcal{G}\|_{\Psi_\alpha} = 0$ *a.s. iff $X = 0$ a.s.*

3. ***Triangle inequality:*** $\|X + Y \mid \mathcal{G}\|_{\Psi_\alpha} \leq \|X \mid \mathcal{G}\|_{\Psi_\alpha} + \|Y \mid \mathcal{G}\|_{\Psi_\alpha}$ *a.s.*

4. ***Normalization:***

$$\mathbb{E}\left[\exp\left(\left|\frac{X}{\|X \mid \mathcal{G}\|_{\Psi_\alpha}}\right|^\alpha\right) \ \Big| \ \mathcal{G}\right] \leq 2 \qquad \text{a.s.}$$

*Proof of Theorem G.1.* Items (i) and the reverse implication of (ii) are immediate from the definition. For the converse direction of (ii), assume $\|X \mid \mathcal{G}\|_{\Psi_\alpha} = 0$ a.s. By definition of essential infimum, for every $\varepsilon > 0$ there exists a $\mathcal{G}$-measurable $\phi_\varepsilon \in (0, \varepsilon]$ such that $\mathbb{E}[\exp(|X/\phi_\varepsilon|^\alpha) \mid \mathcal{G}] \leq 2$ a.s. On the event $\{|X| > 0\}$, we have $|X/\phi_\varepsilon| \to \infty$ as $\varepsilon \downarrow 0$, so $\exp(|X/\phi_\varepsilon|^\alpha) \to \infty$. By conditional Fatou, $\mathbb{E}[\liminf_{\varepsilon\downarrow 0} \exp(|X/\phi_\varepsilon|^\alpha) \mid \mathcal{G}] \leq \liminf_{\varepsilon\downarrow 0} \mathbb{E}[\exp(|X/\phi_\varepsilon|^\alpha) \mid \mathcal{G}] \leq 2$, which forces $\mathbb{P}(|X| > 0) = 0$, i.e., $X = 0$ a.s.

For (iii), fix $\varepsilon > 0$ and set $u := \|X \mid \mathcal{G}\|_{\Psi_\alpha} + \varepsilon$ and $v := \|Y \mid \mathcal{G}\|_{\Psi_\alpha} + \varepsilon$. By definition of essential infimum, $u, v$ are admissible scales, hence $\mathbb{E}[\exp(|X/u|^\alpha) \mid \mathcal{G}] \leq 2$ and $\mathbb{E}[\exp(|Y/v|^\alpha) \mid \mathcal{G}] \leq 2$ a.s. Let $s := u/(u+v)$ (note $s$ is $\mathcal{G}$-measurable and $s \in (0, 1)$). Then

$$\frac{|X + Y|}{u + v} \ \leq \ s \cdot \frac{|X|}{u} + (1 - s) \cdot \frac{|Y|}{v}.$$

Since $t \mapsto t^\alpha$ and $\exp(\cdot)$ are convex and increasing on $\mathbb{R}_+$, we have

$$\exp\left(\big(sA + (1-s)B\big)^\alpha\right) \ \leq \ \exp(sA^\alpha + (1-s)B^\alpha) \ \leq \ s\,e^{A^\alpha} + (1-s)\,e^{B^\alpha} \qquad (A, B \geq 0).$$

Applying this with $A = |X|/u$ and $B = |Y|/v$ and taking conditional expectations yields

$$\mathbb{E}\left[\exp\left(\left|\frac{X+Y}{u+v}\right|^\alpha\right) \ \Big| \ \mathcal{G}\right] \leq s\,\mathbb{E}\left[e^{(|X|/u)^\alpha} \mid \mathcal{G}\right] + (1-s)\,\mathbb{E}\left[e^{(|Y|/v)^\alpha} \mid \mathcal{G}\right] \leq 2.$$

Thus $u + v \in \Phi_\alpha(X + Y \mid \mathcal{G})$, so by taking the essential infimum over admissible scales and sending $\varepsilon \downarrow 0$ we obtain (iii).

For (iv), note that $\Phi_\alpha(X \mid \mathcal{G})$ is directed downward (equivalently, $-\Phi_\alpha$ is directed upward), so there exists a decreasing sequence $\{\phi_n\}_{n\geq 1} \subset \Phi_\alpha(X \mid \mathcal{G})$ with $\phi_n \downarrow \|X \mid \mathcal{G}\|_{\Psi_\alpha}$ a.s. By conditional Fatou,

$$\mathbb{E}\left[\exp\left(\left|\frac{X}{\|X \mid \mathcal{G}\|_{\Psi_\alpha}}\right|^\alpha\right) \ \Big| \ \mathcal{G}\right] \leq \liminf_{n\to\infty} \mathbb{E}\left[\exp\left(\left|\frac{X}{\phi_n}\right|^\alpha\right) \ \Big| \ \mathcal{G}\right] \leq 2,$$

as claimed. $\qquad\square$

For a random vector $\boldsymbol{Z} \in \mathbb{R}^d$, we use the standard sub-Gaussian extension

$$\|\boldsymbol{Z} \mid \mathcal{G}\|_{\Psi_2} := \sup_{\boldsymbol{u} \in \mathbb{S}^{d-1}} \|\langle \boldsymbol{u}, \boldsymbol{Z} \rangle \mid \mathcal{G}\|_{\Psi_2}.$$

For a random matrix $\boldsymbol{M} \in \mathbb{R}^{d \times d}$, define similarly $\|\boldsymbol{M} \mid \mathcal{G}\|_{\Psi_1} := \sup_{\boldsymbol{u}, \boldsymbol{v} \in \mathbb{S}^{d-1}} \|\boldsymbol{u}^\top \boldsymbol{M} \boldsymbol{v} \mid \mathcal{G}\|_{\Psi_1}$.

**Lemma G.1** (Conditional "sub-Gaussian $\times$ sub-Gaussian $\Rightarrow$ sub-exponential"). *Let $\boldsymbol{X}, \boldsymbol{Y} \in \mathbb{R}^d$ be random vectors and $\mathcal{G} \subseteq \mathcal{F}$. Then*

$$\|\boldsymbol{X}\boldsymbol{Y}^\top \mid \mathcal{G}\|_{\Psi_1} \leq \|\boldsymbol{X} \mid \mathcal{G}\|_{\Psi_2} \cdot \|\boldsymbol{Y} \mid \mathcal{G}\|_{\Psi_2} \qquad a.s.$$

*Proof of Lemma G.1.* We first prove the scalar case. Let $U, V$ be real-valued random variables and write $A := \|U \mid \mathcal{G}\|_{\Psi_2}$ and $B := \|V \mid \mathcal{G}\|_{\Psi_2}$. We show that $\|UV \mid \mathcal{G}\|_{\Psi_1} \leq AB$ almost surely.

Fix $\varepsilon > 0$. By the essential-infimum property of the conditional Luxemburg norm, there exist positive $\mathcal{G}$-measurable random variables $A_\varepsilon, B_\varepsilon$ satisfying $A_\varepsilon \leq A + \varepsilon$ and $B_\varepsilon \leq B + \varepsilon$ almost surely, and

$$\mathbb{E}\left[\exp\left(\frac{U^2}{A_\varepsilon^2}\right) \mid \mathcal{G}\right] \leq 2, \quad \mathbb{E}\left[\exp\left(\frac{V^2}{B_\varepsilon^2}\right) \mid \mathcal{G}\right] \leq 2 \quad \text{a.s.}$$

Apply Young's inequality $2ab \leq a^2 + b^2$ with $a = |U|/A_\varepsilon$ and $b = |V|/B_\varepsilon$ to obtain $|UV|/(A_\varepsilon B_\varepsilon) \leq U^2/(2A_\varepsilon^2) + V^2/(2B_\varepsilon^2)$. Exponentiate both sides and apply the convexity inequality $e^{(x+y)/2} \leq (e^x + e^y)/2$ to get

$$\exp\left(\frac{|UV|}{A_\varepsilon B_\varepsilon}\right) \leq \frac{1}{2}\exp\left(\frac{U^2}{A_\varepsilon^2}\right) + \frac{1}{2}\exp\left(\frac{V^2}{B_\varepsilon^2}\right).$$

Taking conditional expectation given $\mathcal{G}$ yields $\mathbb{E}[\exp(|UV|/(A_\varepsilon B_\varepsilon)) \mid \mathcal{G}] \leq 2$ almost surely. By the definition of the conditional $\Psi_1$-norm, $\|UV \mid \mathcal{G}\|_{\Psi_1} \leq A_\varepsilon B_\varepsilon \leq (A + \varepsilon)(B + \varepsilon)$ almost surely. Letting $\varepsilon \downarrow 0$ gives $\|UV \mid \mathcal{G}\|_{\Psi_1} \leq AB = \|U \mid \mathcal{G}\|_{\Psi_2} \|V \mid \mathcal{G}\|_{\Psi_2}$ almost surely.

For the matrix case, observe that for any $\boldsymbol{u}, \boldsymbol{v} \in \mathbb{S}^{d-1}$, we have $\boldsymbol{u}^\top \boldsymbol{X}\boldsymbol{Y}^\top \boldsymbol{v} = \langle \boldsymbol{u}, \boldsymbol{X} \rangle \langle \boldsymbol{v}, \boldsymbol{Y} \rangle$. Applying the scalar result with $U = \langle \boldsymbol{u}, \boldsymbol{X} \rangle$ and $V = \langle \boldsymbol{v}, \boldsymbol{Y} \rangle$ gives $\|\boldsymbol{u}^\top \boldsymbol{X}\boldsymbol{Y}^\top \boldsymbol{v} \mid \mathcal{G}\|_{\Psi_1} \leq \|\langle \boldsymbol{u}, \boldsymbol{X} \rangle \mid \mathcal{G}\|_{\Psi_2} \|\langle \boldsymbol{v}, \boldsymbol{Y} \rangle \mid \mathcal{G}\|_{\Psi_2}$. By the induced vector conditional $\Psi_2$-norm definition, $\|\langle \boldsymbol{u}, \boldsymbol{X} \rangle \mid \mathcal{G}\|_{\Psi_2} \leq \|\boldsymbol{X} \mid \mathcal{G}\|_{\Psi_2}$ and $\|\langle \boldsymbol{v}, \boldsymbol{Y} \rangle \mid \mathcal{G}\|_{\Psi_2} \leq \|\boldsymbol{Y} \mid \mathcal{G}\|_{\Psi_2}$. Therefore, for every $\boldsymbol{u}, \boldsymbol{v} \in \mathbb{S}^{d-1}$, we have $\|\boldsymbol{u}^\top \boldsymbol{X}\boldsymbol{Y}^\top \boldsymbol{v} \mid \mathcal{G}\|_{\Psi_1} \leq \|\boldsymbol{X} \mid \mathcal{G}\|_{\Psi_2} \|\boldsymbol{Y} \mid \mathcal{G}\|_{\Psi_2}$ almost surely. Taking the supremum over $\boldsymbol{u}, \boldsymbol{v} \in \mathbb{S}^{d-1}$ completes the proof. $\qquad \square$

We will repeatedly use the following consequences of the definition and the tower property of conditional expectation.

**Lemma G.2** (Monotonicity and conditioning). *We have the following rules for conditioning and monotonicity:*

1. *If $\mathcal{G}_1 \subseteq \mathcal{G}_2$ and $\|X \mid \mathcal{G}_2\|_{\Psi_2} \leq K$ a.s. for a constant $K$, then $\|X \mid \mathcal{G}_1\|_{\Psi_2} \leq K$ a.s.*

2. *If $|Y| \leq K$ a.s. for a constant $K$, then $\|XY \mid \mathcal{G}\|_{\Psi_\alpha} \leq K\|X \mid \mathcal{G}\|_{\Psi_\alpha}$ a.s.*

3. *If $X$ is independent of $\mathcal{G}$, then $\|X \mid \mathcal{G}\|_{\Psi_\alpha} = \|X\|_{\Psi_\alpha}$ a.s.*

4. *If $X \perp Y \mid \mathcal{G}$, then $\mathbb{E}[XY \mid \mathcal{G}] = \mathbb{E}[X \mid \mathcal{G}] \mathbb{E}[Y \mid \mathcal{G}]$ a.s., and $\mathbb{E}[X \mid Y, \mathcal{G}] = \mathbb{E}[X \mid \mathcal{G}]$ a.s.*

*Proof of Lemma G.2.* (i) By assumption, $\mathbb{E}[\exp(|X/K|^2) \mid \mathcal{G}_2] \leq 2$ a.s.; taking $\mathcal{G}_1$-conditional expectations and using the tower property yields $\mathbb{E}[\exp(|X/K|^2) \mid \mathcal{G}_1] \leq 2$ a.s., hence $\|X \mid \mathcal{G}_1\|_{\Psi_2} \leq K$. (ii) If $\phi$ is admissible for $X$ given $\mathcal{G}$, then $K\phi$ is admissible for $XY$ given $\mathcal{G}$ since $|XY|/(K\phi) \leq |X|/\phi$. Taking essential infima yields the claim. (iii) If $X \perp \mathcal{G}$, then $\mathbb{E}[\exp(|X/\phi|^\alpha) \mid \mathcal{G}] = \mathbb{E}[\exp(|X/\phi|^\alpha)]$ for any constant $\phi$, so the conditional and unconditional admissible scales coincide a.s. (iv) this follows from the standard properties of conditional independence. $\qquad \square$

**Lemma G.3.** *Let $\mathcal{G}_1 \subseteq \mathcal{G}_2$. If $\|X \mid \mathcal{G}_1\|_{\Psi_2} \leq \lambda$ a.s. and $\|\boldsymbol{\xi} \mid \mathcal{G}_2\|_{\Psi_2} \leq K$ a.s. for constants $\lambda, K$, then*

$$\|\boldsymbol{\xi}X \mid \mathcal{G}_1\|_{\Psi_1} \leq K\lambda \qquad a.s.$$

*Proof of Lemma G.3.* By Lemma G.2(i), $\|\boldsymbol{\xi} \mid \mathcal{G}_1\|_{\Psi_2} \leq K$ a.s. Apply Lemma G.1 (in the scalar case) conditioned on $\mathcal{G}_1$ to obtain $\|\boldsymbol{\xi}X \mid \mathcal{G}_1\|_{\Psi_1} \leq \|\boldsymbol{\xi} \mid \mathcal{G}_1\|_{\Psi_2} \cdot \|X \mid \mathcal{G}_1\|_{\Psi_2} \leq K\lambda$ a.s. $\qquad \square$

**Lemma G.4** (Conditional $\Psi_2$ control implies conditional second moments). *Let $\mathcal{F}$ be a $\sigma$-field and let $K_{\mathcal{F}} > 0$ be $\mathcal{F}$-measurable.*

1. ***(Scalar).*** *If $\mathbb{E}\big[\exp\big(|X|^2/K_{\mathcal{F}}^2\big) \mid \mathcal{F}\big] \leq 2$ a.s, then $\mathbb{E}[|X|^2 \mid \mathcal{F}] \leq K_{\mathcal{F}}^2$ a.s*

2. ***(Vector).*** *Let $\boldsymbol{X} \in \mathbb{R}^d$. If $\sup_{\substack{\boldsymbol{u} \in \mathbb{S}^{d-1} \\ \boldsymbol{u} \ \mathcal{F}\text{-measurable}}} \mathbb{E}\big[\exp\big(|\boldsymbol{u}^\top \boldsymbol{X}|^2/K_{\mathcal{F}}^2\big) \mid \mathcal{F}\big] \leq 2$ a.s , then $\mathbb{E}[\boldsymbol{X}\boldsymbol{X}^\top \mid \mathcal{F}] \preceq K_{\mathcal{F}}^2 \mathbf{I}_d$ a.s. and hence $\mathbb{E}[\|\boldsymbol{X}\|_2^2 \mid \mathcal{F}] \leq dK_{\mathcal{F}}^2$ a.s.*

*Proof of Lemma G.4.* We first prove the scalar case. By $e^y \geq 1 + y$ for $y \geq 0$,

$$2 \geq \mathbb{E}\left[1 + \frac{|X|^2}{K_{\mathcal{F}}^2} \ \Big| \ \mathcal{F}\right] = 1 + \frac{1}{K_{\mathcal{F}}^2}\mathbb{E}[|X|^2 \mid \mathcal{F}] \quad \text{a.s.,}$$

so $\mathbb{E}[|X|^2 \mid \mathcal{F}] \leq K_{\mathcal{F}}^2$. The vector case follows very similarly. Fix any $\mathcal{F}$-measurable $\boldsymbol{u} \in \mathbb{S}^{d-1}$ and apply the scalar part to $\boldsymbol{u}^\top \boldsymbol{X}$ to get $\mathbb{E}[(\boldsymbol{u}^\top \boldsymbol{X})^2 \mid \mathcal{F}] \leq K_{\mathcal{F}}^2$ a.s. Since $\boldsymbol{u}^\top \mathbb{E}[\boldsymbol{X}\boldsymbol{X}^\top \mid \mathcal{F}]\boldsymbol{u} = \mathbb{E}[(\boldsymbol{u}^\top \boldsymbol{X})^2 \mid \mathcal{F}]$, this yields $\boldsymbol{u}^\top \mathbb{E}[\boldsymbol{X}\boldsymbol{X}^\top \mid \mathcal{F}]\boldsymbol{u} \leq K_{\mathcal{F}}^2$ for all $\boldsymbol{u} \in \mathbb{S}^{d-1}$, i.e., $\mathbb{E}[\boldsymbol{X}\boldsymbol{X}^\top \mid \mathcal{F}] \preceq K_{\mathcal{F}}^2 \mathbf{I}_d$. Taking traces gives $\mathbb{E}[\|\boldsymbol{X}\|_2^2 \mid \mathcal{F}] = \operatorname{tr}(\mathbb{E}[\boldsymbol{X}\boldsymbol{X}^\top \mid \mathcal{F}]) \leq \operatorname{tr}(K_{\mathcal{F}}^2 \mathbf{I}_d) = dK_{\mathcal{F}}^2$. $\qquad\square$

## G.2. Martingale Concentration Inequalities

**Lemma G.5** (Bernstein inequality for sub-exponential martingale differences). *Let $(\mathcal{F}_t)_{t=0}^T$ be a filtration and let $Z_1, \ldots, Z_T$ be real-valued random variables such that $\mathbb{E}[Z_t \mid \mathcal{F}_{t-1}] = 0$ a.s. $(t = 1, \ldots, T)$. Assume that $Z_t \mid \mathcal{F}_{t-1}$ is conditionally sub-exponential in the sense that $\big\|Z_t \mid \mathcal{F}_{t-1}\big\|_{\Psi_1} \leq K_t$ a.s., $(t = 1, \ldots, T)$, where each $K_t$ is a deterministic constant. Then there exist absolute constants $c, C > 0$ such that for all $s \geq 0$,*

$$\mathbb{P}\left(\sum_{t=1}^T Z_t \geq s\right) \leq \exp\left(-\min\left\{\frac{s^2}{C\sum_{t=1}^T K_t^2}, \ \frac{s}{C\max_{1 \leq t \leq T} K_t}\right\}\right).$$

*Proof of Lemma G.5.* Fix $\lambda > 0$. By Markov's inequality,

$$\mathbb{P}\left(\sum_{t=1}^T Z_t \geq s\right) = \mathbb{P}\left(\exp\left(\lambda \sum_{t=1}^T Z_t\right) \geq e^{\lambda s}\right) \leq e^{-\lambda s}\, \mathbb{E}\exp\left(\lambda \sum_{t=1}^T Z_t\right). \tag{234}$$

Let $S_t := \sum_{i=1}^t Z_i$ with $S_0 := 0$. Using tower property and $\mathcal{F}_{T-1}$-measurability of $S_{T-1}$,

$$\mathbb{E}e^{\lambda S_T} = \mathbb{E}\big[e^{\lambda S_{T-1}} \mathbb{E}\big[e^{\lambda Z_T} \mid \mathcal{F}_{T-1}\big]\big].$$

We now use the standard sub-exponential mgf bound in conditional form: there exist absolute constants $c, C > 0$ such that for any random variable $X$ with $\|X\|_{\Psi_1} \leq K$,

$$\mathbb{E}[e^{\lambda X}] \leq \exp(C\lambda^2 K^2) \quad \text{for all } 0 \leq \lambda \leq c/K.$$

Applying this conditional on $\mathcal{F}_{T-1}$ (and using $\|Z_T \mid \mathcal{F}_{T-1}\|_{\Psi_1} \leq K_T$ a.s.), we obtain that for $0 \leq \lambda \leq c/K_T$,

$$\mathbb{E}\big[e^{\lambda Z_T} \mid \mathcal{F}_{T-1}\big] \leq \exp(C\lambda^2 K_T^2) \quad \text{a.s.}$$

Hence, for $0 \leq \lambda \leq c/K_T$,

$$\mathbb{E}e^{\lambda S_T} \leq \exp(C\lambda^2 K_T^2)\, \mathbb{E}e^{\lambda S_{T-1}}.$$

Iterating this argument for $t = T, T-1, \ldots, 1$ yields that for

$$0 \leq \lambda \leq \frac{c}{\max_{1 \leq t \leq T} K_t},$$

we have

$$\mathbb{E}\exp\left(\lambda \sum_{t=1}^T Z_t\right) = \mathbb{E}e^{\lambda S_T} \leq \exp\left(C\lambda^2 \sum_{t=1}^T K_t^2\right). \tag{235}$$

Combining (234) and (235) gives, for all admissible $\lambda$,

$$\mathbb{P}\left(\sum_{t=1}^{T} Z_t \geq s\right) \leq \exp\left(-\lambda s + C\lambda^2 \sum_{t=1}^{T} K_t^2\right).$$

Choose

$$\lambda := \min\left\{\frac{s}{2C\sum_{t=1}^{T} K_t^2}, \ \frac{c}{\max_{1\leq t\leq T} K_t}\right\}.$$

Substituting this choice into the previous bound yields

$$\mathbb{P}\left(\sum_{t=1}^{T} Z_t \geq s\right) \leq \exp\left(-\min\left\{\frac{s^2}{C\sum_{t=1}^{T} K_t^2}, \ \frac{s}{C\max_{1\leq t\leq T} K_t}\right\}\right),$$

after adjusting absolute constants, which proves the claim. $\qquad\square$

### G.3. Optional stopping

**Lemma G.6** (Optional stopping / sampling theorem)**.** *Let $(\Omega, \mathcal{F}, \mathbb{P})$ be a probability space with a filtration $(\mathcal{F}_t)_{t\geq 0}$, and let $(M_t)_{t\geq 0}$ be an integrable $(\mathcal{F}_t)$-martingale, i.e. $\mathbb{E}[|M_t|] < \infty$ and $\mathbb{E}[M_{t+1} \mid \mathcal{F}_t] = M_t$ a.s. for all $t \geq 0$. Let $\tau$ be an $(\mathcal{F}_t)$-stopping time.*

1. *(**Bounded stopping time**). If $\tau \leq T$ a.s. for some deterministic $T < \infty$, then*

$$\mathbb{E}[M_\tau] = \mathbb{E}[M_0].$$

2. *(**Unbounded stopping time via truncation**). Assume $\mathbb{E}[|M_{\tau\wedge n}|] < \infty$ for all $n$ and that $\{M_{\tau\wedge n}\}_{n\geq 1}$ is uniformly integrable. Then*

$$\mathbb{E}[M_\tau] = \mathbb{E}[M_0].$$

*More generally, if $(M_t)$ is an integrable supermartingale (resp. submartingale), then in* either *(i) or (ii) we have $\mathbb{E}[M_\tau] \leq \mathbb{E}[M_0]$ (resp. $\mathbb{E}[M_\tau] \geq \mathbb{E}[M_0]$).*

*Proof of Lemma G.6.* We first show that the stopped process is a martingale:

**Step 1: Show that the stopped process is a martingale.** Fix a stopping time $\tau$ and define the stopped process

$$M_t^\tau := M_{t\wedge\tau}, \qquad t \geq 0.$$

We claim $(M_t^\tau)_{t\geq 0}$ is an $(\mathcal{F}_t)$-martingale (assuming integrability). Fix $t \geq 0$. Since $\{\tau \leq t\} \in \mathcal{F}_t$, we can split on the events $\{\tau \leq t\}$ and $\{\tau > t\}$:

$$\begin{aligned}
\mathbb{E}[M_{(t+1)\wedge\tau} \mid \mathcal{F}_t] &= \mathbb{E}[M_\tau \mathbf{1}\{\tau \leq t\} + M_{t+1}\mathbf{1}\{\tau > t\} \mid \mathcal{F}_t] \\
&= M_\tau \mathbf{1}\{\tau \leq t\} + \mathbf{1}\{\tau > t\}\mathbb{E}[M_{t+1} \mid \mathcal{F}_t] \\
&= M_\tau \mathbf{1}\{\tau \leq t\} + \mathbf{1}\{\tau > t\}M_t \\
&= M_{t\wedge\tau}.
\end{aligned}$$

Thus, $\mathbb{E}[M_{(t+1)\wedge\tau} \mid \mathcal{F}_t] = M_{t\wedge\tau}$ a.s., so $(M_{t\wedge\tau})$ is a martingale.

**Step 2: Bounded $\tau$.** If $\tau \leq T$ a.s., then $(T \wedge \tau) = \tau$ and by the martingale property of $(M_{t\wedge\tau})$ from Step 1,

$$\mathbb{E}[M_\tau] = \mathbb{E}[M_{T\wedge\tau}] = \mathbb{E}[M_0],$$

which proves (i).

**Step 3: Unbounded $\tau$ under uniform integrability.** Let $\tau_n := \tau \wedge n$. Each $\tau_n$ is a bounded stopping time, so by (i),

$$\mathbb{E}[M_{\tau_n}] = \mathbb{E}[M_0] \qquad \text{for all } n \geq 1.$$

Moreover, $\tau_n \uparrow \tau$ and $M_{\tau_n} \to M_\tau$ a.s. (since $\tau_n = \tau$ for all large $n$ on $\{\tau < \infty\}$). If $\{M_{\tau_n}\}_{n\geq 1}$ is uniformly integrable, then $\mathbb{E}[M_{\tau_n}] \to \mathbb{E}[M_\tau]$ as $n \to \infty$, hence

$$\mathbb{E}[M_\tau] = \lim_{n\to\infty} \mathbb{E}[M_{\tau_n}] = \mathbb{E}[M_0],$$

which proves (ii). The super/submartingale case follows identically. $\qquad\square$

### G.4. Information Theory

**Lemma G.7** (Mutual information bounded by average pairwise KL). *Let $\mathcal{U} = \{\boldsymbol{u}^{(1)}, \ldots, \boldsymbol{u}^{(M)}\}$ be a finite set and let $\boldsymbol{U}$ be uniformly distributed on $\mathcal{U}$. For each $\boldsymbol{u} \in \mathcal{U}$, let $P_{\boldsymbol{u}}$ be a probability distribution on a measurable space $(\mathcal{Z}, \mathcal{A})$, and let $\boldsymbol{Z}$ be a random variable such that $(\boldsymbol{U}, \boldsymbol{Z})$ is generated by $\boldsymbol{U} \sim \mathrm{Unif}(\mathcal{U})$ and $\boldsymbol{Z} \mid (\boldsymbol{U} = \boldsymbol{u}) \sim P_{\boldsymbol{u}}$. Define the mixture distribution*

$$\bar{P} := \frac{1}{M} \sum_{\boldsymbol{u} \in \mathcal{U}} P_{\boldsymbol{u}}.$$

*Then we have the following bound for the mutual information:*

$$I(\boldsymbol{U}; \boldsymbol{Z}) = \frac{1}{M} \sum_{\boldsymbol{u} \in \mathcal{U}} D_{\mathrm{KL}}(P_{\boldsymbol{u}} \,\|\, \bar{P}) \leq \frac{1}{M^2} \sum_{\boldsymbol{u}, \boldsymbol{v} \in \mathcal{U}} D_{\mathrm{KL}}(P_{\boldsymbol{u}} \,\|\, P_{\boldsymbol{v}}). \tag{236}$$

*Proof of Lemma G.7.* Write $M := |\mathcal{U}|$. By definition of mutual information and the uniform prior,

$$I(\boldsymbol{U}; \boldsymbol{Z}) = \sum_{\boldsymbol{u} \in \mathcal{U}} \frac{1}{M} \int \log\left(\frac{dP_{\boldsymbol{u}}}{d\bar{P}}(z)\right) dP_{\boldsymbol{u}}(z) = \frac{1}{M} \sum_{\boldsymbol{u} \in \mathcal{U}} D_{\mathrm{KL}}(P_{\boldsymbol{u}} \,\|\, \bar{P}), \tag{237}$$

where $\bar{P} = \frac{1}{M} \sum_{\boldsymbol{v} \in \mathcal{U}} P_{\boldsymbol{v}}$ is the marginal law of $\boldsymbol{Z}$. Fix $\boldsymbol{u} \in \mathcal{U}$. Since $\bar{P} = \frac{1}{M} \sum_{\boldsymbol{v} \in \mathcal{U}} P_{\boldsymbol{v}}$, we have for $P_{\boldsymbol{u}}$-a.e. $z$,

$$\log \bar{p}(z) = \log\left(\frac{1}{M} \sum_{\boldsymbol{v} \in \mathcal{U}} p_{\boldsymbol{v}}(z)\right),$$

where $p_{\boldsymbol{v}}$ are densities with respect to any common dominating measure (or Radon–Nikodym derivatives). By the concavity of $\log(\cdot)$, Jensen's inequality yields

$$\log\left(\frac{1}{M} \sum_{\boldsymbol{v} \in \mathcal{U}} p_{\boldsymbol{v}}(z)\right) \geq \frac{1}{M} \sum_{\boldsymbol{v} \in \mathcal{U}} \log p_{\boldsymbol{v}}(z). \tag{238}$$

Equivalently, $-\log \bar{p}(z) \leq \frac{1}{M} \sum_{\boldsymbol{v} \in \mathcal{U}} (-\log p_{\boldsymbol{v}}(z))$. Plugging this into the KL definition gives

$$
\begin{aligned}
D_{\mathrm{KL}}(P_{\boldsymbol{u}} \,\|\, \bar{P}) &= \int \left(\log p_{\boldsymbol{u}}(z) - \log \bar{p}(z)\right) p_{\boldsymbol{u}}(z) \, dz \\
&\leq \int \left(\log p_{\boldsymbol{u}}(z) - \frac{1}{M} \sum_{\boldsymbol{v} \in \mathcal{U}} \log p_{\boldsymbol{v}}(z)\right) p_{\boldsymbol{u}}(z) \, dz \\
&= \frac{1}{M} \sum_{\boldsymbol{v} \in \mathcal{U}} \int \left(\log p_{\boldsymbol{u}}(z) - \log p_{\boldsymbol{v}}(z)\right) p_{\boldsymbol{u}}(z) \, dz = \frac{1}{M} \sum_{\boldsymbol{v} \in \mathcal{U}} D_{\mathrm{KL}}(P_{\boldsymbol{u}} \,\|\, P_{\boldsymbol{v}}).
\end{aligned} \tag{239}
$$

Averaging (239) over $\boldsymbol{u} \sim \mathrm{Unif}(\mathcal{U})$ and using (237) yields

$$I(\boldsymbol{U}; \boldsymbol{Z}) = \frac{1}{M} \sum_{\boldsymbol{u} \in \mathcal{U}} D_{\mathrm{KL}}(P_{\boldsymbol{u}} \,\|\, \bar{P}) \leq \frac{1}{M} \sum_{\boldsymbol{u} \in \mathcal{U}} \frac{1}{M} \sum_{\boldsymbol{v} \in \mathcal{U}} D_{\mathrm{KL}}(P_{\boldsymbol{u}} \,\|\, P_{\boldsymbol{v}}) = \frac{1}{M^2} \sum_{\boldsymbol{u}, \boldsymbol{v} \in \mathcal{U}} D_{\mathrm{KL}}(P_{\boldsymbol{u}} \,\|\, P_{\boldsymbol{v}}),$$

which is exactly (236). $\qquad\square$

## G.5. Converting between tracking error and dynamic regret

In this section, we introduce a few technical results that will facilitate converting between tracking error and dynamic regret. Recall for a sequence of functions $G_t : \Theta \to \mathbb{R}$ with $\Theta \subset \mathbb{R}^d$ convex, dynamic regret for a policy $\pi$ is defined as

$$\mathcal{R}_T^\pi(G) := \mathbb{E}^\pi \left[ \sum_{t=0}^{T-1} \left( G_{t+1}(\boldsymbol{\theta}_t) - G_{t+1}(\boldsymbol{\theta}_{t+1}^\star) \right) \right].$$

We first show using smoothness of $G_t$, bounding the tracking error immediately implies a bound on dynamic regret:

**Lemma G.8** (Bounding tracking error implies a bound on dynamic regret). *Assume each $G_t : \Theta \to \mathbb{R}$ is $L$-smooth. Then for any policy $\pi$,*

$$\mathcal{R}_T^\pi(G) \leq L \sum_{t=0}^{T-1} \mathbb{E}\|\boldsymbol{\theta}_t - \boldsymbol{\theta}_t^\star\|^2 + L \sum_{t=0}^{T-1} \mathbb{E}\|\boldsymbol{\Delta}_t\|^2.$$

*where $\boldsymbol{\Delta}_t = \boldsymbol{\theta}_t^\star - \boldsymbol{\theta}_{t+1}^\star$*

*Proof of Lemma G.8.* Fix $t \in \{0, \ldots, T-1\}$. Since $G_{t+1}$ is $L$-smooth, we have the following:

$$G_{t+1}(\boldsymbol{\theta}_t) - G_{t+1}(\boldsymbol{\theta}_{t+1}^\star) \leq \langle \nabla_{\boldsymbol{\theta}} G_{t+1}(\boldsymbol{\theta}_{t+1}^\star), \boldsymbol{\theta}_t - \boldsymbol{\theta}_{t+1}^\star \rangle + \frac{L}{2}\|\boldsymbol{\theta}_t - \boldsymbol{\theta}_{t+1}^\star\|^2$$

$$= \frac{L}{2}\|\boldsymbol{\theta}_t - \boldsymbol{\theta}_{t+1}^\star\|^2, \tag{240}$$

where the last equality follows from the optimality of $\boldsymbol{\theta}_{t+1}^\star \in \arg\min_{\boldsymbol{\theta} \in \Theta} G_{t+1}(\boldsymbol{\theta})$. Now we decompose $\boldsymbol{\theta}_t - \boldsymbol{\theta}_{t+1}^\star$ as follows: $\boldsymbol{\theta}_t - \boldsymbol{\theta}_{t+1}^\star = \boldsymbol{\theta}_t - \boldsymbol{\theta}_t^\star + \boldsymbol{\theta}_t^\star - \boldsymbol{\theta}_{t+1}^\star = \boldsymbol{\theta}_t - \boldsymbol{\theta}_t^\star + \boldsymbol{\Delta}_t$. Using the inequality $\|\boldsymbol{a} + \boldsymbol{b}\|^2 \leq 2\|\boldsymbol{a}\|^2 + 2\|\boldsymbol{b}\|^2$ with (240), we find

$$G_{t+1}(\boldsymbol{\theta}_t) - G_{t+1}(\boldsymbol{\theta}_{t+1}^\star) \leq L\|\boldsymbol{\theta}_t - \boldsymbol{\theta}_t^\star\|^2 + L\|\boldsymbol{\Delta}_t\|^2.$$

Taking expectations and summing over $t = 0, \ldots, T-1$ yields

$$\mathcal{R}_T^\pi(G) \leq L \sum_{t=0}^{T-1} \mathbb{E}\|\boldsymbol{\theta}_t - \boldsymbol{\theta}_t^\star\|^2 + L \sum_{t=0}^{T-1} \mathbb{E}\|\boldsymbol{\Delta}_t\|^2.$$

This completes the proof. $\qquad \square$

Using strong convexity, we can also convert between dynamic regret and tracking error:

**Lemma G.9** (Dynamic regret implies tracking). *Assume each $G_t : \Theta \to \mathbb{R}$ is $\mu$-strongly convex. Then for any policy $\pi$,*

$$\sum_{t=0}^{T-1} \mathbb{E}\|\boldsymbol{\theta}_t - \boldsymbol{\theta}_t^\star\|^2 \leq \frac{4}{\mu}\mathcal{R}_T^\pi(G) + 2\sum_{t=0}^{T-1} \mathbb{E}\|\boldsymbol{\Delta}_t\|^2.$$

*Proof of Lemma G.9.* Fix $t \in \{0, \ldots, T-1\}$. Since $G_{t+1}$ is $\mu$-strongly convex and minimized at $\boldsymbol{\theta}_{t+1}^\star$, we have

$$G_{t+1}(\boldsymbol{\theta}_t) - G_{t+1}(\boldsymbol{\theta}_{t+1}^\star) \geq \frac{\mu}{2}\|\boldsymbol{\theta}_t - \boldsymbol{\theta}_{t+1}^\star\|^2. \tag{241}$$

Now decompose $\boldsymbol{\theta}_t - \boldsymbol{\theta}_t^\star$ as follows: $\boldsymbol{\theta}_t - \boldsymbol{\theta}_t^\star = (\boldsymbol{\theta}_t - \boldsymbol{\theta}_{t+1}^\star) + (\boldsymbol{\theta}_{t+1}^\star - \boldsymbol{\theta}_t^\star) = (\boldsymbol{\theta}_t - \boldsymbol{\theta}_{t+1}^\star) - \boldsymbol{\Delta}_t$. Using the inequality $\|\boldsymbol{a} + \boldsymbol{b}\|^2 \leq 2\|\boldsymbol{a}\|^2 + 2\|\boldsymbol{b}\|^2$, we obtain

$$\|\boldsymbol{\theta}_t - \boldsymbol{\theta}_t^\star\|^2 \leq 2\|\boldsymbol{\theta}_t - \boldsymbol{\theta}_{t+1}^\star\|^2 + 2\|\boldsymbol{\Delta}_t\|^2.$$

Rearranging yields

$$\|\boldsymbol{\theta}_t - \boldsymbol{\theta}_{t+1}^\star\|^2 \geq \frac{1}{2}\|\boldsymbol{\theta}_t - \boldsymbol{\theta}_t^\star\|^2 - \|\boldsymbol{\Delta}_t\|^2.$$

Substituting into (241) gives

$$G_{t+1}(\boldsymbol{\theta}_t) - G_{t+1}(\boldsymbol{\theta}_{t+1}^\star) \geq \frac{\mu}{4}\|\boldsymbol{\theta}_t - \boldsymbol{\theta}_t^\star\|^2 - \frac{\mu}{2}\|\boldsymbol{\Delta}_t\|^2.$$

Taking expectations and summing over $t = 0, \dots, T-1$ yields

$$\mathcal{R}_T^\pi(G) \geq \frac{\mu}{4}\sum_{t=0}^{T-1}\mathbb{E}\|\boldsymbol{\theta}_t - \boldsymbol{\theta}_t^\star\|^2 - \frac{\mu}{2}\sum_{t=0}^{T-1}\mathbb{E}\|\boldsymbol{\Delta}_t\|^2.$$

Rearranging completes the proof. $\qquad\square$

**Corollary G.1** (Dynamic regret bound from the SGDM tracking estimate under uniformly spread gradient drift). *Assume the setting of Theorem 3.3. Write $c_{\gamma,\beta} = \exp(-\gamma\mu/(1-\beta))$. Suppose the conclusion of Theorem 3.3 simplifies to*

$$\mathbb{E}\|\boldsymbol{\theta}_t - \boldsymbol{\theta}_t^\star\|^2 \;\lesssim\; \frac{c_{\gamma,\beta}^t}{(1-\beta)^2}\|\boldsymbol{\theta}_0 - \boldsymbol{\theta}_0^\star\|^2 \;+\; \frac{\Delta^2}{\gamma^2\mu^2} \;+\; \frac{\sigma^2\gamma}{\mu(1-\beta)} \quad \textit{for all } t \geq 0, \tag{242}$$

*then we have the following:*

$$\mathcal{R}_T^\pi(G) \;\lesssim\; \frac{L}{(1-\beta)\gamma\mu}\|\boldsymbol{\theta}_0 - \boldsymbol{\theta}_0^\star\|^2 \;+\; \frac{LT\Delta^2}{\gamma^2\mu^2} \;+\; \frac{LT\sigma^2\gamma}{\mu(1-\beta)} \;+\; L\sum_{t=0}^{T-1}\|\boldsymbol{\Delta}_t\|^2. \tag{243}$$

*If, moreover, the gradient drift satisfies $v_t := \sup_{\boldsymbol{\theta}}\|\nabla G_{t+1}(\boldsymbol{\theta}) - \nabla G_t(\boldsymbol{\theta})\| \lesssim \mathbb{V}_T/T$ for all $t$, then $\|\boldsymbol{\Delta}_t\| \leq v_t/\mu \lesssim \mathbb{V}_T/(\mu T)$, yielding*

$$\mathcal{R}_T^\pi(G) \;\lesssim\; \frac{L}{(1-\beta)\gamma\mu}\|\boldsymbol{\theta}_0 - \boldsymbol{\theta}_0^\star\|^2 \;+\; \frac{L\mathbb{V}_T^2}{\gamma^2\mu^4 T} \;+\; \frac{LT\sigma^2\gamma}{\mu(1-\beta)} \;+\; \frac{L\mathbb{V}_T^2}{\mu^2 T}. \tag{244}$$

*Choosing $\gamma_\star \asymp ((1-\beta)\mathbb{V}_T^2/(\sigma^2\mu^3 T^2))^{1/3}$ gives the steady-state bound $\mathcal{R}_T^\pi(G) \lesssim L(1-\beta)^{-2/3}\sigma^{4/3}\mu^{-2}\mathbb{V}_T^{2/3}T^{1/3}$, up to initialization and lower-order terms.*

*Proof of Corollary G.1.* By Lemma G.8,

$$\mathcal{R}_T^\pi(G) \;\leq\; L\sum_{t=0}^{T-1}\mathbb{E}\|\boldsymbol{\theta}_t - \boldsymbol{\theta}_t^\star\|^2 \;+\; L\sum_{t=0}^{T-1}\|\boldsymbol{\Delta}_t\|^2. \tag{245}$$

Theorem 3.3 gives us $\mathbb{E}\|\boldsymbol{\theta}_t - \boldsymbol{\theta}_t^\star\|^2 \lesssim c_{\gamma,\beta}^t\|\boldsymbol{\theta}_0 - \boldsymbol{\theta}_0^\star\|^2/(1-\beta)^2 + \Delta^2/(\gamma^2\mu^2) + \sigma^2\gamma/(\mu(1-\beta))$ for all $t \geq 0$. Summing over $t = 0, \dots, T-1$ and substituting into (245) yields

$$\mathcal{R}_T^\pi(G) \;\lesssim\; \frac{L}{(1-\beta)^2}\Big(\sum_{t=0}^{T-1}c_{\gamma,\beta}^t\Big)\|\boldsymbol{\theta}_0 - \boldsymbol{\theta}_0^\star\|^2 \;+\; \frac{LT\Delta^2}{\gamma^2\mu^2} \;+\; \frac{LT\sigma^2\gamma}{\mu(1-\beta)} \;+\; L\sum_{t=0}^{T-1}\|\boldsymbol{\Delta}_t\|^2.$$

Since $\sum_{t=0}^{T-1}c_{\gamma,\beta}^t \leq (1-c_{\gamma,\beta})^{-1}$ with $c_{\gamma,\beta} = \exp(-\gamma\mu/(1-\beta))$, and $1 - c_{\gamma,\beta} \asymp \gamma\mu/(1-\beta)$ when $\gamma\mu/(1-\beta) \leq 1$, we have $(1-\beta)^{-2}(1-c_{\gamma,\beta})^{-1} \lesssim ((1-\beta)\gamma\mu)^{-1}$, proving (243).

Under the uniformly spread gradient-drift condition, optimality gives $\nabla G_t(\boldsymbol{\theta}_t^\star) = \nabla G_{t+1}(\boldsymbol{\theta}_{t+1}^\star) = \boldsymbol{0}$, so $\|\nabla G_{t+1}(\boldsymbol{\theta}_t^\star)\| = \|\nabla G_{t+1}(\boldsymbol{\theta}_t^\star) - \nabla G_t(\boldsymbol{\theta}_t^\star)\| \leq v_t$. By strong monotonicity, $\mu\|\boldsymbol{\theta}_t^\star - \boldsymbol{\theta}_{t+1}^\star\| \leq \|\nabla G_{t+1}(\boldsymbol{\theta}_t^\star) - \nabla G_{t+1}(\boldsymbol{\theta}_{t+1}^\star)\| = \|\nabla G_{t+1}(\boldsymbol{\theta}_t^\star)\|$, hence $\|\boldsymbol{\Delta}_t\| \leq v_t/\mu \lesssim \mathbb{V}_T/(\mu T)$. Taking $\Delta \asymp \mathbb{V}_T/(\mu T)$ gives $\sum_{t=0}^{T-1}\|\boldsymbol{\Delta}_t\|^2 \lesssim \mathbb{V}_T^2/(\mu^2 T)$. Substituting into (243) yields (244).

Finally, balancing the $\gamma$-dependent terms $L\mathbb{V}_T^2/(\gamma^2\mu^4 T)$ and $LT\sigma^2\gamma/(\mu(1-\beta))$ in (244) gives $\gamma^3 \asymp (1-\beta)\mathbb{V}_T^2/(\sigma^2\mu^3 T^2)$. Substituting $\gamma = \gamma_\star$ yields the balanced term $L(1-\beta)^{-2/3}\sigma^{4/3}\mu^{-2}\mathbb{V}_T^{2/3}T^{1/3}$, completing the proof. $\qquad\square$

