# OpenReview forum: "On the Provable Suboptimality of Momentum SGD in Nonstationary Stochastic Optimization"
_ICML.cc/2026/Conference — ICML 2026 regular_

### Official Review · Reviewer_fcHz · 2026-02-22

**Soundness:** 3
**Presentation:** 2
**Significance:** 2
**Originality:** 2
**Overall Recommendation:** 4
**Confidence:** 3

**Summary:**

This paper investigates SGD and momentum SGD in non-stationary stochastic optimization. The authors establish theoretical guarantees for strongly convex and non-convex smooth loss functions. The upper bounds indicate that momentum leads to worse results. The authors complement these upper bounds with minimax lower bounds. The experimental results validate the theoretical findings.

**Compliance With Llm Reviewing Policy:**

Affirmed.

**Final Justification:**

The authors addressed my concerns. Therefore I increased my score.

**Key Questions For Authors:**

1. The current theoretical analysis is primarily restricted to strongly convex and non-convex settings. Could the authors justify the decision to exclude general convex loss functions?
2. The authors claim that 'For a variety of strongly convex and nonconvex settings, it is widely known SGD with momentum (SGDM) provides faster convergence and better generalization compared to SGD without momentum'. However, previous work [1] proves that SGDM converges as fast as SGD for smooth objectives under both strongly convex and nonconvex settings. Could the authors clarify this?



__Reference__

[1] An Improved Analysis of Stochastic Gradient Descent with Momentum. NeurIPS, 2020.

**Limitations:**

yes

**Strengths And Weaknesses:**

__Strengths__
1. The paper investigates the convergence of SGD and momentum SGD in non-stationary environments for both strongly convex and non-convex functions.
2. The authors provide both high-probability upper bounds and minimax lower bounds for this problem.
3. Numerical experiments are included to supplement the theoretical claims.

__Weaknesses__
1. The motivation for studying non-stationary stochastic optimization in this specific context is poorly articulated.
2. In my opinion, the choice of the terminal parameter distance $\|\|\theta_ T - \theta^* _T\|\|$ as the primary performance measure is problematic for non-stationary optimization. Since the optimal parameter $\theta^*_t$ varies over time, this "last-iterate" focus only evaluates the algorithm's adaptation to the final environment, ignoring the cumulative error or tracking ability during the transition. A more robust metric, such as dynamic regret, would be far more appropriate to capture the algorithm's performance across the entire sequence of environments.
3. The explanation regarding the suboptimality of momentum SGD lacks technical novelty. The observation that the momentum term $\beta$ accumulates outdated gradient information is straightforward.
4. The paper is limited to a theoretical analysis of existing standard methods (SGD and Momentum SGD). It does not propose any new methods.

---

> ### Author Rebuttal · Authors · 2026-03-27
>
> Thank you for your careful reading and constructive feedback! We answer your comments below.
>
> **W1: Motivation for studying non-stationary optimization.**
>
> We appreciate the feedback and will clarify the motivation with examples to better contextualize our theoretical results. Our paper is motivated by a basic tension: momentum is valuable in stationary stochastic optimization because averaging past gradients reduces variance, but under distribution shift, averaging introduces bias (staleness) and tracking lag. This is important because momentum-based updates remain ubiquitous in practice, both explicitly (SGDM) and implicitly via widely used adaptive optimizers such as Adam, Adabelief, and Lion across drifting, data-dependent settings including policy optimization/RL, recommendation/ads/bandits, continual learning, and federated learning with non-stationary clients. In these problems, the population objective and gradient distribution evolve over time. Our work addresses the key question: when does temporal averaging help, and when does it instead degrade adaptation by relying on stale gradients? We aim to make this tradeoff precise in a nonstationary stochastic setting.
>
> **W2: Terminal parameter distance vs. dynamic regret.**
>
> We thank the reviewer for this comment! We agree that dynamic regret is a natural metric for non-stationary optimization. That said, our result is not intended as a purely last-iterate guarantee. The tracking bounds are formulated for a generic time index and are meant to control the trajectory uniformly over a finite horizon $t \in [T]$; we will revise the notation to make this explicit, since the display with $\|\|\theta_T-\theta_T^\star\|\|$ may have created the misleading impression that we only study the final iterate. Under our standing assumptions that each $G_t$ is uniformly strongly convex and $L$-smooth, the pointwise tracking bound also immediately implies a dynamic-regret guarantee, e.g. $R_T \lesssim L\sum_t \mathbb{E}\|\|\theta_t-\theta_t^\star\|\|^2 + L\sum_t \mathbb{E}\Delta_t^2$, where $\Delta_t := \|\|\theta_t^\star-\theta_{t+1}^\star\|\|$. We will make this explicit in the revision. We also note that dynamic regret already appears in our minimax lower-bound discussion.
>
> **W3: Suboptimality explanation lacks technical novelty.**
>
> We agree the qualitative intuition is classical: momentum stores information from past gradients, which may become uninformative under distribution shift. Our contribution is to give the first sharp nonasymptotic characterization of how this stale-history mechanism degrades SGDM in nonstationary stochastic optimization. Specifically, we derive a three-term decomposition into transient decay, drift lag, and noise floor, and show how momentum interacts quantitatively with both optimizer noise and a moving minimizer through explicit penalties of order $(1-\beta)^{-2}$ and $(1-\beta)^{-3}$. These terms are not apparent from the momentum update or the usual geometric-memory intuition. We further establish time-resolved high-probability upper bounds and the first minimax lower bound over the fixed-$\beta$ SGDM policy class. Thus, the novelty is not the intuition itself, but the first quantitative theory making its consequences precise in the nonstationary stochastic setting.
>
> **W4: No new methods proposed.**
>
> Our paper is primarily a theoretical contribution rather than an algorithmic one. We aim to precisely characterize when and why momentum breaks down under distribution shift, and extract principled mitigation strategies from the analysis. Due to space constraints, we refer to our responses to Reviewer NsSu (W2) and Reviewer TkqQ (Q1), where we discuss theorem-motivated restart, shift detection, memory-horizon tuning, and finite-memory windowing heuristics. We hope this motivates new drift-adaptive methods.
>
> **Q1: Why exclude general convex losses?**
>
> Our proof relies on the strict contraction $\|\|I-\eta_t H_t\|\|_{\mathrm{op}} \le 1-\eta_t\mu$ around a unique optimizer, which disappears when $\mu=0$. A general convex extension would therefore require a different argument, likely based on dynamic regret or averaged suboptimality against a moving comparator rather than pointwise tracking. Due to space constraints, we refer to our responses to Reviewer NsSu (Q1) and Reviewer JqDy (L1), where we discuss potential generalizations to less restrictive settings on convexity and smoothness.
>
> **Q2: Prior work [1] shows SGDM converges as fast as SGD; clarify the claim.**
>
> We agree our wording was too strong. Under broad stationary smooth assumptions, [1] shows SGDM matches SGD in order, not that it is uniformly faster. Our point was only that SGDM is widely used because it can be beneficial in practice and enjoys provable benefits in some settings, not that it dominates uniformly. We will revise the introduction accordingly.
>
> [1] Liu, Y., Y. Gao, and W. Yin. "An Improved Analysis of Stochastic Gradient Descent with Momentum." NeurIPS 33 (2020).

---

> > ### Author Rebuttal · Reviewer_fcHz · 2026-04-01
> >
> > I would like to thank the authors for their detailed and thoughtful rebuttal. The responses have successfully addressed my initial concerns and have increased my score to support the acceptance of this paper.
> >
> >
> > As a suggestion for the revised version, I recommend expanding the discussion on dynamic regret. For instance, it would be beneficial to briefly discuss existing literature on dynamic regret bounds in dynamic environments, such as [1] which first established the optimal dynamic regret bound and lower bound.
> >
> > __References__
> >
> > [1] Adaptive Online Learning in Dynamic Environments. NeurIPS, 2018.

---

### Official Review · Reviewer_JqDy · 2026-03-05

**Soundness:** 4
**Presentation:** 3
**Significance:** 3
**Originality:** 4
**Overall Recommendation:** 5
**Confidence:** 3

**Summary:**

This paper provides a theoretical analysis of the tracking performance of SGD and its momentum variants (Heavy-Ball and Nesterov) in non-stationary environments. The study decomposes the tracking error into three distinct components: transient part, noise-induced error, and drift-induced error. It details convergence rates in expectation, establishes high-probability bounds, and derives minimax lower bounds. A core finding is that while momentum is conventionally used to smooth gradient noise, it incurs a significant drift amplification penalty in the presence of distribution shifts, resulting in a systematic tracking lag. Finally, experimental results validate these theoretical findings.

**Compliance With Llm Reviewing Policy:**

Affirmed.

**Final Justification:**

My questions have been addressed. I recommend the acceptance.

**Key Questions For Authors:**

1. The convergence rates presented for SGDM (e.g., in Theorem 3.3) seem loose when reduced to the deterministic, stationary case. Specifically, if we set the noise $\sigma = 0$ and the drift $\Delta = 0$, the resulting rate is noticeably slower than the optimal accelerated rate known for deterministic settings. Could the authors clarify the theoretical bottlenecks preventing the recovery of the tight deterministic rate?


2. In the proof of Theorem D.1, the transition from "Bounding part (b)" to "Concluding the bound" appears to implicitly rely on mathematical induction over the time step $t$. If so, it is strongly recommended to explicitly state this induction step; otherwise, it reads as a logical leap. The exact same issue also seems to be present in the proof of the high-probability bound for SGDM.

**Limitations:**

The theoretical analysis is confined to strongly convex objectives. It remains unknown whether the observed drift amplification penalty and the suboptimality of momentum extend to general convex or non-convex settings.

**Strengths And Weaknesses:**

Strengths:

This is a technically solid paper with rigorous theoretical foundations. The authors provide a clear proof sketch that effectively guides the reader through the mathematical derivations. The study offers a comprehensive investigation into momentum by analyzing both Heavy-Ball and Nesterov variants. Furthermore, the theoretical contributions are rich, providing convergence rates in expectation, high-probability bounds, and minimax lower bounds. The precise formulations of the rates allows for an intuitive analysis of each factor's individual impact, as well as the transient time.

Weaknesses:
1. The time-varying distribution shift setting investigated in this paper appears relatively uncommon in conventional optimization literature. To strengthen the motivation, it would be beneficial if the authors could provide concrete, real-world examples (rather than simplified toy examples) to demonstrate the potential application scenarios and the practical relevance of this specific setting.

2. The theoretical analysis yields identical convergence rates for both HB and Nesterov momentum. This appears somewhat counterintuitive and misaligned with existing classical optimization theory, where Nesterov's accelerated gradient typically achieves a strictly better rate than HB (at least in the noiseless, strongly convex setting). Does this suggest that the derived bound for Nesterov momentum is loose and has room for further improvement?

---

> ### Author Rebuttal · Authors · 2026-03-27
>
> Thank you for your careful reading and constructive feedback! We answer your comments below.
>
> **W1: Motivation for the time-varying distribution shift setting.**
>
> We agree the paper would benefit from more concrete motivation. This setting arises naturally in several domains: (i) policy optimization/RL, where the data distribution is the state-action occupancy measure induced by the current policy and may also change with a drifting environment, so each update is computed from a different underlying distribution; (ii) online recommendation/ads/bandits, where user populations, click-through behavior, and feedback distributions drift over time due to seasonality, user churn, evolving preferences, etc.; (iii) continual learning, where the task or domain changes over time and the model must track a moving target without catastrophic forgetting; and (iv) federated learning with non-stationary clients, where the set of participating devices and their local data distributions vary across rounds. In all of these, the gradient noise distribution and the population objective vary with time, which is exactly the regime modeled by our time-varying $\Pi_t$. We will add these examples in the revised manuscript.
>
> **W2: Why do HB and Nesterov have the same rate? Is the Nesterov bound loose?**
>
> Thank you for this thoughtful comment! Our bounds are designed for the stochastic, drifting, strongly convex regime, where the main difficulty lies in tracking a moving optimum under noise and stale-gradient memory. In that regime, both HB and Nesterov induce a similar two-state recursion, and our Lyapunov analysis yields the same $(1-\beta)^{-2}$, $(1-\beta)^{-3}$ nonstationary penalties for both. The Nesterov bound is likely loose: our proof does not exploit the estimate-sequence / intermediate-iterate structure that gives Nesterov its sharper deterministic accelerated rate. Sharpening the Nesterov analysis to separate it from HB in the nonstationary stochastic setting is an important direction for future work. We will clarify this explicitly.
>
> **Q1: Why does the theorem not recover the tight deterministic accelerated rate when noise/drift are zero?**
>
> This is an excellent observation! The main bottleneck is the proof technique. Our analysis views momentum through a 2D state-space / robust Lyapunov recursion, which is well-suited to drift and noise but does not exploit the estimate-sequence / intermediate-iterate structure that yields deterministic acceleration. The 2D reduction treats $\theta_t$ and $\psi_t$ symmetrically as a coupled system, which is natural for bounding stochastic perturbations and drift but sacrifices the fine-grained iterate structure needed for tight acceleration results. Thus when $\sigma=0$ and $\Delta=0$, our theorem does not recover the optimal accelerated rate.
>
> Encouragingly, [1] develops a generalized SGDM analysis recovering the optimal $O(1/t^2)$ deterministic rate and an $O(1/t^2 + \sigma/\sqrt{t})$ stochastic rate, using time-varying parameters and Lyapunov functions built on intermediate iterates. This makes us optimistic that similar ideas could sharpen our deterministic, stationary case while preserving the main nonstationary conclusions. We will make this explicit in the revised manuscript.
>
> **Q2: Missing induction step in Theorem D.1.**
>
> Thank you for catching this! The transition from "Bounding part (b)" to "Concluding the bound" does rely on an induction over the time index, and we agree it should be stated explicitly. We will revise the proof of Theorem D.1 to make the induction hypothesis and closing step fully explicit, and will do the same for the analogous high-probability SGDM argument.
>
> **L1: Analysis confined to strongly convex objectives.**
>
> Our current theory requires uniform strong convexity because the pointwise tracking analysis relies on the contraction
> $\|\|I-\eta_t H_t\|\|_{\mathrm{op}} \le 1-\eta_t\mu$.
> When $\mu=0$, this contraction disappears and the minimizer may be non-unique, so the argument does not extend verbatim. That said, we expect the qualitative inertia penalty to persist more broadly. Under weaker conditions such as the Polyak--Łojasiewicz (PL) condition, quadratic growth, or local strong convexity along the optimization path, the same Lyapunov-based proof strategy should extend with additional technical work. In convex smooth settings, the natural metric becomes dynamic regret or averaged suboptimality against a moving comparator. In non-convex settings, the literature studies stationarity-based measures, dynamic regret under additional assumptions, and cumulative nonconvexity regret. We believe the same two-state Lyapunov recursion can be adapted using potential-based rather than contraction-based arguments. Extending our analysis to these settings is an important direction for future work.
>
> [1] Wang, Zimeng, and Alp Yurtsever. "Generalized Stochastic Gradient Descent with Momentum Methods for Smooth Optimization." arXiv preprint arXiv:2602.23444 (2026).

---

> > ### Author Rebuttal · Reviewer_JqDy · 2026-04-01
> >
> > Thanks for the response. I have no further comments and will keep my recommendation to accept.

---

### Official Review · Reviewer_TkqQ · 2026-03-09

**Soundness:** 3
**Presentation:** 3
**Significance:** 3
**Originality:** 3
**Overall Recommendation:** 5
**Confidence:** 3

**Summary:**

This paper sutdies the momentum SGD (which is one of the most popular algorithms widely used in modern machine leanring). It specifically focuses on the nonstationary stochastic optimization setting. The analysis indicates that the distribution shift may potentially result in suboptimality of momentum SGD. Numerical experiments are taken to validate theoretical results.

**Compliance With Llm Reviewing Policy:**

Affirmed.

**Final Justification:**

As I have mentioned in the review, I will keep my positive rating for supporting this submission.

**Key Questions For Authors:**

1. What is the solution proposed in this paper to reduce the negative impact from distribution shift? In Contribution 2, the author states "motivating restart and windowing schemes that discard stale history"; is it just a comment or a well-supported statement?

2. All theoretical results are for the strongly convex function class. Is there any function class in which the momentum SGD is not hurted by distribution shift?

3. How can these theroetical results be extended to Adam or AdamW? These algorithms might be more popular in recent years.

---

Overall, I am satisfied with this submission and I rated 5 for a clear acceptance. Those questions are just for helping me further understand the overall picture of this work and they don't affect my score.

**Limitations:**

yes

**Strengths And Weaknesses:**

Soundness: By intuition, the momentum is obtained from the past trajectory and it is used to update the current parameter; therefore, it may potentially result in suboptimility. This main result sounds to me.

Presentation: This paper is well written. The problem setup is clearly described at the earliest stage of this paper. All concepts are clearly defined. The whole paper is orgnanzed well, which helps me understand where to find the most important results.

Significance: As Momentum SGD is widely used in machine learning, it is nearly the default choice in all cases. It is of great significance to identify the suboptimility of using this method in the non-stationary stochastic optimization case. Therefore, I believe this work presents a significant result and deserves the attention from the ML community.

Originality: It may not be surprised that the Momentum SGD doesn't perform perfectly well in some specific setting. But this paper indeed presents the first result in showing the suboptimility of this algorithm in the nonstationary stochastic optimization setting, which is new in this field.


**Weaknessess**: I didn't find any major weaknesses for this paper.

---

> ### Author Rebuttal · Authors · 2026-03-27
>
> Thank you for your careful reading and constructive feedback! We answer your comments below.
>
> **Q1: Is the restart/windowing suggestion well-supported or just a comment?**
>
> While we do not have a comprehensive solution that completely eliminates the negative effects of distribution shift, we can suggest a few practical guidelines. The statement regarding restart is well supported, with one important distinction. Theorem 3.4 studies a step-decay schedule that restarts the momentum buffer at epoch boundaries ($v=0$), removing stale velocity and provably reaching the tracking floor in finite time. The burn-in time after restart scales as $(1-\beta)/(\mu\gamma)$, making the cost of stale momentum quantitatively explicit.
>
> Windowing/forgetting is strongly motivated by the high-probability bounds: Theorem 3.5 contains exponentially weighted drift terms, so older perturbations are geometrically discounted and only recent drift materially affects the guarantee. This structure directly motivates truncating or downweighting stale history after regime changes. In practice, one can replace long-horizon EMA with a sliding window of the most recent $H \sim (\gamma\mu)^{-1}$ gradients, retaining variance reduction while limiting inertia.
>
> Additionally, our bounds suggest interpreting momentum through its effective memory length $\sim(1-\beta)^{-1}$ versus the adaptation timescale $\sim(\gamma\mu)^{-1}$. Rather than tuning $\beta$ in isolation, one can set $\beta \approx 1-1/H$ where $H$ is an explicit memory horizon that should not exceed $O((\gamma\mu)^{-1})$; otherwise, momentum averages gradients that are no longer informative. The stability condition $\gamma \le \mu(1-\beta)^2/(4L^2)$ from Theorem 3.3 further shows that large $\beta$ forces a smaller admissible stepsize. For shift detection, one can monitor $S_t := 1 - \frac{\langle \nabla g(\psi_t, X_{t+1}), v_t \rangle}{\|\|\nabla g(\psi_t, X_{t+1})\|\| \cdot \|\|v_t\|\| + \epsilon}$ with $\epsilon > 0$ and reset $v_t \leftarrow 0$ when $S_t$ remains large for several consecutive iterations. We view these detection and tuning rules as practical heuristics motivated by the theory, and will include them in the revised manuscript.
>
> **Q2: Is there any function class where momentum SGD is not hurt by distribution shift?**
>
> This is a great question! To our knowledge, no broad structural function class exists for which momentum is generically unaffected by distribution shift. The stale-history mechanism underlying our result is not specific to strong convexity; rather, strong convexity is what allows us to make the effect explicit via tracking bounds. The core issue is that any method placing nontrivial weight on past gradients will likely incur lag after a regime shift, because those gradients become stale.
>
> We expect related penalties to persist in broader classes (convex, Polyak–Łojasiewicz (PL) condition, Hölder-smooth, weakly convex, nonconvex), although the quantitative scaling and the appropriate performance metric (e.g., dynamic regret rather than pointwise tracking) may differ outside the strongly convex setting. Under PL or quadratic growth conditions, we believe our Lyapunov proof strategy extends with modifications; in the convex case ($\mu=0$), the uniform contraction $\|\|I - \eta_t H_t\|\|_{\mathrm{op}} \le 1-\eta_t\mu$ disappears, so the natural metric becomes dynamic regret against a moving comparator. We believe the core Lyapunov recursion structure, particularly the two-state coupling between iterates and momentum buffer, can be adapted using potential-based arguments in place of the contraction-based bounds we currently use.
>
> **Q3: Extension to Adam/AdamW.**
>
> We believe our tracking-error framework can be extended to Adam/AdamW. These optimizers also use EMA memory of past gradients, so they should inherit the same stale-history effect under drift. The main difference is that Adam/AdamW add adaptive preconditioning via a second moment state, so the analysis would require an enlarged Lyapunov recursion tracking both (i) the first-moment EMA ($\beta_1$), which induces a momentum-like lag penalty, and (ii) the second-moment EMA ($\beta_2$), which can introduce additional inertia through the running average of squared gradients. We expect analogous nonstationary penalties involving the effective memory lengths $(1-\beta_1)^{-1}$ and $(1-\beta_2)^{-1}$, and note that Adam's bias correction $1/(1-\beta^t)$ effectively shortens the memory early after initialization or restart, which may partially mitigate the lag penalty in the transient phase.
>
> For AdamW specifically, it remains unclear how decoupled weight decay interacts with these drift penalties, as the regularization acts directly on the iterates rather than the gradient, potentially offering a mild stabilizing effect against drift. Formalizing these extensions, particularly disentangling the contributions of the two EMA timescales to the overall tracking error, is an important direction for future work.

---

> > ### Author Rebuttal · Reviewer_TkqQ · 2026-03-31
> >
> > I appreciate the author's response. As I have mentioned in the review, I will keep my positive rating for supporting this submission.

---

### Official Review · Reviewer_NsSu · 2026-03-11

**Soundness:** 4
**Presentation:** 3
**Significance:** 3
**Originality:** 3
**Overall Recommendation:** 5
**Confidence:** 4

**Summary:**

This paper studies SGD with momentum under nonstationary stochastic optimization, where the objective drifts over time and the goal is to track a moving minimizer rather than converge to a fixed one. The primary contribution is that the authors proved plain SGD achieve a natural tracking bound with three terms: transient bias, variance and drift. For momentum methods (heavy-ball/Nesterov), the paper shows these bounds worsen by explicit factors depending on $(1-\beta)^{-1}$, quantifying the idea that momentum introduces lag under drift. The paper also gives high-probability bounds and lower bounds suggesting this slowdown is not just a proof artifact. Empirical results on drifting quadratics, linear/logistic regression, and a small neural network are consistent with the theory.

**Compliance With Llm Reviewing Policy:**

Affirmed.

**Final Justification:**

The paper is well motivated with significant theoretical contributions. The authors have addressed all my concerns in the rebuttal. Therefore, I will keep my score of 5 - Accept, to support their publication.

**Key Questions For Authors:**

(1) Related to my comments above, could the authors discuss the potential generalization to less restrictive settings on convexity and smoothness?

(2) Can the conditional sub-Gaussian assumption be further relaxed?

(3) For the suboptimality discovered, is it just for the standard fixed-parameter momentum, or for general momentum?

**Limitations:**

Yes.

**Strengths And Weaknesses:**

Strengths:

(1) The problem is well motivated, which aims to extend the standard momentum theory to nonstationary learning settings;

(2) The theoretical parts are solid and clear. I do appreciate the explicit role of momentum which is quantified explicitly through $(1-\beta)$-dependent penalties;

(3) The lower bounds, different from the typical upper bounds arguments in the literature, makes the negative findings quite convincing;

(4) Beyond the asymptotic sense, the authors also provided finite-time analysis;

(5) The experiments are fruitful and consistent with the theoretical findings.

Weaknesses:

(1) My major concern is that the current manuscript is somewhat restrictive in its scope, because the main theorems are derived under the conditions of uniform strongly monotone/smooth drifting objectives, which seem hard to extend the current conlusions to modern deep nonconvex training;

(2) It might be beneficial to suggest some practical guidance given the derived theory. For example, how should people do the restart, decaying momentum, or forgetting training etc., such that the problem diagnosed can be somewhat eliminated;

(3) The paper focuses on the fixed $\beta$ and fixed-step momentum. Will the lower bounds derived also applicable to other adaptive momentum schemes, or maybe there exists some other schemes that can avoid this issue?

---

> ### Author Rebuttal · Authors · 2026-03-27
>
> Thank you for your careful reading and constructive feedback! We answer your comments below.
>
> **W1: Restrictive scope (uniform strong monotonicity/smoothness).**
>
> We agree that extending beyond globally strongly convex objectives is important. Recent work suggests our analysis is likely more broadly applicable: [1] proves regularized ReLU network losses are piecewise strongly convex on open sets containing all global minimizers, and empirically shows SGD trajectories on standard architectures (LeNet-5, ResNet22) reside in such regions for nearly the entire training run. [2] similarly shows efficient minimizers of shallow analytic networks admit locally strongly convex neighborhoods. Once optimization enters such regions, the loss is locally well-approximated by a $(\mu,L)$-objective, and our three-term decomposition with the momentum penalties $(1-\beta)^{-2}$, $(1-\beta)^{-3}$ should provide useful local guidance. Formalizing this by combining our tracking framework with piecewise strong convexity guarantees is a compelling future direction.
>
> **W2: Practical guidance from the theory.**
>
> Thank you for this suggestion! Below are theorem-motivated heuristics rather than formal guarantees:
> 1. **Restart after regime shifts.** Theorem 3.4 shows burn-in scales as $(1-\beta)/(\mu\gamma)$, so stale momentum can persist for many iterations. Use $S_t := 1 - \frac{\langle \nabla g(\psi_t, X_{t+1}), v_t \rangle}{\|\|\nabla g(\psi_t, X_{t+1})\|\| \cdot \|\|v_t\|\| + \epsilon}$ with $\epsilon > 0$ and restart by setting $v_t \leftarrow 0$ when $S_t$ remains large for several consecutive iterations.
> 2. **Choose $\beta$ via a memory horizon.** Set $\beta \approx 1-1/H$ where $H \lesssim O((\gamma\mu)^{-1})$. The stability condition $\gamma \le \mu(1-\beta)^2/(4L^2)$ from Theorem 3.3 shows large $\beta$ forces a smaller admissible stepsize; reducing $\beta$ permits a larger stable stepsize.
> 3. **Finite-memory averaging.** Theorems 3.5–3.6 show recent drift matters most. Replace long-horizon EMA with window averaging over $H \sim (\gamma\mu)^{-1}$ recent gradients, reducing inertia while retaining variance reduction.
>
> We will add this in the revision.
>
> **W3/Q3: Applicability to adaptive momentum and general momentum suboptimality.**
>
> Our lower bound and upper bounds are for fixed-$\beta$ SGDM, and the penalties $(1-\beta)^{-2}$, $(1-\beta)^{-3}$ are specific to fixed geometric memory. These do not directly cover adaptive schemes with time-varying $\beta_t$, bias correction, or state-dependent weighting. However, we believe any method weighting past gradients will likely incur lag when those gradients become stale. Adaptive schemes (e.g., Adam's bias-corrected EMA, SGDF [3]) might mitigate but likely cannot fully eliminate this by shortening effective memory. Our restart result (Theorem 3.4) also shows that explicitly clearing stale history improves adaptation. We conjecture the lower bound would depend on the method's effective averaging horizon at each shift rather than the fixed window $(1-\beta)^{-1}$. Extending the lower bound to general and adaptive momentum is an interesting open direction.
>
> **Q1: Generalization beyond strong convexity/smoothness.**
>
> Our Lyapunov argument uses $\|\|I - \eta_t H_t\|\|_{\mathrm{op}} \le 1-\eta_t\mu$, requiring strong convexity. Under weaker conditions (Polyak–Łojasiewicz (PL) condition, quadratic growth, local strong convexity along the optimization path), our proof strategy should extend with modifications. For convex settings ($\mu=0$), this contraction disappears and the minimizer may be non-unique, so the natural metric becomes dynamic regret against a moving comparator. We believe the core Lyapunov recursion structure, particularly the two-state coupling between iterates and momentum buffer, can be adapted using potential-based arguments in place of the contraction-based bounds we currently use.
>
> **Q2: Relaxing sub-Gaussian noise.**
>
> Yes. Under sub-exponential ($\Psi_1$) noise, concentration in Theorems 3.5–3.6 changes from $\sigma^2$-scaling to mixed Bernstein form, and $\log(2T/\delta)$ becomes $\max\\{ \log(2T/\delta), \log^2(2T/\delta) \\}$. Under bounded $p$-th moment noise ($p>2$), one gets polynomial confidence bounds of order $O(T^{1/p}/\delta^{1/p})$ via Burkholder/Freedman-type inequalities. Crucially, the three-term decomposition and momentum penalties $(1-\beta)^{-2}$, $(1-\beta)^{-3}$ should remain unchanged. We adopted sub-Gaussian as standard in the tracking literature and will add a remark on these relaxations.
>
> [1] Milne, Tristan. "Piecewise strong convexity of neural networks." Advances in neural information processing systems 32 (2019).
>
> [2] Benning, Felix, and Steffen Dereich. "In almost all shallow analytic neural network optimization landscapes, efficient minimizers have strongly convex neighborhoods." arXiv preprint arXiv:2504.08867 (2025).
>
> [3] Yao, Zhipeng, et al. "Signal processing meets SGD: From momentum to filter." arXiv preprint arXiv:2311.02818 (2023).

---

> > ### Author Rebuttal · Reviewer_NsSu · 2026-03-31
> >
> > Thank you very much for addressing my concerns. The proposed revisions sound good, especially the ones on the potential relaxation of the current conditions. Please add them to the final version to strength the paper. I will raise my score to support your publication.

---

### Decision · Program_Chairs · 2026-04-30

**Decision:**

Accept (regular)

**Comment:**

This paper theoretically analyzes the tracking performance of SGD and its momentum variants in non-stationary settings. It decomposes the tracking error into transient, noise-induced, and drift-induced components, and derives expectation convergence rates, high-probability bounds, and minimax lower bounds. A key result shows that although momentum helps smooth gradient noise, it amplifies drift under distribution shifts, leading to persistent tracking lag.

The paper is well written, well motivated and the empirical results validate the finding. All reviewers agreed that the paper merits acceptance, and after carefully reviewing the rebuttal and discussion, I share this view. That said, I recommend that the authors incorporate the reviewers’ feedback into the final version of the paper, with particular attention to:

1- Add read world examples to make to motivation section more stronger

2- Discuss about the possibility of using dynamic regret as the sub optimality measure